

# Monodromies of CFT correlators on the Lorentzian cylinder

**Suman Kundu[1⋆], Shiraz Minwalla[2†] and Abhishek Navhal[2‡]**

**1** Department of Particle Physics and Astrophysics,
Weizmann Institute of Science, Rehovot 76100, Israel
**2** Department of Theoretical Physics, Tata Institute of Fundamental Research,
1, Homi Bhabha Road, Mumbai 400005, India

⋆ suman.kundu@wiezmann.ac.il , † shiraz.minwalla@gmail.com , ‡ abhisheknavhal@gmail.com

## Abstract

While correlators of a CFT are single valued in Euclidean Space, they are multi-valued - and have a complicated sheet structure - in Lorentzian space. Correlators on $R^{1,1}$ are well known to access a finite number of these sheets. In this paper, we demonstrate the spiral nature of lightcones on $S^1 \times$ time, which allows the time-ordered correlators of a $CFT_2$ on this spacetime - the Lorentzian cylinder - to access an infinite number of sheets of the correlator. We present a complete classification, both of the sheets accessed as well as of the various distinct causal configurations that lie on a particular sheet. Our construction provides a physical interpretation for an infinite number of sheets of the correlator, while, however, leaving a larger infinity of these sheets uninterpreted.



# 1  Introduction

Whereas correlation functions in a conformal field theory are single-valued in Euclidean space, they are usually multi-valued in Lorentzian space (see e.g. [1–5]). Consider, for example, a four-point function $C$ of four primary operators in a two-dimensional CFT. After division by a suitable normalization factor $N$ (that accounts for the conformal transformation properties of the inserted operators)[1] such a correlator takes the form [6, 7]

$$C/N = \sum_{ij} G^i(z) P_{ij} \bar{G}^j(\bar{z}), \tag{1}$$

where the conformal blocks $G^i(z)$ and $\bar{G}^j(\bar{z})$ (see e.g. [8] and references therein), are, respectively, holomorphic functions of their arguments, the left and right moving conformal cross-ratios $z$ and $\bar{z}$ (defined in eq (25), (26)) respectively. The matrix of numbers, $P_{ij}$, constitutes the "pairing matrix" that glues these blocks together. The summation over $i$ and $j$ run over a finite range when the theory under study is rational, but over an infinite set of values when the theory is irrational.[2] As the conformal blocks $G^i$ and $\bar{G}^j$ generically have branch cuts at $z$ (or $\bar{z}$) $= (0, 1, \infty)$, $C/N$ is, generically, a multi-valued (branch covered) holomorphic function of the two independent complex variables $z$ and $\bar{z}$.

The (two complex dimensional) space spanned by $z$ and $\bar{z}$ has at least two physically interesting (two real dimensional) sections, the Euclidean and the Lorentzian sections. The Euclidean section is obtained by setting $\bar{z} = z^*$ (i.e. by setting $z$ and $\bar{z}$ to be complex conjugates of each other). On this section, $C$ computes correlators of the field theory in Euclidean space. The single valuedness of these correlators - hence of $C$ evaluated on the Euclidean sheet - imposes stringent constraints on the pairing matrix $P_{ij}$ in (1) (see, for example, the discussion in section 15.4 in [9]).

Lorentzian correlators, in contrast, are obtained by evaluating $C/N$ at (generically distinct) real values of $z$ and $\bar{z}$. Unlike their Euclidean counterparts, however, correlators on this "Lorentzian section" are generically multivalued. It is, for instance, possible to start with a real

---

[1]In the special case where operator dimensions are equal pairwise, $N$ is proportional to the product of the two-point functions of the pairs. The normalization is fixed by operator dimensions and contains no dynamics. Though $N$ is multivalued, its branching is simple (see §2).

[2]While the arguments presented in this paper is most rigorous in the former case, we also expect our results to apply to irrational theories.

value of $z = \bar{z}$ (such a configuration lies both on the Lorentzian and the "Euclidean Sheet"), circle the branch cut at $z = 1$ (without making any corresponding move in $\bar{z}$ - this is consistent because $z$ and $\bar{z}$ are independent variables on the Lorentzian sheet) and return to the original real value of $z$. This operation changes the value of the correlator $C/N$. One obtains a definite value for the correlators on the Lorentzian section only after specifying both $z$ and $\bar{z}$ *and a choice of sheet*. It is natural to wonder about the physical interpretation of the function $C/N$ evaluated on each of the infinitely many sheets of the Lorentzian section.

In simple cases (i.e. for sheets that are not too far from the Euclidean sheet, see below), the answer to this question is well understood (see e.g. [1–5] and [10–12] for relevant older literature) and references therein). Operators that are timelike separated do not commute with each other (of course such a separation is impossible in Euclidean space). Consequently, one finds different answers for the correlators of given operators, inserted at given locations, depending on the ordering of the operators; equivalently, on the choice of different Schwinger-Keldysh contours on which these correlators are evaluated. All such correlators are given by the function $C(z, \bar{z})$ - however, they are evaluated on distinct sheets. This yields a beautiful physical interpretation for a small finite number of the sheets ( $\leq n!$ in the case of $n$ point functions) of the infinite number of sheets of the Lorentzian section.

In this paper we generalize the discussion of [1,5] to find a simple physical interpretation for a larger number - this time an infinite number - of the Lorentzian sheets of the correlator $C/N$. Unfortunately our results are not exhaustive. They still leave a larger infinity of sheets un-interpreted.

We now proceed to describe the simple construction that yields this generalization.

## 1.1 An infinite number of sheets from time-ordered correlators on the Lorentzian cylinder

As is familiar from the study of Penrose diagrams [13], the space $R^{1,1}$ (two dimensional Minkowski space) is Weyl equivalent to a finite diamond of $R^{1,1}$, with horizontal vertices identified[3] (see around Fig 1a). As such a Minkowskian diamond has boundaries at a finite distance, it is an incomplete spacetime. The dynamics of a conformal field theory on this spacetime is well posed only upon the specification of boundary conditions. Alternately, one could avoid specifying boundary data by working, instead, with the maximal analytic continuation of the Minkowski diamond, i.e. Minkowskian cylinder $S^1 \times$ time. This analytic continuation is conceptually similar to the maximal analytic continuation of the Schwarzschild geometry to the Kruskal geometry. (The Hilbert space of the theory on the maximally analytically continued cylinder spacetime famously reproduces the Hilbert space of the radial quantization of the Euclidean theory.) We now explain how time-ordered correlators on this maximally extended spacetime explore an infinite number of sheets of $C/N$.

Consider two operators $A$ and $B$ that are initially spacelike separated. By varying the insertion location of the operator $B$, we can (if we choose) move it so that it cuts the future/past light cone of $A$. As this happens, one moves either over or under a branch point of the correlator $C/N$ (whether over or under is determined by the $i\epsilon$ prescription. See §3.2 for details). When $A$ and $B$ both live in Minkowski space one can execute each of these manoeuvres no more than once; thereby taking $B$ to the causal future/past of $A$ (of course zig-zags - that take $B$ across the lightcone of $A$ from past to future, then back, from future to past - undo each other and do not take us to new sheets of the correlator). Clearly, ranging over these possibilities -for each pair of operators - allows us to access only a finite number of sheets of the correlator

---

[3]In the language of Penrose diagrams, the identified points are "left" and "right" spatial infinity. The equivalent identification, in spacetime dimensions $d > 2$, is the now famous "antipodal identification" of the celestial holography programme (see e.g. [14–16] for reviews).

$C/N$. In §5 we present a complete enumeration of these sheets and a complete classification of all the causally inequivalent configurations that lie on any given sheet.

As we have explained above, to study conformal dynamics, the spacetime $S^1 \times R$ may be thought of as the maximal analytic continuation of $R^{1,1}$. Indeed, $S^1 \times R$ may be thought of as being constructed by patching together an infinite number of Minkowskian diamonds (see Fig. 1b and e.g. around Fig.1 in [4] ). On this much larger spacetime, the future (and past) lightcones that emanate from operator $A$, each form two counter-rotating spirals around the cylinder. As we move $B$ to the future, this operator can cross the future lightcone of $A$ any number of times. In other words, the notion of "causality" can be refined on a Lorentzian cylinder: if we know that $B$ lies in the future of $A$, we can seek more detail and ask "How many $A$ lightcones are cut if one starts with $B$ spacelike located with respect to $A$ and then move to the desired location?". Each time $B$ crosses yet another swirl of the left or right moving future lightcone of $A$, the correlator $C/N$ passes over/under a branch point of the correlator. Under favourable conditions (see §6 for details) repeated crossings etch out a path in $z$ and $\bar{z}$ space that repeatedly winds around a branch point. As the number of windings can be arbitrarily large, it follows that time-ordered four-point functions on $S^1 \times$ time access an infinite number of sheets of the function $C/N$.

## 1.2 The sheet for a time ordered correlator for given insertions on the cylinder

Consider the time ordered correlator $\langle O_1(x_1) \dots O_4(x_4) \rangle$, where $x_i$ ($i = 1 \dots 4$) are insertion locations on the Lorentzian cylinder. The cross-ratios $z$ and $\bar{z}$, associated with the insertion locations $x_i$, are easily worked out (see the formulae in §3.1). It follows that our correlator is given by the function $C$ in (1) evaluated at the given values of $z$ and $\bar{z}$ on *some* sheet of this multivalued function. What is not immediately clear is which sheet this correlator lies on.

In this paper, we use the following simple procedure to answer this question. We first insert all operators at some arbitrarily chosen locations on the same spatial slice of the cylinder. At these insertion locations, the time-ordered correlator is the same as the Euclidean correlator, and so lies on the "Euclidean sheet" of the function $C/N$. We then continuously deform all insertion locations from their arbitrarily chosen starting points to the actual final desired locations $x_1 \dots x_4$. In the process, the inserted operators cross several light cones emanating out of the other operator insertions. Consequently, we traverse a path in $z, \bar{z}$ space that passes over (and under) several branch points. By keeping careful track of these crossings, we deduce the final location (in sheet space) that we reach when our insertion locations finally reach the desired endpoints $x_1 \dots x_4$.

The procedure described above has a potential ambiguity, as there are many causally inequivalent paths leading from the (arbitrarily chosen) insertion locations to the final locations of interest. We could, for instance, first move $O_1$ then $O_2 \dots$, or first move $O_3$ halfway to its final destination, then $O_1$ then $O_2$ halfway then $O_4$ .... Each of these distinct choices takes us along a distinct trajectories in sheet space. Below we carefully demonstrate that each of the different choices above yields the same final result for the location on the sheet space of the correlator. This general result is both a simplification as well as a disappointment. It is a simplification as it eases the computation of the sheet location associated with any particular physical location of operators. It is a disappointment because it tells us that operators at all possible locations on the Lorentzian cylinder only access a small fraction of the much larger infinity of available locations in sheet space (this larger infinity has to do with the non-abelian nature of sheet moves around distinct branch points, which our physical situation never explores, precisely because causally distinct paths that lead to the same final configuration, are all associated with the same monodromy). It is also a disappointment because it tells us that the requirement of "path independence" is automatic, and imposes no new general constraints on four point functions of operators with integer spins.

### 1.3 Concrete results of this paper

While the main focus of this paper is on the study of four-point functions, as a warm-up we first study two and three-point functions. Like the four-point function, these correlators (whose form is completely determined by conformal invariance) are also multi-valued in Lorentzian space: the 'sheet ambiguity" of these correlators lies in their phase. In section 2 we give a clear and simple rule that determines the phase of the 2 and 3 point functions for any given insertion locations on the Lorentzian cylinder.

Turning to the study of four-point functions, in section §6 we implement the procedure of §1.2 to derive rules that allow one to determine the sheet location associated with any given insertion locations on the Lorentzian cylinder §6.4. Our final result is given in terms of a list of instructions that one must follow (e.g. start on the Euclidean sheet and then wind 7 times anti-clockwise around the branch point at unity) to correctly evaluate the correlator at the given insertion locations. As correlation functions capture the response of a theory to sources, our results are needed (together, of course, with knowledge of the correlators as analytic function of $z$ and $\bar{z}$), to determine the physical response of a CFT on $S^1 \times$ time to arbitrary sources as a function of angle and time.

Finally, we also view the problem in reverse order, and provide a complete listing (see Tables 1, 2, 3, 4, 5, 6, 7) of all branch structures that are accessed by varying overall insertion locations for time ordered correlators on the Lorentzian cylinder. We also provide an explicit listing of the various distinct causal configurations of operator insertions that lie in any one of our list of accessed Lorentzian sheets. It is physically interesting that several causally distinct configurations sometimes land on the same sheet of the correlator. This tells us that different (and symmetry unrelated) physical experiments sometimes have identical answers. We leave the interesting problem of understanding this observation from a physical viewpoint to future work.

### 1.4 What remains to be done

The construction presented in this paper yields a simple physical interpretation for an infinite number of branches of the sheets of $(C/N)(z, \bar{z})$. However, our work also leaves a much larger infinity of such sheets uninterpreted. The sheets accessed by the construction of this paper are essentially abelian in nature. As we explain in section §6.4, these sheets are all reached either by

- starting with the Euclidean sheet and repeating a given (clockwise) monodromy - around a single branch point- an indefinite number of times, or

- starting on the Euclidean sheet, performing a single clockwise half-monodromy around one of the branch points followed by an indefinite number of clockwise monodromies - around a second branch point - then undoing the original half-monodromy,

- starting on the Euclidean sheet, perform a single clockwise half-monodromy around one of the branch points followed by an indefinite number of clockwise monodromies - around a second branch point - then repeat the original half-monodromy.

In contrast, the most general sheet manipulation is non-abelian, as monodromies around different singularities do not commute. Starting with the Euclidean sheet, if we limit ourselves to $n$ monodromy moves, the total number of non-commuting monodromies - i.e. the total number of distinct sheets we can access - grows exponentially with $n$. Time-ordered correlators on Lorentzian space access a very small fraction of these sheets. This paper throws no light on the physical interpretation of the sheets obtained from these non-commuting monodromy

moves. The physical interpretation of these sheets - if one such exists - remains an interesting open question for the future.

## 1.5 Relation to earlier work

The work presented in this paper builds on the ideas presented in many earlier papers including [1–5, 17–20].

In particular, [21] presents an extensive study of the convergence of conformal blocks in small cross-ratio expansion in various complex sheets. In chapter 23 of this thesis, the author points out that one obtains multiple different monodromies by working on the Lorentzian cylinder. In more detail, the author studies operators with fixed ordering, and explains (in this context) how one obtains a range of monodromies, associated with different insertion locations on the cylinder. Because the operator ordering is fixed, the monodromy results of [21] are relatively simple. In this paper we work with time-ordered correlators. As the operator orderings in our correlators change depending on insertion locations, our monodromy results are much more involved. See around (F.1) for one example of the difference between the fixed ordered monodromies of [21], and our time-ordered monodromies, in the context of a particular example.

We would also like to note that while we have viewed correlators as multi-valued branch functions in this paper, there is an alternative way of viewing this structure. The authors of [22] have defined the function $q(z)$ (see Eq. 5 of that paper). The function $q(z)$ is a multivalued function of $z$, in precisely such a way that the function takes a different value on each sheet of a holomorphic conformal block. In other words, $q(z)$ is a faithful coordinate on the full branched cover of the $z$ plane relevant to the study of conformal blocks. In this way the complex plane parameterized by $q$ can be decomposed into an infinite number of "unit cells", defined so that any given unit cell is the image of one sheet of the $z$ plane. A correlator - which is multi-valued in $z$- can therefore be recast as a single valued function of $q$. In this language, the question posed in this paper is: what is the physical interpretation of the correlator in a generic unit cell in $q(z)$ space.

## 1.6 Organization of this paper

This paper is organized as follows. In §2 we first recall how the Lorentzian cylinder can be tiled with Minkowskian diamonds. We then study two and three-point functions, and (in particular) determine the sheet (phase) of these correlators as a function of insertion locations on the Lorentzian cylinder. In section §2.2 we also verify that the fixed functional forms of two and three-point functions automatically obey the constraints imposed by the requirement of path independence provided all operators have integer spins.

In §3 we begin our study of four-point functions. We define the conformal cross-ratios, $z$ and $\bar{z}$, and work out the rules that determine whether a particular light cone crossing takes us over or under the relevant branch point in cross-ratio space. We also explain that holomorphic and anti-holomorphic cross-ratios commute and monodromy moves in terms of conformal blocks. In §4 we demonstrate that all ways of moving insertion locations (from one configuration to another) give the same result for the monodromy of four-point functions.

In §5 we first explain the protocol we follow in this paper to reach any given physical configuration of interest starting from a Euclidean configuration, and then work out all the sheets accessed by insertions of four operators at arbitrary locations on a single Minkowskian diamond. We demonstrate that every configuration on the Lorentzian cylinder can be reached starting from operators that are either all mutually spacelike or all mutually timelike on a single Minkowskian diamond, and then making the moves $\omega_i \rightarrow \omega_i - n_i\pi$; such moves do not change any conformal cross-ratio, but can change the sheet. In section §6 we work out

the branch moves in cross-ratio space that follow from every such move (focusing on cases in which the cross-ratio is that of a Euclidean configuration).

In section §6.4 we summarize the results of section §6 from the "dual" viewpoint: instead of working out the sheet location as a function of insertion locations, we provide a listing of all insertion locations that correspond to any given sheet in cross-ratio space. While most sheets are implemented by several inequivalent causal configurations, we point out that the Regge sheet is special, in that the causal configuration that corresponds to this sheet is essentially unique. In §7 we end with a discussion of our results and interesting future directions.

## 2 Two and three point functions on the Lorentzian cylinder

As explained in the introduction, in this paper, we study the correlation functions of a 2d CFT on the Lorentzian cylinder $S^1 \times R$ (where $R$ is time).[4] In this section, we first explain how the Lorentzian cylinder can be tiled by Minkowski diamonds.[5] As a warm-up for the study of branch structures of four-point functions (the topic of main interest to this paper), we now present a detailed analysis of the branch structure of two and three-point functions on the Lorentzian cylinders, as a function of insertion locations.

### 2.1 Tiling the Lorentzian cylinder with Minkowski diamonds

While the metric on a 2D Lorentzian cylinder is locally Minkowskian, the spatial direction is a circle of circumference $2\pi$. Through this paper, we use the coordinate $\theta$ (with $0 \le \theta < 2\pi$) to parameterize points on the spatial circle. In contrast, the time coordinate $\tau$ is, of course, non-compact, and varies over the range $-\infty < \tau < \infty$. In equations, the metric on the Lorentzian cylinder is given by

$$ds^2 = -d\tau^2 + d\theta^2, \tag{2}$$

subject to the identification

$$(\tau, \theta) \equiv (\tau, \theta + 2\pi n), \qquad \forall \ n \in \mathbb{Z}, \tag{3}$$

the symbol $\equiv$ means "is the same point as".[6]

Below, we will often find it useful to change coordinates from $(\tau, \theta)$ to the left moving and right moving coordinates $(\omega, \bar{\omega})$ where,

$$\omega = \frac{1}{2}(\theta - \tau), \qquad \bar{\omega} = \frac{1}{2}(\theta + \tau). \tag{5}$$

Through this paper, we often refer to $(\omega, \bar{\omega})$ as light-cone coordinates. In these coordinates, the line element takes the form

$$ds^2 = -4 \, d\omega \, d\bar{\omega}, \tag{6}$$

---

[4]The reader who is accustomed to studying the AdS/CFT correspondence may choose to visualize this Lorentzian cylinder as lying on the boundary of global AdS$_3$. However, the analysis of this paper is purely field-theoretic and will make no use of the AdS/CFT correspondence.

[5]We use this name because each such diamond is Weyl equivalent to Minkowski space.

[6]The embedding space coordinates (see eg appendix B.7.2 of [23]) for our four boundary points are

$$P_i = \left( \cos \tau_i, \sin \tau_i, \cos \theta_i, \sin \theta_i, \vec{0} \right). \tag{4}$$

and the identification (3) becomes[7]

$$(\omega, \bar{\omega}) \equiv (\omega + n\pi, \bar{\omega} + n\pi), \qquad \forall \ n \in \mathbb{Z}. \tag{8}$$

It is well known that the space (6) can be "tiled" by an infinite sequence of Minkowski diamonds (see Fig. 1b).

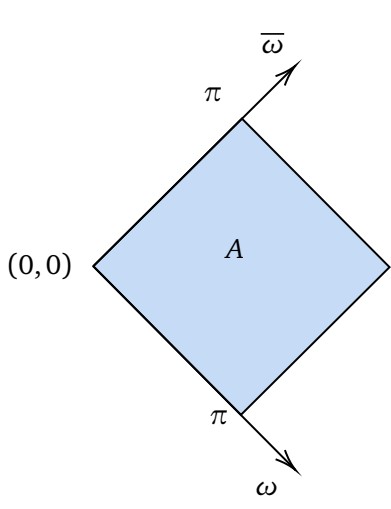

(a) Configuration space can be tiled with Minkowski diamonds.

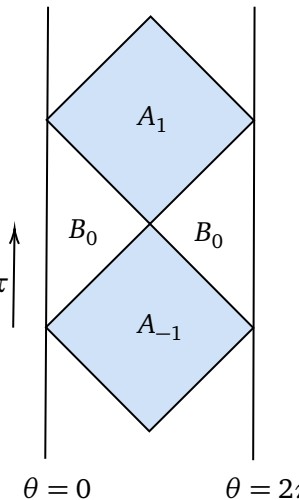

(b) The tiling of the Lorentzian cylinder with an infinite sequence of A-type and B-type Minkowskian diamonds.

Figure 1: Visualizing the tiling of the Lorentzian cylinder with Minkowski diamonds.

In the coordinates $\omega$ and $\bar{\omega}$, the Minkowski diamond is a square, of length (in $\omega$) and breadth (in $\bar{\omega}$) each equal to $\pi$ (see Fig. 1a).

It is not difficult to convince oneself[8] that the operator shifts

$$\begin{aligned} \omega_i &\to \omega_i + n_i \pi, \\ \bar{\omega}_i &\to \bar{\omega}_i + m_i \pi, \end{aligned} \tag{9}$$

(where $m_i$ and $n_i$ are any integers whatsoever) leaves all conformal cross-ratios unchanged.[9]

---

[7]The usual analytic continuation $\tau = -i\tau_E$ turns (5) into

$$\omega = \frac{1}{2}(\theta + i\tau_E), \qquad \bar{\omega} = \frac{1}{2}(\theta - i\tau_E). \tag{7}$$

[8]This point is easily verified from the explicit expressions for cross-ratios presented below. The structural reason for this invariance is most easily seen from embedding space formalism. As reviewed in (4), and e.g. Appendices B.2 and B.5 of [23]), operator insertions are labelled by null vectors $P_i$ in embedding space. As rescaling $P_i$ by a positive real number leaves the location of the inserted operator unchanged, conformal cross-ratios are simply those ratios of dot products of the various $P_i$ that are invariant under separate rescaling of each null vector $P_i$. As a consequence, however, these cross-ratios are also left unchanged when $P_i$ is multiplied by a negative number. Such a rescaling does not leave the insertion point unchanged; instead, it maps the point to its "antipodal image" ($\theta \to \theta \pm \pi$ and $\tau \to \tau \pm \pi$ in (4)). We thus see that such antipodal shifts of insertion points also leave cross-ratios unchanged. In addition to antipodal shifts, the operations $\tau_i \to \tau_i + 2\pi r_i$ and $\theta_i \to \theta_i + 2\pi s_i$ (where $r_i$ and $s_i$ are both integers) do not change the location of the insertion in embedding space, and so (trivially) leave all cross-ratios unchanged. Putting these facts together, the invariance of cross-ratios under (9) follows.

[9]More precisely, this shift leaves cross-ratios invariant only when all $\omega_i$ are chosen to be precisely real. When studying time-ordered correlators, we insert our operators at times that include a small imaginary part (i.e. we make the shift $\tau_i \to \tau_i - i\epsilon\tau_i$). The shifts (9) - which can be accomplished by shifting $\tau$ and $\theta$ according to $\tau_i \to \tau_i + (m_i - n_i)\pi$, and $\theta_i \to \theta_i + (m_i + n_i)\pi$ - sometimes changes the imaginary part of $\tau_i$ hence the effective ordering of operators - in a way that is sometimes physically consequential.

The invariance of cross-ratios under (9) tells us that the various Minkowski diamonds in Fig 1b can each be thought of as "unit cells" (for the Lorentzian cylinder) as far as cross-ratios are concerned. Given any collection of insertion locations on the Lorentzian cylinder, one can use the transformations (9) to find another associated set of insertion locations - all now in the same Minkowski diamond - that carries the same values of all conformal cross-ratios.[10]

## 2.2 Branch moves for two and three point functions

As we review in Appendix A, the functional form of two and three-point functions in the Lorentzian cylinder is completely determined by conformal invariance. One finds that the time-ordered two-point function of an operator with holomorphic and anti-holomorphic dimensions $(h, \bar{h})$ is proportional to

$$\langle \phi_{h,\bar{h}}(\omega_1, \bar{\omega}_1) \, \phi_{h,\bar{h}}(\omega_2, \bar{\omega}_2) \rangle \propto \frac{1}{\zeta_{12}^h \, \bar{\zeta}_{12}^{\bar{h}}} \,, \tag{10}$$

where

$$\begin{aligned} \zeta_{ij} &= \sin^2(\omega_{ij} + i\epsilon \tau_{ij}), \\ \bar{\zeta}_{ij} &= \sin^2(\bar{\omega}_{ij} - i\epsilon \tau_{ij}). \end{aligned} \tag{11}$$

Notice that the cross-ratios (11) are both invariant under the shifts (9) as expected on general grounds. Similarly one finds that the time-ordered three-point function of three operators is given by

$$\langle \phi_{h_1,\bar{h}_1}(\omega_1, \bar{\omega}_1) \, \phi_{h_2,\bar{h}_2}(\omega_2, \bar{\omega}_2) \, \phi_{h_3,\bar{h}_3}(\omega_3, \bar{\omega}_3) \rangle = \frac{C_{123}}{\zeta_{12}^{H_{12}} \zeta_{23}^{H_{23}} \zeta_{31}^{H_{31}} \bar{\zeta}_{12}^{\bar{H}_{12}} \bar{\zeta}_{23}^{\bar{H}_{23}} \bar{\zeta}_{31}^{\bar{H}_{31}}} \,, \tag{12}$$

where

$$H_{ij} = \frac{h_i + h_j - h_k}{2} \,, \qquad \bar{H}_{ij} = \frac{\bar{h}_i + \bar{h}_j - \bar{h}_k}{2} \,. \tag{13}$$

For generic values of $h_i$, the expressions (10) and (12) have branch cuts in the variables $\omega_i$ (and also in the variables $\bar{\omega}_j$), and so the expressions on the RHS of (10) and (12) are multivalued (i.e. have many different sheets). Consequently, the expressions (10) and (12) have an (relatively trivial, pure phase) ambiguity. We can resolve this ambiguity as follows.

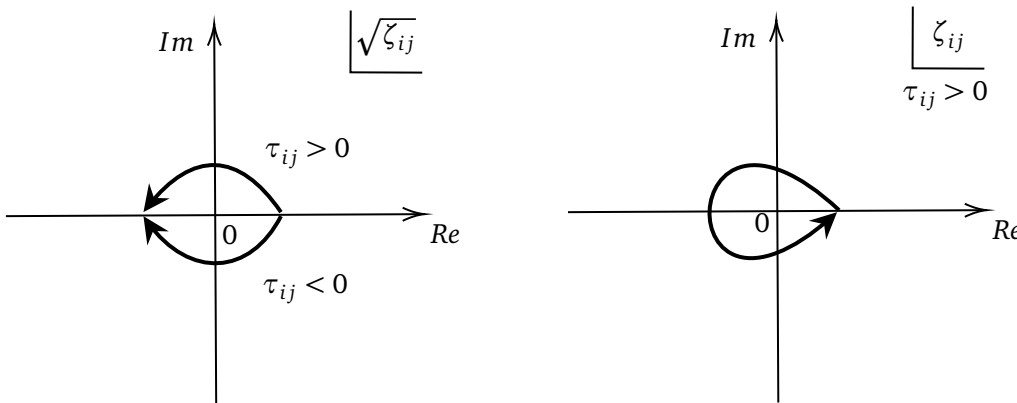

Figure 2: Two-point function cross-ratio space and the monodromies due to crossing light-cones going to the past or future.

---

[10]We emphasize again that while these moves leave cross-ratios unchanged, they, in general, change the value of correlation functions, by moving the correlator to a different sheet, in the manner we will describe in detail in much of the rest of this paper.

Let us suppose we start with insertion locations that are all mutually space-like with respect to one another. For such configurations, all correlators take their Euclidean values and so are unambiguous. We can then continuously vary insertion locations until they reach their final values (which are not necessarily space-like related to one another). By keeping track of all the branch windings that we are forced to make as we move along this continuous path, we thus obtain a definite answer for the phase of the correlator at its final insertion location.

Branch windings happen when $i$ crosses through the "leftmoving/rightmoving" light-cones of $j$, i.e., when, respectively $\zeta_{ij} = 0$ or $\bar{\zeta}_{ij} = 0$. Let us, for definiteness, focus on the passage through rightmoving (or holomorphic) lightcones. These lightcones occur when $\omega_i = \omega_j - n_{ij}\pi$ for some choice of integer $n_{ij}$. Suppose $\omega_i$ starts at a value just larger than $\omega_j - n_{ij}\pi$ and cuts the lightcone towards the future[11] ending up at a value just smaller than $\omega_j - n_{ij}\pi$. In the neighbourhood of this value, $\sqrt{\zeta_{ij}} \simeq \omega_{ij} + n_{ij}\pi + i\epsilon\tau_{ij}$[12] where $\tau_{ij} = \tau_i - \tau_j$ is positive if $i$ lies to the future of $j$, but negative if $i$ lies to the past of $j$. It follows that, for the motion described above $\sqrt{\zeta_{ij}}$ moves,[13] along the upper trajectory of Fig. 2 when $\tau_{ij} > 0$, but moves along the lower trajectory shown of Fig. 2, when $\tau_{ij} < 0$. Of course, motions towards the past execute the opposite trajectories. It follows from this discussion that

- $\zeta_{ij}$ undergoes an anticlockwise monodromy of $2\pi$ when $\omega_i$ moves from past to future, cutting a right-moving future light cone of particle $j$. $\zeta_{ij}$ also undergoes an anticlockwise monodromy of $2\pi$ when $\omega_i$ moves from future to past, cutting a right moving past light cone of particle $j$.

- $\zeta_{ij}$ undergoes a clockwise monodromy of $2\pi$ when $\omega_i$ moves from past to future, cutting a right moving past light cone of particle $j$. $\zeta_{ij}$ also undergoes a clockwise monodromy of $2\pi$ when $\omega_i$ moves from future to past, cutting a right-moving future light cone of particle $j$.

Of course, the rules above are symmetric under the interchange of $i$ and $j$. These pass all relevant consistency checks. For instance, the motion of $i$ toward the future, cutting a future lightcone of $j$ can also be thought of as the motion of $j$, towards the past, cutting a past lightcone of $i$: both these manoeuvres have $\zeta_{ij}$ executing an anticlockwise monodromy of $2\pi$. The rules are also invariant under time reversal (because they are invariant under the uniform replacement future $\longleftrightarrow$ past.

The rules described above allow us to track two and three-point functions as we move insertion locations from Euclidean values to the locations of interest. There is, however, one remaining concern about a potential ambiguity in the procedure described above. The continuous motions (that link spacelike separations to arbitrary separations) can be carried out along many different paths. We will now demonstrate that the procedure outlined in the previous paragraph is unambiguous - i.e. that we get the same answer from all possible paths - provided that the scaling dimensions for all our inserted operators obey the level-matching condition

$$h_i - \bar{h}_i \in \mathbb{Z}. \tag{14}$$

---

[11]Because $\omega = \frac{\theta - \tau}{2}$, motion towards the future corresponds to decreasing $\omega$.

[12]The term proportional to $i\epsilon$ follows from the usual continuation $\tau \to \tau(1 - i\epsilon)$ which ensures that all operators have the same ordering in (infinitesimal) Euclidean time as in Lorentzian time, and so are time ordered. The sign of the $i\epsilon$ term in the main text follows because of the minus sign in the equation $\omega = \frac{\theta - \tau}{2}$.

[13]In this paragraph we keep track of $\sqrt{\zeta_{ij}}$ rather than simply $\zeta_{ij}$ because it turns out that cross-ratios for four-point functions are naturally written in terms of $\sqrt{\zeta_{ij}}$ (see (25)).

## 2.3 Path independence and branch structure of two-point functions

Let us start with the case of the two-point function. Translation invariance lets us fix the location of one of the insertions - let's say to the apex between the two *A* type diamonds in Fig 1b. As in Fig 1b, we now give an integer labelling of all the diamonds in the diagram. The *B* type diamond displayed in the diagram is labelled 0 (and so has been denoted $B_0$ in Fig 1b). The higher of the two *A* type diamonds is labelled 1 (and is denoted $A_1$), the next (higher) *B* diamond is labelled 2 and will be denoted $B_2$, and so on. Similarly, the lower *A* diamond is labelled $-1$ and so is denoted $A_{-1}$ the subsequent lower *B* diamond is labelled $-2$, denoted $B_{-2}$, and so on.

For causal (or branch structure) purposes, the location of the second insertion is completely specified by the integer that labels the diamond in which it is located. If the second particle is located in the (*B* type) diamond 0, then it is spacelike separated with the original insertion, and the correlator is given by simple continuation from the Euclidean value. In going from diamond 0 to diamond 1, the rules listed in the previous section tell us that the two-point function listed in (10) picks up either the additional phase $e^{-2\pi i h}$ or the additional phase $e^{-2\pi i \bar{h}}$, depending on whether one cuts the left or right moving lightcone. Provided (14) is obeyed, these two phases are the same, and so our answer is unambiguous. Iterating this procedure, we see that when the insertion of the second operator lies in the $m^{th}$ diamond with $m > 0$, (10) picks up the additional phase $e^{-2\pi i h m}$ over and above the simple analytic continuation of the Euclidean answer to the given values of $\zeta_{ij}$.[14] The workout above is easily generalized to the case that $m$ is negative: using the rules of the previous subsection, we find that, in the $m^{th}$ diamond, the two-point function equals its value on the Euclidean sheet (diamond 0) times the extra phase $e^{-2\pi i h |m|}$; this answer is time reversal invariant.[15]

In summary, we have thus both demonstrated that our procedure gives us a well-defined result for the phase of the time-ordered two-point function on the Lorentzian cylinder, and also found the simple final formula for the correlator including its phase. Our final answer is the real Euclidean sheet correlator with the same value of $\zeta_{ij}$ (i.e. the correlator with the second insertion in the diamond $B_0$) times the phase $e^{-2\pi i h |m|}$, where $m$ is the "diamond number" of the insertion location of the second operator, as defined above. In equations

$$C_{(\omega_i, \bar{\omega}_i)} = C_{\text{space}-\text{like}} \times e^{-2\pi i h \left| \left[ \frac{\omega_{12}}{\pi} \right] - \left[ \frac{\bar{\omega}_{12}}{\pi} \right] \right|}, \tag{15}$$

here, $[x]$ is a function which spits out the highest integer no greater than $x$, and $\omega_{12}$ is the difference between the $\omega$ values of the insertion points of the operators 1 and 2 ($\bar{\omega}_{12}$ is defined similarly).

## 2.4 Path independence and branch structure of three-point functions

It is possible to analyze three-point functions like two-point functions, by using translation invariance to locate the operator 3 at the vertex between the diamonds $A_1$ and $A_{-1}$ in Fig 1b. Causally distinct configurations can then be specified by two integers, namely the label for a diamond in which the operator 1 is inserted and the label for the diamond in which operator 2 is inserted. The diamond locations of points 1 and 2 completely specify their causal locations

---

[14]Note that the phase of the two-point function (over the phase on the Euclidean sheet) can also be written as $e^{-2\pi i (m_1 h + \bar{m}_1 \bar{h})}$ for any choice of $m_1$ and $\bar{m}_1$ such that $m_1 + \bar{m}_1 = m$: (14) ensures that all values of $m_1$ and $\bar{m}_1$ (that add up to $m$) give the same phase.

[15]If the first insertion is placed at the point where the two blue diamonds meet in Fig. 1b, the shifts $\omega_i \to \omega_i - n_i \pi$ can be used to move the second insertion to the diamond $B_1$. It follows, in other words, that every configuration of two points on the Lorentzian cylinder can be obtained, starting from points that are spacelike separated and then making $\pi$ shifts. We will see that the situation is slightly more complicated in the case of three and four-point functions.

with respect to 3, but only partially specify causal locations with respect to each other. To complete this specification, we must also specify which of the four relative causal orderings 1 and 2 can be in, consistent with their given locations within the diamond structure.[16] With this way of functioning, the set of possible causal orderings makes up a two-dimensional lattice, with each lattice point hosting a "square molecule" (4 possibilities). Local lightcone crossings would give us links on this lattice. We would then be required to prove the path independence of monodromies as we move from one lattice point to another via allowed links.

### 2.4.1 Holomorphic factorized parametrization of operator insertions

While the analysis described in the previous paragraph is not too difficult to carry through in the case of three-point functions, the equivalent analysis is rather messy in the case of 4 point functions. In preparation for that more complicated analysis, we study the case of three-point functions in a manner that is as holomorphically factorized as possible. As above, we use translational invariance to insert operator 3 at the origin. After we have made this choice, we write[17]

$$
\begin{aligned}
\omega_1 = -m_1\pi + \alpha_1 &\implies \left[\frac{\omega_1}{\pi}\right] = -m_1, \\
\bar\omega_1 = \bar m_1\pi + \bar\alpha_1 &\implies \left[\frac{\bar\omega_1}{\pi}\right] = \bar m_1, \\
\omega_2 = -m_2\pi + \alpha_2 &\implies \left[\frac{\omega_2}{\pi}\right] = -m_2, \\
\bar\omega_2 = \bar m_2\pi + \bar\alpha_2 &\implies \left[\frac{\bar\omega_2}{\pi}\right] = \bar m_2,
\end{aligned}
\tag{16}
$$

where $0 \leq \alpha_i < \pi$ and $0 \leq \bar\alpha_i < \pi$. Recall, however, that the shifts (8) of $\omega$ and $\bar\omega$ are a redundancy of description; two different values of $\omega$ and $\bar\omega$ that are related by (8) denote the same point on the Lorentzian cylinder. As a consequence the four integers that appear in (16) are a redundant description; knowing the physical locations of our insertions only unambiguously fixes $m_i + \bar m_i$. To proceed we simply choose any convenient values of $m_i$ and $\bar m_i$ with the given physical difference (our final answer will not depend on this choice).[18]

### 2.4.2 Nearest neighbour moves on the three-point causal lattice

With the conventions of the previous subsection in place, the relative "holomorphic causal ordering" of two points is specified by the integers $m_1$ and $m_2$, and the relative ordering of $\alpha_1$ and $\alpha_2$. Let us denote configurations with integers $m_1$, $m_2$ as $P_{12}^{m_1,m_2}$ if $\alpha_2 > \alpha_1$[19] and $P_{21}^{m_1,m_2}$ if $\alpha_1 > \alpha_2$. Local holomorphic light cone crossings induce the following three motions on this lattice.

- If we start with $P_{12}^{m_1,m_2}$ and 2 crosses a lightcone of 1 (moving from past to future) we end up with $P_{21}^{m_1,m_2}$. The inverse of this lattice move is given by starting with $P_{21}^{m_1,m_2}$ and having 1 cross the lightcone of 2 from past to future.

---

[16]The four possible causal orderings can be thought of as follows. We translate the operator 1, in a purely left-moving (holomorphic manner), by $\omega_1 \to \omega_1 + n_1\pi$, to the diamond in which the operator 2 is located. Once we have done this, 1 can be either in the past, future or "left spacelike" or "right spacelike" related to 2. Note that the two spacelike regions are distinct from each other (one cannot circle the cylinder and go from left spacelike to light spacelike as this requires crossing a lightcone that emanates out of the point 3).

[17]Recall that $\omega = \frac{\theta - \tau}{2}$ so we have put a negative sign with $m$ as a convention so that increasing $m$ would mean moving to the future.

[18]Rotating the insertion point of any operator around the cylinder is a continuous operation that affects a change of $m_i$ and $\bar m_i$ individually while leaving $m_i + \bar m_i$ unchanged. This operation leaves the correlator unchanged precisely because the operator spin $h_i - \bar h_i$ is an integer. See Appendix B for a more detailed discussion.

[19]As $\omega = \frac{\theta - \tau}{2}$, this means that the time coordinate in $\alpha_1$ is larger than that in $\alpha_2$. Consequently, in this configuration, 1 is to the future of 2, as far as the $\alpha$ coordinates are concerned.

- If we start with $P_{12}^{m_1,m_2}$ and 1 crosses a light cone of 3 (moving from past to future), we end up with $P_{21}^{m_1+1,m_2}$. The inverse of this move is to start with $P_{21}^{m_1+1,m_2}$ and have 1 cross a lightcone of 3 moving from future to past.

- If we start with $P_{12}^{m_1,m_2}$ and 2 crosses a light cone of 3 (moving from future to past), we end up with $P_{21}^{m_1,m_2-1}$. The inverse of this move is to start with $P_{21}^{m_1,m_2-1}$ and have 2 cross a lightcone of 3 moving from past to future.

We call the "one crossing" holomorphic moves described as the "interchange" $I$, the "forward push" $F$ or the backward push $B$. Explicitly, the action of these operations on the lattice points $P_{12}^{m_1,m_2}$ are given by

$$
\begin{aligned}
I &: \left(P_{12}^{m_1,m_2}, Q\right) \to \left(P_{21}^{m_1,m_2}, Q\right), \\
I &: \left(P_{21}^{m_1,m_2}, Q\right) \to \left(P_{12}^{m_1,m_2}, Q\right), \\
F &: \left(P_{12}^{m_1,m_2}, Q\right) \to \left(P_{21}^{m_1+1,m_2}, Q\right), \\
F &: \left(P_{21}^{m_1,m_2}, Q\right) \to \left(P_{12}^{m_1,m_2+1}, Q\right), \\
B &: \left(P_{12}^{m_1,m_2}, Q\right) \to \left(P_{21}^{m_1,m_2-1}, Q\right), \\
B &: \left(P_{21}^{m_1,m_2}, Q\right) \to \left(P_{12}^{m_1-1,m_2}, Q\right),
\end{aligned}
\tag{17}
$$

where $Q$ is an arbitrary anti-holomorphic lattice "atom".[20] Let us note that $I^{-1} = I$, $F^{-1} = B$ and $B^{-1} = F$.

The maps $I$, $F$ and $B$ act only on the holomorphic part of the lattice, leaving the anti-holomorphic "atom" unchanged. We can, of course, define similar maps $\bar{I}$, $\bar{F}$ and $\bar{B}$ that leave the holomorphic part of the lattice untouched, but act on the anti-holomorphic part of the lattice according to the mirror image rules

$$
\begin{aligned}
\bar{I} &: \left(P, Q_{12}^{\bar{m}_1,\bar{m}_2}\right) \to \left(P, Q_{21}^{\bar{m}_1,\bar{m}_2}\right), \\
\bar{I} &: \left(P, Q_{21}^{\bar{m}_1,\bar{m}_2}\right) \to \left(P, Q_{12}^{\bar{m}_1,\bar{m}_2}\right), \\
\bar{F} &: \left(P, Q_{12}^{\bar{m}_1,\bar{m}_2}\right) \to \left(P, Q_{21}^{\bar{m}_1+1,\bar{m}_2}\right), \\
\bar{F} &: \left(P, Q_{21}^{\bar{m}_1,\bar{m}_2}\right) \to \left(P, Q_{12}^{\bar{m}_1,\bar{m}_2+1}\right), \\
\bar{B} &: \left(P, Q_{12}^{\bar{m}_1,\bar{m}_2}\right) \to \left(P, Q_{21}^{\bar{m}_1,\bar{m}_2-1}\right), \\
\bar{B} &: \left(P, Q_{21}^{\bar{m}_1,\bar{m}_2}\right) \to \left(P, Q_{12}^{\bar{m}_1-1,\bar{m}_2}\right).
\end{aligned}
\tag{18}
$$

This web of moves detailed above builds a holomorphic causal cubic lattice in two dimensions. Locations on this lattice are labelled by the two integers $(m_1, m_2)$. Each lattice point hosts a "molecule" made of the two "atoms" $P_{12}^{m_1,m_2}$ and $P_{21}^{m_1,m_2}$. The moves (17) can be thought of as links on this lattice (we have one link both between "atoms" on a given "molecule", as well as links between "atoms" on neighbouring molecules).

### 2.4.3 Path independence

In the previous subsection, we have constructed a causal lattice (for three-point functions) together with a partner anti-holomorphic causal lattice. Points on the holomorphic causal lattice are $P_{12}^{m_1,m_2}$ and $P_{21}^{m_1,m_2}$. Every move in (17) connects two lattice points and defines a link on this lattice. A trajectory between two lattice points $A$ and $B$ is defined to be a continuous

---

[20]More precisely, $Q = Q_{21}^{\bar{m}_1,\bar{m}_2}$ or $Q = Q_{12}^{\bar{m}_1,\bar{m}_2}$ for some values of the integers $\bar{m}_1$ and $\bar{m}_2$.

path - always moving along links - that takes us from *A* to *B*. Each such trajectory represents a distinct class[21] of continuous motions of the insertion locations that take us from the initial (*A*) to the final (*B*) insertion locations. Exactly analogous remarks hold for the anti-holomorphic part of the causal lattice.

The process of deforming insertion locations from *A* to *B* typically induces a monodromy. We will now explain that (as in the case of two-point functions) this monodromy is independent of the detailed path traversed between *A* and *B*.

We will demonstrate path independence by showing that the monodromy associated with each closed loop on the lattice vanishes. We will show this result - in turn - by demonstrating that the monodromy around each of the elementary plaquettes ("minimal" loops) - those that can be used to tile any given macroscopic closed loops - vanishes.[22]

The redundant labelling of configurations (16) at first appears to have complicated our task of demonstrating path independence. (16) appears to have doubled the dimensionality of our lattice (from 2 to 4), and so forced us to study with $(4 \times 3)/2 = 6$ distinct orientations of plaquettes on this lattice (corresponding to the number of ways one can choose two directions out of a collection of 4). However, 4 of these 6 distinct plaquettes have one leg in a holomorphic lattice direction and the second leg in an anti-holomorphic lattice direction. In Appendices C.1 and C.2 respectively, we demonstrate

- That these "mixed" monodromies always vanish, and so we only need to worry about the "purely holomorphic" or "purely anti-holomorphic" monodromies.

- That, moreover, the vanishing of mixed monodromies can be used to show that if the monodromy of a "holomorphic unit face" vanishes when we are sitting at a given point on the anti-holomorphic lattice, then the same holomorphic unit face monodromy will also vanish when we sit at any other point on the anti-holomorphic lattice.

### 2.4.4 Vanishing of purely left moving monodromies

All that now remains to be shown is that purely left-moving monodromies also vanish. The basic monodromy loop - or plaquette- in the left-moving causal lattice is given by

$$P_{12}^{m_1 m_2} \xrightarrow{F} P_{21}^{m_1+1,m_2} \xrightarrow{F} P_{12}^{m_1+1,m_2+1} \xrightarrow{I} P_{21}^{m_1+1,m_2+1} \xrightarrow{B} P_{12}^{m_1,m_2+1} \xrightarrow{B} P_{21}^{m_1,m_2} \xrightarrow{I} P_{12}^{m_1 m_2}. \quad (19)$$

The monodromy associated with this sequence of moves is easily evaluated using the rules of section 2.2). We find that it vanishes. Of course, a similar result holds for the anti-holomorphic causal lattice. This completes our demonstration of path independence (in the context of the three-point function).

### 2.4.5 Value of the three-point function at arbitrary insertion locations

With path independence established, we can now proceed to work out the value of the three point function for three arbitrary insertion locations on the Lorentzian cylinder. The computation is not too difficult to perform. In this subsubsection we present our final result.

We first note that any set of three insertions on the Lorentzian cylinder can - by the integer shifts $\omega_i \to \omega_i - n\pi$ - be brought to one of the two single diamond configurations depicted in Fig. 3. The first (*A* type) of these configurations has the three insertion points separated in a

---

[21]We say that two different motions of insertion locations lie in the same class if they can be continuously deformed to each other without any operator cutting any lightcone.

[22]The reader may find the following analogy useful. The monodromy around a closed loop is analogous to the holonomy (of an abstract gauge field) around that path. Demonstrating path independence is equivalent to demonstrating that this abstract gauge field is closed, i.e. its field strength vanishes everywhere. This is the case if the field strength vanishes plaquette by plaquette, which is what we demonstrate below.

spacelike manner on the Minkowski diamond. The second ($D$ type) the configuration has one pair of timelike operators, spacelike separated from the third insertion. We describe our result for the three-point function in each of these cases.

Let us first start with configurations that can be obtained by performing the shifts $\omega_i \to \omega_i - n\pi$ on an $A$ type configuration. On the $A$ type configuration itself, the three-point function lies on the Euclidean sheet and, in particular, is real-valued. The shifts $\omega_i \to \omega_i - n\pi$ generically cause the three-point function to pick up a phase. This phase is given as follows. Let us define the integers

$$n_{ij} = \left[ \frac{|\omega_{ji}|}{\pi} \right], \qquad \bar{n}_{ij} = \left[ \frac{|\bar{\omega}_{ij}|}{\pi} \right], \qquad N_{ij} = n_{ij} + \bar{n}_{ij}, \tag{20}$$

where $[x]$ denotes the greatest integer no greater than $x$. With these definitions in place, we find that the value of the three-point function, at arbitrary insertion locations on the Lorentzian cylinder, is given by the (real-valued) Euclidean or principal value of the three-point function times the phase

$$e^{-2\pi i (N_{12} H_{12} + N_{13} H_{13} + N_{23} H_{23})} . \tag{21}$$

Let us now turn to the $D$ type configuration, (see figure 3) where $(n_i, n_j, n_k)$ are three integers representing the shifts we need to make to reach the final configuration diamond number. In this case, the starting correlator (in the single diamond D type configuration) is itself not real-valued, but instead has the phase $e^{-2\pi i H_{jk}}$. Finally, the total phase due to $\pi$ shifts turns out to be

- $e^{-2\pi i (N_{12} H_{12} + N_{13} H_{13} + N_{23} H_{23} + H_{jk})}$ in the case that $n_j \geq n_k$ (recall the starting $D$ type configuration had $\tau_j > \tau_k$).

- $e^{-2\pi i (N_{12} H_{12} + N_{13} H_{13} + N_{23} H_{23} - H_{jk})}$ in the case that $n_k > n_j$.

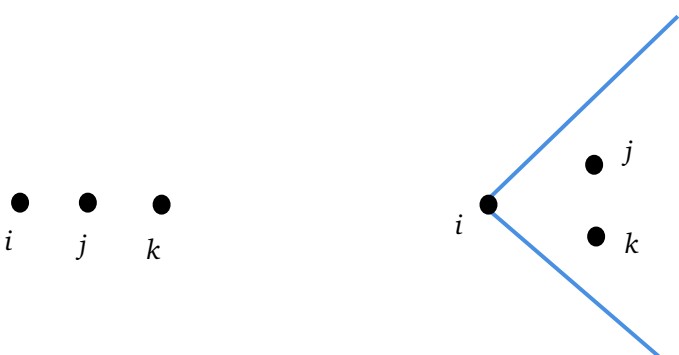

Figure 3: A type and D type configuration for the three point function.

## 3 Four point functions: Kinematics and branch moves

In the rest of this paper we focus on four point correlators, the topic of principal interest to this paper.

### 3.1 Four point cross-ratios

Consider the insertion of four operators, at the locations

$$(\omega_i, \bar{\omega}_i) \quad (i = 1 \ldots 4), \tag{22}$$

on the Lorentzian cylinder (6). The (four insertion) conformal cross-ratios are given by

$$z = \frac{\sin \omega_{12} \, \sin \omega_{34}}{\sin \omega_{13} \, \sin \omega_{24}}, \qquad \bar{z} = \frac{\sin \bar{\omega}_{12} \, \sin \bar{\omega}_{34}}{\sin \bar{\omega}_{13} \, \sin \bar{\omega}_{24}}, \tag{23}$$

where $\omega_{ij} = \omega_i - \omega_j$. It is easy to explicitly verify that the cross-ratios (23) are indeed invariant under the independent shifts (of either $\omega_i$ or $\bar{\omega}_j$ by a multiple of $\pi$ (9)) (as we have argued expected on general grounds, see above).

In the formulae above we have assumed that $\omega_{ij}$ is a real number. As we have mentioned above, correlators have branch point singularities: to detect which branch of the correlator we end up on we need $i\epsilon$ corrections to the formulae (23). The form of the $i\epsilon$ corrections is dictated by physical considerations. If, for instance, we wish to study time-ordered correlators (as is largely the case in this paper) we choose

$$\epsilon_{ij} = \epsilon \tau_{ij}, \quad \epsilon > 0. \tag{24}$$

The $i\epsilon$ corrected formulae for cross-ratios are

$$z = \frac{\sin(\omega_{12} + i\epsilon\tau_{12}) \, \sin(\omega_{34} + i\epsilon\tau_{34})}{\sin(\omega_{13} + i\epsilon\tau_{13}) \, \sin(\omega_{24} + i\epsilon\tau_{24})}, \tag{25}$$

$$\bar{z} = \frac{\sin(\bar{\omega}_{12} - i\epsilon\tau_{12}) \, \sin(\bar{\omega}_{34} - i\epsilon\tau_{34})}{\sin(\bar{\omega}_{13} - i\epsilon\tau_{13}) \, \sin(\bar{\omega}_{24} - i\epsilon\tau_{24})}. \tag{26}$$

### 3.2 Branch moves

Correlation functions develop (branch point) singularities at $z$ and $\bar{z} = (0, 1, \infty)$. Using (23), it is easy to convince oneself that these singularities occur precisely when two points are lightlike separated,[23] as might have been anticipated on general grounds.

Consider a time-ordered correlator. Consider moving the location of insertions in a manner that causes one point to cut the lightcone of another. When this happens the cross-ratio naively becomes 0, 1 or $\infty$. However the $i\epsilon$ in (25) and (26) tells us that our path in configuration space misses the branch point, passing either below or above it. In the rest of this paper, we sometimes use the term "half-monodromy" for the process of passing either over or under one of the branch points. If the motion around the branch point is in the clockwise/anti-clockwise direction, we call the resultant move a clockwise/anti-clockwise half-monodromy around the given branch point. Consider, for instance, starting in the range $z \in (0, 1)$ and moving over the branch point at $z = 0$ towards negative $z$. We refer to this motion as an anti-clockwise half-monodromy around $z = 0$. In this subsection, we present the rules that determine the precise effect of this motion (on conformal cross-ratios, in the complex plane). These rules (which are easily derived using (25) and (26)) will play a key role in the determination of the location of branch space in subsequent sections.

The "half-monodromy" rules are

1. If $P_i$ crosses the future right moving lightcone of $P_j$ from past to future, or if $P_i$ crosses the past right moving lightcone of $P_j$ from future to past, then this results in:
   An anti-clockwise traversal around $z = 0$ for $(i, j) = (1, 2)$ or $(i, j) = (3, 4)$,
   An anti-clockwise traversal around $z = 1$ for $(i, j) = (1, 4)$ or $(i, j) = (2, 3)$,
   An "anti-clockwise" traversal around $z = \infty$ (i.e. an anti-clockwise traversal around $1/z = 0$ in the $1/z$ plane, i.e. a clockwise traversal around both 0 and 1 in the $z$ plane) for $(i, j) = (1, 3)$ or $(i, j) = (2, 4)$.

---

[23]For instance, (23) tells us that $z = 0$ when $\omega_1 = \omega_2$ - i.e. when points 1 and 2 lie on each other's rightmoving lightcone.

2. If $P_i$ crosses the past right moving lightcone of $P_j$ from past to future, or if $P_i$ crosses the future right moving lightcone of $P_j$ from future to past, then this results in:
A clockwise traversal around $z = 0$ for $(i, j) = (1, 2)$ or $(i, j) = (3, 4)$,
A clockwise traversal around $z = 1$ for $(i, j) = (1, 4)$ or $(i, j) = (2, 3)$,
An "clockwise" traversal around $z = \infty$ (i.e. a clockwise traversal around $1/z = 0$ in the $1/z$ plane, i.e. an anti-clockwise traversal around both 0 and 1 in the $z$ plane) for $(i, j) = (1, 3)$ or $(i, j) = (2, 4)$.

3. The rules (1) and (2) above continue to apply if we make the replacements right-moving lightcone $\rightarrow$ left-moving lightcone and $z \rightarrow \bar{z}$.

This set of rules completely determines where in the branch structure we land up when, for instance, starting from an Euclidean configuration we move to any other configuration of interest.

**Notation:** Every branch point is associated with one of two pairs of particles that become light-like at that branch point. For instance, the branch point at $z = 0$ is associated with either particles 1 and 2 or particles 3 and 4 becoming lightlike w.r.t each other. In the rest of this paper, we use the associated pairs to label branch points. In other words, we use the notation

$$z_{12} = z_{34} = 0, \qquad z_{23} = z_{14} = 1, \qquad z_{13} = z_{24} = \infty. \tag{27}$$

In a similar manner, in the $\bar{z}$ complex plane we define,

$$\bar{z}_{12} = \bar{z}_{34} = 0, \qquad \bar{z}_{23} = \bar{z}_{14} = 1, \qquad \bar{z}_{13} = \bar{z}_{24} = \infty. \tag{28}$$

Using the notation developed above, rules presented earlier in this subsection can be rewritten as

1. If $P_i$ crosses the future (past) right-moving lightcone of $P_j$ from past to future, or if $P_i$ crosses the past (future) right-moving lightcone of $P_j$ from future to past, then this results in an anti-clockwise (clockwise) half-monodromy around $z_{ij}$, i.e., $\sqrt{A_{ij}}$ ($\sqrt{C_{ij}}$).

2. The above rule continues to apply if we make the replacements right-moving lightcone $\rightarrow$ left-moving lightcone and $z \rightarrow \bar{z}$.

Also note that, by definition, two consecutive half-monodromies (i.e. half-monodromies with no other monodromy inserted in the middle) yield a full monodromy, i.e., $\sqrt{A_{ij}} \cdot \sqrt{A_{ij}} = A_{ij}$ and $\sqrt{C_{ij}} \cdot \sqrt{C_{ij}} = C_{ij}$.

### 3.3 A matrix representation of branch moves

As we have already explained in the introduction, a CFT correlator can be written in terms of conformal blocks, in the form (1). Individual conformal blocks are multivalued: however, it is always possible to choose convenient bases of blocks that have no branch cuts in any given portion of the real axis. In this subsection, we explain how we can switch between the relevant basis blocks to obtain a matrix representation for any given monodromy operation. The content (and notation) of this section closely follows the classic papers of Seiberg and Moore [24–27, 27, 28].

#### 3.3.1 Basis for blocks

In this subsubsection, we define three different convenient bases for blocks. Our bases are respectively chosen to ensure that all basis elements are free of branch cuts in the range $(0, 1)$, $(1, \infty)$ and $(-\infty, 0)$ respectively.

**Blocks $\alpha_m$ regular in $(0, 1)$**   To start with consider blocks that diagonalize the monodromy of 1 around 2, and 3 around 4. Such basis blocks describe the fusion of 1 with 2 (and so 3 with 4) to an operator with dimensions $h_m, \bar{h}_m$. We call such a block $\alpha_m$ (recall that $m$ is the operator into which 1 and 2 fuse). Near $z = 0$, such blocks behave like $\frac{1}{z^{\Delta_1 + \Delta_2 - \Delta_m}}$, and so, generically, have branch points at $z = 0$. We choose this (relatively simple, phase type) branch cut to run from $z = 0$ to $z = -\infty$ along the negative $z$ axis.

The blocks $\alpha_m$ have a second branch cut at $z = 1$ (this cut can be thought of as a consequence of the fact that the $1 \to 2$ OPE does not converge if 3 lies somewhere on the straight line between 1 and 2). This cut can be chosen to run from 1 to $\infty$ along the real axis. This cut is more complicated because the discontinuity across it is non-abelian: the blocks $\alpha_m$ above the cut are linear combinations of $\alpha_n$ (for all $n$) below the cut.

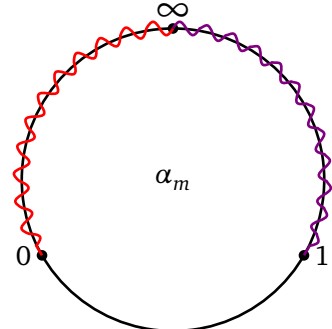

Figure 4: Blocks $\alpha_m$ regular in $(0, 1)$.

The key point for us, however, is that no element of this basis of blocks has a branch cut in the range $z \in (0, 1)$.

**Blocks $\beta_m$ regular in $(1, \infty)$**   A very similar construction yields blocks that are regular in the range $(1, \infty)$. We choose blocks that diagonalize the monodromy as 2 is taken around 3, i.e. blocks $\beta_m$ in which 2 and 3 fuse to $O_m$. We choose the branch cut for the "Abelian" monodromy of this block to run from 1 to 0. The more serious "non-Abelian" monodromy of this block has a branch cut from $-\infty$ to 0. These blocks are regular for $z \in (1, \infty)$.

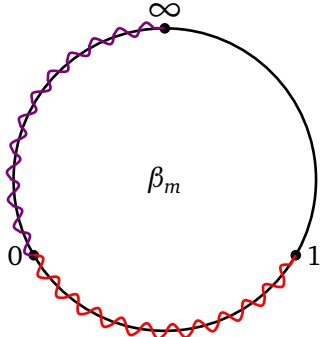

Figure 5: Blocks $\beta_m$ regular in $(1, \infty)$.

**Blocks $\gamma_m$ regular in $(-\infty, 0)$**   Finally, blocks in which 1 and 3 fuse to $O_m$ are called $\gamma_m$. The Abelian cut of this block is taken to run from 1 to $\infty$. The cut of the non-Abelian monodromy is taken to run from $(0, 1)$. These blocks are all free of cuts in the range $(-\infty, 0)$.

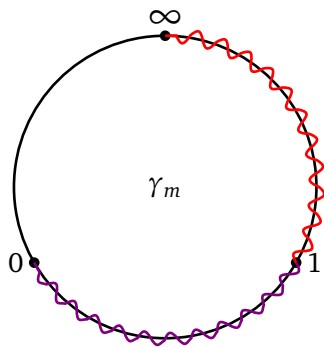

Figure 6: Blocks $\gamma_m$ regular in $(-\infty, 0)$.

### 3.3.2 Change of basis

Each of the collection of blocks, $\{\alpha_m\}$, $\{\beta_m\}$ and $\{\gamma_m\}$ are individually bases for the space of blocks. As a consequence, everywhere in the upper/lower half-plane, we have

$$\beta_m(z) = (F^\pm)^n_m \alpha_n(z), \tag{29}$$

where the "fusion" matrices $F^\pm$ are constants, independent of $z$. The two matrices $F^\pm$ are distinct from each other as moving from the upper to the lower half plane requires either the $\alpha$ block or the $\beta$ block or both to move through a cut. Similarly, in the upper/lower half-plane

$$\gamma_m(z) = (B^\pm)^n_m \alpha_n(z). \tag{30}$$

The relationship between $\beta$ and $\gamma$ can now be deduced from (29) and (30). On the upper/lower half plane we have

$$\gamma = B^\pm (F^\pm)^{-1} \beta. \tag{31}$$

### 3.3.3 Matrix implementation of motion in sheet space

Consider a (in general complicated) trajectory in $z$ space. The trajectory could, for instance, involve loops around the branch points, etc. We are interested in following the evolution of the normalized correlator (1) as we move along this path. This can be conveniently done as follows. Let us adopt the following convention: whenever the real part of $z$ lies between $(-\infty, 0)$, we use the expression (1) with blocks expressed in the $\gamma$ basis. When the real part of $z$ lies in the range $(0, \infty)$, we use the expression (1) with blocks expressed in the $\alpha$ basis. Finally, when the real part of $z$ lies in the range $(1, \infty)$, we use the expression (1) with blocks expressed in the $\beta$ basis.

If we adopt the convention described in the previous paragraph, we are compelled to change the basis whenever the real part of $z$ crosses 0, 1 or $\infty$. The advantage of adopting this convention is then, that we can always cross real $z$ in a completely smooth manner without ever encountering any cuts, as our basis blocks - by construction - are always regular along the real axis.

It follows that moving along any trajectory in $z$ space affects the expression (1) in the following way: the Pairing matrix gets multiplied, from the left, by a series of constant (i.e. $z$ independent) "basis change" matrices every time we move over (or under) any of the branch points $0, 1, \infty$. In subsection 3.2, we introduced the terminology "half-monodromy" for the process of passing over or under any branch point. We see that the discussion of this section has allowed us to represent each of the half-monodromy moves in terms of matrices.

As an example consider a trajectory that starts in the range $z \in (0, 1)$, loops around the branch point at unity in a counterclockwise manner, and then returns to its original location.

This operation turns the expression (1) into another expression of similar sort, but with $P$ replaced by $P'$ where $P' = (F^+)^{-1} F^- P$.

## 3.4 Euclidean single valuedness

In this paper, we are principally interested in correlators on the Lorentzian section ($z$ and $\bar{z}$ are both real). In this subsection, however, we study the simpler Euclidean section, on which $z$ and $\bar{z}$ are complex conjugates of each other. This section computes correlation functions in Euclidean space.

Now Euclidean correlators are single-valued. Suppose we start at some point $(z, \bar{z})$, and then move the locations of our four inserted operators in any way we like, ensuring, however, that the final configuration (at the end of the motion) has the same value of the cross-ratios, $(z, \bar{z})$, as the initial configuration. Single valuedness tells us that the correlator has the same value at the beginning and end of this motion. This condition is easily unpacked. Let us suppose that the motion we have undertaken involves $n$ "half-monodromy" operations (i.e. $n$ different crossings across $\text{Re}(z) = 0, 1, \infty$). According to the conventions of the previous subsection, each such "half-monodromy" move requires a change of basis and is implemented by the appropriate matrix multiplication.

Let the $i^{th}$ basis change matrix be denoted by $M_i$ (on the holomorphic side) and $\tilde{M}_i^*$ on the anti-holomorphic side.[24] It follows that the full motion effectively causes the pairing matrix $P$ to be transformed into

$$\tilde{M}_n^\dagger \cdots \tilde{M}_2^\dagger \tilde{M}_1^\dagger P M_1 M_2 \cdots M_n \,. \tag{32}$$

Single valuedness tells us that $P$ is invariant under this operation. In other words that

$$\tilde{M}_n^\dagger \cdots \tilde{M}_2^\dagger \tilde{M}_1^\dagger P M_1 M_2 \cdots M_n = P \,. \tag{33}$$

(33) can be rewritten as

$$\tilde{M}_n^\dagger \cdots \tilde{M}_2^\dagger \tilde{M}_1^\dagger P = P M_n^{-1} \cdots M_2^{-1} M_1^{-1} \,. \tag{34}$$

(34) tells us that any sequence of half-monodromy operations performed on $\bar{z}$ can be traded for a related sequence of half-monodromy operations on $z$.

Consider a configuration on the Lorentzian cylinder in which our four insertions are all inserted on a single spatial slice. Such a configuration has $z = \bar{z} =$ real and so lies both on the Lorentzian and the Euclidean sections. For this special class of configurations, the time-ordered correlator coincides with the Euclidean correlator.

Let us now move the insertion locations of the operators on the Lorentzian cylinder, away from this special configuration, but in such a way that all points always remain space-like separated with respect to each other. This constraint defines a region of the Lorentzian section that we call the Euclidean patch. Time-ordered correlators of the Euclidean patch are simple analytic continuations of Euclidean correlators.

The strategy we will adopt in this paper is the following. To reach a particular configuration of operator insertions, we will start with a configuration on the Euclidean patch, and then describe the branch moves (monodromies) we need to make to reach the configuration of interest.

In the rest of this section (i.e. in subsection §3.5 below) we study the Euclidean patch in more detail. This study will prove useful in section §6, where we will describe a protocol to move from configurations on the Euclidean patch to arbitrary configurations of interest.

---

[24]In the special case of a diagonal CFT, we can choose our basis of anti-holomorphic blocks to be complex conjugates of the holomorphic blocks. In this case $\tilde{M}_i = M_i$.

## 3.5 Ranges of $z$ and $\bar{z}$ for Euclidean configurations

It is not difficult to verify that - for Euclidean configurations - the following cyclical orderings map always yields conformal cross-ratios in the corresponding ranges as listed below.[25]

$$
\begin{aligned}
(1234) \quad \text{and} \quad (4321) &\to z \in (0,1), \quad && \text{and} \quad \bar{z} \in (0,1), \\
(1324) \quad \text{and} \quad (4231) &\to z \in (1,\infty), \quad && \text{and} \quad \bar{z} \in (1,\infty), \\
(2134) \quad \text{and} \quad (4312) &\to z \in (-\infty,0), \quad && \text{and} \quad \bar{z} \in (-\infty,0).
\end{aligned}
\tag{35}
$$

The rule (35) can be invariantly stated as follows. With any ordering of points (up to cyclical permutations), $(abcd)$, we associate the singular point $z_{ac} \equiv z_{bd} = z$ (see (27) for notation). Note that cyclical permutations and parity reflections of $(abcd)$ do not change this association. To the value of $z$, we then associate the unique range that does not include the point $z$ as one of its endpoints. Consider, for instance, the first of (35). The associated value of $z$ is $z_{24} = \infty$. The unique range that does not include $\infty$ as one of its endpoints is $(0,1)$, explaining the first line of (35).

For future use, it is useful to have names for the intervals of the real line that appear on the RHS of (35). Let us define

$$
\begin{aligned}
(-\infty,0) &= R_{23} = R_{14}, \\
(0,1) &= R_{24} = R_{13}, \\
(1,\infty) &= R_{21} = R_{34}
\end{aligned}
\tag{36}
$$

(the indices associated with particular ranges have been determined by the logic of the previous paragraph).

It is not difficult to verify that as we range over all Euclidean configurations, we obtain a full coverage[26] of the ranges specified in (35). Let us consider, for example, the case of the ordering (1234). In this case one standard configuration

$$
\begin{aligned}
\omega_1 &= 0, \quad \omega_2 = \omega, \quad \omega_3 = \frac{\pi}{4}, \quad \omega_4 = \frac{\pi}{2}, \\
\bar{\omega}_1 &= 0, \quad \bar{\omega}_2 = \bar{\omega}, \quad \bar{\omega}_3 = \frac{\pi}{4}, \quad \bar{\omega}_4 = \frac{\pi}{2}.
\end{aligned}
\tag{37}
$$

Using (25), it is easy to check that for this configuration $z = \tan\omega$ and $\bar{z} = \tan\bar{\omega}$. As $\omega$ ranges between 0 and $\frac{\pi}{4}$, $z$ ranges from 0 to unity, and an analogous statement is true of $\bar{z}$.

# 4 Path independence of four point functions

We would like to find the sheet location of a four-point correlator with the four operators inserted at the coordinates $(\omega_i, \bar{\omega}_i)$ with $(i = 1 \dots 4)$.[27]

As in our analysis of two and three-point functions in section 2, our strategy for this determination is to first evaluate the correlator at four points that are spacelike separated, and

---

[25]It is not possible to reverse the cyclic ordering of operators (along $\theta$) of our four insertion points while staying within the class of configurations for which all points are spacelike separated. Note that the orderings above - which were specified for the variable $\theta$ - are also the orderings of $\omega$ and $\bar{\omega}$ (this follows because all points are spacelike separated).

[26]While we have not attempted a careful proof of this claim, we believe that all mutually spacelike separated points with a given $z$ - and a specified ordering of $\omega_i$ - are related by $SL(2,R) \times SL(2,R) \times P$ transformations, where $P$ is the parity operation that takes $\theta$ to $-\theta$.

[27]As explained subsection 2.1, this labelling is convenient by redundant as

$$
(\omega_i + m_i\pi, \bar{\omega}_i + m_i\pi) \sim (\omega_i, \bar{\omega}_i),
$$

where $\sim$ means "is the same point as".

then continuously deform our insertion points until we have reached the points of interest, tracking half-monodromies in the process. In this section, we will demonstrate that the monodromy obtained via this process is independent of the path we choose to move on from the starting configuration to the configuration of interest (recall that a similar result held for two and three-point functions, see §2). Indeed, our demonstration of path independence closely mimics the corresponding analysis for three-point functions presented in §2.4. In particular we, once again, choose to work in a holographically factorized manner.

## 4.1 Points on the holomorphic causal lattice

We choose to insert operator 4 at the origin, and operators 1, 2, and 3 at

$$
\begin{aligned}
\omega_1 &= -m\pi - \alpha_1, \\
\bar{\omega}_1 &= \bar{m}\pi + \bar{\alpha}_1, \\
\omega_2 &= -n\pi - \alpha_2, \\
\bar{\omega}_2 &= \bar{n}\pi + \bar{\alpha}_2, \\
\omega_3 &= -p\pi - \alpha_3, \\
\bar{\omega}_3 &= \bar{p}\pi + \bar{\alpha}_3.
\end{aligned}
\tag{38}
$$

The relative causal orderings are determined by $m, n, p$ together with the relative orderings of $\alpha_i$ (an element of $S_3$).[28] We denote a configuration with $\alpha_i < \alpha_j < \alpha_k$ as an $(ijk)$ configuration. The holomorphic causal lattice is a 3-dimensional cubic lattice (whose points are labelled by the integers $(m, n, p)$), with a "6 atomized molecule" at each site (the 6 atoms are the 6 permutation elements $(ijk)$).[29] We then use the terminology

$$
\begin{aligned}
A_1^{m,n,p} &= m, n, p, \ (123), \\
A_2^{m,n,p} &= m, n, p, \ (231), \\
A_3^{m,n,p} &= m, n, p, \ (312), \\
B_1^{m,n,p} &= m, n, p, \ (132), \\
B_2^{m,n,p} &= m, n, p, \ (321), \\
B_3^{m,n,p} &= m, n, p, \ (213)
\end{aligned}
\tag{39}
$$

(and similar expressions in the anti-holomorphic sector), to denote the causal configuration (or branch structure) corresponding to the insertions (38).[30, 31]

In summary, the holomorphic causal lattice is a three-dimensional cubic lattice with each lattice site hosting a "benzene molecule" whose atoms are one of the 3 As or one of the 3 Bs, listed in (39).

## 4.2 Links on the holomorphic causal lattice

A deformation of insertion points that results in one (holomorphic) light cone crossing is a link - or a bond - on the holomorphic causal lattice. Four links radiate out of each lattice point. Consider the lattice point $m, n, p, (ijk)$. The links emanating out of this point are the moves

---

[28]We will use the symbols $m, n, p$ rather than $m_1, m_2, m_3$ (the analogue of our notation for 3 point functions), because these symbols will occur in many places below, and we find symbols without subscripts easier to read.

[29]Notice that $\omega_i = -m_i\pi - \alpha_i$ whereas $\bar{\omega}_i = \bar{m}_i\pi + \bar{\alpha}_i$. The minus sign convention for $\omega_i$ is so that increasing $m_i$ and $\alpha_i$ will mean increasing time direction, just like increasing $\bar{m}$ and $\bar{\alpha}$ mean increasing time direction for $\bar{\omega}$. Recall that this is due to definitions $\omega_i = \dfrac{\theta_i - \tau_i}{2}$ and $\bar{\omega}_i = \dfrac{\theta_i + \tau_i}{2}$.

[30]In other words, $A_1$ refers to the atom at cubic lattice site $(m, n, p)$ and with $\alpha_1 < \alpha_2 < \alpha_3$, and so on.

[31]Notice that $A_1, A_2$ and $A_3$ are cyclically related : the same is true of $B_1, B_2, B_3$.

$B, P_1, P_2, F$, which are defined as follows. $B$ moves $O_i$ backwards in time (so forward in $\omega_i$) till it cuts the holomorphic lightcone emanating out of $O_4$. $P_1$ moves $O_i$ forward in time (so backward in $\omega_i$) till it cuts the holomorphic lightcone centered at $O_j$. $P_2$ takes $O_j$ forward in time (so backward in $\omega_j$) till it cuts the holomorphic lightcone centered at $O_k$. And $F$ takes $O_k$ forward in time (so backwards in $\omega_k$) till it cuts the holomorphic lightcone centered at $O_4$.

Each of $B, P_1, P_2, F$ maps the lattice point $(m, n, p), (ijk)$ to another lattice point, whose value we list below. $P_1$ acts as

$$
\begin{aligned}
A_1^{mnp} &\leftrightarrow B_3^{mnp}, \\
A_2^{mnp} &\leftrightarrow B_2^{mnp}, \\
A_3^{mnp} &\leftrightarrow B_1^{mnp},
\end{aligned}
\tag{40}
$$

$P_2$ acts as

$$
\begin{aligned}
A_1^{mnp} &\leftrightarrow B_1^{mnp}, \\
A_2^{mnp} &\leftrightarrow B_3^{mnp}, \\
A_3^{mnp} &\leftrightarrow B_2^{mnp},
\end{aligned}
\tag{41}
$$

The move $F$ that takes the last $\alpha_i$ "ahead" (past the next spiral of the lightcone centered on $\omega_4$) acts as

$$
\begin{aligned}
A_1^{mnp} &\to A_3^{m,n,p+1}, \\
A_2^{mnp} &\to A_1^{m+1,n,p}, \\
A_3^{mnp} &\to A_2^{m,n+1,p}, \\
B_1^{mnp} &\to B_3^{m,n+1,p}, \\
B_2^{mnp} &\to B_1^{m+1,n,p}, \\
B_3^{mnp} &\to B_2^{m,n,p+1}.
\end{aligned}
\tag{42}
$$

Finally, the move $B$ that takes the "first $\alpha_i$ one behind (that crosses the lightcone out of $O_4$ to the past) is simply the inverse of the move above, i.e.

$$
\begin{aligned}
A_1^{mnp} &\to A_2^{m-1,n,p}, \\
A_2^{mnp} &\to A_3^{m,n-1,p}, \\
A_3^{mnp} &\to A_1^{m,n,p-1}, \\
B_1^{mnp} &\to B_2^{m-1,n,p}, \\
B_2^{mnp} &\to B_3^{m,n,p-1}, \\
B_3^{mnp} &\to B_1^{m,n-1,p}.
\end{aligned}
\tag{43}
$$

Fig 7 gives a pictorial representation of the connections or edges between neighbouring vertices on the lattice described in (42) and (43). Note that the connections link one kind of "atom" on a given lattice point, to a related but distinct "atom" on the neighbouring point. As illustrated in Fig 7 for example, $A_1$ type "atoms" link to $A_2$ atoms at one lower value of $m$, but to $A_3$ type atoms at one larger value of $p$.

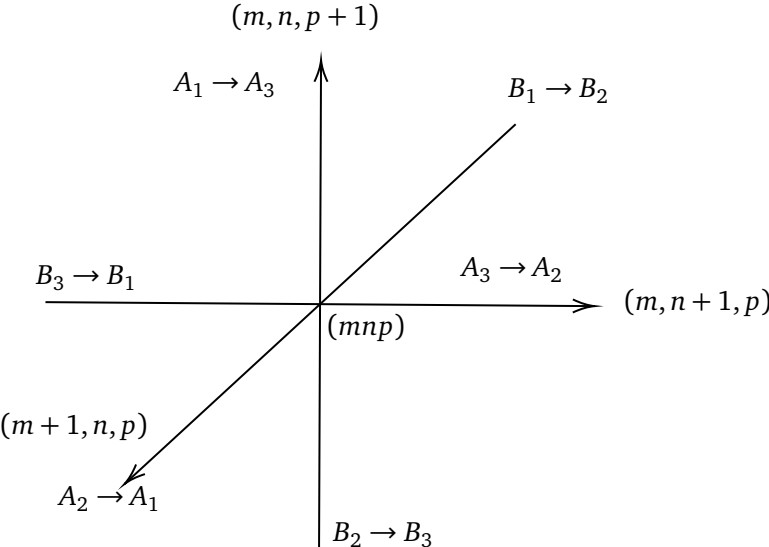

Figure 7: Links on the causal lattice involving "atoms" at the site $(m, n, p)$.

## 4.3 Vanishing of mixed holonomies and triviality of winding

As in our study of three-point functions §2.4.3, we call any continuous collection of links that begins at $A$ and ends at $B$ a trajectory from $A$ to $B$. Any continuous deformation of operator insertion points - from a configuration that lies in the causal class $A$ to a second configuration that lies in the causal class $B$ - travels along some trajectory from $A$ to $B$. In this section, we will demonstrate that the monodromy associated with any such motion of insertion points depends on the endpoints $A$ and $B$, but is independent of the details of the trajectory.[32] Equivalently, we demonstrate that every closed loop produces a trivial monodromy. To establish this result, it is sufficient to demonstrate the vanishing of the monodromy associated with every elementary plaquette on the causal lattice. These plaquettes are of two types; those that lie entirely in the holomorphic (or entirely in the anti-holomorphic) lattice and those that are mixed.

Mixed monodromies vanish if and only if the holonomies associated with left and right moving motions commute with each other. That this is the case follows immediately from subsection 3.3.[33] It remains only to show that the holonomies around purely holomorphic and purely anti-holomorphic plaquettes all vanish. We turn to this point in the next subsection.[34]

## 4.4 The holonomy of a purely holomorphic closed loop vanishes

To complete our demonstration of single valuedness (as mentioned above) it remains to show that the monodromy associated with a purely holomorphic closed loop (or purely anti-holomorphic closed loop) vanishes. In the rest of this section, we present a demonstration of this point.

Recall that the holomorphic causal lattice consists of 6 "atoms" on each site of a cubic lattice. It is useful to view this lattice as a fibration, with the cubic lattice as the base, and the six atoms as the fibre. To establish path independence,

---

[32]I.e. that the gauge field associated with monodromies is flat.

[33]This result - together with the analysis of Appendix C.2 - then tells us that the monodromy associated with a closed loop in $\omega_i$ space at fixed $\bar{\omega}_i$ is independent of the value of $\bar{\omega}_i$.

[34]A motion that increases $m_i$ by an integer, but decreases $\bar{m}_i$ by the same integer also carries trivial monodromy. Such a motion causes the insertion point of $O_i$ to wind around the circle of the cylinder an integer number of times. The single valuedness of correlators under such a winding is guaranteed on general grounds by the fact that $h_i - \bar{h}_i$ is an integer. For completeness, in Appendix B.2 we analyze the monodromy associated with one such winding motion and verify that it is indeed trivial.

1. We first demonstrate that the monodromy vanishes for any closed loop contained entirely in the fibre (at any given base point). Recalling that the fibre is the group manifold $S_3$, proof of this point follows as a special case of the analysis presented in Appendix E.1.

2. We then demonstrate the vanishing of the holonomy associated with plaquettes that are part in the fibre and part in the base - but are trivial when projected down to the base.

3. Finally we study the projection of paths down to the base and demonstrate the vanishing of the holonomy on any convenient representative for plaquettes of this projection.[35]

### 4.4.1 Triviality of loops on a "Hexagon molecule" at a given location

Let us first start with the analysis of paths that are identical on the base but different on the full lattice. Using the fact that the 6 atoms in a fibre are elements of $S_3$, it follows as a special case of the analysis of Appendix E.1 that loops on a given fibre are trivial.

### 4.4.2 Triviality of mixed loops that descend to trivial loops on the base

Let us now consider loops that involve motion in both the fibre and the base, but are trivial (i.e. involve only paths that simply retrace themselves) when projected to the base.

Recall that the cubic lattice points $(m, n, p)$ and $(m+1, n, p)$ are connected by the following two links:

$$F(A_2^{m,n,p}) = A_1^{m+1,n,p}, \qquad F(B_2^{mn,p}) = B_1^{m+1,n,p}. \tag{44}$$

Noting also that

$$\begin{aligned}
P_1(A_2^{m,n,p}) &= B_2^{m,n,p}, & P_1(B_2^{m,n,p}) &= A_2^{m,n,p}, \\
P_2(A_1^{mn,p}) &= B_1^{m,n,p}, & P_2(B_1^{mn,p}) &= A_1^{m,n,p},
\end{aligned} \tag{45}$$

we see that the operation

$$BP_2FP_1(B_2^{m,n,p}), \tag{46}$$

generates a closed loop. When this loop is projected down to the base it appears to reduce to the trivial sequence of operations $(m, n, p) \to (m + 1, n, p) \to (m, n, p)$. On the full lattice, however, the operation takes us around a square - and so is not necessarily trivial. This square, plus its (cube) reflected counterparts

$$BP_2FP_1(A_3^{m,n,p}), \tag{47}$$

and

$$BP_2FP_1(B_3^{m,n,p}), \tag{48}$$

"generate" all nontrivial paths (in the space of trajectories that appear trivial on the cubic base space).[36] In Appendix E.2 we verify, however, that each of these three "generator nontrivial loops" has a trivial monodromy, completing our demonstration of (1) above.

---

[35]Two plaquettes that are different in the full lattice may be identical in projection down to the base. In this situation it is now sufficient to demonstrate the vanishing of holonomies for any one of these cases; the vanishing for all others then follows from the first two points above.

[36]The projection of these two trajectories onto the base cube reduce, respectively, to $(m, n, p) \to (m, n + 1, p) \to (m, n, p)$ and $(m, n, p) \to (m, n, p + 1) \to (m, n, p)$.

### 4.4.3 Triviality of loops on the cubic face

We now turn to the study of nontrivial loops on the base cubic lattice. The traversal around each of the three faces of the cube gives a basis for loops on this lattice: to show path independence we must show that the monodromy associated with each of these face traversals is trivial. One set of operators generates a closed loop that descends, on the base cubic lattice to

$$(m, n, p) \rightarrow (m+1, n, p) \rightarrow (m+1, n+1, p) \rightarrow (m, n+1, p) \rightarrow (m, n, p),$$

is

$$P_1 P_2 P_1 B P_2 B P_1 F P_2 F \left( A_2^{m,n,p} \right).$$

In Appendix E.3 we verify that the monodromy associated with this closed path - as well as those for paths on the two other faces of the cube - are trivial. This completes our demonstration of the path of independence of all monodromies.

## 5 Monodromies for all causal configurations on a single Lorentzian diamond

In this section, we focus on configurations in which all four operators are inserted in a single Minkowski diamond (equivalently, in ordinary Minkowski space). We proceed to determine the sheet monodromy associated with any such insertions. We address this question for two reasons. First, because of its intrinsic physical interest (in practice we are often interested in the dynamics of a conformal field theory in Minkowski space). Second, as useful input for the analysis of the same question on the Lorentzian cylinder (recall that the Minkowski diamond forms a unit cell for this cylinder: see around Fig. 1a).

The procedure we adopt to work out the sheet structure is the following. Consider a Minkowski diamond centered at $\tau = 0$. Say we are interested in insertions at the locations $(\omega_i, \bar{\omega}_i)$. We begin, instead, by inserting the operators at the locations $(\alpha_1, \alpha_i)$ with $\alpha_i = \bar{\omega}_i$. In other words, we insert all operators on the spatial slice $\tau = 0$, and at the correct values of $\bar{\omega}_i$. All operators are spacelike separated on the starting location, and so the correlator starts on the Euclidean sheet. We then continuously deform $\omega_i$ - always staying at constant values of $\bar{\omega}$ - until we reach the locations of interest. In the process of moving from our initial to final configuration, we are forced to cross several lightcones. Each of these crossings results in a leftmoving (or holomorphic) "half-monodromy", whose nature is dictated by the rules of subsection 3.2. By keeping track of all these various half-monodromies (and their order) we obtain our final result for the monodromy location (w.r.t. the Euclidean sheet) associated with any given insertion locations.[37]

In the rest of this section, we simply present the results of the implementation of the algorithm spelt out in the previous paragraph.

### 5.1 Configurations on the Euclidean sheet

It turns out that several causally distinct classes of configurations lie on the Euclidean sheet; these configurations are all enumerated In Figure 8. All of these configurations correspond to

---

[37]While we choose our path (that connects initial to final locations) in a purely left-moving manner, this does not, of course, completely determine the path, as we could first move particle 1, then particle 3, then particle 4... or chose some other order. In the previous section, we demonstrated that the final result does not depend on the details of this choice (see Appendix E.1 for a direct demonstration of this point in the case of the Minkowski Diamond). In the analysis of this section, as a consequence, we choose our order of operator motions in any convenient manner.

values of $z$ and $\bar{z}$ in the same range (e.g. they both lie in the range $(0, 1)$ or both in $(1, \infty)$ or $(-\infty, 0)$; see (35)).

The conventions of Fig 8 are as follows. The figure specifies the causal structure in the Minkowski diamond. Black dots represent operator insertion locations. Blue lines represent light cones emanating out of operators. For instance, the first diagram in Figure 8 denotes a configuration in which all points are spacelike separated w.r.t. each other (i.e. the Euclidean configuration studied in section §3.5). The second diagram in the figure depicts a configuration in which three points are mutually spacelike, and they all lie to the past of the fourth point. The third diagram depicts the time reversal of the second. The fourth and fifth diagrams depict configurations in which three points are timelike separated from each other - but are all spacelike separated w.r.t. the last point. The final diagram in the Figure depicts a configuration in which all four points are timelike separated w.r.t. each other. Correlators in each of these configurations turn out to lie on the Euclidean sheet.

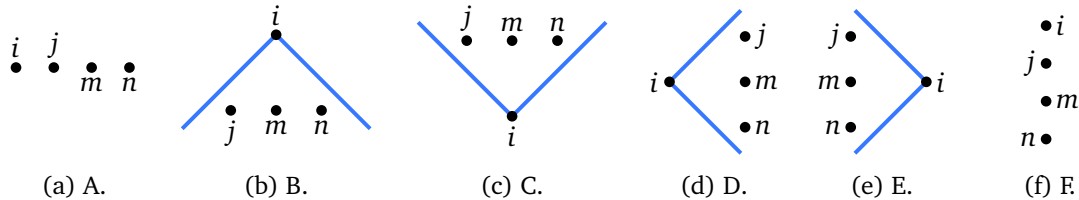

(a) A.     (b) B.     (c) C.     (d) D.     (e) E.     (f) F.

Figure 8: Configurations on the Euclidean sheet.

We have already mentioned that all the configurations in Fig 8 have values of $z$ and $\bar{z}$ that lie in the same ranges (35). The rules that determine what this range is - whether $(-\infty, 0)$, $(0, 1)$ or $(1, \infty)$ are given as follows

- Fig 8a : The rule depends on the cyclical ordering of the points along the spatial, is invariant under reflection, and is presented in (35).

- Fig 8b, 8c : In this case, one translates the future/past operator along the blue line until it is spacelike related to the other three operators, and then uses the rule for Fig 8a presented above.[38]

- Fig 8f : In this case, one translates all the top three operators along leftmoving past lightcones (or rightmoving past lightcones) and moves them until they are spacelike related to each other. After doing this one uses the rule of Fig 8a for the resultant configuration.

- Fig 8d, 8e : In this case, one translates the special operator along the blue line until it is either to the past or to the future of all the other three operators. One then uses the rule for Fig. 8f for the resultant configuration.

## 5.2 Configurations that are one crossing away from the Euclidean sheet

All configurations that are a single crossing away from a Euclidean configuration are listed in Fig. 9 (together with the Euclidean configurations that they are related to by a single crossing). For each of these configurations, we can deduce which final range $(X, Y)$[39] the cross-ratio lies in - and how one transits to this range from the associated nearby Euclidean configuration $(Y, Y)$ as follows.

---

[38] We get the same answer if we translate along either the left or right lightcone because the rule of (35) depends only on the cyclical ordering of operators. Similar comments apply to all the subsequent rules presented below.

[39] Here $X, Y$ are any of $R_i$, $i = 1, 2, 3$.

We start from the Euclidean configuration listed in the first column of Fig. 9. The cross-ratios for this configuration lie in the range $(Y,Y)$ where the value of $Y$ can be deduced from the rules presented in the previous subsection. We then move the special operator along the left moving light cone (i.e. along a line of constant $\theta + \tau$) until it reaches its final position. This motion involves a single crossing of a $z$ lightcone at either $z = 0$ or $z = 1$ or $z = \pm\infty$. The cutting always happens at the value of $z$ that lies at one of the two boundaries of $Y$. (For instance, if $Y = (0,1)$, the light cone we cross in our motion will always be either $z = 0$ or $z = 1$). The range $X$ equals the range that neighbours $Y$ along this boundary. We make this move either above or below the branch point (at the boundary of the ranges $X$ and $Y$): whether above or below is determined by the rules listed in subsection §3.2.

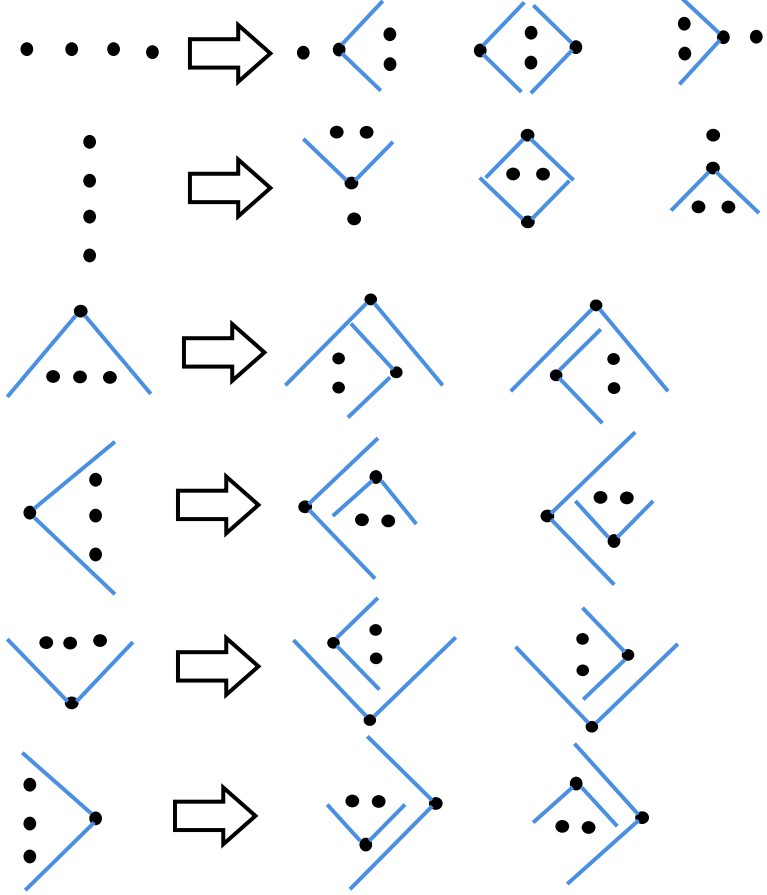

Figure 9: The left column represents the Euclidean configurations presented in §8. The configurations on the right side are configurations that are one crossing away from the corresponding Euclidean sheet.

The rules presented here are very easy to implement. We illustrate this in a couple of examples.

Consider the second diagram on the first row of Fig 9, with the particular choice of the ordering of operators shown in Fig. 10. In this example, the starting Euclidean configuration lies in the range $(R_2, R_2)$. To go from this configuration to the one in interest, one moves operator 4 towards the future along the left-moving light cone. In this process, we cut the rightmoving light cone emanating from operator 3 (see Fig 10). The corresponding branch point lies at $z = z_{34} = 0$. It follows that the final configuration for this figure lies in the range $(R_2, R_1)$. According to the first of the rules in subsection §3.2, we must make the traversal from $(R_2, R_2)$ to $(R_2, R_1)$ in an anticlockwise manner. It follows that, in going from $R_2$ to $R_1$, we cross the branch point at $z = 0$ from above, as illustrated in Fig 11.

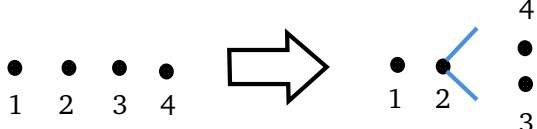

Figure 10: An example of configurations that are one crossing away from Euclidean sheet.

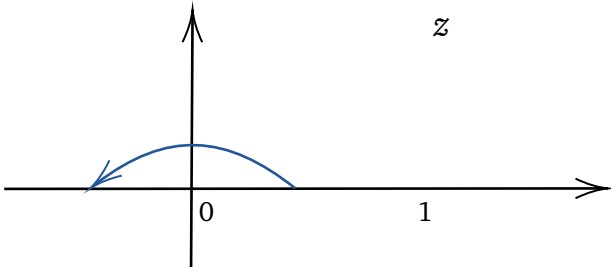

Figure 11: Monodromy related to Fig. 10.

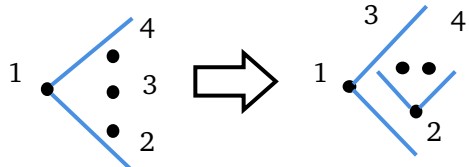

Figure 12: Another example of one crossing away configurations from Euclidean sheet.

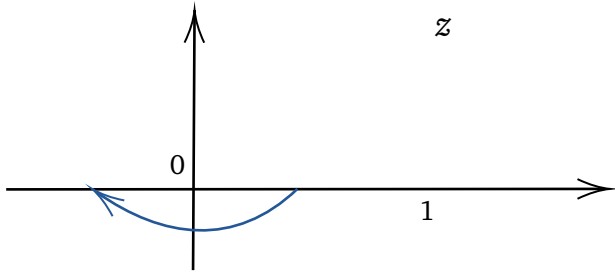

Figure 13: Monodromy related to Fig. 12.

As a second example, consider the third diagram on the fourth row of Fig 9, with the particular choice of the ordering of operators shown in Fig. 12. In this example, the starting Euclidean configuration lies in the same range $(R_2, R_2)$. To go from this configuration to the one in interest, one moves operator 4 towards the past along the left-moving light cone. In this process, we cut the future rightmoving light cone of 3 from future to past. This corresponding branch point (in cross-ratio space) lies at $z = z_{34} = 0$. It follows that the final configuration for this figure lies in the range $(R_2, R_1)$. According to the second of the rules in subsection §3.2, we must make the traversal from $(R_2, R_2)$ to $(R_2, R_1)$ in a clockwise manner. It follows that, in going from $R_2$ to $R_1$, $z$ crosses the branch point at $z = 0$ from below, as illustrated in Fig. 13.

## 5.3   Configurations related to a Euclidean configuration by two crossings

In the previous subsection, we presented a classification of all configurations that lie "one crossing" away from a configuration on the Euclidean sheet. In this subsection, we present a classification of all configurations that lie "two crossings" away from an Euclidean sheet configuration. Fortunately, these are the most complicated configurations we will need to consider: it turns out that all correlators on the Minkowskian diamond are at most two crossings away from an Euclidean sheet configuration.

Consider some cross-ratios $(z, \bar{z})$ that lie on the Euclidean sheet. It follows that $z$ and $\bar{z}$ both lie in the same $R_i$ range. Configurations that lie two crossings away from this Euclidean sheet configurations are all depicted in Fig. 14, and of two qualitatively different types. Configurations of the first type are depicted in the first two diagrams in Fig 14. We will call the configurations in the first (14a) and second (14b) diagrams Regge and scattering configurations respectively. It is easy to check that both these configurations have cross-ratios with values of $z$ and $\bar{z}$ that both lie in the range $R_{im}$ (see (36) for definitions). This is the same range of $z$ and $\bar{z}$ variables that we find for appropriate Euclidean configurations.

Configurations depicted in subfigures 14c and 14d, on the other hand, are of the second type. These have cross-ratios $z$ and $\bar{z}$ in different $R_i$ ranges.

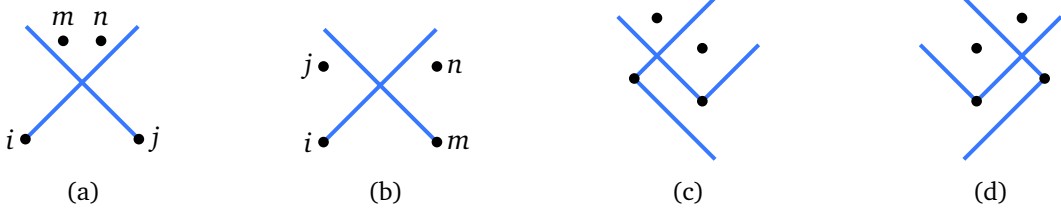

Figure 14: Configurations which are two monodromy away from the Euclidean sheet. The first configuration is known as the "scattering configuration" whereas the 2$^{\text{nd}}$ configuration is known as the "Regge type" configuration.

Let us first study configurations of the first sort, i.e. the configurations depicted in the first two figures of Fig. 14. Even though $z$ and $\bar{z}$ lie in the same $R_i$ region in this case, it is not difficult to verify that in this case, the correlators do not lie on the Euclidean sheet. The sheet these correlators lie on is obtained by starting on the Euclidean sheet and making either a single clockwise or a single anticlockwise rotation (around the appropriate branch point), depending on whether it is scattering or Regge configuration respectively. We now explain this point in more detail.

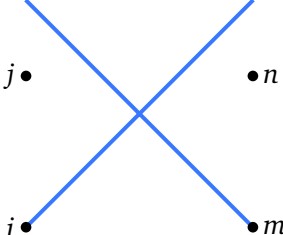

Figure 15: Regge configuration.

**Regge configurations:**   The Regge configurations depicted in figure 15 can be constructed from the all space-like configuration ($ijmn$). Starting from this space-like configuration we move $P_j$ along a left-moving light cone, in the process crossing the future right-moving light

cone of $P_i$ from past to future. Using the rules of §3.2 the effect of this move, in cross-ratio space, is an anticlockwise rotation around the point $z_{ij}$ by an angle $\pi$, i.e. the half-monodromy $\sqrt{A_{ij}}$. Next move $P_n$ to the future along its left-moving light cone. In the process $P_n$ cuts the future right-moving light cone emanating at $P_m$ from past to future. Once again, the effect of this move, in cross-ratio space, is a second half-monodromy $\sqrt{A_{ij}}$. It follows that the net effect of the two motions described above is an $A_{ij}$ i.e. a complete $2\pi$ anticlockwise rotation around $z_{ij}$.

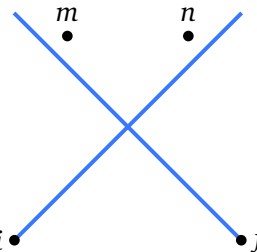

Figure 16: Scattering configurations.

**Scattering configurations:**    The scattering configurations depicted in figure 16 can also be constructed from the all space-like configuration ($ijmn$). Unlike the Regge case, the motion that takes us from the Euclidean configuration to the scattering configuration involves (at least) four light crossings. Starting from the space-like configuration we move $P_m$ along its left-moving light cone, causing it to cross the future of the right-moving light cone of $P_j$ and $P_i$ from past to future consecutively. Using §3.2 rules, the impact of this motion in cross-ratio space is a $\sqrt{A_{jm}}$ followed by a $\sqrt{A_{im}}$. Next, we take $P_n$ to the future along its left moving light cone, causing it to cross the right moving light cones of $P_j$ and $P_i$ in that order. The impact of this motion on cross-ratio space is a $\sqrt{A_{jn}}$ followed by a $\sqrt{A_{in}}$.[40] So, the net effect (in cross-ratio space) of these motions is $\sqrt{A_{jm}} \cdot \sqrt{A_{im}} \cdot \sqrt{A_{jn}} \cdot \sqrt{A_{in}}$. The two middle half-monodromies combine to give an $A_{im}$, so our net monodromy operation simplifies to $\sqrt{A_{jm}} \cdot A_{im} \cdot \sqrt{A_{jm}} = C_{ij}$ (the reader will find it easy to verify the last equality once she puts pen to paper, for instance by choosing any convenient specific values for $i$, $j$, $m$, $n$).

**Configurations of the second type:**    The remaining configurations in figure 14 do not go through a full monodromy rather they change the range of the cross-ratios.[41] The locations associated with these configurations (in branch ratio sheet space) are easily worked out along the lines of the discussion for Regge and Scattering configurations. We leave the detailed explication to the interested reader.

---

[40]Through this paper, when we list a sequence of half-monodromies in the form $H_1.H_2.H_3 \ldots H_r$ we mean the half-monodromy $H_1$ followed by $H_2 \ldots$ followed by $H_r$.

[41]We emphasize that moves described above generically change the values of cross-ratios (the cross-ratios in the final configuration never equal those in the initial starting configuration). This fact is obvious for configurations of the second type (as the moves above change the ranges of $z$ and $\bar{z}$), but is also true for configurations of the first type. The left-moving motion of a single operator always changes cross-ratios ( unless the motion changes $\omega$ by a multiple of $\pi$, but such moves always take one out of the Minkowskian diamond). While the simultaneous left-moving motion of two operators can leave $z$ invariant, this requires fine-tuning of the final operator locations within the Regge/scattering configurations.

# 6 Moving to arbitrary configurations

In the previous section, we have given a complete classification of all configurations for which our four insertion points all lie within a Minkowskian diamond. Of course, not every configuration (of insertion of four operators) on the Minkowskian cylinder has all four insertion points within a single Minkowskian diamond. As the Minkowski diamond forms a unit cell for the Lorentzian cylinder (see around Fig 1b) however, every collection of insertion points on the cylinder *can* be obtained starting from some configuration within a single Minkowskian diamond and making shifts $\omega_i \to \omega_i \pm n_i \pi$, with all values of $\bar{\omega}_i$ held constant.[42] Such moves do not change the cross-ratios associated with the collection of insertion points (see (23)), but, in general, result in monodromy operations. In this section we compute the monodromies that result from any such move - and so for any given choices for insertion locations on the Lorentzian cylinder.

## 6.1 Notation for monodromies

In this subsection, we briefly remind the reader of our notation for monodromies (already briefly introduced in §3.2). We denote clockwise half-monodromy around the branch point $z_{ij}$ by $\sqrt{C_{ij}}$. Similarly, an anticlockwise half-monodromy around $z_{ij}$ is denoted by $\sqrt{A_{ij}}$. We use the notation $\sqrt{M_1} \cdot \sqrt{M_2} \cdot \sqrt{M_3} \cdots \sqrt{M_n}$ to denote the operation of the half-monodromy $\sqrt{M_1}$ followed by the half-monodromy $\sqrt{M_2}$ followed by the half-monodromy $\sqrt{M_3}$, etc.

We define a full clockwise monodromy, $C_{ij}$ around the branch point $z_{ij}$ as two half-monodromies around $z_{ij}$ that follow immediately after each other (i.e. without any other half-monodromies occurring in between). In equations

$$\sqrt{C_{ij}} \cdot \sqrt{C_{ij}} = C_{ij}. \tag{49}$$

Similarly

$$\sqrt{A_{ij}} \cdot \sqrt{A_{ij}} = A_{ij}, \tag{50}$$

where $\sqrt{A_{ij}}$ is the anticlockwise half-monodromy around $z_{ij}$. On the other hand

$$\sqrt{C_{ij}} \cdot \sqrt{A_{ij}} = \sqrt{A_{ij}} \cdot \sqrt{C_{ij}} = \phi, \tag{51}$$

where $\phi$ is the trivial monodromy. Note, of course, that half-monodromies do not commute with each other. For example, starting at a value of $z$ in $(1, \infty)$, and executing the motion $\sqrt{A_{23}} \cdot C_{12} \cdot \sqrt{C_{23}}$ takes us along the path depicted in Fig 17.

---

[42]Rewritten in terms of $\theta_i$ and $\tau_i$, this motion amounts to $\tau_i \to \tau_i - n_i \pi$, $\theta_i \to \theta_i + n_i \pi$. In the special case that $n_i$ is even, $n_i = 2m_i$, this motion gives the same monodromy as $\tau_i \to \tau_i + m_i(2\pi)$ at fixed $\theta_i$ (recall that winding $\theta_i$ around the circle does not result in a monodromy).

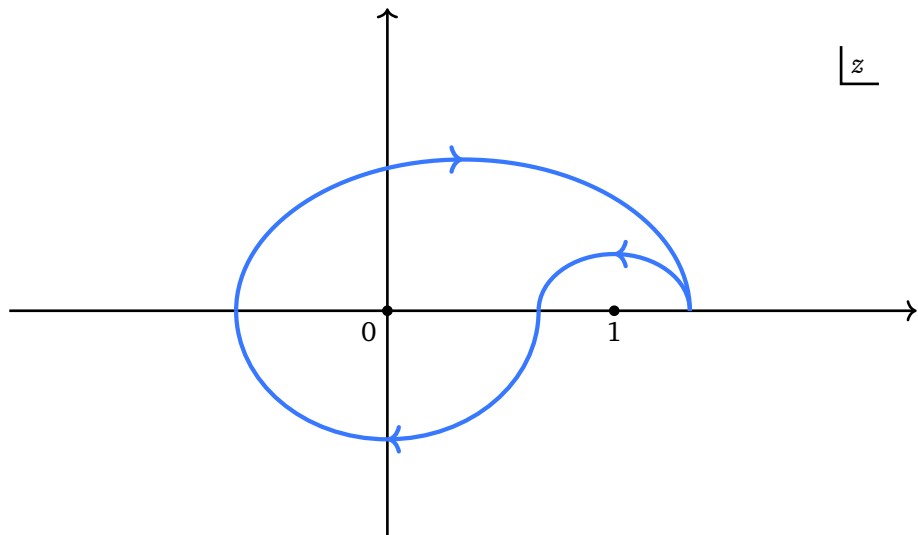

Figure 17: Executing the motion $\sqrt{A_{23}} \cdot C_{12} \cdot \sqrt{C_{23}}$ starting from the range $(1, \infty)$.

This path is clearly different from $C_{12}$; indeed one cannot even execute a $C_{12}$ starting from $z \in (1, \infty)$ as the starting range does not border $z_{12} = 0$.

## 6.2  Monodromies induced by $\omega_i \to \omega_i \pm \pi$

If a set of insertion locations can be obtained starting from some given insertions on a given Minkowski diamond and then making the shifts

$$\omega_i \to \omega_i - m_i \pi, \qquad \bar{\omega}_i \to \bar{\omega}_i + \bar{m}_i \pi, \qquad i = 1, 2, 3, 4,$$

it follows from the periodicity of $\theta$ that the same insertion locations can also be obtained starting from the same initial condition and making the purely holomorphic shifts,

$$\omega_i \to \omega_i - (m_i + \bar{m}_i)\pi, \qquad \bar{\omega}_i \to \bar{\omega}_i, \qquad i = 1, 2, 3, 4. \tag{52}$$

In this subsection, we will derive the rules for monodromies induced by shifts of the form (52), i.e. for shifts of $\omega_i$ by integral multiples of $\pi$, with $\bar{\omega}_i$ are held fixed.

In making the $\omega_i$ shift $\omega_i \to \omega_i \pm \pi$, we cross the right moving lightcone emanating out of each of the particles $P_j$ ($j \neq i$) exactly once. We could cut the lightcone emanating out of any of the other particles - say the one at $P_m$ - either to the past of $P_m$ or to the future of $P_m$. In other words, we could (depending on details) cut either the future or the past rightmoving lightcone of $P_m$. This is true for each of the three values of $m$. We thus have many possibilities. Using the rules listed in section §3.2, it is easy to work out the following set of rules for the (future directed) move $\omega_i \to \omega_i - \pi$ with all other $\omega_j$ (and all $\bar{\omega}_m$) held fixed.[43]

1. If the crossed lightcones, emerging from all $P_j$ insertions are all future or all past lightcones (so that all three crossings happen with past lightcones or all three crossings happen with future lightcones) then the resulting move results in no monodromy i.e. the identity monodromy $\phi$.

2. If the move results in one crossing of the past lightcone emerging from $P_j$, but only crossing of future lightcones emerging from the other two insertions, the resultant monodromy depends on the order of these crossings.

---

[43]Note that the motion $\omega \to \omega - \pi$ is a future directed left moving shift, as $\omega = \frac{\theta - \tau}{2}$.

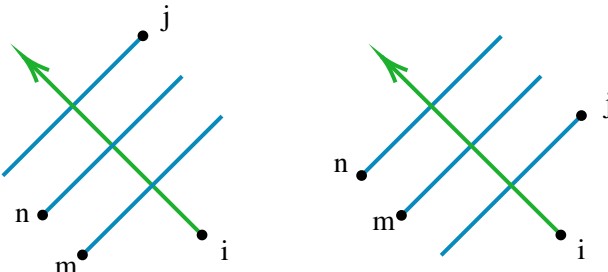

Figure 18: Crossing of two consecutive future light cones from past to future.

If we first cross the two future lightcones and then the past lightcone, or first cross the past lightcone and then the two future lightcones, then the resultant monodromy is a clockwise circle around $z_{ij}$, i.e, $C_{ij}$.

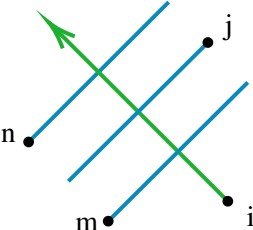

Figure 19: Crossing of two future light cones with a past light cone in the middle from past to future.

On the other hand, if we first cut a future light-cone emanating from $P_m$, then past light-cone emanating from $P_j$, future light-cone emanating from $P_n$ the resultant monodromy is first $\sqrt{A_{im}}$ then $C_{ij}$ then $\sqrt{C_{im}}$. Note that this sequence of moves cannot be represented as a single monodromy around any branch point. So the answer is

$$\sqrt{A_{im}} \cdot C_{ij} \cdot \sqrt{C_{im}} \equiv \sqrt{C_{in}} \cdot C_{ij} \cdot \sqrt{A_{in}}. \tag{53}$$

3. If the move results in one crossing of the future lightcone emerging from $P_j$, but only crossing of past lightcones emerging from the other two insertions, the resultant monodromy again depends on the order of these crossings.

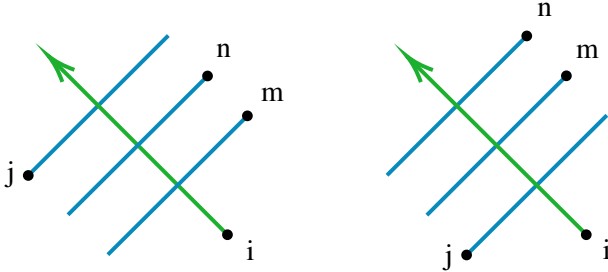

Figure 20: Crossing of two consecutive past light cones from past to future.

If we first cross the two past lightcones and then the future lightcone, or first cross the future lightcone and then the two past lightcones, then the resultant monodromy is an anticlockwise circle around $z_{ij}$, i.e, $A_{ij}$.

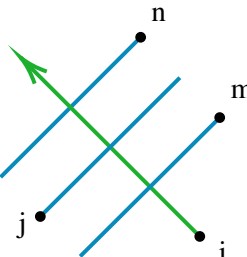

Figure 21: Crossing of two past light cones with a future light cone in the middle from past to future.

If, on the other hand, the order of crossings is first past of $P_m$ then future of $P_j$ and then past of $P_n$ then the resultant monodromy is $\sqrt{C_{im}}$ followed by $A_{ij}$ followed by $\sqrt{A_{im}}$, i.e.,

$$\sqrt{C_{im}} \cdot A_{ij} \cdot \sqrt{A_{im}} \equiv \sqrt{A_{in}} \cdot A_{ij} \cdot \sqrt{C_{in}}. \tag{54}$$

The (past directed) reverse motion, $\omega_i \to \omega_i + \pi$, undoes the monodromies described above, and so results in the following monodromies:

1'. If the crossed lightcones, emerging from all $z_j$ insertions are all future or all past light-cones (so that all three crossings happen with past lightcones or all three crossings happen with future lightcones) then the resulting move results in no monodromy i.e. the monodromy $\phi$.

2'. If the move results in one crossing of the past light cone emerging from $P_j$, but only the crossing of future light cones emerging from the other two insertions, the resultant monodromy depends on the order of these crossings.

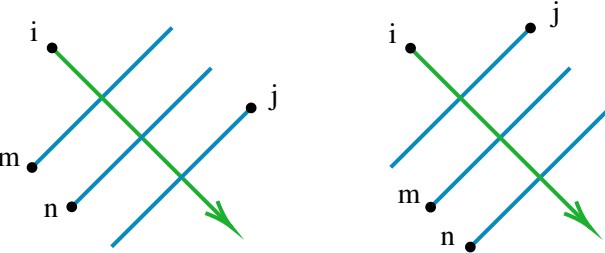

Figure 22: Crossing of two consecutive future light cones from future to past.

If we first cross the two future lightcones and then the past lightcone, or first cross the past lightcone and then the two future lightcones, then the resultant monodromy is an anticlockwise circle around $z_{ij}$, i.e, $A_{ij}$.

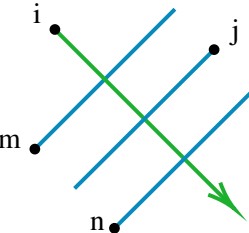

Figure 23: Crossing of two future light cones with a past light cone in the middle from future to past.

On the other hand, if we first cut a future light cone emanating from $P_m$, then a past light cone emanating from $P_j$, future light cone emanating from $P_n$ the resultant monodromy is first $\sqrt{C_{im}}$ then $A_{ij}$ then $\sqrt{A_{im}}$. Note that this sequence of moves cannot be represented as a single monodromy around any branch point. So the answer is

$$\sqrt{C_{im}} \cdot A_{ij} \cdot \sqrt{A_{im}} \equiv \sqrt{A_{in}} \cdot A_{ij} \cdot \sqrt{C_{in}}. \tag{55}$$

3′. If the move results in one crossing of the future light cone emerging from $P_j$, but only the crossing of past light cones emerging from the other two insertions, the resultant monodromy depends on the order of these crossings.

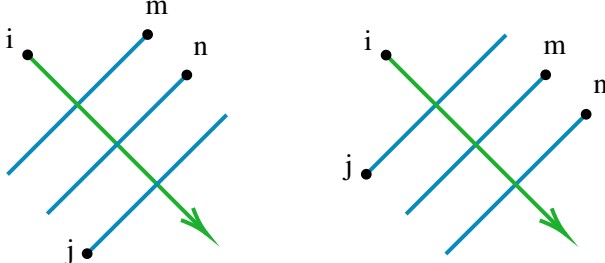

Figure 24: Crossing of two consecutive past light cones from future to past.

If we first cross the two past lightcones and then the future lightcone, or first cross the future lightcone and then the two past lightcones, then the resultant monodromy is a clockwise circle around $z_{ij}$, i.e, $C_{ij}$.

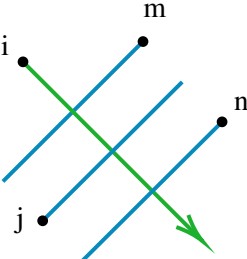

Figure 25: Crossing of two past light cones with a future light cone in the middle from future to past.

If, on the other hand, the order of crossings is first past of $P_m$ then future of $P_j$ and then past of $P_n$ then the resultant monodromy is $\sqrt{A_{im}}$ followed by $C_{ij}$ followed by $\sqrt{C_{im}}$, i.e.,

$$\sqrt{A_{im}} \cdot C_{ij} \cdot \sqrt{C_{im}} \equiv \sqrt{C_{in}} \cdot C_{ij} \cdot \sqrt{A_{in}}. \tag{56}$$

Note that these rules are invariant under time reversal. For instance, we get the same monodromy from the future-directed translation $\omega_i \to \omega_i - \pi$ that cuts one past and two future lightcones (rule 2) and the past directed translation $\omega_i \to \omega_i + \pi$ that cuts one future and two past lightcones (rule 3′). We will use this fact extensively below.[44]

---

[44]This observation may be understood as follows. Time reversal interchanges $z$ and $\bar{z}$. However, the principal of Euclidean single valuedness assures us that, e.g., a $C_{ij}$ monodromy in $z$ is the same as a $C_{ij}$ monodromy in $\bar{z}$.

### 6.3 Configurations obtained from $\omega_i \to \omega_i - n_i\pi$ starting from Minkowski diamond configurations with $z$ and $\bar{z}$ in the same $R_i$ range

In the rest of this paper, we focus on configurations for which $z$ and $\bar{z}$ both lie in the same range (36). As the Minkowski diamond constitutes a unit cell for insertion locations, it follows that all insertion locations of the type above can be obtained by performing $\omega_i \to \omega_i - n_i\pi$ shifts, starting with one of

- the 6 inequivalent Euclidean sheet configurations (see Fig. 8),

- the Regge configuration (see 15),

- the scattering configuration (see Fig. 16)

– all of which lie on a single Minkowski diamond.

To determine the sheet location of the most general configuration for which $z$ and $\bar{z}$ lie in the same $R_i$ range, we could proceed to enumerate the sheet location of the configurations obtained from every possible set of $\pi$ shifts $\omega_i \to \omega_i - n_i\pi$ starting from any of these 8 configurations. For completeness, in Appendix F we have, indeed, presented the derivation and result of the enumeration described above in full detail.

One can, however, get away with much less work. The key point here is to recognize that the 8 starting configurations listed above can themselves be lumped into two groups. The first group consists of the Euclidean $A$, $B$, $C$ (see Fig. 8) and scattering (see Fig. 16) configurations. The second group consists of The Euclidean $D$, $E$, $F$ (see Fig. 8) and Regge (see 15) configurations. The point of these groupings is that any configuration in one group can be related to any other by appropriate $\pi$ shifts.

The last claim may, at first, seem surprising. If we start with a configuration with all insertion locations in a given diamond, and then make a shift of the form $\omega_i \to \omega_i - n_i\pi$, the $i^{th}$ operator - by definition - leaves the original diamond. For some choices of starting locations and $n_i$, however, it turns out to be possible to find a new diamond (i.e. a diamond centered around a new point on the cylinder) which contains all insertion points. We now see how this works in various examples.

Consider starting with the Euclidean $A$ type configuration in Fig. 8. The shift $\omega \to \omega - \pi$, performed on the coordinate of the last (rightmost) operator in Fig. 8 A, moves this insertion upwards and to the left (diagonally upwards). It is easy to see that the resultant configuration is of the Euclidean $B$ type in a re-centered red diamond. Similarly, if we start with the Euclidean $A$ type configuration, and move the insertion location of the first (leftmost) operator by $\omega \to \omega + \pi$ (this move takes this operator downward and to the right), and re-centering our diamond, we find a configuration of the Euclidean $C$ type. Finally, if we once again start with the Euclidean $A$ type configuration, but move the locations of each of the last two operators by $\omega \to \omega - \pi$ (diagonally upwards and to the left) we obtain a scattering-type configuration.

We can perform similar manipulations within the second group of configurations. If we start with the Euclidean $F$ type configuration (Fig. 8), and perform the shift $\omega \to \omega - \pi$ on the bottommost operator (move it upwards and to the left) we obtain a configuration of the $D$ type. Similarly, if we start with the Euclidean $F$ type configuration, and move the insertion location of the topmost operator by $\omega \to \omega + \pi$ (downward and to the right) we obtain a Euclidean configuration of the $E$ type. Finally, if we start with the Euclidean $F$ type configuration, move the bottommost insertion $\omega \to \omega - \pi$ (diagonally upwards and to the left), and also move the topmost insertion $\omega \to \omega + \pi$ (diagonally downwards and to the right), we obtain the Regge type configuration, with the middle two operators and the top and bottom operators making up the two pairs that are mutually spacelike with respect to each other.

It follows from the discussion above that every configuration on the Euclidean cylinder - with $z$ and $\bar{z}$ in the same range - can be obtained by performing the shifts $\omega_i \to \omega_i \pm n_i\pi$ starting

from either an $A$ type or an $F$ type Euclidean configuration. To enumerate all sheets accessed by time-ordered correlators on the Lorentzian cylinder (with $z$ and $\bar{z}$ in the same $R_i$ range), it remains only to enumerate all the sheets one obtains starting with either a Euclidean $A$ or a Euclidean $F$ type configuration. As mentioned above, we have provided a detailed derivation of these results in Appendix F. Here we simply summarize our final results.

### 6.3.1 Monodromies from shifts of $A$ type configurations

In this subsubsection we study the configurations that are obtained by the moves $\omega_i \to \omega_i - n_i \pi$, starting with configurations of the form depicted in Fig. 8 A. The rule for the monodromy of this final configuration turns out to be rather simple and can be summarized as follows.

Consider the set of four numbers $n_i$. Let its four elements, arranged in non-ascending order be $n_{i_1}, n_{i_2}, n_{i_3}, n_{i_4}$, so that

$$n_{i_1} \geq n_{i_2} \geq n_{i_3} \geq n_{i_4}. \tag{57}$$

In words, $n_{i_1}$ is the largest of the $n_i$, $n_{i_2}$ is the second largest, and so on.

Now recall that operators in Fig. 8 appear with a given cyclical ordering on the spatial circle (the cyclical motion proceeds in the direction of increasing $\theta$). The monodromy that one finds from the $n_i$ shifts above, turns out to depend on the relation between the $n_i$ ordering above and the $\theta$ cyclical ordering of the operators in their original $A$ type configuration

It turns out that when the operators $i_1$ and $i_2$ (defined in (57)) neighbour each other (from the viewpoint of the $\theta$ cyclical ordering), then the relevant shifts take us to a configuration (see Appendix F.1 for details) with monodromy equal to $\mathbf{C}_{\mathbf{i_1 i_2}}^{\mathbf{n_{i_2} - n_{i_3}}}$.

When, on the other hand, the $i_1$ and $i_2$ do not neighbour each other (but are diagonally opposite to each other in the sense of $\theta$ cyclical ordering), the rule is a bit different, and is given as follows. Let $a$ be the counterclockwise (i.e. in the direction of increasing $\theta$) neighbour of $i_i$ (in the sense of $\theta$ cyclical ordering). Then the monodromy for this case turns out to be $\sqrt{\mathbf{C_{i_1 a}}} \cdot \mathbf{C_{i_1 i_2}^{n_{i_2} - n_{i_3}}} \cdot \sqrt{\mathbf{A_{i_1 a}}}$.

The rules presented above meet a simple consistency check. The replacement $\theta \to -\theta$ and $\tau \to -\tau$ (i.e. parity plus time reversal) take $\omega \to -\omega$ and $\bar{\omega} \to -\bar{\omega}$. Consequently, this operation takes $n_i \to -n_i$ (and so reverses the ordering of the $n_i$) and also interchanges anticlockwise with clockwise. The reader can easily check that our monodromy rules are invariant under this combined operation.

Note that the fact that our monodromy rules depend in detail only on $n_{i_2} - n_{i_3}$ (and in particular are independent of the precise values of $n_{i_1}$ and $n_{i_4}$) can be understood from the OPE. We obtain nontrivial monodromies only when the OPE channel (equivalently, insertion of a complete set of states in the Hamiltonian picture) is nontrivial, in the sense that it allows the running of multiple intermediate operators.

### 6.3.2 Monodromies from shifts of $F$ type configurations

Any $F$ type configuration is characterized by the temporal ordering of its operators. Once we perform $\omega_i \to \omega_i - n_i \pi$ shifts on the locations of these operators, the integers $n_i$ give us a second ordering for these operators (through the symbols $i_m$ defined in the equation (57)).

As in the previous subsection, the rules for the monodromy shifts resulting from these translations depend on the relationship between the temporal ordering (from future to past) of the operators and the ordering defined by the symbols $i_n$ (see (57)). In the rest of this subsection, we summarize the final results - derived in detail in Appendix F.6 - for the monodromies that follow from these shifts.

1. When either the ordering $(i_1, i_2, i_3, i_4)$ or the the once cyclically rotated orderings $(i_4, i_1, i_2, i_3)$ or $(i_2, i_3, i_4, i_1)$ or any of the complete reversal of these three configurations $(i_4, i_3, i_2, i_1)$ or $(i_3, i_2, i_1, i_4)$ or $(i_1, i_4, i_3, i_2)$ matches the temporal ordering of the operators (always listed from future to past), then the monodromy again turns out to be $\mathbf{C}_{\mathbf{i_1 i_2}}^{\mathbf{n_{i_2} - n_{i_3}}}$.

2. When the ordering $(i_2, i_1, i_3, i_4)$ (obtained by flipping top two) or the ordering $(i_1, i_2 i_4, i_3)$ (obtained by flipping the last two) or the orderings $(i_4, i_2, i_3, i_1)$ (obtained by flipping the first and the last) or the ordering $(i_2, i_4, i_3, i_1)$ (obtained by flipping the first and the second in the last ordering) matches the temporal ordering of operators then the monodromy turns out to be $\mathbf{C}_{\mathbf{i_1 i_2}}^{\mathbf{n_{i_2} - n_{i_3} + 1}}$. On the other hand, if the reversal of any of the orderings listed above, namely $(i_4, i_3, i_1, i_2)$, $(i_3, i_4, i_2, i_1)$, $(i_1, i_3, i_2, i_4)$ or $(i_1, i_3, i_4, i_2)$ matches the temporal ordering of the operators after flipping either $(i_1, i_2)$ or $(i_3, i_4)$ then the monodromy turns out to be $\mathbf{C}_{\mathbf{i_1 i_2}}^{\mathbf{n_{i_2} - n_{i_3} - 1}}$.

3. When the ordering $(i_1, i_2, i_3, i_4)$ matches the temporal ordering of operators after flipping both $(i_1, i_2)$ and $(i_3, i_4)$ then the monodromy turns out to be $\mathbf{C}_{\mathbf{i_1 i_2}}^{\mathbf{n_{i_2} - n_{i_3} + 2}}$. On the other hand, if the reversed ordering $(i_4, i_3, i_2, i_1)$ matches the temporal ordering of the operators after flipping then the monodromy turns out to be $\mathbf{C}_{\mathbf{i_1 i_2}}^{\mathbf{n_{i_2} - n_{i_3} - 2}}$.

4. When any of the orderings $(i_1, i_3, i_2, i_4)$ or $(i_4, i_1, i_3, i_2)$ or $(i_3, i_2, i_4, i_1)$ matches the temporal order of the operators (ordered from future to past), the monodromy is given by $\sqrt{\mathbf{C_{i_1 i_4}}} \cdot \mathbf{C}_{\mathbf{i_1 i_2}}^{\mathbf{n_{i_2} - n_{i_3}}} \cdot \sqrt{\mathbf{C_{i_1 i_4}}}$.

5. When any of the orderings $(i_1, i_4, i_2, i_3)$ or $(i_4, i_2, i_3, i_1)$ or $(i_2, i_3, i_1, i_4)$ matches the temporal order of the operators (ordered from future to past), the monodromy is given by $\sqrt{\mathbf{C_{i_1 i_3}}} \cdot \mathbf{C}_{\mathbf{i_1 i_2}}^{\mathbf{n_{i_2} - n_{i_3} + 1}} \cdot \sqrt{\mathbf{C_{i_1 i_3}}}$.

6. When the ordering $(i_2, i_4, i_1, i_3)$ matches the temporal order of the operators (ordered from future to past), the monodromy is given by $\sqrt{\mathbf{C_{i_1 i_4}}} \cdot \mathbf{C}_{\mathbf{i_1 i_2}}^{\mathbf{n_{i_2} - n_{i_3} + 2}} \cdot \sqrt{\mathbf{C_{i_1 i_4}}}$.

7. When the ordering $(i_3, i_1, i_4, i_2)$ matches the temporal order of the operators (ordered from future to past), the monodromy is given by $\sqrt{\mathbf{C_{i_1 i_3}}} \cdot \mathbf{C}_{\mathbf{i_1 i_2}}^{\mathbf{n_{i_2} - n_{i_3} - 1}} \cdot \sqrt{\mathbf{C_{i_1 i_4}}}$.

### 6.4 Sheets and corresponding causal configurations

The rules of the previous subsection tell us that all insertion locations on the Lorentzian cylinder lie on a sheet obtained starting from the Euclidean sheet and making one of the following monodromy moves

1. $C_{ij}^q$ where $q$ is an integer, $q \geq -1$,

2. $\sqrt{C_{ij}} \cdot C_{im}^q \cdot \sqrt{A_{ij}}$, $q \geq 0$,

3. $\sqrt{C_{ij}} \cdot C_{im}^q \cdot \sqrt{C_{ij}}$, $q \geq 0$,

where $i, j, m$ are distinct elements of the set $\{1, 2, 3, 4\}$.

Several causally distinct configuration locations (of operator insertions) turn out to be evaluated on the same sheet of the correlator. Below we present an exhaustive tabulation of all causally distinct operator configurations, together with the sheets on which they lie.. The reader who is interested in tracking down all causal configurations that lie on a particular sheet can easily read this information from a glance through the tables below.

In the tables below, the integer $q$ refers to the power to which $C_{ij}$ is raised. The tables list sequences of configurations, and are organized depending on whether the integer $q$ in these sequences ranges from $-1\ldots$, or from $0\ldots$ or from $1\ldots$ or from $2\ldots$

Table 1: Single branch point towers with $q \geq -1$.

| $q \geq -1$ | Monodromy | Configuration | Condition |
|---|---|---|---|
| 1 | $C_{ij}^{n_n-n_i-2}$ | Euclidean - F case 17 | $n_n > n_i$ |

Table 2: Single branch point towers with $q \geq 0$.

| $q \geq 0$ | Monodromy | Configuration | Condition |
|---|---|---|---|
| 1 | $C_{ij}^{n_i-n_m}$ | Euclidean - A case 7 | $n_i \geq n_m$ |
| 2 | $C_{ij}^{n_i-n_n}$ | Euclidean - A case 8 | $n_i \geq n_n$ |
| 3 | $C_{ij}^{n_j-n_m}$ | Euclidean - A case 1 | $n_j \geq n_m$ |
| | | Euclidean - F case 1 | Same as above |
| 4 | $C_{ij}^{n_j-n_n}$ | Euclidean - A case 2 | $n_j \geq n_n$ |
| 5 | $C_{ij}^{n_m-n_i-1}$ | Euclidean - F case 23 | $n_m > n_i$ |
| 6 | $C_{ij}^{n_m-n_i}$ | Euclidean - A case 23 | $n_m \geq n_i$ |
| 7 | $C_{ij}^{n_m-n_j}$ | Euclidean - A case 24 | $n_m \geq n_j$ |
| 8 | $C_{ij}^{n_n-n_i}$ | Euclidean - A case 17 | $n_n \geq n_i$ |
| 9 | $C_{ij}^{n_n-n_j-1}$ | Euclidean - F case 18 | $n_n > n_j$ |
| 10 | $C_{ij}^{n_n-n_j}$ | Euclidean - A case 18 | $n_n \geq n_j$ |
| 11 | $C_{in}^{n_i-n_j}$ | Euclidean - A case 19 | $n_i \geq n_j$ |
| | | Euclidean - F case 19 | Same as above |
| 12 | $C_{in}^{n_i-n_m}$ | Euclidean - A case 20 | $n_i \geq n_m$ |
| 13 | $C_{in}^{n_j-n_i}$ | Euclidean - A case 15 | $n_j \geq n_i$ |
| 14 | $C_{in}^{n_j-n_n}$ | Euclidean - A case 16 | $n_j \geq n_n$ |
| 15 | $C_{in}^{n_m-n_i-1}$ | Euclidean - F case 9 | $n_m > n_i$ |
| 16 | $C_{in}^{n_m-n_i}$ | Euclidean - A case 9 | $n_m \geq n_i$ |
| 17 | $C_{in}^{n_m-n_n}$ | Euclidean - A case 10 | $n_m \geq n_n$ |
| | | Euclidean - F case 10 | Same as above |
| 18 | $C_{in}^{n_n-n_j-1}$ | Euclidean - F case 5 | $n_n > n_j$ |
| 19 | $C_{in}^{n_n-n_j}$ | Euclidean - A case 5 | $n_n \geq n_j$ |
| 20 | $C_{in}^{n_n-n_m}$ | Euclidean - A case 6 | $n_n \geq n_m$ |

Table 3: Single branch point towers with $q \geq 1$.

| $q \geq 1$ | Monodromy | Configuration | Condition |
|---|---|---|---|
| 1 | $C_{ij}^{n_i - n_m + 1}$ | Euclidean - F case 7 | $n_i \geq n_m$ |
| 2 | $C_{ij}^{n_j - n_n + 1}$ | Euclidean - F case 2 | $n_j \geq n_n$ |
| 3 | $C_{ij}^{n_m - n_j}$ | Euclidean - F case 24 | $n_m > n_j$ |
| 4 | $C_{in}^{n_i - n_m + 1}$ | Euclidean - F case 20 | $n_i \geq n_m$ |
| 5 | $C_{in}^{n_j - n_i}$ | Euclidean - F case 15 | $n_j > n_i$ |
| 6 | $C_{in}^{n_j - n_n + 1}$ | Euclidean - F case 16 | $n_j \geq n_n$ |
| 7 | $C_{in}^{n_n - n_m}$ | Euclidean - F case 6 | $n_n > n_m$ |

Table 4: Single branch point towers with $q \geq 2$.

| $q \geq 2$ | Monodromy | Configuration | Condition |
|---|---|---|---|
| 1 | $C_{ij}^{n_i - n_n + 2}$ | Euclidean - F case 8 | $n_i \geq n_n$ |

Table 5: Double branch point towers with $q \geq 0$.

| $q \geq 0$ | Monodromy | Configuration | Condition |
|---|---|---|---|
| 1 | $\sqrt{C_{ij}} \cdot C_{im}^{n_i - n_j} \cdot \sqrt{A_{ij}}$ | Euclidean - A case 13 | $n_i \geq n_j$ |
| 2 | $\sqrt{C_{ij}} \cdot C_{im}^{n_i - n_n} \cdot \sqrt{A_{ij}}$ | Euclidean - A case 14 | $n_i \geq n_n$ |
| 3 | $\sqrt{C_{ij}} \cdot C_{im}^{n_m - n_j} \cdot \sqrt{A_{ij}}$ | Euclidean - A case 3 | $n_m \geq n_j$ |
| 4 | $\sqrt{C_{ij}} \cdot C_{im}^{n_m - n_n} \cdot \sqrt{A_{ij}}$ | Euclidean - A case 4 | $n_m \geq n_n$ |
| 5 | $\sqrt{C_{in}} \cdot C_{im}^{n_j - n_i} \cdot \sqrt{A_{in}}$ | Euclidean - A case 21 | $n_j \geq n_i$ |
| 6 | $\sqrt{C_{in}} \cdot C_{im}^{n_j - n_m} \cdot \sqrt{A_{in}}$ | Euclidean - A case 22 | $n_j \geq n_m$ |
| 7 | $\sqrt{C_{in}} \cdot C_{im}^{n_n - n_i} \cdot \sqrt{A_{in}}$ | Euclidean - A case 11 | $n_n \geq n_i$ |
| 8 | $\sqrt{C_{in}} \cdot C_{im}^{n_n - n_m} \cdot \sqrt{A_{in}}$ | Euclidean - A case 12 | $n_n \geq n_m$ |
| 9 | $\sqrt{C_{ij}} \cdot C_{im}^{n_n - n_i - 1} \cdot \sqrt{C_{ij}}$ | Euclidean - F case 11 | $n_n > n_i$ |

Table 6: Double branch point towers with $q \geq 1$.

| $q \geq 1$ | Monodromy | Configuration | Condition |
|:---:|:---:|:---:|:---:|
| 1 | $\sqrt{C_{ij}} \cdot C_{im}^{n_j - n_i} \cdot \sqrt{C_{ij}}$ | Euclidean - F case 21 | $n_j > n_i$ |
| 2 | $\sqrt{C_{ij}} \cdot C_{im}^{n_j - n_m + 1} \cdot \sqrt{C_{ij}}$ | Euclidean - F case 22 | $n_j \geq n_m$ |
| 3 | $\sqrt{C_{ij}} \cdot C_{im}^{n_n - n_m} \cdot \sqrt{C_{ij}}$ | Euclidean - F case 12 | $n_n > n_m$ |
| 4 | $\sqrt{C_{in}} \cdot C_{im}^{n_i - n_j + 1} \cdot \sqrt{C_{in}}$ | Euclidean - F case 13 | $n_i \geq n_j$ |
| 5 | $\sqrt{C_{in}} \cdot C_{im}^{n_m - n_j} \cdot \sqrt{C_{in}}$ | Euclidean - F case 3 | $n_m > n_j$ |
| 6 | $\sqrt{C_{in}} \cdot C_{im}^{n_m - n_n + 1} \cdot \sqrt{C_{in}}$ | Euclidean - F case 4 | $n_m \geq n_n$ |

Table 7: Double branch point towers with $q \geq 2$.

| $q \geq 2$ | Monodromy | Configuration | Condition |
|:---:|:---:|:---:|:---:|
| 1 | $\sqrt{C_{in}} \cdot C_{im}^{n_i - n_n + 2} \cdot \sqrt{C_{in}}$ | Euclidean - F case 14 | $n_i \geq n_n$ |

## 6.5 "Uniqueness" on the Regge sheet

As the tables above make clear (and as we have already emphasized above) most sheets evaluate the correlator on several non-trivially different causal configurations. This is the case even for the Euclidean Sheet. As we have already seen (see Fig. 8) several distinct causal configurations in a single Minkowski diamond already lie on the Euclidean sheet. The tables in the previous subsection list several additional configurations (which cannot be accommodated on a single Minkowski diamond) that also lie on the Euclidean sheet.

There is, however, a single monodromy, namely the Regge monodromy $C_{ij}^{-1} = A_{ij}$, that figures exactly once in these tables (this mention is the case $n_n = n_i + 1$) in Table 1). We see, as a consequence, that the configurations that give rise to the Regge monodromy are essentially causally unique[45] and is given by the single diamond configuration displayed in Fig 15 and discussed there.

## 7 Conclusion

In this paper, we have studied the branch structure of time-ordered four-point functions in $1 + 1$ dimensional CFTs on a Lorentzian cylinder. The locations of the four insertions on the cylinder determine the conformal cross-ratios $z$ and $\bar{z}$ in a simple and well-understood manner. However Lorentzian correlators are multi-valued functions, so specification of the conformal cross-ratios does not completely determine the correlation function: one also needs to know which sheet in cross-ratio space the correlator is evaluated on. In this paper we have provided a complete answer to this question: we have determined which sheet the correlator lies on for every set of insertion locations of the time ordered correlator.

As we scan over all possible insertion locations (in §6), we find that we access three qualitatively different infinite sequences of sheets (as listed in §6.4). In the first sequence one starts

---

[45]The monodromy for any configuration is left unchanged if we perform future directed $\pi$ translations on the future most insertion, or past directed $\pi$ translations on the past most point. In the discussion of this subsection, we are treating configurations related by such moves as causally equivalent.

from the Euclidean sheet and then makes an arbitrary number of clockwise[46] monodromies around exactly one of the three branch points (at zero, one or infinity) (see tables 1, 2, 3, and 4). In the second sequence, one first makes a single clockwise half-monodromy around one branch point and then makes an arbitrary number of clockwise monodromies around the second branch point followed by a single anticlockwise half-monodromy around the first branch point (see tables 5, 6, and 7). In the third sequence, one first makes a single clockwise half-monodromy around one branch point and then makes an arbitrary number of clockwise monodromies around the second branch point followed by a single clockwise half-monodromy around the first branch point (again tables 5, 6, and 7).

These infinite sequence - which form a small subset of the set of all possible branch moves that one can mathematically make - are the only ones that time ordered four point functions on the cylinder explore. It follows, in other words, that while the construction described in this paper has given a physical interpretation of an infinite number of branch sheets, it has also left a much larger infinity of sheets uninterpreted. It would be very interesting to search for another interpretation of this larger infinity of sheets. We leave this to future work.[47]

In this paper, we have only studied time-ordered correlators. It would be interesting - and not too difficult - to generalize our results to correlators on the Lorentzian cylinder with other orderings. The analysis of a simple alternate class correlators - those whose $i\epsilon$ values have fixed values, independent of insertion locations - was performed in Chapter 23.6 of [21], and yields a much simpler answer than the time-ordered correlators studied in this paper (the sheets accessed in that study are a subset of those accessed by the time-ordered correlators of this paper). It would be interesting to study other generalizations of time ordering that could allow us to access a larger number of infinite sequences of sheets of the correlator. We leave such a study to future work.

In the process of obtaining our results, we have, in particular, presented a complete classification of all causally distinct (and so, potentially, sheet distinct) configurations of a collection of four points on the Lorentzian cylinder. This classification turns out to be rather simple. Restricting to points whose $z$ and $\bar{z}$ values lie in the same ranges $R_i$, we find that all such configurations can be obtained by starting configuration that consist of four points that are all mutually spacelike or all mutually timelike on a single Minkowskian diamond and then performing the shift operations $\omega_i \to \omega_i - n_i \pi$ on the points.

Another interesting aspect of our results is the following. We find that the same sheet and value of cross-ratios often describe several symmetry inequivalent (and causally distinct) configurations. This is the case for every sheet that appears in our classification except for one; the "Regge" sheet (which plays in key role in the famous bound on Chaos) is associated with only a single causal configuration. We find both the "uniqueness" of the Regge sheet, as well as the "non-uniqueness" of all other sheets interesting. The fact that distinct causal configurations give rise to the same correlator suggests that interesting features of the correlation function on the relevant sheets (e.g. bulk point singularities on the scattering sheet) could admit multiple physical interpretations. It would be interesting to investigate this point further.

It would be interesting to use the constructions presented in this paper to make predictions for the physical features of the correlator in relevant situations. For instance, when the CFT under study has a bulk dual, configurations on the so-called "scattering sheet" are well known to have "bulk point singularities" that describe bulk scattering [19, 23]. In analogy (and for

---

[46]We obtain towers of $C_{ij}$ - rather than towers of $A_{ij}$ - because we are studying time ordered (rather than anti-time ordered) correlators.

[47]Other sequences of sheets, similar to those we have studied, can be obtained by modifying our construction in obvious ways. For instance, we could study "anti-time ordered" correlators: this would interchange clockwise and anticlockwise monodromies in this paper. The study of OTOCs would enlarge the canvas somewhat: recall, however, that all four point correlators lie on at most 2 (nontrivial) time folds, so this enlargement is not very substantial. The study of the (non-abelian) exponential infinity of sheets seems to require new ideas.

similar reasons) we expect configurations on several of the sheets studied above to have new "repeated bulk point" singularities describing scattering processes that follow earlier scattering processes on the Lorentzian cylinder. It would be very interesting to study this further.

The braiding, and fusions matrices that characterize holomorphic conformal blocks of rational CFTs are well known to obey nontrivial pentagon and hexagon identities [27]. While the discussion of §3.3 below touched on (analogues of) these matrices, we never had occasion to make nontrivial use of the identities these objects obey. It would be interesting to explore the interplay (if any) of these identities with the study of Lorentzian correlators, along the lines of this paper.[48]

It would also be interesting to generalize the analysis of this paper to the study of correlators in CFTs with defects. In this situation, even two-point functions can have nontrivial cross-ratios (depending on the details). Obtaining an understanding of the branch structures (even of two-point functions) in this case could have interesting physical applications.[49]

In this paper, we have focused on the study of correlators in $1 + 1$ dimensional CFTs. Focusing on 4-point functions, it should be possible - and would hopefully not be too difficult - to generalize this study to higher dimensional CFTs. Of course, all the sheets we have described above will also exist in higher dimensions (this follows as we can simply choose to restrict attention to configurations that lie on an effective 2 d cylinder - i.e. on one particular equator on $S^{d-1}$). We expect that several of the results presented in this paper will generalize in a straightforward manner to higher dimensional CFTs.[50] We note, however, that correlators in higher dimensions have new features (for instance, the Lorentzian cross-ratios $z$ and $\bar{z}$ can be either independent real numbers or complex conjugates of each other, depending on the details of the insertion locations). It is thus possible that the higher dimensional study will encounter qualitatively new features. We leave an investigation of this point to future work.

Of course, the differences in analytic structure between correlators in higher dimension and two dimensions will grow when considering higher point functions. This follows from the simple kinematical fact that while the total number of cross-ratios for $n$- point functions in two dimensions is $n - 2$, the corresponding number in $d$-dimensions equals, for instance, $n(n-3)/2$ in the case that $n \leq d + 2$.[51] Consequently, the analytic structure of correlators presumably has significant differences.

In this paper, we have attempted to present a physical interpretation for an infinite number of sheets of the four point correlator in a CFT. A similar question can be asked for S-matrices in non-conformal theories, which also have multi-sheeted structure (this time in the kinematical variables $s$ and $t$). It would be very interesting to find a physical interpretations of a sequence of sheets of the S-matrix. Perhaps the AdS/CFT correspondence (which, very roughly speaking, relates bulk S-matrices to boundary correlators) could be of use here. We also leave further contemplation of this point to future work.

## Acknowledgments

We would like to thank S. Biswas, A. Gadde, I. Haldar, D. Jain, O. Parrikar, S. Raju, and S. Trivedi for very useful discussions. SK, SM and AN would also like to acknowledge their debt to the people of India for their steady support of the study of the basic sciences.

---

[48]We thank the referee for the question.

[49]We thank the referee for making this point.

[50]For instance, we verify in Appendix D that higher dimensional correlators described by the D function (which arise out of tree level contact interactions in the bulk of AdS/CFT) have the property that left and right moving monodromies commute with each other.

[51]We thank the reviewer for mentioning this point.

**Funding information**   The work of AN and SM was supported by the Infosys Endowment for the study of the Quantum Structure of Spacetime. The work of SM and AN is supported by the J C Bose Fellowship JCB/2019/000052. The work of SK was supported in part by an ISF, centre for excellence grant (grant number 2289/18), Simons Foundation grant 994296, and Koshland Fellowship.

# A   Two and three point functions on the Lorentzian cylinder

Consider a two-point function of an operator with holomorphic and anti-holomorphic dimensions $(h, \bar{h})$. On a complex plane (parameterized by the complex coordinate $u$), the Euclidean two-point function takes the form

$$G_2(u_{12}, \bar{u}_{12}) = \frac{1}{u_{12}^{2h}} \frac{1}{\bar{u}_{12}^{2\bar{h}}}. \tag{A.1}$$

Similarly, the three-point function of three operators with holomorphic dimensions $h_1, h_2, h_3$ and anti-holomorphic dimensions $\bar{h}_1, \bar{h}_2$ and $\bar{h}_3$ is given by

$$G(u_i, \bar{u}_i) = C_{123} \frac{1}{u_{12}^{h_1+h_2-h_3} u_{23}^{h_2+h_3-h_1} u_{13}^{h_3+h_1-h_2}} \frac{1}{\bar{u}_{12}^{\bar{h}_1+\bar{h}_2-\bar{h}_3} \bar{u}_{23}^{\bar{h}_2+\bar{h}_3-\bar{h}_1} \bar{u}_{13}^{\bar{h}_3+\bar{h}_1-\bar{h}_2}}. \tag{A.2}$$

Under the variable change[52]

$$u = e^{-2i\omega}, \qquad \bar{u} = e^{2i\bar{\omega}}, \tag{A.4}$$

the line element becomes

$$ds^2 = du d\bar{u} = 4e^{4\text{Im}\omega} d\omega d\bar{\omega}, \tag{A.5}$$

and the coordinates $\omega$ and $\bar{\omega}$ obey (8). (A.5) reflects a well-known fact: the complex plane is Weyl equivalent to a Euclidean cylinder. Stripping off the Weyl factor $e^{4\text{Im}\omega}$ turns (A.5) into the Euclidean cylinder. Finally, the analytic continuation $\tau_E = i\tau$ takes us to the Lorentzian cylinder, and $\omega$ and $\bar{\omega}$ into independent real variables.

Tracing through this series of operations on (A.1), and using the standard formula

$$\phi_{\omega\bar{\omega}} = (\partial_w z)^h (\partial_{\bar{w}} \bar{z})^{\bar{h}} \phi_{z\bar{z}},$$

we obtain the following formula for the two-point function of our operator on the Lorentzian cylinder

$$
\begin{aligned}
G(\omega_i, \bar{\omega}_i) &= \left(2ie^{2i\omega_1}\right)^h \left(-2ie^{-2i\bar{\omega}_1}\right)^{\bar{h}} \left(2ie^{2i\omega_2}\right)^h \left(-2ie^{-2i\bar{\omega}_2}\right)^{\bar{h}} \\
&\quad \times \frac{1}{(e^{2i\omega_1} - e^{2i\omega_2})^{2h}} \frac{1}{(e^{-2i\bar{\omega}_1} - e^{-2i\bar{\omega}_2})^{2\bar{h}}} \\
&= (-4)^{h+\bar{h}} e^{2i[h(\omega_1+\omega_2)-\bar{h}(\bar{\omega}_1+\bar{\omega}_2)]} \frac{1}{(e^{2i\omega_1} - e^{2i\omega_2})^{2h}} \frac{1}{(e^{-2i\bar{\omega}_1} - e^{-2i\bar{\omega}_2})^{2\bar{h}}} \\
&= 2^{2h+2\bar{h}} \frac{1}{(2\sin\omega_{12})^{2h}} \frac{1}{(2\sin\bar{\omega}_{12})^{2\bar{h}}} \\
&= \frac{1}{(\sin\omega_{12})^{2h}(\sin\bar{\omega}_{12})^{2\bar{h}}}.
\end{aligned} \tag{A.6}
$$

---

[52]In terms of the variables defined in (7), (A.4) is

$$u = e^{\tau_E - i\theta}, \qquad \bar{u} = e^{\tau_E + i\theta}. \tag{A.3}$$

In particular, $\tau_E = -\infty$ maps to the origin of the $u$ plane.

After introducing the proper $i\epsilon$ we get

$$G(\omega_i, \bar{\omega}_i) = \frac{1}{(\sin(\omega_{12} + i\epsilon\tau_{ij}))^{2h}(\sin(\bar{\omega}_{12} - i\epsilon\tau_{ij}))^{2\bar{h}}}. \tag{A.7}$$

A very similar manipulation turns the three-point correlator (A.2) into

$$G(\omega, \bar{\omega}) = \frac{C_{123}(\sqrt{2})^{\sum_i (h_i + \bar{h}_i)}}{\zeta_{12}^{H_{12}} \zeta_{23}^{H_{23}} \zeta_{31}^{H_{31}} \bar{\zeta}_{12}^{\bar{H}_{12}} \bar{\zeta}_{23}^{\bar{H}_{23}} \bar{\zeta}_{31}^{\bar{H}_{31}}}, \tag{A.8}$$

where

$$H_{ij} = \frac{h_i + h_j - h_k}{2}, \qquad \bar{H}_{ij} = \frac{\bar{h}_i + \bar{h}_j - \bar{h}_k}{2}. \tag{A.9}$$

# B Taking an operator around the cylinder leaves the correlator unchanged

## B.1 Single valuedness of the three-point function under winding

The shift $\omega_i \to \omega_i + \pi n_i$, $\bar{\omega}_i \to \bar{\omega}_i + \pi n_i$ (which effectively shifts $m_i$ by $-n_i$ and $\bar{m}_i$ by $n_i$ and so leaves $m_i + \bar{m}_i$ invariant)[53] generates the coordinate shift $\theta_i \to \theta_i + 2\pi n_i$, $\tau_i \to \tau_i$. In other words, we can achieve this shift by winding the insertion of our operator $O_i$ $n_i$ times around the Lorentzian cylinder (at a fixed value of Lorentzian time). This should change $O_i$ by $O_i \to O_i e^{2\pi i(h_i - \bar{h}_i)}$, so should leave correlators invariant because every operator carries integer values of the spin $h_i - \bar{h}_i$.

We pause to illustrate the fact that shift $\omega_i \to \omega_i + \pi$, $\bar{\omega}_i \to \bar{\omega}_i + \pi$ affects the transformation $O_i \to O_i e^{2\pi i(h_i - \bar{h}_i)}$ - while obvious using the monodromy rules across cuts in three-point functions. Suppose that the point $j$ lies to the past of $i$. Keeping $\omega_j$ fixed, let us move the $i^{th}$ point around the Lorentzian cylinder, i.e. take $\theta_i \to \theta_i + 2\pi$, i.e. $\omega_i \to \omega_i + \pi$ and $\bar{\omega}_i \to \bar{\omega} + \pi$. In the process of undertaking this motion, we cut the future rightmoving lightcone of $j$ from future to past and also cut the future leftmoving lightcone of $j$ from past to future. It follows from the rules of subsection 2.4 that under this motion $\zeta_{ij}$ undergoes a clockwise monodromy of $2\pi$, while $\bar{\zeta}_{ij}$ undergoes an anticlockwise monodromy of the same magnitude. The reader can easily verify that the same final result for monodromies also holds when $j$ is to the future (rather than the past) of $i$. It follows that under this motion $\zeta_{12}^{H_{12}} \zeta_{23}^{H_{23}} \zeta_{31}^{H_{31}} \bar{\zeta}_{12}^{\bar{H}_{12}} \bar{\zeta}_{23}^{\bar{H}_{23}} \bar{\zeta}_{31}^{\bar{H}_{31}}$ picks up the net phase $e^{2\pi i(\bar{H}_{13} + \bar{H}_{12} - H_{13} - H_{12})} = e^{2\pi i(\bar{h}_1 - h_1)} = 1$ as expected.

In summary, we have established that correlators depend on the integers $m_i$ defined in (16) only through the "gauge invariant" combination

$$m_i + \bar{m}_i, \tag{B.1}$$

for each value of $i$.

## B.2 Single valuedness of the four-point function under winding

On general grounds, we expect the motion that takes the insertion point of a correlator around the spatial circle to leave the correlator invariant, provided the operator under study has integral angular momentum, i.e. provided $h_i - \bar{h}_i$ is an integer. In this Appendix, we pause to illustrate how this works in detail, in one example involving the four-point function.

---

[53]In terms of the coordinates, this shift leaves $\omega_i - \bar{\omega}_i$ invariant.

Consider a configuration in which particles $1, 3, 4$ located at $\tau = 0$ and at $\theta_1$, $\theta_3$ and $\theta_4$, with $\theta_1 < \theta_3 < \theta_4$. Let the $\tau$ coordinate for the second insertion be fixed to $\epsilon$ (where $\epsilon$ is very small) and let $\theta_2$ vary from 0 to $2\pi$. As we follow $\theta_2$ on this trajectory, we successively cross the future leftmoving lightcone associated with particle 1, from past to future, the future rightmoving lightcone associated with particle 1 from future to past, the future leftmoving lightcone associated with particle 3, from past to future, the future rightmoving lightcone associated with particle 3 from future to past, the future leftmoving lightcone associated with particle 4, from past to future, the future rightmoving lightcone associated with particle 4 from future to past. The monodromies for these moves are easily computed using the rules of 3.2. We find that the $z$ monodromies are $\sqrt{C_0} \cdot \sqrt{C_1} \cdot \sqrt{C_\infty}$ and $\bar{z}$ monodromies are $\sqrt{A_0} \cdot \sqrt{A_1} \cdot \sqrt{A_\infty}$. The monodromy matrix associated with this series of moves is given (see (33)) we write

$$\sqrt{A_\infty} \cdot \sqrt{A_1} \cdot \sqrt{A_0} \cdot P \cdot \sqrt{C_0} \cdot \sqrt{C_1} \cdot \sqrt{C_\infty} \,. \tag{B.2}$$

However it follows from Euclidean single valuedness (see (34)) that this monodromy matrix can equivalently be written as

$$\begin{aligned}
\sqrt{A_\infty} \cdot \sqrt{A_1} &\cdot \sqrt{A_0} \cdot P \cdot \sqrt{C_0} \cdot \sqrt{C_1} \cdot \sqrt{C_\infty} \\
&= P \cdot \sqrt{C_0} \cdot \sqrt{C_1} \cdot \sqrt{C_\infty} \cdot \sqrt{A_\infty} \cdot \sqrt{A_1} \cdot \sqrt{A_0} \\
&= P \,,
\end{aligned} \tag{B.3}$$

so that the monodromies associated with winding is trivial.

# C  Commutation of path moves between left and right movers

In this appendix, we will explicitly demonstrate the path independence of the monodromy. We will show it in three steps. First, we will show the path-independence of a plaquette (unit face) where one side of the plaquette is extended in the holomorphic direction and the other side is extended in the anti-holomorphic direction. Second, we will show the path independence in a plaquette where both sides are in either fully holomorphic or anti-holomorphic direction. Here we explicitly check that the path independence of such a plaquette doesn't depend on the position of it in the anti-holomorphic directions. Finally, we show that it is true for a cube where two directions are in the holomorphic direction and one direction is in the anti-holomorphic direction. The same is true if two directions are in the anti-holomorphic direction and one is in the holomorphic direction. This establishes the complete path independence of monodromy.

## C.1  Vanishing of monodromies on mixed holomorphic/anti-holomorphic unit squares

Let $A$ represent any of the maps $I$, $F$, $B$ and let $\bar{A}$ represent any of the maps $\bar{I}$, $\bar{F}$ and $\bar{B}$. In the rest of this section, we will now consider the sequence of moves[54]

$$A^{-1} \bar{A}^{-1} A \bar{A} \,. \tag{C.1}$$

(C.1) describes an elementary move in the anti-holomorphic part, followed by an elementary move in the holomorphic part, and then the inverse move on the anti-holomorphic part, followed by the inverse move in the holomorphic part. This sequence clearly describes the most general elementary "mixed" closed loop on the lattice, with two legs in the holomorphic lattice,

---

[54]Our convention is that maps to the right always act before maps to the left.

and the other two in the anti-holomorphic lattice. In the rest of this section, we will explain that the monodromy associated with the sequence of moves (C.1) always vanishes.

The argument proceeds as follows. The operation $A$ always involves crossing a particular holomorphic light cone, let us say the $ij$ lightcone (associated with operator $i$ crossing a holomorphic lightcone of $j$ or vice-versa). For instance, the map $I$ causes the operator 1 to cross the lightcone of the operator 2. On the other hand, map $F$ causes either operators 1 or 2 (depending on details) to cross the lightcone emanating from operator 3. This crossing gives rise to a "half-monodromy" (see around §3.2 ). If immediately after acting with the map $A$ we then act (on the resultant lattice point) with the map $A^{-1}$ we clearly undo the light crossing and undo the corresponding half-monodromy. Now consider inserting the operation $\bar{A}^{-1}$ in between acting with $A$ and acting with $A^{-1}$, i.e. consider the map $A^{-1}\bar{A}^{-1}A$. Since $\bar{A}^{-1}$ leaves the holomorphic part of the lattice untouched, it may, at first, appear that the insertion of $\bar{A}^{-1}$ between $A$ and $A^{-1}$ changes nothing, i.e. the half-monodromies associated with $A$ and $A^{-1}$ continue to cancel. This is indeed generically the case. However, it fails in precisely one situation; when $\bar{A}^{-1}$ reverses the (global) time ordering of the operators $i$ and $j$. In this case, the half-monodromies associated with $A$ and $A^{-1}$ add rather than cancelling.[55]

We have explained above that the insertion of $\bar{A}^{-1}$ between $A^{-1}$ and $A$ sometimes obstructs the cancellation of the monodromy between $A$ and $A^{-1}$. Now the operation listed in (C.1) does involve such an insertion (of $\bar{A}^{-1}$ ). However, it turns that in those situations (and only in those situations) that the $A$ and $A^{-1}$ cancellation is obstructed, there is an equal and opposite lack of cancellation between $\bar{A}$ and $\bar{A}^{-1}$. The net result is that the sequence of moves (C.1) is always monodromy-free.

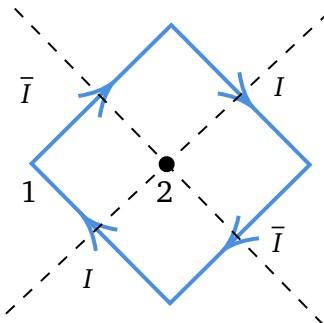

Figure 26: Illustration of the sequence of moves $I\bar{I}I\bar{I}$ in a situation in which the monodromies associated with the two $I$ moves add with each other, but are cancelled by the monodromies associated with the two $\bar{I}$ moves (which also add with each other). The two $I$ monodromies add rather than cancelling because while the top right $I$ move cuts a *future* lightcone of 2, the bottom left $I$ move cuts a *past* lightcone of 2 in the reverse direction. A similar analysis is applied for the move $\bar{I}$. The phases associated with $I$ motion cancel the phases associated with $\bar{I}$ motion because $h - \bar{h}$ is an integer for all operators.

We now illustrate that last point in two examples. Let us first choose the case $A = A^{-1} = I$ and study the action of $I\bar{A}I\bar{A}^{-1}$ on the lattice point $P_{12}^{m_1,m_2}$. As we have explained, the cancellation between the two factors of $I$ above is obstructed if and only if the action of $\bar{A}$ interchanges the relative time order of 1 and 2. Recall that - at the "moment" of the holomorphic crossing

---

[55]Let us, for example, suppose that $A$ takes $i$, from past to future, through a future lightcone of $j$. Then - in the generic case, $A^{-1}$ takes $i$, from future to past, through a future lightcone of $j$. According to the rules presented in §2.2, the "half-monodromies" (phase shifts) associated with these moves cancel. If, however $\bar{A}$ flips the order of $i$ and $j$, then $A^{-1}$ results in taking $i$ from future to past, through a *past* lightcone of $j$. In this case, the phases associated with $A^{-1}$ add to (instead of cancelling from) the phase associated with $A$.

(where $\alpha_1 = \alpha_2$), this relative ordering is determined by the quantity

$$((\bar{m}_1 - m_1) - (\bar{m}_2 - m_2)) \pi + \bar{\alpha}_1 - \bar{\alpha}_2. \tag{C.2}$$

The sign of this quantity can be changed by a single anti-holomorphic light crossing only if $(\bar{m}_1 - m_1) - (\bar{m}_2 - m_2) = 0$ and then if the move $\bar{A}$ is one of $\bar{I}$ or its inverse. In every other case, the monodromy associated with the two $I$ operations vanishes trivially. In the special case that $(\bar{m}_1 - m_1) - (\bar{m}_2 - m_2) = 0$ and $\bar{A} = \bar{I}$, it is also true that the insertion of $I$ between $\bar{A}$ and $\bar{A}^{-1}$ obstructs the cancellation of monodromies of $\bar{A}$ and $\bar{A}^{-1}$ (indeed the condition for lack of this obstruction is, by symmetry, clearly identical to that (C.2), so the last statement is of the "if and only if" variety). In this potentially problematic situation, the sequence of moves $I\bar{I}I\bar{I}$ generates the motion on the Lorentzian cylinder that can be enclosed in a single Poincare Patch. In Fig 26 we depict the action of $I\bar{I}I\bar{I}$ on the configuration $(P_{12}^{m,m}, Q_{21}^{\bar{m},\bar{m}})$ (the leftmost vertex in Fig 26). Note that the initial configuration for this motion has $m_1 = m_2 = m$ and $\bar{m}_1 = \bar{m}_2 = \bar{m}$. It follows, in other words, that the motion depicted in 26 occurs entirely in a single Minkowski diamond of the cylinder (if choose to tile the cylinder with Minkowski diamonds centered at the operator 3).

Using the explicit form (12) (in the case of the 3 point function)[56] or the half-monodromy rules listed in section 3.2[57] (when we turn to the study of 4 point, or more general point functions) it is easy to see that the net monodromy associated with the moves in Fig 26 vanishes even in this potentially nontrivial case.

We can now repeat this analysis for the unit face of moves given by

$$F \, \bar{B} \, B \, \bar{F}, \tag{C.3}$$

acting on $\left(P_{21}^{m_1,m_2}\right)$.[58] This sequence of moves is non-trivial in two cases. The first of these is when $m_1 = 0$ and $\bar{m}_1 = -1$[59] so that the starting point is $(P_{21}^{0,m_2}, Q_{12}^{-1,\bar{m}_2})$ (see (16) for definitions). In this case, the sequence of moves (C.3) causes the insertion operator 1 to execute a motion on the Lorentzian cylinder depicted in Fig 27 below.

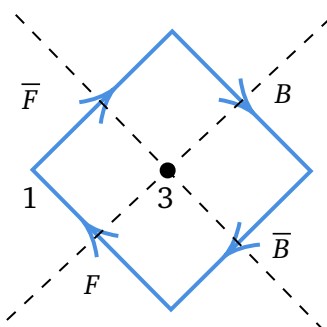

Figure 27: In this figure, we depict the motion of the insertion of the operator 1 for the sequence of moves $F\bar{B}B\bar{F}$ acting on $(P_{21}^{0,m_2}, Q_{12}^{-1,\bar{m}_2})$. We start at the left corner of this diagram. The operations, $\bar{F}$, $B$, $\bar{B}$ and $F$ then respectively move us along the legs of this diamond. As we explain in the main text, the total monodromy along this path also is zero.

It is easily verified that the net monodromy vanishes for the motion depicted in Fig 27.

The second case in which the moves (C.3) are nontrivial is when these moves act on $(P_{12}^{m_1,0}, Q_{21}^{\bar{m}_1,-1})$. This is the starting configuration considered in the previous paragraph but

---

[56] Together with the fact that $h_i - \bar{h}_i$ are integers.

[57] Together with Euclidean single valuedness, see (33).

[58] In this situation, the operation $F$ acts on $\left(P_{12}^{m_1-1,m_2}, Q^{\bar{m}_1,\bar{m}_2}\right)$.

[59] More invariantly, we must choose $m_1 - \bar{m}_1 = -1$, see the para below (16).

with $1 \leftrightarrow 2$. The resultant motion is thus again given the motion in Fig. 27 but with $1 \leftrightarrow 2$, and is depicted in Fig 28. Of course, the monodromy for this sequence of moves also vanishes.

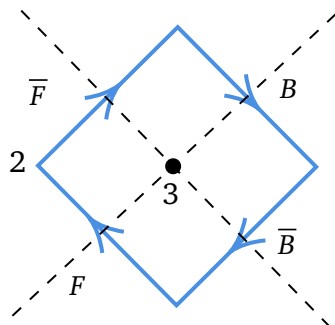

Figure 28: This figure is the $1 \leftrightarrow 2$ version of the previous figure. We depict the motion of the insertion of the operator 2 for the sequence of moves $F\bar{B}B\bar{F}$ acting on $(P_{12}^{m_1,0}, Q_{21}^{\tilde{m}_1,-1})$. We start at the left corner of this diagram. The operations, $\bar{F}$, $B$, $\bar{B}$ and $F$ then respectively move us along the legs of this diamond. The total monodromy vanishes.

## C.2 Purely leftmoving holonomies are independent of the rightmoving location

Consider a configuration with insertions at some given leftmoving and some given rightmoving locations. Consider a closed loop in purely leftmoving space (i.e. at constant values of rightmoving coordinates). Any such loop is associated with a (potentially nontrivial) monodromy. We will now demonstrate that this monodromy is independent of the rightmoving locations of our insertions. This completes our demonstration that (for computing monodromies) leftmoving and rightmoving coordinates completely decouple from each other.

The proof that follows is built on the following intuition. Consider a cube like that depicted in Fig. 29. The vertical axis in this cube represents a motion on the rightmoving lattice of causally distinct configurations, while the horizontal directions of this cube represent motions on the corresponding leftmoving lattice. We wish to show that the (purely left-moving) monodromy associated with traversing the lower horizontal face of this cube is the same as the (purely left-moving) monodromy associated with traversing the upper horizontal face of this cube.[60]

The problem discussed above has a simple gauge theory analogue, whose study helps build intuition. Let the monodromy around any loop around a lattice be thought of as a Wilson line of a particular (latticized) gauge field. Stokes' law then tells us that the monodromy associated with the lower and upper horizontal surfaces is given by the "field strengths" $F_{12}$ at neighbouring values of $\bar{m}_1$ (see the cube in Fig. 29). So the statement we want to demonstrate is the lattice version of the equation $\partial_{\bar{1}} F_{12} = 0$.

Now, in the previous subsection, we have already argued that monodromies associated with holomorphic and anti-holomorphic moves commute. Since $\bar{1}$ is an anti-holomorphic direction, while 1 and 2 are holomorphic directions, we have argued that monodromies on the faces $1\bar{1}$ and $2\bar{1}$ (see Fig. 29) vanish. In the intuitive language of the previous paragraph, we have, therefore argued that $F_{1\bar{1}} = F_{2\bar{1}} = 0$. Since this is true everywhere, it follows that $\partial_2 F_{1\bar{1}} = \partial_1 F_{2\bar{1}} = 0$. Now our field strength should obey the Bianchi identity $\partial_{\bar{1}} F_{12} + \partial_2 F_{\bar{1}1} + \partial_1 F_{2\bar{1}} = 0$. Plugging $\partial_2 F_{1\bar{1}} = \partial_1 F_{2\bar{1}} = 0$ into this identity, we conclude that $\partial_{\bar{1}} F_{12} = 0$ as desired.

---

[60]Thus establishing that left-moving monodromies are independent of right-moving location.

The intuitive argument of the previous paragraph may be made precise as follows following the discussion of [29]. We consider the trajectory 29, which we will call $r$. A brief perusal of Fig. 30 will convince the reader that the path drawn in Fig. 29 is both equal to the composition of the moves $r_3$, $r_2$ and $r_1$, and separately equal to the composition of the moves $r_6$, $r_5$ and $r_4$. In equations

$$r = r_3 \times r_2 \times r_1 = (r_6 \times r_5 \times r_4)^{-1}. \tag{C.4}$$

However, it follows from the discussion of the previous subsection that each of the contours $r_1$, $r_3$, $r_4$ and $r_6$ have trivial monodromies. It follows that the monodromies associated with $r_2$ and $r_5$ are equal, as we set out to prove.

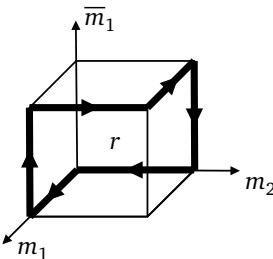

Figure 29: A closed contour in the lattice of causal configurations, which we decompose into a product of simple contours in two inequivalent ways in the next figure.

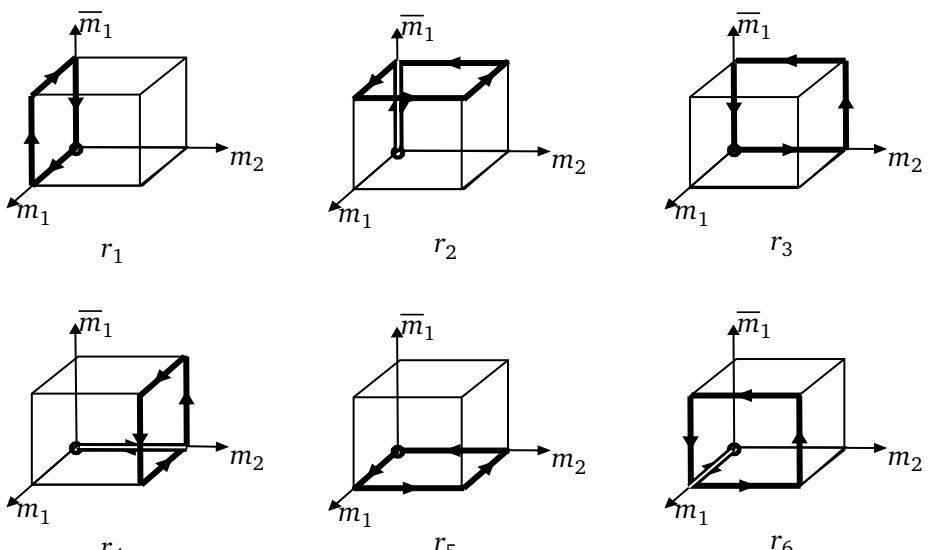

Figure 30: The closed contour of the previous figure can be equal to the product of motions $r_1 r_2 r_3$ and also the product of $r_4 r_5 r_6$ (with the convention that the move listed last is performed first). Note that the moves $r_1$, $r_3$, $r_4$ and $r_6$ capture the commutativity of holomorphic and anti-holomorphic moves, and so were demonstrated to vanish in the previous subsection. The argument presented in this diagram is adapted from [29].

In this section, we have demonstrated that the "12" monodromy is independent of the $\bar{1}$ location. 1 and 2 could represent any of the many possible holomorphic moves, while $\bar{1}$ could represent any anti-holomorphic direction. We have demonstrated in great generality (i.e., both for three, four and higher point functions) that the values of holomorphic monodromies are independent of anti-holomorphic locations. Of course, the values of anti-holomorphic monodromies are also independent of holomorphic locations.

# D  Analysis using D function

D-function is defined as follows,

$$D(z,\bar{z}) = \frac{z\bar{z}}{z-\bar{z}}\left[2\mathrm{Li}_2(z) - 2\mathrm{Li}_2(\bar{z}) + \log(z\bar{z})\log\frac{1-z}{1-\bar{z}}\right]. \tag{D.1}$$

For simplicity, we will suppress the $\frac{z\bar{z}}{z-\bar{z}}$ factor outside. Let us consider,

$$
\begin{aligned}
v &= \{1, \log z, \log(1-z), \log z \log(1-z), \mathrm{Li}_2(z)\}, \\
\bar{v} &= \{1, \log\bar{z}, \log(1-\bar{z}), \log\bar{z}\log(1-\bar{z}), \mathrm{Li}_2(\bar{z})\}.
\end{aligned}
\tag{D.2}
$$

Then, we can represent the $D$ function as following

$$D = \bar{v}^T.P.v, \tag{D.3}$$

where,

$$P = \begin{pmatrix} 0 & 0 & 0 & 1 & 2 \\ 0 & 0 & 1 & 0 & 0 \\ 0 & -1 & 0 & 0 & 0 \\ -1 & 0 & 0 & 0 & 0 \\ -2 & 0 & 0 & 0 & 0 \end{pmatrix}. \tag{D.4}$$

We can also write the matrix form of the transformations related to each monodromy. In the functional form,

Table 8: Monodromy effects on D function.

| monodromy | non-trivial effect |
|---|---|
| $C_0$ | $\log z \to \log z - 2\pi i$ |
| $A_0$ | $\log z \to \log z + 2\pi i$ |
| $\bar{C}_0$ | $\log \bar{z} \to \log \bar{z} - 2\pi i$ |
| $\bar{A}_0$ | $\log \bar{z} \to \log \bar{z} + 2\pi i$ |
| $C_1$ | $\mathrm{Li}_2(z) \to \mathrm{Li}_2(z) + 2\pi i \log z,\ \log(1-z) \to \log(1-z) - 2\pi i$ |
| $A_1$ | $\mathrm{Li}_2(z) \to \mathrm{Li}_2(z) - 2\pi i \log z,\ \log(1-z) \to \log(1-z) + 2\pi i$ |
| $\bar{C}_1$ | $\mathrm{Li}_2(\bar{z}) \to \mathrm{Li}_2(\bar{z}) + 2\pi i \log \bar{z},\ \log(1-\bar{z}) \to \log(1-\bar{z}) - 2\pi i$ |
| $\bar{A}_1$ | $\mathrm{Li}_2(\bar{z}) \to \mathrm{Li}_2(\bar{z}) - 2\pi i \log \bar{z},\ \log(1-\bar{z}) \to \log(1-\bar{z}) + 2\pi i$ |

The same effects can be captured using matrices as follows,

$$v \to C_0.v, \qquad v \to C_1.v, \qquad \bar{v} \to \bar{C}_0.\bar{v}, \qquad \text{etc.} \tag{D.5}$$

Then, we can assign the following matrix form for the monodromies

$$
C_0 = \bar{C}_0 = \begin{pmatrix} 1 & 0 & 0 & 0 & 0 \\ -2i\pi & 1 & 0 & 0 & 0 \\ 0 & 0 & 1 & 0 & 0 \\ 0 & 0 & -2i\pi & 1 & 0 \\ 0 & 0 & 0 & 0 & 1 \end{pmatrix}, \qquad
C_1 = \bar{C}_1 = \begin{pmatrix} 1 & 0 & 0 & 0 & 0 \\ 0 & 1 & 0 & 0 & 0 \\ -2i\pi & 0 & 1 & 0 & 0 \\ 0 & -2i\pi & 0 & 1 & 0 \\ 0 & 2i\pi & 0 & 0 & 1 \end{pmatrix},
$$

$$
A_0 = \bar{A}_0 = \begin{pmatrix} 1 & 0 & 0 & 0 & 0 \\ 2i\pi & 1 & 0 & 0 & 0 \\ 0 & 0 & 1 & 0 & 0 \\ 0 & 0 & 2i\pi & 1 & 0 \\ 0 & 0 & 0 & 0 & 1 \end{pmatrix}, \qquad
A_1 = \bar{A}_1 = \begin{pmatrix} 1 & 0 & 0 & 0 & 0 \\ 0 & 1 & 0 & 0 & 0 \\ 2i\pi & 0 & 1 & 0 & 0 \\ 0 & 2i\pi & 0 & 1 & 0 \\ 0 & -2i\pi & 0 & 0 & 1 \end{pmatrix}.
\tag{D.6}
$$

The following identities are true,

$$
\begin{aligned}
A_1.C_1 &= A_0.C_0 = \mathbb{1}\,, \\
\bar{v}^T.P.v &= D\,, \\
\bar{C}_1^T.P.A_1 &= P\,, \\
\bar{C}_0^T.P.A_0 &= P\,, \\
\bar{C}_0^T.\bar{C}_1^T.P.A_1.A_0 &= P\,, \ \text{etc.}\,, \\
P.A_0.A_1.A_0 = \bar{A}_0^T.\bar{A}_1^T.\bar{A}_0^T.P &= \bar{A}_0^T.P.A_1.A_0\,.
\end{aligned}
\tag{D.7}
$$

The only confusing part is, realization of the last identity can be realised through equivalence of three motions,

- motion only in $z$ gives, $A_0 \to A_1 \to A_0$,

- motion only in $\bar{z}$ gives, $A_0 \to A_1 \to A_0$,

- straight motion gives, $A_1 \to A_0$ in $z$ and $A_0$ in $\bar{z}$ plane,

here, "$\to$" implies "then".

Also, if we use the direct rules from the table to $D$ function, and we act the monodromies in the order as we get from our way, we get a perfect match.

We parenthetically note that the $D$ function is the result of a correlator computed using contact diagram in the bulk of AdS (via the AdS/CFT correspondence). It would be interesting to perform an analysis similar to this appendix, on the more complicated analytic structures that arise out of exchange or loop computations in the bulk.

# E  Path Independence of 4 point functions

## E.1  Path independence on a Minkowski diamond

As a warm-up, let us first consider four operators inserted on the Minkowski diamond. We have 4! possible different "leftmoving" orderings, and 4! different rightmoving orderings of these operators. We have already seen above that monodromies in $z$ and $\bar{z}$ do not talk to each other (this follows because $z$ monodromies are represented by right multiplications on $P$, while $\bar{z}$ monodromies are represented by left multiplications on $P$). As a consequence, we can deal separately with trajectories in $z$ and $\bar{z}$.

The 4! different leftmoving orderings are in one-to-one correspondence with elements of the permutation group. Starting with any one element of the permutation group, one can reach every one of the 4! elements using only three local moves: interchanging the first and second element, the second and third element, and the third and fourth element. Physically, this reflects the fact that we can reach any "leftmoving time ordering" starting from an arbitrary initial configuration, by interchanging neighbouring (in time ordering) insertions.

It is convenient to draw a lattice diagram in which every node is a distinct leftmoving time ordering (distinct element of the permutation group) and every link represents one of these three basic adjacent flips in time ordering. The resultant diagram takes the form represented in Fig. 31.

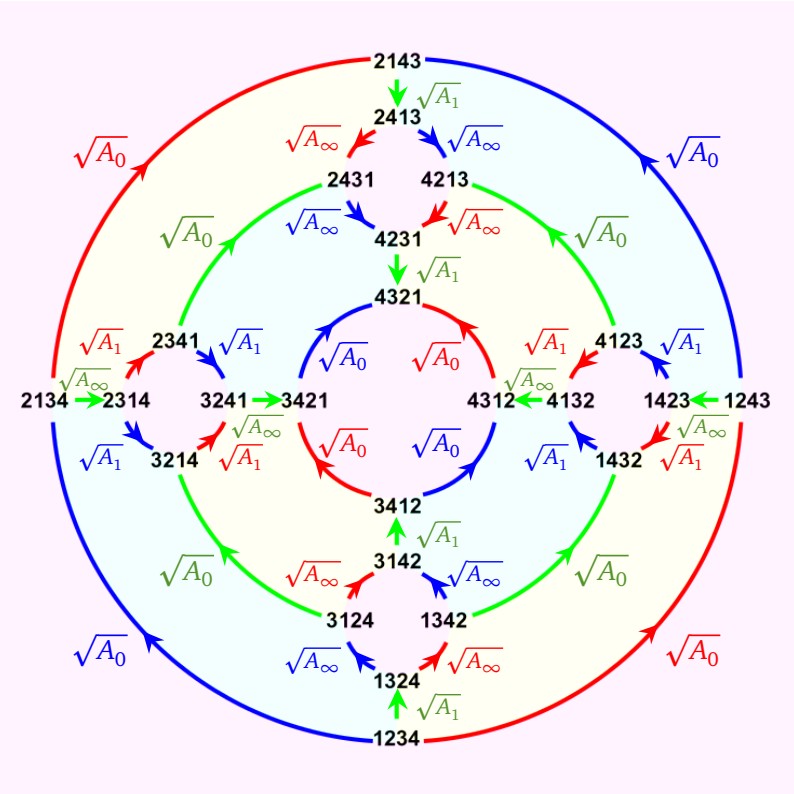

Figure 31: In this figure, we denote the causal lattice for configurations on Minkowski space, or, equivalently, a single Minkowski diamond. Points on the lattice are elements of $S_4$ as explained in the main text. Links are moves that involve crossing a single lightcone and are depicted by solid-coloured lines above. The monodromy over each plaquette of this diagram vanishes.

Now, any closed path in the space of insertion points maps to a closed path in Fig. 31. One starts at a given vertex of this graph, moves along different edges, and eventually comes back to the starting vertex. We want to argue that the motion along any such closed trajectory returns the correlator to itself. To make this argument, it is sufficient to argue that the traversal along each of the "elementary or generator" faces of the graph in Fig. 31 leaves the correlator unchanged. The graph in Fig. 31 has two distinct kinds of elementary simplices, one four-sided and the second six-sided. The four-sided faces are generated by the permutation sequence $P_{12}P_{34}P_{12}P_{34}$, while the six-sided faces are generated by either of the permutation sequences $P_{12}P_{23}P_{12}P_{23}P_{12}P_{23}$ or $P_{23}P_{34}P_{23}P_{34}P_{23}P_{34}$.

To verify path independence, we must now check that the monodromy associated with each of the basic faces listed above vanishes. It might, at first, seem that this path independence can easily be verified using "half-monodromy" rules developed in section 3.2. However, we encounter a subtlety at this point. The half-monodromy rules of section 3.2 are different depending on whether we pass lightcones from past to future or from future to past, and whether the lightcone we cut is a past or future lightcone.

Let's say that we are changing $\omega_i$ values at fixed $\bar{\omega}_i$. The first question (if we are moving from past to future or the converse) is completely determined by the relative orderings of $\omega_i$ in the initial and final configurations. However, the second question (whether we are cutting a past or future lightcone) is determined by the relative values of $\bar{\omega}_i$ at the time of crossing of $\omega_i$ lightcones. For this reason, the determination of monodromies does not obviously factorize between left and right motion. One has to remember that the full configuration space (as far

as monodromies are concerned) is a direct product of two of the "footballs" displayed in Fig. 31. The first "football" keeps track of the relative $\omega_i$ orderings, while the second one keeps track of $\bar{\omega}_i$ orderings.

Before worrying about the path independence for motion in $\omega_i$, consequently, we must first verify the commutation of monodromies in $\omega_i$ and $\bar{\omega}_i$. This is easily done as follows. Suppose that, initially, $\omega_i > \omega_j$ and, also, $\bar{\omega}_i < \bar{\omega}_j$. Say we want to move to $\omega_i < \omega_j$ and $\bar{\omega}_i > \bar{\omega}_j$. We can move in two distinct ways: either by first moving to $\omega_i < \omega_j$ keeping $\bar{\omega}$ fixed and then to $\bar{\omega}_i > \bar{\omega}_j$ keeping $\omega$ fixed, or by doing these moves in the converse order.

The first option involves $\omega_i$ first crossing a past holomorphic lightcone (centered at $\omega_j$) and $\bar{\omega}_i$ then crossing a future anti-holomorphic lightcone (again centered at $\bar{\omega}_j$ from past to future. We thus get the monodromies $\sqrt{C_{ij}}$ and $\sqrt{\bar{A}_{ij}}$. The second option switches the order of crossings, and so involves $\bar{\omega}_i$ first crossing a past anti-holomorphic lightcone (centered at $\omega_j$) and $\omega_i$ then crossing a future holomorphic lightcone (again centered at $\bar{\omega}_j$ from past to future. We get the monodromies $\sqrt{\bar{C}_{ij}}$ and $\sqrt{A_{ij}}$. As in the discussion around (32), these two orders of operations, respectively, effectively replace the pairing matrix by $\sqrt{A_{ij}}^\dagger \cdot P \cdot \sqrt{C_{ij}}$ and $\sqrt{C_{ij}}^\dagger \cdot P \cdot \sqrt{A_{ij}}$ respectively. Note that $\sqrt{C_{ij}}$ and $\sqrt{A_{ij}}$ are inverses of each other. It follows that the two new effective pairing matrices are identical if and only if

$$A_{ij}^\dagger \cdot P \cdot C_{ij} = P = C_{ij}^\dagger \cdot P \cdot A_{ij}. \tag{E.1}$$

But $C_{ij}$ represents a full monodromy, and so (E.1) indeed holds as a consequence of (33).

To verify that moving along any of the edges in Fig. 31 induces the half-monodromy listed under each of the edges in the figure (moving in the opposite direction reverses this motion, and so, for instance, replaces $\sqrt{A_0}$ - an anticlockwise half-monodromy around 0 - with $\sqrt{C_0}$ - a clockwise half-monodromy around the same branch point.). Using these rules - together with the obvious fact that $\sqrt{A_0} \cdot \sqrt{A_1} \cdot \sqrt{A_\infty} = \sqrt{C_0} \cdot \sqrt{C_1} \cdot \sqrt{C_\infty} = \phi$, it is easy to check that the motion around both the four-sided as well as a six-sided simplex induces trivial monodromy. This completes the proof of path independence on a Minkowski diamond.

## E.2 Triviality of loops on cubic edges

In this subsection we demonstrate that the three loops which are lying on the edges of the cubic lattice give trivial monodromy. We will be using equations (40), (41), (42) and (43). Let's see the first edge from (46).

$$\begin{aligned}
BP_2FP_1(B_2^{m,n,p}) &= BP_2F(A_2^{m,n,p}) \\
&= BP_2(A_1^{m+1,n,p}) \\
&= B(B_1^{m+1,n,p}) \\
&= B_2^{m,n,p}.
\end{aligned} \tag{E.2}$$

As we can see, it gives a trivial monodromy. Now similarly the second edge from (47).

$$\begin{aligned}
BP_2FP_1(A_3^{m,n,p}) &= BP_2F(B_1^{m,n,p}) \\
&= BP_2(B_3^{m,n+1,p}) \\
&= B(A_2^{m,n+1,p}) \\
&= A_3^{m,n,p}.
\end{aligned} \tag{E.3}$$

Finally the third edge from (48).

$$
\begin{aligned}
BP_2FP_1(B_3^{m,n,p}) &= BP_2F(A_1^{m,n,p}) \\
&= BP_2(A_3^{m,n,p+1}) \\
&= B(B_2^{m,n,p+1}) \\
&= B_3^{m,n,p} .
\end{aligned} \tag{E.4}
$$

### E.3 Triviality of loops on cubic faces

In this subsection we demonstrate that the three loops which are lying on the faces of the cubic lattice give trivial monodromy. We will be using equations (40), (41), (42) and (43).

$$
\begin{aligned}
P_1P_2P_1BP_2BP_1FP_2F(A_2^{m,n,p}) &= P_1P_2P_1BP_2BP_1FP_2(A_1^{m+1,n,p}) \\
&= P_1P_2P_1BP_2BP_1F(B_1^{m+1,n,p}) \\
&= P_1P_2P_1BP_2BP_1(B_3^{m+1,n+1,p}) \\
&= P_1P_2P_1BP_2B(A_1^{m+1,n+1,p}) \\
&= P_1P_2P_1BP_2(A_2^{m,n+1,p}) \\
&= P_1P_2P_1B(B_3^{m,n+1,p}) \\
&= P_1P_2P_1(B_1^{m,n,p}) \\
&= P_1P_2(A_3^{m,n,p}) \\
&= P_1(B_2^{m,n,p}) \\
&= A_2^{m,n,p} .
\end{aligned} \tag{E.5}
$$

## F  Detailed calculations of various configurations

### F.1  *A type configurations*

1. $n_i \geq n_j \geq n_m \geq n_n$

    In this case we first move $\omega_i \to \omega_i - n_i\pi$. It crosses 3 future light cones which according to rule (1) of §6.2 gives $\phi$. We then move $\omega_j \to \omega_j - n_j\pi$. It crosses 1 past and 2 future light cones which according to rule (2) of §6.2 gives $C_{ij}^{n_j}$. We then move $\omega_m \to \omega_m - n_m\pi$. It crosses 2 past and 1 future light cones which as per rule (3) gives $A_{mn}^{n_m} \equiv A_{ij}^{n_m}$. Finally we move $\omega_n \to \omega_n - n_n\pi$. It crosses 3 past light cones and hence as per rule (1) does nothing.

    So in this configuration we get to the relevant sheet by starting from the Euclidean sheet and doing $(n_j - n_m)$ clockwise circles around $z_{ij}$, i.e., $C_{ij}^{n_j-n_m}$.

2. $n_i \geq n_j \geq n_n \geq n_m$

    In this case we first move $\omega_i \to \omega_i - n_i\pi$. It crosses 3 future light cones. This move induces no monodromy. We then make the shifts $\omega_j \to \omega_j - n_j\pi$. It crosses 1 past and 2 future lightcones which according to rule 2 of §6.2 gives $C_{ij}^{n_j}$.

    We then move $\omega_n \to \omega_n - n_n\pi$. It crosses 1 future and 2 past lightcones which as per rule (3) of §6.2 gives $A_{nm}^{n_n} \equiv A_{ij}^{n_n}$. Finally we move $\omega_m \to \omega_m - n_m\pi$. It crosses 3 past light cones and gives $\phi$.

    So in this configuration we get to the relevant sheet by starting from the Euclidean sheet and doing $(n_j - n_n)$ clockwise circles around $z_{ij}$, i.e., $C_{ij}^{n_j-n_n}$.

3. $n_i \geq n_m \geq n_j \geq n_n$

In this case we first move $\omega_i \to \omega_i - n_i \pi$. It crosses 3 future light cones. This move induces no monodromy. We then make the shifts $\omega_m \to \omega_m - n_m \pi$. It crosses future, past, future lightcone configuration which according to rule 2 of §6.2 gives $\sqrt{A_{mj}} \cdot C_{mi}^{\,n_m} \cdot \sqrt{C_{mj}} \equiv \sqrt{A_{in}} \cdot C_{im}^{\,n_m} \cdot \sqrt{C_{in}}$.

We then move $\omega_j \to \omega_j - n_j \pi$. It crosses past, future, past lightcone configuration which as per rule (3) of §6.2 gives $\sqrt{C_{ji}} \cdot A_{jn}^{\,n_j} \cdot \sqrt{A_{ji}} \equiv \sqrt{C_{ij}} \cdot A_{im}^{\,n_j} \cdot \sqrt{A_{ij}}$.

Finally we move $\omega_n \to \omega_n - n_n \pi$. It crosses 3 past light cones and gives $\phi$. Putting it all together, the final monodromy is

$$\sqrt{A_{in}} \cdot C_{im}^{\,n_m} \cdot \sqrt{C_{in}} \cdot \sqrt{C_{ij}} \cdot A_{im}^{\,n_j} \cdot \sqrt{A_{ij}}$$
$$= \sqrt{C_{ij}} \cdot \sqrt{C_{im}} \cdot C_{im}^{\,n_m} \cdot \sqrt{A_{im}} \cdot A_{im}^{\,n_j} \cdot \sqrt{A_{ij}}$$
$$= \sqrt{C_{ij}} \cdot C_{im}^{\,n_m - n_j} \cdot \sqrt{A_{ij}} .$$

In words, we get to the relevant sheet by starting from the Euclidean sheet, doing a half-clockwise monodromy around $z_{ij}$, then $(n_m - n_j)$ clockwise circles around $z_{im}$ and finally followed by a half-anticlockwise monodromy around $z_{ij}$, i.e., $\sqrt{C_{ij}} \cdot C_{im}^{\,n_m - n_j} \cdot \sqrt{A_{ij}}$.

4. $n_i \geq n_m \geq n_n \geq n_j$

In this case we first move $\omega_i \to \omega_i - n_i \pi$. It crosses 3 future light cones. This move induces no monodromy. We then make the shifts $\omega_m \to \omega_m - n_m \pi$. It crosses future, past, future lightcone configuration which according to rule 2 of §6.2 gives $\sqrt{A_{mj}} \cdot C_{mi}^{\,n_m} \cdot \sqrt{C_{mj}} \equiv \sqrt{A_{in}} \cdot C_{im}^{\,n_m} \cdot \sqrt{C_{in}}$.

We then move $\omega_n \to \omega_n - n_n \pi$. It crosses past, future, past lightcone configuration which as per rule (3) of §6.2 gives $\sqrt{C_{nm}} \cdot A_{nj}^{\,n_n} \cdot \sqrt{A_{nm}} \equiv \sqrt{C_{ij}} \cdot A_{im}^{\,n_n} \cdot \sqrt{A_{ij}}$.

Finally we move $\omega_j \to \omega_j - n_j \pi$. It crosses 3 past light cones and gives $\phi$. Putting it all together, the final monodromy is

$$\sqrt{A_{in}} \cdot C_{im}^{\,n_m} \cdot \sqrt{C_{in}} \cdot \sqrt{C_{ij}} \cdot A_{im}^{\,n_n} \cdot \sqrt{A_{ij}}$$
$$= \sqrt{C_{ij}} \cdot \sqrt{C_{im}} \cdot C_{im}^{\,n_m} \cdot \sqrt{A_{im}} \cdot A_{im}^{\,n_n} \cdot \sqrt{A_{ij}}$$
$$= \sqrt{C_{ij}} \cdot C_{im}^{\,n_m - n_n} \cdot \sqrt{A_{ij}} .$$

In words, we get to the relevant sheet by starting from the Euclidean sheet, doing a half-clockwise monodromy around $z_{ij}$, then $(n_m - n_n)$ clockwise circles around $z_{im}$ and finally followed by a half-anticlockwise monodromy around $z_{ij}$, i.e., $\sqrt{C_{ij}} \cdot C_{im}^{\,n_m - n_n} \cdot \sqrt{A_{ij}}$.

5. $n_i \geq n_n \geq n_j \geq n_m$

In this case we first move $\omega_i \to \omega_i - n_i \pi$. It crosses 3 future light cones. This move induces no monodromy. We then make the shifts $\omega_n \to \omega_n - n_n \pi$. It crosses 2 future and 1 past lightcones which according to rule 2 of §6.2 gives $C_{in}^{\,n_n}$.

We then move $\omega_j \to \omega_j - n_j \pi$. It crosses 2 past and 1 future lightcones which as per rule (3) of §6.2 gives $A_{jm}^{\,n_j} \equiv A_{in}^{\,n_j}$. Finally we move $\omega_m \to \omega_m - n_m \pi$. It crosses 3 past light cones and gives $\phi$.

So in this configuration we get to the relevant sheet by starting from the Euclidean sheet and doing $(n_n - n_j)$ clockwise circles around $z_{in}$, i.e., $C_{in}^{\,n_n - n_j}$.

6. $n_i \geq n_n \geq n_m \geq n_j$

In this case we first move $\omega_i \to \omega_i - n_i \pi$. It crosses 3 future light cones. This move induces no monodromy. We then make the shifts $\omega_n \to \omega_n - n_n \pi$. It crosses 2 future and 1 past lightcones which according to rule 2 of §6.2 gives $C_{in}^{n_n}$.

We then move $\omega_m \to \omega_m - n_m \pi$. It crosses 1 future and 2 past lightcones which as per rule (3) of §6.2 gives $A_{mj}^{n_m} \equiv A_{in}^{n_m}$. Finally we move $\omega_j \to \omega_j - n_j \pi$. It crosses 3 past light cones and gives $\phi$.

So in this configuration we get to the relevant sheet by starting from the Euclidean sheet and doing $(n_n - n_m)$ clockwise circles around $z_{in}$, i.e., $C_{in}^{n_n - n_m}$.

7. $n_j \geq n_i \geq n_m \geq n_n$

In this case we first move $\omega_j \to \omega_j - n_j \pi$. It crosses 3 future light cones. This move induces no monodromy. We then make the shifts $\omega_i \to \omega_i - n_i \pi$. It crosses 2 future and 1 past lightcones which according to rule 2 of §6.2 gives $C_{ij}^{n_i}$.

We then move $\omega_m \to \omega_m - n_m \pi$. It crosses 2 past and 1 future lightcones which as per rule (3) of §6.2 gives $A_{mn}^{n_m} \equiv A_{ij}^{n_m}$. Finally we move $\omega_n \to \omega_n - n_n \pi$. It crosses 3 past light cones and gives $\phi$.

So in this configuration we get to the relevant sheet by starting from the Euclidean sheet and doing $(n_i - n_m)$ clockwise circles around $z_{ij}$, i.e., $C_{ij}^{n_i - n_m}$.

8. $n_j \geq n_i \geq n_n \geq n_m$

In this case we first move $\omega_j \to \omega_j - n_j \pi$. It crosses 3 future light cones. This move induces no monodromy. We then make the shifts $\omega_i \to \omega_i - n_i \pi$. It crosses 2 future and 1 past lightcones which according to rule 2 of §6.2 gives $C_{ij}^{n_i}$.

We then move $\omega_n \to \omega_n - n_n \pi$. It crosses 1 future and 2 past lightcones which as per rule (3) of §6.2 gives $A_{nm}^{n_n} \equiv A_{ij}^{n_n}$. Finally we move $\omega_m \to \omega_m - n_m \pi$. It crosses 3 past light cones and gives $\phi$.

So in this configuration we get to the relevant sheet by starting from the Euclidean sheet and doing $(n_i - n_n)$ clockwise circles around $z_{ij}$, i.e., $C_{ij}^{n_i - n_n}$.

9. $n_j \geq n_m \geq n_i \geq n_n$

In this case we first move $\omega_j \to \omega_j - n_j \pi$. It crosses 3 future light cones. This move induces no monodromy. We then make the shifts $\omega_m \to \omega_m - n_m \pi$. It crosses 1 past and 2 future lightcones which according to rule 2 of §6.2 gives $C_{mj}^{n_m} \equiv C_{in}^{n_m}$.

We then move $\omega_i \to \omega_i - n_i \pi$. It crosses 1 future and 2 past lightcones which as per rule (3) of §6.2 gives $A_{in}^{n_i}$. Finally we move $\omega_n \to \omega_n - n_n \pi$. It crosses 3 past light cones and gives $\phi$.

So in this configuration we get to the relevant sheet by starting from the Euclidean sheet and doing $(n_m - n_i)$ clockwise circles around $z_{in}$, i.e., $C_{in}^{n_m - n_i}$.

10. $n_j \geq n_m \geq n_n \geq n_i$

In this case we first move $\omega_j \to \omega_j - n_j \pi$. It crosses 3 future light cones. This move induces no monodromy. We then make the shifts $\omega_m \to \omega_m - n_m \pi$. It crosses 1 past and 2 future lightcones which according to rule 2 of §6.2 gives $C_{mj}^{n_m} \equiv C_{in}^{n_m}$.

We then move $\omega_n \to \omega_n - n_n \pi$. It crosses 2 past and 1 future lightcones which as per rule (3) of §6.2 gives $A_{in}^{n_n}$. Finally we move $\omega_i \to \omega_i - n_i \pi$. It crosses 3 past light cones and gives $\phi$.

So in this configuration we get to the relevant sheet by starting from the Euclidean sheet and doing $(n_m - n_n)$ clockwise circles around $z_{in}$, i.e., $C_{in}^{n_m - n_n}$.

11. $n_j \geq n_n \geq n_i \geq n_m$

   In this case we first move $\omega_j \to \omega_j - n_j\pi$. It crosses 3 future light cones. This move induces no monodromy. We then make the shifts $\omega_n \to \omega_n - n_n\pi$. It crosses future, past, future lightcone configuration which according to rule 2 of §6.2 gives $\sqrt{A_{nm}} \cdot C_{nj}^{n_n} \cdot \sqrt{C_{nm}} \equiv \sqrt{A_{ij}} \cdot C_{im}^{n_n} \cdot \sqrt{C_{ij}}$.

   We then move $\omega_i \to \omega_i - n_i\pi$. It crosses past, future, past lightcone configuration which as per rule (3) of §6.2 gives $\sqrt{C_{in}} \cdot A_{im}^{n_i} \cdot \sqrt{A_{in}}$.

   Finally we move $\omega_m \to \omega_m - n_m\pi$. It crosses 3 past light cones and gives $\phi$. Putting it all together, the final monodromy is

$$
\begin{aligned}
& \sqrt{A_{ij}} \cdot C_{im}^{n_n} \cdot \sqrt{C_{ij}} \cdot \sqrt{C_{in}} \cdot A_{im}^{n_i} \cdot \sqrt{A_{in}} \\
& = \sqrt{C_{in}} \cdot \sqrt{C_{im}} \cdot C_{im}^{n_n} \cdot \sqrt{A_{im}} \cdot A_{im}^{n_i} \cdot \sqrt{A_{in}} \\
& = \sqrt{C_{in}} \cdot C_{im}^{n_n - n_i} \cdot \sqrt{A_{in}}.
\end{aligned}
$$

   In words, we get to the relevant sheet by starting from the Euclidean sheet, doing a half-clockwise monodromy around $z_{in}$, then $(n_n - n_i)$ clockwise circles around $z_{im}$ and finally followed by a half-anticlockwise monodromy around $z_{in}$, i.e., $\sqrt{C_{in}} \cdot C_{im}^{n_n - n_i} \cdot \sqrt{A_{in}}$.

12. $n_j \geq n_n \geq n_m \geq n_i$

   In this case we first move $\omega_j \to \omega_j - n_j\pi$. It crosses 3 future light cones. This move induces no monodromy. We then make the shifts $\omega_n \to \omega_n - n_n\pi$. It crosses future, past, future lightcone configuration which according to rule 2 of §6.2 gives $\sqrt{A_{nm}} \cdot C_{nj}^{n_n} \cdot \sqrt{C_{nm}} \equiv \sqrt{A_{ij}} \cdot C_{im}^{n_n} \cdot \sqrt{C_{ij}}$.

   We then move $\omega_m \to \omega_m - n_m\pi$. It crosses past, future, past lightcone configuration which as per rule (3) of §6.2 gives $\sqrt{C_{mj}} \cdot A_{mi}^{n_m} \cdot \sqrt{A_{mj}} \equiv \sqrt{C_{in}} \cdot A_{im}^{n_m} \cdot \sqrt{A_{in}}$.

   Finally we move $\omega_i \to \omega_i - n_i\pi$. It crosses 3 past light cones and gives $\phi$. Putting it all together, the final monodromy is

$$
\begin{aligned}
& \sqrt{A_{ij}} \cdot C_{im}^{n_n} \cdot \sqrt{C_{ij}} \cdot \sqrt{C_{in}} \cdot A_{im}^{n_m} \cdot \sqrt{A_{in}} \\
& = \sqrt{C_{in}} \cdot \sqrt{C_{im}} \cdot C_{im}^{n_n} \cdot \sqrt{A_{im}} \cdot A_{im}^{n_m} \cdot \sqrt{A_{in}} \\
& = \sqrt{C_{in}} \cdot C_{im}^{n_n - n_m} \cdot \sqrt{A_{in}}.
\end{aligned}
$$

   In words, we get to the relevant sheet by starting from the Euclidean sheet, doing a half-clockwise monodromy around $z_{in}$, then $(n_n - n_m)$ clockwise circles around $z_{im}$ and finally followed by a half-anticlockwise monodromy around $z_{in}$, i.e., $\sqrt{C_{in}} \cdot C_{im}^{n_n - n_m} \cdot \sqrt{A_{in}}$.

13. $n_m \geq n_i \geq n_j \geq n_n$

   In this case we first move $\omega_m \to \omega_m - n_m\pi$. It crosses 3 future light cones. This move induces no monodromy. We then make the shifts $\omega_i \to \omega_i - n_i\pi$. It crosses future, past, future lightcone configuration which according to rule 2 of §6.2 gives $\sqrt{A_{in}} \cdot C_{im}^{n_i} \cdot \sqrt{C_{in}}$.

   We then move $\omega_j \to \omega_j - n_j\pi$. It crosses past, future, past lightcone configuration which as per rule (3) of §6.2 gives $\sqrt{C_{ji}} \cdot A_{jn}^{n_j} \cdot \sqrt{A_{ji}} \equiv \sqrt{C_{ij}} \cdot A_{im}^{n_j} \cdot \sqrt{A_{ij}}$.

Finally we move $\omega_n \to \omega_n - n_n \pi$. It crosses 3 past light cones and gives $\phi$. Putting it all together, the final monodromy is

$$\sqrt{A_{in}} \cdot C_{im}^{n_i} \cdot \sqrt{C_{in}} \cdot \sqrt{C_{ij}} \cdot A_{im}^{n_j} \cdot \sqrt{A_{ij}}$$
$$= \sqrt{C_{ij}} \cdot \sqrt{C_{im}} \cdot C_{im}^{n_i} \cdot \sqrt{A_{im}} \cdot A_{im}^{n_j} \cdot \sqrt{A_{ij}}$$
$$= \sqrt{C_{ij}} \cdot C_{im}^{n_i - n_j} \cdot \sqrt{A_{ij}}.$$

In words, we get to the relevant sheet by starting from the Euclidean sheet, doing a half-clockwise monodromy around $z_{ij}$, then $(n_i - n_j)$ clockwise circles around $z_{im}$ and finally followed by a half-anticlockwise monodromy around $z_{ij}$, i.e., $\sqrt{C_{ij}} \cdot C_{im}^{n_i - n_j} \cdot \sqrt{A_{ij}}$.

**Comments on [21]:** For the same configuration [21] (Lemma 23.6.1, equation 23.6.7) concludes the monodromy should be

$$z \to z \, e^{2\pi i (n_3 - n_2)} \qquad \Rightarrow \qquad C_0^{n_3 - n_2}, \tag{F.1}$$

whereas we get $\sqrt{C_{ij}} \cdot C_{im}^{n_i - n_j} \cdot \sqrt{A_{ij}}$.

Note, the $(n_3 - n_2)$ factor in (F.1) is fixed for all configurations, since $(n_1 - n_2) + (n_3 - n_4) - (n_1 - n_3) - (n_2 - n_4) = 2(n_3 - n_2)$. This is coming from from the definition of the cross-ratio. In comparison, for us, broadly the monodromy comes from the diamond number difference between the middle two operators in time ordering.

This difference is showing up simply because the ordering of the correlators in [21] is assumed to be fixed and specifically they are not time ordered (see eq 23.6.2 in [21]). Hence, there are only trivial monodromies unlike the time ordered case we considered. This comparison could be done with any of the examples presented in this paper.

14. $n_m \geq n_i \geq n_n \geq n_j$

In this case we first move $\omega_m \to \omega_m - n_m \pi$. It crosses 3 future light cones. This move induces no monodromy. We then make the shifts $\omega_i \to \omega_i - n_i \pi$. It crosses future, past, future lightcone configuration which according to rule 2 of §6.2 gives $\sqrt{A_{in}} \cdot C_{im}^{n_i} \cdot \sqrt{C_{in}}$.

We then move $\omega_n \to \omega_n - n_n \pi$. It crosses past, future, past lightcone configuration which as per rule (3) of §6.2 gives $\sqrt{C_{nm}} \cdot A_{nj}^{n_n} \cdot \sqrt{A_{nm}} \equiv \sqrt{C_{ij}} \cdot A_{im}^{n_n} \cdot \sqrt{A_{ij}}$.

Finally we move $\omega_j \to \omega_j - n_j \pi$. It crosses 3 past light cones and gives $\phi$. Putting it all together, the final monodromy is

$$\sqrt{A_{in}} \cdot C_{im}^{n_i} \cdot \sqrt{C_{in}} \cdot \sqrt{C_{ij}} \cdot A_{im}^{n_n} \cdot \sqrt{A_{ij}}$$
$$= \sqrt{C_{ij}} \cdot \sqrt{C_{im}} \cdot C_{im}^{n_i} \cdot \sqrt{A_{im}} \cdot A_{im}^{n_n} \cdot \sqrt{A_{ij}}$$
$$= \sqrt{C_{ij}} \cdot C_{im}^{n_i - n_n} \cdot \sqrt{A_{ij}}.$$

In words, we get to the relevant sheet by starting from the Euclidean sheet, doing a half-clockwise monodromy around $z_{ij}$, then $(n_i - n_n)$ clockwise circles around $z_{im}$ and finally followed by a half-anticlockwise monodromy around $z_{ij}$, i.e., $\sqrt{C_{ij}} \cdot C_{im}^{n_i - n_n} \cdot \sqrt{A_{ij}}$.

15. $n_m \geq n_j \geq n_i \geq n_n$

In this case we first move $\omega_m \to \omega_m - n_m \pi$. It crosses 3 future light cones. This move induces no monodromy. We then make the shifts $\omega_j \to \omega_j - n_j \pi$. It crosses 2 future and 1 past lightcones which according to rule 2 of §6.2 gives $C_{jm}^{n_j} \equiv C_{in}^{n_j}$.

We then move $\omega_i \to \omega_i - n_i \pi$. It crosses 1 future and 2 past lightcones which as per rule (3) of §6.2 gives $A_{in}^{n_i}$. Finally we move $\omega_n \to \omega_n - n_n \pi$. It crosses 3 past light cones and gives $\phi$.

So in this configuration we get to the relevant sheet by starting from the Euclidean sheet and doing $(n_j - n_i)$ clockwise circles around $z_{in}$, i.e., $C_{in}^{n_j - n_i}$.

16. $n_m \geq n_j \geq n_n \geq n_i$

In this case we first move $\omega_m \to \omega_m - n_m \pi$. It crosses 3 future light cones. This move induces no monodromy. We then make the shifts $\omega_j \to \omega_j - n_j \pi$. It crosses 2 future and 1 past lightcones which according to rule 2 of §6.2 gives $C_{jm}^{n_j} \equiv C_{in}^{n_j}$.

We then move $\omega_n \to \omega_n - n_n \pi$. It crosses 2 past and 1 future lightcones which as per rule (3) of §6.2 gives $A_{in}^{n_n}$. Finally we move $\omega_i \to \omega_i - n_i \pi$. It crosses 3 past light cones and gives $\phi$.

So in this configuration we get to the relevant sheet by starting from the Euclidean sheet and doing $(n_j - n_n)$ clockwise circles around $z_{in}$, i.e., $C_{in}^{n_j - n_n}$.

17. $n_m \geq n_n \geq n_i \geq n_j$

In this case we first move $\omega_m \to \omega_m - n_m \pi$. It crosses 3 future light cones. This move induces no monodromy. We then make the shifts $\omega_n \to \omega_n - n_n \pi$. It crosses 1 past and 2 future lightcones which according to rule 2 of §6.2 gives $C_{nm}^{n_n} \equiv C_{ij}^{n_n}$.

We then move $\omega_i \to \omega_i - n_i \pi$. It crosses 2 past and 1 future lightcones which as per rule (3) of §6.2 gives $A_{ij}^{n_i}$. Finally we move $\omega_j \to \omega_j - n_j \pi$. It crosses 3 past light cones and gives $\phi$.

So in this configuration we get to the relevant sheet by starting from the Euclidean sheet and doing $(n_n - n_i)$ clockwise circles around $z_{ij}$, i.e., $C_{ij}^{n_n - n_i}$.

18. $n_m \geq n_n \geq n_j \geq n_i$

In this case we first move $\omega_m \to \omega_m - n_m \pi$. It crosses 3 future light cones. This move induces no monodromy. We then make the shifts $\omega_n \to \omega_n - n_n \pi$. It crosses 1 past and 2 future lightcones which according to rule 2 of §6.2 gives $C_{nm}^{n_n} \equiv C_{ij}^{n_n}$.

We then move $\omega_j \to \omega_j - n_j \pi$. It crosses 1 future and 2 past lightcones which as per rule (3) of §6.2 gives $A_{ij}^{n_j}$. Finally we move $\omega_i \to \omega_i - n_i \pi$. It crosses 3 past light cones and gives $\phi$.

So in this configuration we get to the relevant sheet by starting from the Euclidean sheet and doing $(n_n - n_j)$ clockwise circles around $z_{ij}$, i.e., $C_{ij}^{n_n - n_j}$.

19. $n_n \geq n_i \geq n_j \geq n_m$

In this case we first move $\omega_n \to \omega_n - n_n \pi$. It crosses 3 future light cones. This move induces no monodromy. We then make the shifts $\omega_i \to \omega_i - n_i \pi$. It crosses 1 past and 2 future lightcones which according to rule 2 of §6.2 gives $C_{in}^{n_i}$.

We then move $\omega_j \to \omega_j - n_j \pi$. It crosses 2 past and 1 future lightcones which as per rule (3) of §6.2 gives $A_{jm}^{n_j} \equiv A_{in}^{n_j}$. Finally we move $\omega_m \to \omega_m - n_m \pi$. It crosses 3 past light cones and gives $\phi$.

So in this configuration we get to the relevant sheet by starting from the Euclidean sheet and doing $(n_i - n_j)$ clockwise circles around $z_{in}$, i.e., $C_{in}^{n_i - n_j}$.

20. $n_n \geq n_i \geq n_m \geq n_j$

In this case we first move $\omega_n \to \omega_n - n_n \pi$. It crosses 3 future light cones. This move induces no monodromy. We then make the shifts $\omega_i \to \omega_i - n_i \pi$. It crosses 1 past and 2 future lightcones which according to rule 2 of §6.2 gives $C_{in}^{n_i}$.

We then move $\omega_m \to \omega_m - n_m \pi$. It crosses 1 future and 2 past lightcones which as per rule (3) of §6.2 gives $A_{mj}^{n_m} \equiv A_{in}^{n_m}$. Finally we move $\omega_j \to \omega_j - n_j \pi$. It crosses 3 past light cones and gives $\phi$.

So in this configuration we get to the relevant sheet by starting from the Euclidean sheet and doing $(n_i - n_m)$ clockwise circles around $z_{in}$, i.e., $C_{in}^{n_i - n_m}$.

21. $n_n \geq n_j \geq n_i \geq n_m$

In this case we first move $\omega_n \to \omega_n - n_n \pi$. It crosses 3 future light cones. This move induces no monodromy. We then make the shifts $\omega_j \to \omega_j - n_j \pi$. It crosses future, past, future lightcone configuration which according to rule 2 of §6.2 gives $\sqrt{A_{ji}} \cdot C_{jn}^{n_j} \cdot \sqrt{C_{ji}} \equiv \sqrt{A_{ij}} \cdot C_{im}^{n_j} \cdot \sqrt{C_{ij}}$.

We then move $\omega_i \to \omega_i - n_i \pi$. It crosses past, future, past lightcone configuration which as per rule (3) of §6.2 gives $\sqrt{C_{in}} \cdot A_{im}^{n_i} \cdot \sqrt{A_{in}}$.

Finally we move $\omega_m \to \omega_m - n_m \pi$. It crosses 3 past light cones and gives $\phi$. Putting it all together, the final monodromy is

$$\sqrt{A_{ij}} \cdot C_{im}^{n_j} \cdot \sqrt{C_{ij}} \cdot \sqrt{C_{in}} \cdot A_{im}^{n_i} \cdot \sqrt{A_{in}}$$
$$= \sqrt{C_{in}} \cdot \sqrt{C_{im}} \cdot C_{im}^{n_j} \cdot \sqrt{A_{im}} \cdot A_{im}^{n_i} \cdot \sqrt{A_{in}}$$
$$= \sqrt{C_{in}} \cdot C_{im}^{n_j - n_i} \cdot \sqrt{A_{in}}.$$

In words, we get to the relevant sheet by starting from the Euclidean sheet, doing a half-clockwise monodromy around $z_{in}$, then $(n_j - n_i)$ clockwise circles around $z_{im}$ and finally followed by a half-anticlockwise monodromy around $z_{in}$, i.e., $\sqrt{C_{in}} \cdot C_{im}^{n_j - n_i} \cdot \sqrt{A_{in}}$.

22. $n_n \geq n_j \geq n_m \geq n_i$

In this case we first move $\omega_n \to \omega_n - n_n \pi$. It crosses 3 future light cones. This move induces no monodromy. We then make the shifts $\omega_j \to \omega_j - n_j \pi$. It crosses future, past, future lightcone configuration which according to rule 2 of §6.2 gives $\sqrt{A_{ji}} \cdot C_{jn}^{n_j} \cdot \sqrt{C_{ji}} \equiv \sqrt{A_{ij}} \cdot C_{im}^{n_j} \cdot \sqrt{C_{ij}}$.

We then move $\omega_m \to \omega_m - n_m \pi$. It crosses past, future, past lightcone configuration which as per rule (3) of §6.2 gives $\sqrt{C_{mj}} \cdot A_{mi}^{n_m} \cdot \sqrt{A_{mj}} \equiv \sqrt{C_{in}} \cdot A_{im}^{n_m} \cdot \sqrt{A_{in}}$.

Finally we move $\omega_i \to \omega_i - n_i \pi$. It crosses 3 past light cones and gives $\phi$. Putting it all together, the final monodromy is

$$\sqrt{A_{ij}} \cdot C_{im}^{n_j} \cdot \sqrt{C_{ij}} \cdot \sqrt{C_{in}} \cdot A_{im}^{n_m} \cdot \sqrt{A_{in}}$$
$$= \sqrt{C_{in}} \cdot \sqrt{C_{im}} \cdot C_{im}^{n_j} \cdot \sqrt{A_{im}} \cdot A_{im}^{n_m} \cdot \sqrt{A_{in}}$$
$$= \sqrt{C_{in}} \cdot C_{im}^{n_j - n_m} \cdot \sqrt{A_{in}}.$$

In words, we get to the relevant sheet by starting from the Euclidean sheet, doing a half-clockwise monodromy around $z_{in}$, then $(n_j - n_m)$ clockwise circles around $z_{im}$ and finally followed by a half-anticlockwise monodromy around $z_{in}$, i.e., $\sqrt{C_{in}} \cdot C_{im}^{n_j - n_m} \cdot \sqrt{A_{in}}$.

23. $n_n \geq n_m \geq n_i \geq n_j$

In this case we first move $\omega_n \to \omega_n - n_n \pi$. It crosses 3 future light cones. This move induces no monodromy. We then make the shifts $\omega_m \to \omega_m - n_m \pi$. It crosses 2 future and 1 past lightcones which according to rule 2 of §6.2 gives $C_{mn}^{n_m} \equiv C_{ij}^{n_m}$.

We then move $\omega_i \to \omega_i - n_i \pi$. It crosses 2 past and 1 future lightcones which as per rule (3) of §6.2 gives $A_{ij}^{n_i}$. Finally we move $\omega_j \to \omega_j - n_j \pi$. It crosses 3 past light cones and gives $\phi$.

So in this configuration we get to the relevant sheet by starting from the Euclidean sheet and doing $(n_m - n_i)$ clockwise circles around $z_{ij}$, i.e., $C_{ij}^{n_m - n_i}$.

24. $n_n \geq n_m \geq n_j \geq n_i$

   In this case we first move $\omega_n \to \omega_n - n_n \pi$. It crosses 3 future light cones. This move induces no monodromy. We then make the shifts $\omega_m \to \omega_m - n_m \pi$. It crosses 2 future and 1 past lightcones which according to rule 2 of §6.2 gives $C_{mn}^{n_m} \equiv C_{ij}^{n_m}$.

   We then move $\omega_j \to \omega_j - n_j \pi$. It crosses 1 future and 2 past lightcones which as per rule (3) of §6.2 gives $A_{ij}^{n_j}$. Finally we move $\omega_i \to \omega_i - n_i \pi$. It crosses 3 past light cones and gives $\phi$.

   So in this configuration we get to the relevant sheet by starting from the Euclidean sheet and doing $(n_m - n_j)$ clockwise circles around $z_{ij}$, i.e., $C_{ij}^{n_m - n_j}$.

## F.2 $B$ type configurations

Consider the $B$ type configurations listed in Fig. 8b. This configuration has one special operator (the one that is in the causal future of all the other three). Let us call this the $i^{th}$ operator, located at position $P_i$.

1. $n_i \geq n_j \geq n_m \geq n_n$

   In this case we first move $\omega_i \to \omega_i - n_i \pi$. It crosses 3 future light cones which according to rule (1) of §6.2 gives $\phi$. We then move $\omega_j \to \omega_j - n_j \pi$. It crosses 1 past and 2 future light cones which according to rule (2) of §6.2 gives $C_{ij}^{n_j}$.

   We then move $\omega_m \to \omega_m - n_m \pi$. It crosses 2 past and 1 future light cones which as per rule (3) gives $A_{mn}^{n_m} \equiv A_{ij}^{n_m}$. Finally we move $\omega_n \to \omega_n - n_n \pi$. It crosses 3 past light cones and hence as per rule (1) does nothing.

   So in this configuration we get to the relevant sheet by starting from the Euclidean sheet and doing $(n_j - n_m)$ clockwise circles around $z_{ij}$, i.e., $C_{ij}^{n_j - n_m}$.

2. $n_i \geq n_j \geq n_n \geq n_m$

   In this case we first move $\omega_i \to \omega_i - n_i \pi$. It crosses 3 future light cones. This move induces no monodromy. We then make the shifts $\omega_j \to \omega_j - n_j \pi$. It crosses 1 past and 2 future lightcones which according to rule 2 of §6.2 gives $C_{ij}^{n_j}$.

   We then move $\omega_n \to \omega_n - n_n \pi$. It crosses 1 future and 2 past lightcones which as per rule (3) of §6.2 gives $A_{nm}^{n_n} \equiv A_{ij}^{n_n}$. Finally we move $\omega_m \to \omega_m - n_m \pi$. It crosses 3 past light cones and gives $\phi$.

   So in this configuration we get to the relevant sheet by starting from the Euclidean sheet and doing $(n_j - n_n)$ clockwise circles around $z_{ij}$, i.e., $C_{ij}^{n_j - n_n}$.

3. $n_i \geq n_m \geq n_j \geq n_n$

   In this case we first move $\omega_i \to \omega_i - n_i \pi$. It crosses 3 future light cones. This move induces no monodromy. We then make the shifts $\omega_m \to \omega_m - n_m \pi$. It crosses future, past, future lightcone configuration which according to rule 2 of §6.2 gives $\sqrt{A_{mj}} \cdot C_{mi}^{n_m} \cdot \sqrt{C_{mj}} \equiv \sqrt{A_{in}} \cdot C_{im}^{n_m} \cdot \sqrt{C_{in}}$.

   We then move $\omega_j \to \omega_j - n_j \pi$. It crosses past, future, past lightcone configuration which as per rule (3) of §6.2 gives $\sqrt{C_{ji}} \cdot A_{jn}^{n_j} \cdot \sqrt{A_{ji}} \equiv \sqrt{C_{ij}} \cdot A_{im}^{n_j} \cdot \sqrt{A_{ij}}$.

Finally we move $\omega_n \to \omega_n - n_n \pi$. It crosses 3 past light cones and gives $\phi$. Putting it all together, the final monodromy is

$$\sqrt{A_{in}} \cdot C_{im}^{\,n_m} \cdot \sqrt{C_{in}} \cdot \sqrt{C_{ij}} \cdot A_{im}^{\,n_j} \cdot \sqrt{A_{ij}}$$
$$= \sqrt{C_{ij}} \cdot \sqrt{C_{im}} \cdot C_{im}^{\,n_m} \cdot \sqrt{A_{im}} \cdot A_{im}^{\,n_j} \cdot \sqrt{A_{ij}}$$
$$= \sqrt{C_{ij}} \cdot C_{im}^{\,n_m - n_j} \cdot \sqrt{A_{ij}}.$$

In words, we get to the relevant sheet by starting from the Euclidean sheet, doing a half-clockwise monodromy around $z_{ij}$, then $(n_m - n_j)$ clockwise circles around $z_{im}$ and finally followed by a half-anticlockwise monodromy around $z_{ij}$, i.e., $\sqrt{C_{ij}} \cdot C_{im}^{\,n_m - n_j} \cdot \sqrt{A_{ij}}$.

4. $n_i \geq n_m \geq n_n \geq n_j$

In this case we first move $\omega_i \to \omega_i - n_i \pi$. It crosses 3 future light cones. This move induces no monodromy. We then make the shifts $\omega_m \to \omega_m - n_m \pi$. It crosses future, past, future lightcone configuration which according to rule 2 of §6.2 gives $\sqrt{A_{mj}} \cdot C_{mi}^{\,n_m} \cdot \sqrt{C_{mj}} \equiv \sqrt{A_{in}} \cdot C_{im}^{\,n_m} \cdot \sqrt{C_{in}}$.

We then move $\omega_n \to \omega_n - n_n \pi$. It crosses past, future, past lightcone configuration which as per rule (3) of §6.2 gives $\sqrt{C_{nm}} \cdot A_{nj}^{\,n_n} \cdot \sqrt{A_{nm}} \equiv \sqrt{C_{ij}} \cdot A_{im}^{\,n_n} \cdot \sqrt{A_{ij}}$.

Finally we move $\omega_j \to \omega_j - n_j \pi$. It crosses 3 past light cones and gives $\phi$. Putting it all together, the final monodromy is

$$\sqrt{A_{in}} \cdot C_{im}^{\,n_m} \cdot \sqrt{C_{in}} \cdot \sqrt{C_{ij}} \cdot A_{im}^{\,n_n} \cdot \sqrt{A_{ij}}$$
$$= \sqrt{C_{ij}} \cdot \sqrt{C_{im}} \cdot C_{im}^{\,n_m} \cdot \sqrt{A_{im}} \cdot A_{im}^{\,n_n} \cdot \sqrt{A_{ij}}$$
$$= \sqrt{C_{ij}} \cdot C_{im}^{\,n_m - n_n} \cdot \sqrt{A_{ij}}.$$

In words, we get to the relevant sheet by starting from the Euclidean sheet, doing a half-clockwise monodromy around $z_{ij}$, then $(n_m - n_n)$ clockwise circles around $z_{im}$ and finally followed by a half-anticlockwise monodromy around $z_{ij}$, i.e., $\sqrt{C_{ij}} \cdot C_{im}^{\,n_m - n_n} \cdot \sqrt{A_{ij}}$.

5. $n_i \geq n_n \geq n_j \geq n_m$

In this case we first move $\omega_i \to \omega_i - n_i \pi$. It crosses 3 future light cones. This move induces no monodromy. We then make the shifts $\omega_n \to \omega_n - n_n \pi$. It crosses 2 future and 1 past lightcones which according to rule 2 of §6.2 gives $C_{in}^{\,n_n}$.

We then move $\omega_j \to \omega_j - n_j \pi$. It crosses 2 past and 1 future lightcones which as per rule (3) of §6.2 gives $A_{jm}^{\,n_j} \equiv A_{in}^{\,n_j}$. Finally we move $\omega_m \to \omega_m - n_m \pi$. It crosses 3 past light cones and gives $\phi$.

So in this configuration we get to the relevant sheet by starting from the Euclidean sheet and doing $(n_n - n_j)$ clockwise circles around $z_{in}$, i.e., $C_{in}^{\,n_n - n_j}$.

6. $n_i \geq n_n \geq n_m \geq n_j$

In this case we first move $\omega_i \to \omega_i - n_i \pi$. It crosses 3 future light cones. This move induces no monodromy. We then make the shifts $\omega_n \to \omega_n - n_n \pi$. It crosses 2 future and 1 past lightcones which according to rule 2 of §6.2 gives $C_{in}^{\,n_n}$.

We then move $\omega_m \to \omega_m - n_m \pi$. It crosses 1 future and 2 past lightcones which as per rule (3) of §6.2 gives $A_{mj}^{\,n_m} \equiv A_{in}^{\,n_m}$. Finally we move $\omega_j \to \omega_j - n_j \pi$. It crosses 3 past light cones and gives $\phi$.

So in this configuration we get to the relevant sheet by starting from the Euclidean sheet and doing $(n_n - n_m)$ clockwise circles around $z_{in}$, i.e., $C_{in}^{\,n_n - n_m}$.

7. $n_j > n_i \geq n_m \geq n_n$

   In this case we first move $\omega_j \to \omega_j - \pi$. It crosses 1 past and 2 future light cones which according to rule 2 of §6.2 gives $C_{ij}$. We then move $\omega_j \to \omega_j - (n_j - 1)\pi$. It crosses 3 future light cones. This move induces no monodromy. We then make the shifts $\omega_i \to \omega_i - n_i \pi$. It crosses 2 future and 1 past lightcones which according to rule 2 of §6.2 gives $C_{ij}^{n_i}$.

   We then move $\omega_m \to \omega_m - n_m \pi$. It crosses 2 past and 1 future lightcones which as per rule (3) of §6.2 gives $A_{mn}^{n_m} \equiv A_{ij}^{n_m}$. Finally we move $\omega_n \to \omega_n - n_n \pi$. It crosses 3 past light cones and gives $\phi$.

   So in this configuration we get to the relevant sheet by starting from the Euclidean sheet and doing $(n_i - n_m + 1)$ clockwise circles around $z_{ij}$, i.e., $C_{ij}^{n_i - n_m + 1}$.

8. $n_j > n_i \geq n_n \geq n_m$

   In this case we first move $\omega_j \to \omega_j - \pi$. It crosses 1 past and 2 future light cones which according to rule 2 of §6.2 gives $C_{ij}$. We then move $\omega_j \to \omega_j - (n_j - 1)\pi$. It crosses 3 future light cones. This move induces no monodromy. We then make the shifts $\omega_i \to \omega_i - n_i \pi$. It crosses 2 future and 1 past lightcones which according to rule 2 of §6.2 gives $C_{ij}^{n_i}$.

   We then move $\omega_n \to \omega_n - n_n \pi$. It crosses 1 future and 2 past lightcones which as per rule (3) of §6.2 gives $A_{nm}^{n_n} \equiv A_{ij}^{n_n}$. Finally we move $\omega_m \to \omega_m - n_m \pi$. It crosses 3 past light cones and gives $\phi$.

   So in this configuration we get to the relevant sheet by starting from the Euclidean sheet and doing $(n_i - n_n + 1)$ clockwise circles around $z_{ij}$, i.e., $C_{ij}^{n_i - n_n + 1}$.

9. $n_j \geq n_m > n_i \geq n_n$

   In this case we first move $\omega_j \to \omega_j - \pi$. It crosses 1 past and 2 future light cones which according to rule 2 of §6.2 gives $C_{ij}$. We then move $\omega_j \to \omega_j - (n_j - 1)\pi$. It crosses 3 future light cones. This move induces no monodromy. We then make the shifts $\omega_m \to \omega_m - \pi$. It crosses 2 past and 1 future light cones which according to rule 3 of §6.2 gives $A_{mn} \equiv A_{ij}$. We then move $\omega_m \to \omega_m - (n_m - 1)\pi$. It crosses 1 past and 2 future light cones which according to rule 2 of §6.2 gives $C_{jm}^{(n_m - 1)} \equiv C_{in}^{(n_m - 1)}$.

   We then move $\omega_i \to \omega_i - n_i \pi$. It crosses 1 future and 2 past lightcones which as per rule (3) of §6.2 gives $A_{in}^{n_i}$. Finally we move $\omega_n \to \omega_n - n_n \pi$. It crosses 3 past light cones and gives $\phi$.

   So in this configuration we get to the relevant sheet by starting from the Euclidean sheet and doing $(n_m - n_i - 1)$ clockwise circles around $z_{in}$, i.e., $C_{in}^{n_m - n_i - 1}$.

10. $n_j \geq n_m \geq n_n > n_i$

    In this case we first move $\omega_j \to \omega_j - \pi$. It crosses 1 past and 2 future light cones which according to rule 2 of §6.2 gives $C_{ij}$. We then move $\omega_j \to \omega_j - (n_j - 1)\pi$. It crosses 3 future light cones. This move induces no monodromy. We then make the shifts $\omega_m \to \omega_m - \pi$. It crosses 2 past and 1 future light cones which according to rule 3 of §6.2 gives $A_{mn} \equiv A_{ij}$. We then move $\omega_m \to \omega_m - (n_m - 1)\pi$. It crosses 1 past and 2 future light cones which according to rule 2 of §6.2 gives $C_{jm}^{(n_m - 1)} \equiv C_{in}^{(n_m - 1)}$.

    We then make the shifts $\omega_n \to \omega_n - \pi$. It crosses 3 past light cones which according to rule 1 gives $\phi$. We then move $\omega_n \to \omega_n - (n_n - 1)\pi$. It crosses 2 past and 1 future light cones which according to rule 3 of §6.2 gives $A_{in}^{(n_n - 1)}$. Finally we move $\omega_i \to \omega_i - n_i \pi$. It crosses 3 past light cones and gives $\phi$.

So in this configuration we get to the relevant sheet by starting from the Euclidean sheet and doing $(n_m - n_n)$ clockwise circles around $z_{in}$, i.e., $\boldsymbol{C_{in}^{n_m - n_n}}$.

11. $\boldsymbol{n_j \geq n_n > n_i \geq n_m}$

In this case we first move $\omega_j \to \omega_j - \pi$. It crosses 1 past and 2 future light cones which according to rule 2 of §6.2 gives $C_{ij}$. We then move $\omega_j \to \omega_j - (n_j - 1)\pi$. It crosses 3 future light cones. This move induces no monodromy. We then make the shifts $\omega_n \to \omega_n - \pi$. It crosses 1 future and 2 past lightcones which according to rule 3 of §6.2 gives $A_{nm} \equiv A_{ij}$. We then make the shifts $\omega_n \to \omega_n - (n_n - 1)\pi$. It crosses future, past, future lightcone configuration which according to rule 2 of §6.2 gives $\sqrt{A_{nm}} \cdot C_{nj}^{n_n - 1} \cdot \sqrt{C_{nm}} \equiv \sqrt{A_{ij}} \cdot C_{im}^{n_n - 1} \cdot \sqrt{C_{ij}}$.

We then move $\omega_i \to \omega_i - n_i \pi$. It crosses past, future, past lightcone configuration which as per rule (3) of §6.2 gives $\sqrt{C_{in}} \cdot A_{im}^{n_i} \cdot \sqrt{A_{in}}$.

Finally we move $\omega_m \to \omega_m - n_m \pi$. It crosses 3 past light cones and gives $\phi$. Putting it all together, the final monodromy is

$$
\begin{aligned}
C_{ij} \cdot A_{ij} \cdot &\sqrt{A_{ij}} \cdot C_{im}^{n_n - 1} \cdot \sqrt{C_{ij}} \cdot \sqrt{C_{in}} \cdot A_{im}^{n_i} \cdot \sqrt{A_{in}} \\
&= \sqrt{C_{in}} \cdot \sqrt{C_{im}} \cdot C_{im}^{n_n - 1} \cdot \sqrt{A_{im}} \cdot A_{im}^{n_i} \cdot \sqrt{A_{in}} \\
&= \sqrt{C_{in}} \cdot C_{im}^{n_n - n_i - 1} \cdot \sqrt{A_{in}} \,.
\end{aligned}
$$

In words, we get to the relevant sheet by starting from the Euclidean sheet, doing a half-clockwise monodromy around $z_{in}$, then $(n_n - n_i - 1)$ clockwise circles around $z_{im}$ and finally followed by a half-anticlockwise monodromy around $z_{in}$, i.e., $\boldsymbol{\sqrt{C_{in}} \cdot C_{im}^{n_n - n_i - 1} \cdot \sqrt{A_{in}}}$.

12. $\boldsymbol{n_j \geq n_n \geq n_m > n_i}$

In this case we first move $\omega_j \to \omega_j - \pi$. It crosses 1 past and 2 future light cones which according to rule 2 of §6.2 gives $C_{ij}$. We then move $\omega_j \to \omega_j - (n_j - 1)\pi$. It crosses 3 future light cones. This move induces no monodromy. We then make the shifts $\omega_n \to \omega_n - \pi$. It crosses 1 future and 2 past lightcones which according to rule 3 of §6.2 gives $A_{nm} \equiv A_{ij}$. We then make the shifts $\omega_n \to \omega_n - (n_n - 1)\pi$. It crosses future, past, future lightcone configuration which according to rule 2 of §6.2 gives $\sqrt{A_{nm}} \cdot C_{nj}^{n_n - 1} \cdot \sqrt{C_{nm}} \equiv \sqrt{A_{ij}} \cdot C_{im}^{n_n - 1} \cdot \sqrt{C_{ij}}$.

We then move $\omega_m \to \omega_m - \pi$. It crosses 3 past light cones which as per rule (1) gives $\phi$. We then move $\omega_m \to \omega_m - (n_m - 1)\pi$. It crosses past, future, past lightcone configuration which as per rule (3) of §6.2 gives $\sqrt{C_{mj}} \cdot A_{mi}^{n_m - 1} \cdot \sqrt{A_{mj}} \equiv \sqrt{C_{in}} \cdot A_{im}^{n_m - 1} \cdot \sqrt{A_{in}}$.

Finally we move $\omega_i \to \omega_i - n_i \pi$. It crosses 3 past light cones and gives $\phi$. Putting it all together, the final monodromy is

$$
\begin{aligned}
C_{ij} \cdot A_{ij} \cdot &\sqrt{A_{ij}} \cdot C_{im}^{n_n - 1} \cdot \sqrt{C_{ij}} \cdot \sqrt{C_{in}} \cdot A_{im}^{n_m - 1} \cdot \sqrt{A_{in}} \\
&= \sqrt{C_{in}} \cdot \sqrt{C_{im}} \cdot C_{im}^{n_n - 1} \cdot \sqrt{A_{im}} \cdot A_{im}^{n_m - 1} \cdot \sqrt{A_{in}} \\
&= \sqrt{C_{in}} \cdot C_{im}^{n_n - n_m} \cdot \sqrt{A_{in}} \,.
\end{aligned}
$$

In words, we get to the relevant sheet by starting from the Euclidean sheet, doing a half-clockwise monodromy around $z_{in}$, then $(n_n - n_m)$ clockwise circles around $z_{im}$ and finally followed by a half-anticlockwise monodromy around $z_{in}$, i.e., $\boldsymbol{\sqrt{C_{in}} \cdot C_{im}^{n_n - n_m} \cdot \sqrt{A_{in}}}$.

13. $n_m > n_i \geq n_j \geq n_n$

    In this case we first move $\omega_m \to \omega_m - \pi$. It crosses future, past, future lightcone configuration which as per rule (2) of §6.2 gives $\sqrt{A_{mj}} \cdot C_{mi} \cdot \sqrt{C_{mj}} \equiv \sqrt{A_{in}} \cdot C_{im} \cdot \sqrt{C_{in}}$. We then move $\omega_m \to \omega_m - (n_m - 1)\pi$. It crosses 3 future light cones. This move induces no monodromy.

    We then make the shifts $\omega_i \to \omega_i - n_i \pi$. It crosses future, past, future lightcone configuration which according to rule 2 of §6.2 gives $\sqrt{A_{in}} \cdot C_{im}^{n_i} \cdot \sqrt{C_{in}}$.

    We then move $\omega_j \to \omega_j - n_j \pi$. It crosses past, future, past lightcone configuration which as per rule (3) of §6.2 gives $\sqrt{C_{ji}} \cdot A_{jn}^{n_j} \cdot \sqrt{A_{ji}} \equiv \sqrt{C_{ij}} \cdot A_{im}^{n_j} \cdot \sqrt{A_{ij}}$.

    Finally we move $\omega_n \to \omega_n - n_n \pi$. It crosses 3 past light cones and gives $\phi$. Putting it all together, the final monodromy is

    $$\sqrt{A_{in}} \cdot C_{im} \cdot \sqrt{C_{in}} \cdot \sqrt{A_{in}} \cdot C_{im}^{n_i} \cdot \sqrt{C_{in}} \cdot \sqrt{C_{ij}} \cdot A_{im}^{n_j} \cdot \sqrt{A_{ij}}$$
    $$= \sqrt{C_{ij}} \cdot \sqrt{C_{im}} \cdot C_{im} \cdot C_{im}^{n_i} \cdot \sqrt{A_{im}} \cdot A_{im}^{n_j} \cdot \sqrt{A_{ij}}$$
    $$= \sqrt{C_{ij}} \cdot C_{im}^{n_i - n_j + 1} \cdot \sqrt{A_{ij}}.$$

    In words, we get to the relevant sheet by starting from the Euclidean sheet, doing a half-clockwise monodromy around $z_{ij}$, then $(n_i - n_j + 1)$ clockwise circles around $z_{im}$ and finally followed by a half-anticlockwise monodromy around $z_{ij}$, i.e., $\sqrt{C_{ij}} \cdot C_{im}^{n_i - n_j + 1} \cdot \sqrt{A_{ij}}$.

14. $n_m > n_i \geq n_n \geq n_j$

    In this case we first move $\omega_m \to \omega_m - \pi$. It crosses future, past, future lightcone configuration which as per rule (2) of §6.2 gives $\sqrt{A_{mj}} \cdot C_{mi} \cdot \sqrt{C_{mj}} \equiv \sqrt{A_{in}} \cdot C_{im} \cdot \sqrt{C_{in}}$. We then move $\omega_m \to \omega_m - (n_m - 1)\pi$. It crosses 3 future light cones. This move induces no monodromy.

    We then make the shifts $\omega_i \to \omega_i - n_i \pi$. It crosses future, past, future lightcone configuration which according to rule 2 of §6.2 gives $\sqrt{A_{in}} \cdot C_{im}^{n_i} \cdot \sqrt{C_{in}}$.

    We then move $\omega_n \to \omega_n - n_n \pi$. It crosses past, future, past lightcone configuration which as per rule (3) of §6.2 gives $\sqrt{C_{nm}} \cdot A_{nj}^{n_n} \cdot \sqrt{A_{nm}} \equiv \sqrt{C_{ij}} \cdot A_{im}^{n_n} \cdot \sqrt{A_{ij}}$.

    Finally we move $\omega_j \to \omega_j - n_j \pi$. It crosses 3 past light cones and gives $\phi$. Putting it all together, the final monodromy is

    $$\sqrt{A_{in}} \cdot C_{im} \cdot \sqrt{C_{in}} \cdot \sqrt{A_{in}} \cdot C_{im}^{n_i} \cdot \sqrt{C_{in}} \cdot \sqrt{C_{ij}} \cdot A_{im}^{n_n} \cdot \sqrt{A_{ij}}$$
    $$= \sqrt{C_{ij}} \cdot \sqrt{C_{im}} \cdot C_{im} \cdot C_{im}^{n_i} \cdot \sqrt{A_{im}} \cdot A_{im}^{n_n} \cdot \sqrt{A_{ij}}$$
    $$= \sqrt{C_{ij}} \cdot C_{im}^{n_i - n_n + 1} \cdot \sqrt{A_{ij}}.$$

    In words, we get to the relevant sheet by starting from the Euclidean sheet, doing a half-clockwise monodromy around $z_{ij}$, then $(n_i - n_n + 1)$ clockwise circles around $z_{im}$ and finally followed by a half-anticlockwise monodromy around $z_{ij}$, i.e., $\sqrt{C_{ij}} \cdot C_{im}^{n_i - n_n + 1} \cdot \sqrt{A_{ij}}$.

15. $n_m \geq n_j > n_i \geq n_n$

    In this case we first move $\omega_m \to \omega_m - \pi$. It crosses future, past, future lightcone configuration which as per rule (2) of §6.2 gives $\sqrt{A_{mj}} \cdot C_{mi} \cdot \sqrt{C_{mj}} \equiv \sqrt{A_{in}} \cdot C_{im} \cdot \sqrt{C_{in}}$. We then move $\omega_m \to \omega_m - (n_m - 1)\pi$. It crosses 3 future light cones. This move induces no monodromy.

We then move $\omega_j \to \omega_j - \pi$. It crosses past, future, past lightcone configuration which as per rule (3) of §6.2 gives $\sqrt{C_{ji}} \cdot A_{jn} \cdot \sqrt{A_{ji}} \equiv \sqrt{C_{ij}} \cdot A_{im} \cdot \sqrt{A_{ij}}$. We then make the shifts $\omega_j \to \omega_j - (n_j - 1)\pi$. It crosses 2 future and 1 past lightcones which according to rule (2) of §6.2 gives $C_{jm}^{n_j-1} \equiv C_{in}^{n_j-1}$.

We then move $\omega_i \to \omega_i - n_i \pi$. It crosses 1 future and 2 past lightcones which as per rule (3) of §6.2 gives $A_{in}^{n_i}$. Finally we move $\omega_n \to \omega_n - n_n \pi$. It crosses 3 past light cones and gives $\phi$. Putting it all together, the final monodromy is

$$\sqrt{A_{in}} \cdot C_{im} \cdot \sqrt{C_{in}} \cdot \sqrt{C_{ij}} \cdot A_{im} \cdot \sqrt{A_{ij}} \cdot C_{in}^{n_j-1} \cdot A_{in}^{n_i}$$
$$= \sqrt{C_{ij}} \cdot \sqrt{C_{im}} \cdot C_{im} \cdot \sqrt{A_{im}} \cdot A_{im} \cdot \sqrt{A_{ij}} \cdot C_{in}^{n_j-1} \cdot A_{in}^{n_i}$$
$$= C_{in}^{n_j-n_i-1} .$$

So in this configuration we get to the relevant sheet by starting from the Euclidean sheet and doing $(n_j - n_i - 1)$ clockwise circles around $z_{in}$, i.e., $C_{in}^{n_j-n_i-1}$.

16. $n_m \geq n_j \geq n_n > n_i$

In this case we first move $\omega_m \to \omega_m - \pi$. It crosses future, past, future lightcone configuration which as per rule (2) of §6.2 gives $\sqrt{A_{mj}} \cdot C_{mi} \cdot \sqrt{C_{mj}} \equiv \sqrt{A_{in}} \cdot C_{im} \cdot \sqrt{C_{in}}$. We then move $\omega_m \to \omega_m - (n_m - 1)\pi$. It crosses 3 future light cones. This move induces no monodromy.

We then move $\omega_j \to \omega_j - \pi$. It crosses past, future, past lightcone configuration which as per rule (3) of §6.2 gives $\sqrt{C_{ji}} \cdot A_{jn} \cdot \sqrt{A_{ji}} \equiv \sqrt{C_{ij}} \cdot A_{im} \cdot \sqrt{A_{ij}}$. We then make the shifts $\omega_j \to \omega_j - (n_j - 1)\pi$. It crosses 2 future and 1 past lightcones which according to rule (2) of §6.2 gives $C_{jm}^{n_j-1} \equiv C_{in}^{n_j-1}$.

We then move $\omega_n \to \omega_n - \pi$. It crosses 3 past light cones which as per rule (1) gives $\phi$. We then move $\omega_n \to \omega_n - (n_n - 1)\pi$. It crosses 2 past and 1 future lightcones which as per rule (3) of §6.2 gives $A_{in}^{n_n-1}$. Finally we move $\omega_i \to \omega_i - n_i \pi$. It crosses 3 past light cones and gives $\phi$. Putting it all together, the final monodromy is

$$\sqrt{A_{in}} \cdot C_{im} \cdot \sqrt{C_{in}} \cdot \sqrt{C_{ij}} \cdot A_{im} \cdot \sqrt{A_{ij}} \cdot C_{in}^{n_j-1} \cdot A_{in}^{n_n-1}$$
$$= \sqrt{C_{ij}} \cdot \sqrt{C_{im}} \cdot C_{im} \cdot \sqrt{A_{im}} \cdot A_{im} \cdot \sqrt{A_{ij}} \cdot C_{in}^{n_j-1} \cdot A_{in}^{n_n-1}$$
$$= C_{in}^{n_j-n_n} .$$

So in this configuration we get to the relevant sheet by starting from the Euclidean sheet and doing $(n_j - n_n)$ clockwise circles around $z_{in}$, i.e., $C_{in}^{n_j-n_n}$.

17. $n_m \geq n_n > n_i \geq n_j$

In this case we first move $\omega_m \to \omega_m - \pi$. It crosses future, past, future lightcone configuration which as per rule (2) of §6.2 gives $\sqrt{A_{mj}} \cdot C_{mi} \cdot \sqrt{C_{mj}} \equiv \sqrt{A_{in}} \cdot C_{im} \cdot \sqrt{C_{in}}$. We then move $\omega_m \to \omega_m - (n_m - 1)\pi$. It crosses 3 future light cones. This move induces no monodromy.

We then move $\omega_n \to \omega_n - \pi$. It crosses past, future, past lightcone configuration which as per rule (3) of §6.2 gives $\sqrt{C_{nm}} \cdot A_{nj} \cdot \sqrt{A_{nm}} \equiv \sqrt{C_{ij}} \cdot A_{im} \cdot \sqrt{A_{ij}}$. We then make the shifts $\omega_n \to \omega_n - (n_n - 1)\pi$. It crosses 1 past and 2 future lightcones which according to rule (2) of §6.2 gives $C_{nm}^{n_n-1} \equiv C_{ij}^{n_n-1}$.

We then move $\omega_i \to \omega_i - n_i \pi$. It crosses 2 past and 1 future lightcones which as per rule (3) of §6.2 gives $A_{ij}^{n_i}$. Finally we move $\omega_j \to \omega_j - n_j \pi$. It crosses 3 past light cones and gives $\phi$. Putting it all together, the final monodromy is

$$
\sqrt{A_{in}} \cdot C_{im} \cdot \sqrt{C_{in}} \cdot \sqrt{C_{ij}} \cdot A_{im} \cdot \sqrt{A_{ij}} \cdot C_{ij}^{n_n-1} \cdot A_{ij}^{n_i}
$$
$$
= \sqrt{C_{ij}} \cdot \sqrt{C_{im}} \cdot C_{im} \cdot \sqrt{A_{im}} \cdot A_{im} \cdot \sqrt{A_{ij}} \cdot C_{ij}^{n_n-1} \cdot A_{ij}^{n_i}
$$
$$
= C_{ij}^{n_n-n_i-1} .
$$

So in this configuration we get to the relevant sheet by starting from the Euclidean sheet and doing $(n_n - n_i - 1)$ clockwise circles around $z_{ij}$, i.e., $C_{ij}^{n_n-n_i-1}$.

18. $n_m \geq n_n \geq n_j > n_i$

In this case we first move $\omega_m \to \omega_m - \pi$. It crosses future, past, future lightcone configuration which as per rule (2) of §6.2 gives $\sqrt{A_{mj}} \cdot C_{mi} \cdot \sqrt{C_{mj}} \equiv \sqrt{A_{in}} \cdot C_{im} \cdot \sqrt{C_{in}}$. We then move $\omega_m \to \omega_m - (n_m - 1)\pi$. It crosses 3 future light cones. This move induces no monodromy.

We then move $\omega_n \to \omega_n - \pi$. It crosses past, future, past lightcone configuration which as per rule (3) of §6.2 gives $\sqrt{C_{nm}} \cdot A_{nj} \cdot \sqrt{A_{nm}} \equiv \sqrt{C_{ij}} \cdot A_{im} \cdot \sqrt{A_{ij}}$. We then make the shifts $\omega_n \to \omega_n - (n_n - 1)\pi$. It crosses 1 past and 2 future lightcones which according to rule (2) of §6.2 gives $C_{nm}^{n_n-1} \equiv C_{ij}^{n_n-1}$.

We then move $\omega_j \to \omega_j - \pi$. It crosses 3 past light cones which as per rule (1) gives $\phi$. We then move $\omega_j \to \omega_j - (n_j - 1)\pi$. It crosses 1 future and 2 past lightcones which as per rule (3) of §6.2 gives $A_{ij}^{n_j-1}$. Finally we move $\omega_i \to \omega_i - n_i \pi$. It crosses 3 past light cones and gives $\phi$. Putting it all together, the final monodromy is

$$
\sqrt{A_{in}} \cdot C_{im} \cdot \sqrt{C_{in}} \cdot \sqrt{C_{ij}} \cdot A_{im} \cdot \sqrt{A_{ij}} \cdot C_{ij}^{n_n-1} \cdot A_{ij}^{n_j-1}
$$
$$
= \sqrt{C_{ij}} \cdot \sqrt{C_{im}} \cdot C_{im} \cdot \sqrt{A_{im}} \cdot A_{im} \cdot \sqrt{A_{ij}} \cdot C_{ij}^{n_n-1} \cdot A_{ij}^{n_j-1}
$$
$$
= C_{ij}^{n_n-n_j} .
$$

So in this configuration we get to the relevant sheet by starting from the Euclidean sheet and doing $(n_n - n_j)$ clockwise circles around $z_{ij}$, i.e., $C_{ij}^{n_n-n_j}$.

19. $n_n > n_i \geq n_j \geq n_m$

In this case we first move $\omega_n \to \omega_n - \pi$. It crosses 2 future and 1 past lightcones which according to rule 2 of §6.2 gives $C_{in}$. We then move $\omega_n \to \omega_n - (n_n - 1)\pi$. It crosses 3 future light cones. This move induces no monodromy. We then make the shifts $\omega_i \to \omega_i - n_i \pi$. It crosses 1 past and 2 future lightcones which according to rule (2) of §6.2 gives $C_{in}^{n_i}$.

We then move $\omega_j \to \omega_j - n_j \pi$. It crosses 2 past and 1 future lightcones which as per rule (3) of §6.2 gives $A_{jm}^{n_j} \equiv A_{in}^{n_j}$. Finally we move $\omega_m \to \omega_m - n_m \pi$. It crosses 3 past light cones and gives $\phi$.

So in this configuration we get to the relevant sheet by starting from the Euclidean sheet and doing $(n_i - n_j + 1)$ clockwise circles around $z_{in}$, i.e., $C_{in}^{n_i-n_j+1}$.

20. $n_n > n_i \geq n_m \geq n_j$

In this case we first move $\omega_n \to \omega_n - \pi$. It crosses 2 future and 1 past lightcones which according to rule 2 of §6.2 gives $C_{in}$. We then move $\omega_n \to \omega_n - (n_n - 1)\pi$. It

crosses 3 future light cones. This move induces no monodromy. We then make the shifts $\omega_i \to \omega_i - n_i\pi$. It crosses 1 past and 2 future lightcones which according to rule (2) of §6.2 gives $C_{in}^{n_i}$.

We then move $\omega_m \to \omega_m - n_m\pi$. It crosses 1 future and 2 past lightcones which as per rule (3) of §6.2 gives $A_{mj}^{n_m} \equiv A_{in}^{n_m}$. Finally we move $\omega_j \to \omega_j - n_j\pi$. It crosses 3 past light cones and gives $\phi$.

So in this configuration we get to the relevant sheet by starting from the Euclidean sheet and doing $(n_i - n_m + 1)$ clockwise circles around $z_{in}$, i.e., $\boldsymbol{C_{in}^{n_i - n_m + 1}}$.

21. $\boldsymbol{n_n \geq n_j > n_i \geq n_m}$

In this case we first move $\omega_n \to \omega_n - \pi$. It crosses 2 future and 1 past lightcones which according to rule (2) of §6.2 gives $C_{in}$. We then move $\omega_n \to \omega_n - (n_n - 1)\pi$. It crosses 3 future light cones. This move induces no monodromy. We then make the shifts $\omega_j \to \omega_j - \pi$. It crosses 2 past and 1 future lightcones which according to rule (3) of §6.2 gives $A_{jm} \equiv A_{in}$. We then make the shifts $\omega_j \to \omega_j - (n_j - 1)\pi$. It crosses future, past, future lightcone configuration which according to rule (2) of §6.2 gives $\sqrt{A_{ji}} \cdot C_{jn}^{n_j - 1} \cdot \sqrt{C_{ji}} \equiv \sqrt{A_{ij}} \cdot C_{im}^{n_j - 1} \cdot \sqrt{C_{ij}}$.

We then move $\omega_i \to \omega_i - n_i\pi$. It crosses past, future, past lightcone configuration which as per rule (3) of §6.2 gives $\sqrt{C_{in}} \cdot A_{im}^{n_i} \cdot \sqrt{A_{in}}$.

Finally we move $\omega_m \to \omega_m - n_m\pi$. It crosses 3 past light cones and gives $\phi$. Putting it all together, the final monodromy is

$$C_{in} \cdot A_{in} \cdot \sqrt{A_{ij}} \cdot C_{im}^{n_j - 1} \cdot \sqrt{C_{ij}} \cdot \sqrt{C_{in}} \cdot A_{im}^{n_i} \cdot \sqrt{A_{in}}$$
$$= \sqrt{C_{in}} \cdot \sqrt{C_{im}} \cdot C_{im}^{n_j - 1} \cdot \sqrt{A_{im}} \cdot A_{im}^{n_i} \cdot \sqrt{A_{in}}$$
$$= \sqrt{C_{in}} \cdot C_{im}^{n_j - n_i - 1} \cdot \sqrt{A_{in}}.$$

In words, we get to the relevant sheet by starting from the Euclidean sheet, doing a half-clockwise monodromy around $z_{in}$, then $(n_j - n_i - 1)$ clockwise circles around $z_{im}$ and finally followed by a half-anticlockwise monodromy around $z_{in}$, i.e., $\boldsymbol{\sqrt{C_{in}} \cdot C_{im}^{n_j - n_i - 1} \cdot \sqrt{A_{in}}}$.

22. $\boldsymbol{n_n \geq n_j \geq n_m > n_i}$

In this case we first move $\omega_n \to \omega_n - \pi$. It crosses 2 future and 1 past lightcones which according to rule (2) of §6.2 gives $C_{in}$. We then move $\omega_n \to \omega_n - (n_n - 1)\pi$. It crosses 3 future light cones. This move induces no monodromy. We then make the shifts $\omega_j \to \omega_j - \pi$. It crosses 2 past and 1 future lightcones which according to rule (3) of §6.2 gives $A_{jm} \equiv A_{in}$. We then make the shifts $\omega_j \to \omega_j - (n_j - 1)\pi$. It crosses future, past, future lightcone configuration which according to rule (2) of §6.2 gives $\sqrt{A_{ji}} \cdot C_{jn}^{n_j - 1} \cdot \sqrt{C_{ji}} \equiv \sqrt{A_{ij}} \cdot C_{im}^{n_j - 1} \cdot \sqrt{C_{ij}}$.

We then move $\omega_m \to \omega_m - \pi$. It crosses 3 past light cones which as per rule (1) gives $\phi$. We then move $\omega_m \to \omega_m - (n_m - 1)\pi$. It crosses past, future, past lightcone configuration which as per rule (3) of §6.2 gives $\sqrt{C_{mj}} \cdot A_{mi}^{n_m - 1} \cdot \sqrt{A_{mj}} \equiv \sqrt{C_{in}} \cdot A_{im}^{n_m - 1} \cdot \sqrt{A_{in}}$.

Finally we move $\omega_i \to \omega_i - n_i\pi$. It crosses 3 past light cones and gives $\phi$. Putting it all together, the final monodromy is

$$C_{in} \cdot A_{in} \cdot \sqrt{A_{ij}} \cdot C_{im}^{n_j - 1} \cdot \sqrt{C_{ij}} \cdot \sqrt{C_{in}} \cdot A_{im}^{n_m - 1} \cdot \sqrt{A_{in}}$$
$$= \sqrt{C_{in}} \cdot \sqrt{C_{im}} \cdot C_{im}^{n_j - 1} \cdot \sqrt{A_{im}} \cdot A_{im}^{n_m - 1} \cdot \sqrt{A_{in}}$$
$$= \sqrt{C_{in}} \cdot C_{im}^{n_j - n_m} \cdot \sqrt{A_{in}}.$$

In words, we get to the relevant sheet by starting from the Euclidean sheet, doing a half-clockwise monodromy around $z_{in}$, then $(n_j - n_m)$ clockwise circles around $z_{im}$ and finally followed by a half-anticlockwise monodromy around $z_{in}$, i.e., $\sqrt{C_{in}} \cdot C_{im}^{n_j - n_m} \cdot \sqrt{A_{in}}$.

23. $n_n \geq n_m > n_i \geq n_j$

In this case we first move $\omega_n \to \omega_n - \pi$. It crosses 2 future and 1 past light cones which according to rule (2) of §6.2 gives $C_{in}$. We then move $\omega_n \to \omega_n - (n_n - 1)\pi$. It crosses 3 future light cones. This move induces no monodromy. We then make the shifts $\omega_m \to \omega_m - \pi$. It crosses 1 future and 2 past lightcones which as per rule (3) of §6.2 gives $A_{mj} \equiv A_{in}$. We then move $\omega_m \to \omega_m - (n_m - 1)\pi$. It crosses 2 future and 1 past lightcones which according to rule (2) of §6.2 gives $C_{mn}^{n_m - 1} \equiv C_{ij}^{n_m - 1}$.

We then move $\omega_i \to \omega_i - n_i \pi$. It crosses 2 past and 1 future lightcones which as per rule (3) of §6.2 gives $A_{ij}^{n_i}$. Finally we move $\omega_j \to \omega_j - n_j \pi$. It crosses 3 past light cones and gives $\phi$.

So in this configuration we get to the relevant sheet by starting from the Euclidean sheet and doing $(n_m - n_i - 1)$ clockwise circles around $z_{ij}$, i.e., $C_{ij}^{n_m - n_i - 1}$.

24. $n_n \geq n_m \geq n_j > n_i$

In this case we first move $\omega_n \to \omega_n - \pi$. It crosses 2 future and 1 past light cones which according to rule (2) of §6.2 gives $C_{in}$. We then move $\omega_n \to \omega_n - (n_n - 1)\pi$. It crosses 3 future light cones. This move induces no monodromy. We then make the shifts $\omega_m \to \omega_m - \pi$. It crosses 1 future and 2 past lightcones which as per rule (3) of §6.2 gives $A_{mj} \equiv A_{in}$. We then move $\omega_m \to \omega_m - (n_m - 1)\pi$. It crosses 2 future and 1 past lightcones which according to rule (2) of §6.2 gives $C_{mn}^{n_m - 1} \equiv C_{ij}^{n_m - 1}$.

We then move $\omega_j \to \omega_j - \pi$. It crosses 3 past light cones which as per rule (1) gives $\phi$. We then move $\omega_j \to \omega_j - (n_j - 1)\pi$. It crosses 1 future and 2 past lightcones which as per rule (3) of §6.2 gives $A_{ij}^{n_j - 1}$. Finally we move $\omega_i \to \omega_i - n_i \pi$. It crosses 3 past light cones and gives $\phi$.

So in this configuration we get to the relevant sheet by starting from the Euclidean sheet and doing $(n_m - n_j)$ clockwise circles around $z_{ij}$, i.e., $C_{ij}^{n_m - n_j}$.

### F.3 $C$ type configurations

Time reversal turns configurations of the type depicted in Fig. 8b into configurations of type depicted in Fig. 8c. For this reason, the rules for $C$ type configurations can be obtained rather simply from those listed in the previous subsection. In order to see this, we first note, from (25) and (26), that time reversal interchanges $z$ and $\bar{z}$ (including keeping track of the $i\epsilon$). Now suppose we were to start with a configuration of the type Fig. 8b and perform the moves $\omega_a \to \omega_a - n_a \pi$ (for $a = i, j, n, m$) exactly in the manner described in the previous subsubsection, and then time reverse this entire process. It follows that this time reversed process traverses the same path in $\bar{z}$ space (at fixed $z$) that the original process traversed in $z$ space (at fixed $\bar{z}$). As we have explained above, however, the principle of Euclidean single valuedness tells us that these two processes lead us to the same eventual location in cross-ratio space.

It follows, in summary, that starting with configurations of the type depicted in Fig. 8c, and then making the shift moves $\omega_a \to \omega_a + n_a \pi$, leads us to exactly the same sheet in cross-ratio space as we land up in by starting with the time reversed Fig. 8b, configuration and making the shifts $\omega_a \to \omega_a - n_a \pi$. The rules for this move can be read off from the previous subsubsection.

We can me more concrete. Let us start with a Fig. 8c configuration, name the special operator (the one in the causal past of all the others) as the $i^{th}$ operator, choose the terminology for the remaining operators so that $n_j \leq n_m \leq n_n$. Then if $n_i < n_j$, rule §1 of the previous subsubsection applies, but with all $n_a$ replaced by $-n_a$. Similarly rules §7, §9 and §10 respectively apply when $n_j \leq n_i < n_m \leq n_n$, $n_j \leq n_m \leq n_i < n_n$ and $n_j \leq n_m \leq n_n \leq n_i$ respectively (in every case the rules have to be applied with all $n_a$ replaced by $-n_a$). We are not going to give full details of all the cases but since this is the only type in which we will be moving to the past, i.e., making the shifts $\omega_i \rightarrow \omega_i + n_i \pi$, as an example we will give details of 4 cases.

1. $n_n \geq n_m \geq n_j > n_i$

   In this case we first move $\omega_n \rightarrow \omega_n + \pi$. It crosses 1 future and 2 past light cones which according to rule (3') of §6.2 gives $C_{in}$. We then move $\omega_n \rightarrow \omega_n + (n_n - 1)\pi$. It crosses 3 past light cones which as per rule (1') of §6.2 gives no monodromy, i.e. $\phi$. We then move $\omega_m \rightarrow \omega_m + \pi$. It crosses 2 future and 1 past light cones which according to rule (2') gives $A_{mj} \equiv A_{in}$. We then move $\omega_m \rightarrow \omega_m + (n_m - 1)\pi$. It crosses 1 future and 2 past light cones which as per rule (3') gives $C_{mn}^{n_m-1} \equiv C_{ij}^{n_m-1}$.

   We then move $\omega_j \rightarrow \omega_j + \pi$. It crosses 3 future light cones which according to rule (1') gives $\phi$. We then move $\omega_j \rightarrow \omega_j + (n_j - 1)\pi$. It crosses 2 future and 1 past light cones which as per rule (2') of §6.2 gives $A_{ij}^{n_j-1}$. Finally we move $\omega_i \rightarrow \omega_i + n_i \pi$. It crosses 3 future light cones and hence as per rule (1') does nothing.

   So in this configuration we start from the Euclidean sheet and do $(n_m - n_j)$ clockwise circles around $z_{ij}$, i.e., $C_{ij}^{n_m-n_j}$. As you may notice, it is the time reversal of B case 1.

2. $n_n \geq n_m > n_i \geq n_j$

   In this case we first move $\omega_n \rightarrow \omega_n + \pi$. It crosses 1 future and 2 past light cones which according to rule (3') of §6.2 gives $C_{in}$. We then move $\omega_n \rightarrow \omega_n + (n_n - 1)\pi$. It crosses 3 past light cones which as per rule (1') of §6.2 gives no monodromy, i.e. $\phi$. We then move $\omega_m \rightarrow \omega_m + \pi$. It crosses 2 future and 1 past light cones which according to rule (2') gives $A_{mj} \equiv A_{in}$. We then move $\omega_m \rightarrow \omega_m + (n_m - 1)\pi$. It crosses 1 future and 2 past light cones which as per rule (3') gives $C_{mn}^{n_m-1} \equiv C_{ij}^{n_m-1}$.

   We then move $\omega_i \rightarrow \omega_i + n_i \pi$. It crosses 1 past and 2 future light cones which according to rule (2') of §6.2 gives $A_{ij}^{n_i}$. Finally we move $\omega_j \rightarrow \omega_j + n_j \pi$. It crosses 3 future light cones and hence as per rule (1') does nothing.

   So in this configuration we start from the Euclidean sheet and do $(n_m - n_i - 1)$ clockwise circles around $z_{ij}$, i.e., $C_{ij}^{n_m-n_i-1}$. As you may notice, it is the time reversal of B case 7.

3. $n_n > n_i \geq n_m \geq n_j$

   In this case we first move $\omega_n \rightarrow \omega_n + \pi$. It crosses 1 future and 2 past light cones which according to rule (3') of §6.2 gives $C_{in}$. We then move $\omega_n \rightarrow \omega_n + (n_n - 1)\pi$. It crosses 3 past light cones which as per rule (1') of §6.2 gives no monodromy, i.e. $\phi$. We then move $\omega_i \rightarrow \omega_i + n_i \pi$ which crosses 2 past and 1 future light cones and according to rule (3') gives $C_{in}^{n_i}$.

   We then move $\omega_m \rightarrow \omega_m + n_m \pi$. It crosses 2 future and 1 past light cones which according to rule (2') of §6.2 gives $A_{mj}^{n_m} \equiv A_{in}^{n_m}$. Finally we move $\omega_j \rightarrow \omega_j + n_j \pi$. It crosses 3 future light cones and hence as per rule (1') does nothing.

   So in this configuration we start from the Euclidean sheet and do $(n_i - n_m + 1)$ clockwise circles around $z_{in}$, i.e., $C_{in}^{n_i-n_m+1}$. As you may notice, it is the time reversal of B case 9.

4. $n_i \geq n_n \geq n_m \geq n_j$

In this case we first move $\omega_i \to \omega_i + n_i \pi$. It crosses 3 past light cones which according to rule (1') of §6.2 gives $\phi$. We then move $\omega_n \to \omega_n + n_n \pi$. It crosses 2 past and 1 future light cones which according to rule (3') gives $C_{in}^{n_n}$.

We then move $\omega_m \to \omega_m + n_m \pi$. It crosses 2 future and 1 past light cones which according to rule (2') gives $A_{mj}^{n_m} \equiv A_{in}^{n_m}$. Finally we move $\omega_j \to \omega_j + n_j \pi$. It crosses 3 future light cones and hence as per rule (1') does nothing.

So in this configuration we start from the Euclidean sheet and do $(n_n - n_m)$ clockwise circles around $z_{in}$, i.e., $C_{in}^{n_n - n_m}$. As you may notice, it is the time reversal of B case 10.

### F.4 *D* type configurations

1. $n_i \geq n_j \geq n_m \geq n_n$

   Let us first assume $n_i > n_j$. In this case we first move $\omega_i \to \omega_i - n_i \pi$. It crosses 3 future light cones which as per rule (1) of §6.2 gives no monodromy, i.e. $\phi$. We then move $\omega_j \to \omega_j - n_j \pi$. It crosses 1 past and 2 future light cones which according to rule (2) of §6.2 gives $C_{ij}^{n_j}$.

   We then move $\omega_m \to \omega_m - n_m \pi$. It crosses 2 past and 1 future light cones which as per rule (3) gives $A_{mn}^{n_m} \equiv A_{ij}^{n_m}$. Finally we move $\omega_n \to \omega_n - n_n \pi$. It crosses 3 past light cones and hence as per rule (1) does nothing.

   The special case $n_i = n_j$ has to be dealt with separately. One way to do this is to make use of translational invariance to set $n_i = n_j = 0$, and $n_m \to n_m - n_j$, $n_n \to n_n - n_j$. Then one makes the moves $\omega_n \to \omega_n - (n_j - n_n)\pi$ (this has no monodromy) followed by $\omega_m \to \omega_m - (n_j - n_m)\pi$ (this gives monodromy of $(n_j - n_m)$ clockwise rotations around $z_{ij}$).

   In summary the configuration described in this item is located in cross-ratio space as follows: we start on the Euclidean sheet and make $(n_j - n_m)$ clockwise rotations around $z_{ij}$, i.e., $C_{ij}^{n_j - n_m}$.

2. $n_j \geq n_m \geq n_n \geq n_i$

   After permuting labels $(j, m, n) \leftrightarrow (n, m, j)$ we recognize this configuration as the time reversal of that of item §1. Following the discussion presented at the beginning of subsection §F.3, we conclude that this configuration is given by starting on the Euclidean sheet and making $((-n_n) - (-n_m) = n_m - n_n$[61] clockwise rotations around $z_{in}$, i.e., $C_{in}^{n_m - n_n}$.

3. $n_i \geq n_m > n_j \geq n_n$

   Let us first assume $n_i > n_m$. In this case we first move $\omega_i \to \omega_i - n_i \pi$. It crosses 3 future light cones. This move induces no monodromy. We then we move $\omega_m \to \omega_m - \pi$. It crosses 2 past and 1 future light cones which according to rule (3) of §6.2 gives $A_{mn} \equiv A_{ij}$. We then make the shifts $\omega_m \to \omega_m - (n_m - 1)\pi$. It crosses future, past, future lightcone configuration which according to rule 2 of §6.2 gives $\sqrt{A_{mj}} \cdot C_{im}^{n_m - 1} \cdot \sqrt{C_{mj}} \equiv \sqrt{A_{in}} \cdot C_{im}^{n_m - 1} \cdot \sqrt{C_{in}}$.

   We then make the moves $\omega_j \to \omega_j - n_j \pi$. It crosses past, future, past lightcone configuration which as per rule (3) of §6.2 gives $\sqrt{C_{ij}} \cdot A_{jn}^{n_j} \cdot \sqrt{A_{ij}} \equiv \sqrt{C_{ij}} \cdot A_{im}^{n_j} \cdot \sqrt{A_{ij}}$.

---

[61]Note that under time reversal, $n_a \leftrightarrow -n_a$.

Finally we move $\omega_n \to \omega_n - n_n\pi$. It crosses 3 past light cones and gives $\phi$. Although we have presented the analysis for the case $n_i > n_m$, the reader can verify (e.g. following the discussion in item §1 above) that the final result also applies if $n_i = n_m$.

$$A_{ij} \cdot \sqrt{A_{in}} \cdot C_{im}^{n_m-1} \cdot \sqrt{C_{in}} \cdot \sqrt{C_{ij}} \cdot A_{im}^{n_j} \cdot \sqrt{A_{ij}}$$
$$= A_{ij} \cdot \sqrt{C_{ij}} \cdot \sqrt{C_{im}} \cdot C_{im}^{n_m-1} \cdot \sqrt{A_{im}} \cdot A_{im}^{n_j} \cdot \sqrt{C_{im}} \cdot \sqrt{C_{in}}$$
$$= \sqrt{A_{ij}} \cdot C_{im}^{n_m-n_j-\frac{1}{2}} \cdot \sqrt{C_{in}}$$
$$= \sqrt{C_{in}} \cdot \sqrt{C_{im}} \cdot C_{im}^{n_m-n_j-\frac{1}{2}} \cdot \sqrt{C_{in}}$$
$$= \sqrt{C_{in}} \cdot C_{im}^{n_m-n_j} \cdot \sqrt{C_{in}} \,.$$

So in this configuration we start from the Euclidean sheet and first do a half-clockwise monodromy around $z_{in}$, then do $(n_m - n_j)$ clockwise circles around $z_{im}$ followed by a half-clockwise monodromy around $z_{in}$, i.e., $\sqrt{C_{in}} \cdot C_{im}^{n_m-n_j} \cdot \sqrt{C_{in}}$.

4. $n_j \geq n_n > n_m \geq n_i$

In this case we first move $\omega_j \to \omega_j - n_j\pi$. It crosses 3 future light cones which according to rule (1) of §6.2 gives $\phi$. We then move $\omega_n \to \omega_n - \pi$ which crosses 2 past and 1 future light cones and according to rule (3) gives $A_{in}$. We then move $\omega_n \to \omega_n - (n_n-1)\pi$. It crosses future, past, future light cones configuration which according to rule (2) of §6.2 gives $\sqrt{A_{nm}} \cdot C_{nj}^{n_n-1} \cdot \sqrt{C_{nm}} \equiv \sqrt{A_{ij}} \cdot C_{im}^{n_n-1} \cdot \sqrt{C_{ij}}$.

We then move $\omega_m \to \omega_m - n_m\pi$. It crosses past, future, past lightcone configuration which as per rule (3) gives $\sqrt{C_{jm}} \cdot A_{im}^{n_m} \cdot \sqrt{A_{jm}} \equiv \sqrt{C_{in}} \cdot A_{im}^{n_m} \cdot \sqrt{A_{in}}$.

Finally we move $\omega_i \to \omega_i - n_i\pi$. It crosses 3 past light cones and hence as per rule (1) does nothing.

$$A_{in} \cdot \sqrt{A_{ij}} \cdot C_{im}^{n_n-1} \cdot \sqrt{C_{ij}} \cdot \sqrt{C_{in}} \cdot A_{im}^{n_m} \cdot \sqrt{A_{in}}$$
$$= A_{in} \cdot \sqrt{C_{in}} \cdot \sqrt{C_{im}} \cdot C_{im}^{n_n-1} \cdot \sqrt{A_{im}} \cdot A_{im}^{n_m} \cdot \sqrt{C_{im}} \cdot \sqrt{C_{ij}}$$
$$= \sqrt{A_{in}} \cdot C_{im}^{n_n-n_m-\frac{1}{2}} \cdot \sqrt{C_{ij}} = \sqrt{C_{ij}} \cdot \sqrt{C_{im}} \cdot C_{im}^{n_n-n_m-\frac{1}{2}} \cdot \sqrt{C_{ij}}$$
$$= \sqrt{C_{ij}} \cdot C_{im}^{n_n-n_m} \cdot \sqrt{C_{ij}} \,.$$

So in this configuration we start from the Euclidean sheet and first do a half-clockwise monodromy around $z_{ij}$, then do $(n_n - n_m)$ clockwise circles around $z_{im}$ followed by a half-clockwise monodromy around $z_{ij}$, i.e., $\sqrt{C_{ij}} \cdot C_{im}^{n_n-n_m} \cdot \sqrt{C_{ij}}$.

Also notice that after permuting labels $(j, m, n) \leftrightarrow (n, m, j)$ we recognize this configuration as the time reversal of that of item §3. Following the discussion presented at the beginning of subsection §F.3, we conclude that this configuration is given by starting on the Euclidean sheet then make one half-clockwise monodromy around $z_{ij}$, then do $(n_n - n_m)$ clockwise circles around $z_{im}$ followed by a half-clockwise monodromy around $z_{ij}$, i.e., $\sqrt{C_{ij}} \cdot C_{im}^{n_n-n_m} \cdot \sqrt{C_{ij}}$.

5. $n_i \geq n_j \geq n_n > n_m$

Let us first assume $n_i > n_j$. In this case we first move $\omega_i \to \omega_i - n_i\pi$. It crosses 3 future light cones which as per rule (1) of §6.2 gives no monodromy, i.e. $\phi$. We then move $\omega_j \to \omega_j - n_j\pi$. It crosses 1 past and 2 future light cones which according to rule (2) of §6.2 gives $C_{ij}^{n_j}$.

We then move $\omega_n \to \omega_n - \pi$. It crosses 3 past light cones which as per rule (1) gives $\phi$. We then move $\omega_n \to \omega_n - (n_n - 1)\pi$. It crosses 1 future and 2 past light cones which as per rule (3) gives $A_{mn}^{n_n-1} \equiv A_{ij}^{n_n-1}$. Finally we move $\omega_m \to \omega_m - n_m\pi$. It crosses 3 past light cones and hence as per rule (1) does nothing.

In summary the configuration described in this item is located in cross-ratio space as follows: we start on the Euclidean sheet and make $(n_j - n_n + 1)$ clockwise rotations around $z_{ij}$, i.e., $C_{ij}^{n_j-n_n+1}$. The reader can verify that the result obtained above applies even when $n_i = n_j$.

6. $n_m > n_j \geq n_n \geq n_i$

   Once again, the relabeling $(j, m, n) \leftrightarrow (n, m, j)$ turns this case to the time reversal of item §5. It follows that the net result in this case is to start from the Euclidean sheet and make $(-n_n) - (-n_j) + 1 = (n_j - n_n + 1)$ clockwise circles around $z_{in}$, i.e., $C_{in}^{n_j-n_n+1}$.

7. $n_i \geq n_m \geq n_n > n_j$

   In this case we first move $\omega_i \to \omega_i - n_i\pi$. It crosses 3 future light cones. This move induces no monodromy. We then we move $\omega_m \to \omega_m - \pi$. It crosses 2 past and 1 future light cones which according to rule (3) of §6.2 gives $A_{mn} \equiv A_{ij}$. We then make the shifts $\omega_m \to \omega_m - (n_m - 1)\pi$. It crosses future, past, future lightcone configuration which according to rule 2 of §6.2 gives $\sqrt{A_{mj}} \cdot C_{im}^{n_m-1} \cdot \sqrt{C_{mj}} \equiv \sqrt{A_{in}} \cdot C_{im}^{n_m-1} \cdot \sqrt{C_{in}}$.

   We then make the moves $\omega_n \to \omega_n - \pi$. It crosses 3 past light cones which as per rule (1) of §6.2 gives $\phi$. Then we move $\omega_n \to \omega_n - (n_n - 1)\pi$. It crosses past, future, past lightcone configuration which as per rule (3) of §6.2 gives $\sqrt{C_{nm}} \cdot A_{nj}^{n_n-1} \cdot \sqrt{A_{nm}} \equiv \sqrt{C_{ij}} \cdot A_{im}^{n_n-1} \cdot \sqrt{A_{ij}}$.

   Finally we move $\omega_j \to \omega_j - n_j\pi$. It crosses 3 past light cones and gives $\phi$.

   $$
   \begin{aligned}
   & A_{ij} \cdot \sqrt{A_{in}} \cdot C_{im}^{n_m-1} \cdot \sqrt{C_{in}} \cdot \sqrt{C_{ij}} \cdot A_{im}^{n_n-1} \cdot \sqrt{A_{ij}} \\
   &= A_{ij} \cdot \sqrt{C_{ij}} \cdot \sqrt{C_{im}} \cdot C_{im}^{n_m-1} \cdot \sqrt{A_{im}} \cdot A_{im}^{n_n-1} \cdot \sqrt{C_{im}} \cdot \sqrt{C_{in}} \\
   &= \sqrt{A_{ij}} \cdot C_{im}^{n_m-n_n+\frac{1}{2}} \cdot \sqrt{C_{in}} \\
   &= \sqrt{C_{in}} \cdot \sqrt{C_{im}} \cdot C_{im}^{n_m-n_n+\frac{1}{2}} \cdot \sqrt{C_{in}} \\
   &= \sqrt{C_{in}} \cdot C_{im}^{n_m-n_n+1} \cdot \sqrt{C_{in}}.
   \end{aligned}
   $$

   So in this configuration we start from the Euclidean sheet and first do a half-clockwise monodromy around $z_{in}$, then do $(n_m - n_j + 1)$ clockwise circles around $z_{im}$ followed by a half-clockwise monodromy around $z_{in}$, i.e., $\sqrt{C_{in}} \cdot C_{im}^{n_m-n_n+1} \cdot \sqrt{C_{in}}$.

8. $n_n > n_j \geq n_m \geq n_i$

   Once again, the relabeling $(j, m, n) \leftrightarrow (n, m, j)$ turns this case to the time reversal of item §7. It follows that the net result in this case is to start from the Euclidean sheet and first do a half-clockwise monodromy around $z_{ij}$, then do $(-n_m) - (-n_j) + 1 = (n_j - n_m + 1)$ clockwise circles around $z_{im}$ followed by a half-clockwise monodromy around $z_{in}$., i.e., $\sqrt{C_{ij}} \cdot C_{im}^{n_j-n_m+1} \cdot \sqrt{C_{ij}}$.

9. $n_i \geq n_n > n_j \geq n_m$

   In this case we first move $\omega_i \to \omega_i - n_i\pi$. It crosses 3 future light cones which as per rule (1) of §6.2 gives no monodromy, i.e. $\phi$. We then move $\omega_n \to \omega_n - \pi$. It crosses 3 past light cones which according to rule (1) gives $\phi$. We then move $\omega_n \to \omega_n - (n_n - 1)\pi$. It crosses 2 future and 1 past light cones which as per rule (2) gives $C_{in}^{n_n-1}$.

We then move $\omega_j \to \omega_j - n_j \pi$. It crosses 2 past and 1 future light cones which as per rule (3) gives $A_{jm}^{n_j} \equiv A_{in}^{n_j}$. Finally we move $\omega_m \to \omega_m - n_m \pi$. It crosses 3 past light cones and hence as per rule (1) does nothing.

In summary the configuration described in this item is located in cross-ratio space as follows: we start on the Euclidean sheet and make $(n_n - n_j - 1)$ clockwise rotations around $z_{in}$, i.e., $C_{in}^{n_n - n_j - 1}$.

10. $n_m \geq n_n > n_j \geq n_i$

Once again, the relabeling $(j, m, n) \leftrightarrow (n, m, j)$ turns this case to the time reversal of item §9. The final result in this case is to start from the Euclidean sheet and make $((-n_j) - (-n_n) - 1) = (n_n - n_j - 1)$ clockwise monodromies around $z_{ij}$, i.e., $C_{ij}^{n_n - n_j - 1}$.

11. $n_i \geq n_n > n_m > n_j$

In this case we first move $\omega_i \to \omega_i - n_i \pi$. It crosses 3 future light cones which as per rule (1) of §6.2 gives no monodromy, i.e. $\phi$. We then move $\omega_n \to \omega_n - \pi$. It crosses 3 past light cones which according to rule (1) gives $\phi$. We then move $\omega_n \to \omega_n - (n_n - 1)\pi$. It crosses 2 future and 1 past light cones which as per rule (2) gives $C_{in}^{n_n - 1}$.

We then move $\omega_m \to \omega_m - \pi$. It crosses 3 past light cones which as per rule (1) gives $\phi$. We then move $\omega_m \to \omega_m - (n_m - 1)\pi$. It crosses 1 future and 2 past light cones which as per rule (3) of §6.2 gives $A_{jm}^{n_m - 1} \equiv A_{in}^{n_m - 1}$. Finally we move $\omega_j \to \omega_j - n_j \pi$. It crosses 3 past light cones and hence as per rule (1) does nothing.

In summary the configuration described in this item is located in cross-ratio space as follows: we start on the Euclidean sheet and make $(n_n - n_m)$ clockwise rotations around $z_{in}$, i.e., $C_{in}^{n_n - n_m}$.

12. $n_n > n_m > n_j \geq n_i$

Once again, the relabeling $(j, m, n) \leftrightarrow (n, m, j)$ turns this case to the time reversal of item §11. The final result in this case is to start from the Euclidean sheet and make $(-n_j) - (-n_m) = (n_m - n_j)$ clockwise monodromies around $z_{ij}$, i.e., $C_{ij}^{n_m - n_j}$.

As a check, let's see this case explicitly:-

We first move $\omega_n \to \omega_n - \pi$. It crosses 2 past and 1 future light cones which according to rule (3) of §6.2 gives $A_{in}$. We then move $\omega_n \to \omega_n - (n_n - 1)\pi$ which crosses 3 future light cones and according to rule (1) of §6.2 gives no monodromy. We then move $\omega_m \to \omega_m - \pi$. It crosses past, future, past lightcone configuration which as per rule (3) gives $\sqrt{C_{mj}} \cdot A_{mi} \cdot \sqrt{A_{mj}} \equiv \sqrt{C_{in}} \cdot A_{im} \cdot \sqrt{A_{in}}$.

We then move $\omega_m \to \omega_m - (n_m - 1)\pi$. It crosses 2 future and 1 past light cones which according to rule (2) gives $C_{mn}^{n_m - 1} \equiv C_{ij}^{n_m - 1}$. We now move $\omega_j \to \omega_j - n_j \pi$. It crosses 1 future and 2 past light cones which as per rule (3) gives $A_{ij}^{n_j}$. Finally we move $\omega_i \to \omega_i - n_i \pi$. It crosses 3 past light cones and hence as per rule (1) does nothing.

$$
\begin{aligned}
A_{in} \cdot \sqrt{C_{in}} \cdot A_{im} &\cdot \sqrt{A_{in}} \cdot C_{ij}^{n_m - 1} \cdot A_{ij}^{n_j} \\
&= \sqrt{A_{in}} \cdot A_{im} \cdot \sqrt{C_{im}} \cdot \sqrt{C_{ij}} \cdot C_{ij}^{n_m - n_j - 1} \\
&= \sqrt{A_{in}} \cdot \sqrt{A_{im}} \cdot \sqrt{C_{ij}} \cdot C_{ij}^{n_m - n_j - 1} \\
&= \sqrt{C_{ij}} \cdot C_{ij}^{n_m - n_j - \frac{1}{2}} \\
&= C_{ij}^{n_m - n_j} .
\end{aligned}
$$

So in this configuration we start from the Euclidean sheet and do $(n_m - n_j)$ clockwise circles around $z_{ij}$, i.e., $C_{ij}^{n_m - n_j}$.

13. $n_j \geq n_i \geq n_m \geq n_n$

In this case we first move $\omega_j \to \omega_j - n_j\pi$. It crosses 3 future light cones which as per rule (1) of §6.2 gives no monodromy, i.e. $\phi$. We then move $\omega_i \to \omega_i - n_i\pi$. It crosses 2 future and 1 past light cones which as per rule (2) gives $C_{ij}^{n_i}$.

We then move $\omega_m \to \omega_m - n_m\pi$. It crosses 2 past and 1 future light cones which as per rule (3) of §6.2 gives $A_{mn}^{n_m} \equiv A_{ij}^{n_m}$. Finally we move $\omega_n \to \omega_n - n_n\pi$. It crosses 3 past light cones and hence as per rule (1) does nothing.

In summary the configuration described in this item is located in cross-ratio space as follows: we start on the Euclidean sheet and make $(n_i - n_m)$ clockwise rotations around $z_{ij}$, i.e., $C_{ij}^{n_i - n_m}$.

14. $n_j \geq n_m \geq n_i \geq n_n$

Once again, the relabeling $(j, m, n) \leftrightarrow (n, m, j)$ turns this case to the time reversal of item §13. The final result in this case is to start from the Euclidean sheet and make $(-n_i) - (-n_m) = (n_m - n_i)$ clockwise monodromies around $z_{in}$, i.e., $C_{in}^{n_m - n_i}$.

15. $n_j \geq n_i \geq n_n > n_m$

In this case we first move $\omega_j \to \omega_j - n_j\pi$. It crosses 3 future light cones which as per rule (1) of §6.2 gives no monodromy, i.e. $\phi$. We then move $\omega_i \to \omega_i - n_i\pi$. It crosses 2 future and 1 past light cones which as per rule (2) gives $C_{ij}^{n_i}$.

We then move $\omega_n \to \omega_n - \pi$. It crosses 3 past light cones which as per rule (1) gives $\phi$. We then move $\omega_n \to \omega_n - (n_n - 1)\pi$. It crosses 1 future and 2 past light cones which as per rule (3) of §6.2 gives $A_{mn}^{n_n - 1} \equiv A_{ij}^{n_n - 1}$. Finally we move $\omega_m \to \omega_m - n_m\pi$. It crosses 3 past light cones and hence as per rule (1) does nothing.

In summary the configuration described in this item is located in cross-ratio space as follows: we start on the Euclidean sheet and make $(n_i - n_n + 1)$ clockwise rotations around $z_{ij}$, i.e., $C_{ij}^{n_i - n_n + 1}$.

16. $n_m > n_j \geq n_i \geq n_n$

Once again, the relabeling $(j, m, n) \leftrightarrow (n, m, j)$ turns this case to the time reversal of item §15. The final result in this case is to start from the Euclidean sheet and make $(-n_i - (-n_j)) + 1 = (n_j - n_i + 1)$ clockwise monodromies around $z_{in}$, i.e., $C_{in}^{n_j - n_i + 1}$.

17. $n_m \geq n_i \geq n_j \geq n_n$, $\quad n_m > n_j$

In this case we first move $\omega_m \to \omega_m - \pi$. It crosses 1 past and 2 future light cones which as per rule (2) of §6.2 gives $C_{jm} \equiv C_{in}$. Then we move $\omega_m \to \omega_m - (n_m - 1)\pi$. It crosses 3 future light cones. This move induces no monodromy. We then make the shifts $\omega_i \to \omega_i - n_i\pi$. It crosses future, past, future lightcone configuration which according to rule 2 of §6.2 gives $\sqrt{A_{in}} \cdot C_{im}^{n_i} \cdot \sqrt{C_{in}}$.

We then make the moves $\omega_j \to \omega_j - n_j\pi$. It crosses past, future, past lightcone configuration which as per rule (3) of §6.2 gives $\sqrt{C_{ij}} \cdot A_{jn}^{n_j} \cdot \sqrt{A_{ij}} \equiv \sqrt{C_{ij}} \cdot A_{im}^{n_j} \cdot \sqrt{A_{ij}}$.

Finally we move $\omega_n \to \omega_n - n_n\pi$. It crosses 3 past light cones and gives $\phi$.

$$C_{in} \cdot \sqrt{A_{in}} \cdot C_{im}^{n_i} \cdot \sqrt{C_{in}} \cdot \sqrt{C_{ij}} \cdot A_{im}^{n_j} \cdot \sqrt{A_{ij}}$$
$$= \sqrt{C_{in}} \cdot C_{im}^{n_i} \cdot \sqrt{A_{im}} \cdot A_{im}^{n_j} \cdot \sqrt{C_{im}} \cdot \sqrt{C_{in}}$$
$$= \sqrt{C_{in}} \cdot C_{im}^{n_i - n_j} \cdot \sqrt{C_{in}}.$$

So in this configuration we start from the Euclidean sheet and first do a half-clockwise monodromy around $z_{in}$, then do $(n_i - n_j)$ clockwise circles around $z_{im}$ followed by a half-clockwise monodromy around $z_{in}$, i.e., $\sqrt{C_{in}} \cdot C_{im}^{n_i-n_j} \cdot \sqrt{C_{in}}$.

18. $n_j \geq n_n \geq n_i \geq n_m$, $\quad n_n > n_m$

Once again, the relabeling $(j, m, n) \leftrightarrow (n, m, j)$ turns this case to the time reversal of item §17. The final result in this case is to start from the Euclidean sheet and first do a half-clockwise monodromy around $z_{ij}$ followed by $(-n_i - (-n_n)) = (n_n - n_i)$ clockwise circles around $z_{im}$ followed by a half-clockwise monodromy around $z_{ij}$, i.e., $\sqrt{C_{ij}} \cdot C_{im}^{n_n-n_i} \cdot \sqrt{C_{ij}}$.

19. $n_m \geq n_i \geq n_n > n_j$

In this case we first move $\omega_m \to \omega_m - \pi$. It crosses 1 past and 2 future light cones which as per rule (2) of §6.2 gives $C_{jm} \equiv C_{in}$. Then we move $\omega_m \to \omega_m - (n_m - 1)\pi$. It crosses 3 future light cones. This move induces no monodromy. We then make the shifts $\omega_i \to \omega_i - n_i\pi$. It crosses future, past, future lightcone configuration which according to rule 2 of §6.2 gives $\sqrt{A_{in}} \cdot C_{im}^{n_i} \cdot \sqrt{C_{in}}$.

Now we move $\omega_n \to \omega_n - \pi$. It crosses 3 past light cones which as per rule (1) of §6.2 gives $\phi$. We then make the moves $\omega_n \to \omega_n - (n_n - 1)\pi$. It crosses past, future, past lightcone configuration which as per rule (3) of §6.2 gives $\sqrt{C_{nm}} \cdot A_{nj}^{n_n-1} \cdot \sqrt{A_{nm}} \equiv \sqrt{C_{ij}} \cdot A_{im}^{n_n-1} \cdot \sqrt{A_{ij}}$.

Finally we move $\omega_j \to \omega_j - n_j\pi$. It crosses 3 past light cones and gives $\phi$.

$$
\begin{aligned}
&C_{in} \cdot \sqrt{A_{in}} \cdot C_{im}^{n_i} \cdot \sqrt{C_{in}} \cdot \sqrt{C_{ij}} \cdot A_{im}^{n_n-1} \cdot \sqrt{A_{ij}} \\
&= \sqrt{C_{in}} \cdot C_{im}^{n_i} \cdot \sqrt{A_{im}} \cdot A_{im}^{n_n-1} \cdot \sqrt{C_{im}} \cdot \sqrt{C_{in}} \\
&= \sqrt{C_{in}} \cdot C_{im}^{n_i-n_n+1} \cdot \sqrt{C_{in}}.
\end{aligned}
$$

So in this configuration we start from the Euclidean sheet and first do a half-clockwise monodromy around $z_{in}$, then do $(n_i - n_n + 1)$ clockwise circles around $z_{im}$ followed by a half-clockwise monodromy around $z_{in}$, i.e., $\sqrt{C_{in}} \cdot C_{im}^{n_i-n_n+1} \cdot \sqrt{C_{in}}$.

20. $n_n > n_j \geq n_i \geq n_m$

Once again, the relabeling $(j, m, n) \leftrightarrow (n, m, j)$ turns this case to the time reversal of item §19. The final result in this case is to start from the Euclidean sheet and first make a half-clockwise monodromy around $z_{ij}$ followed by $(-n_i - (-n_j)) + 1 = (n_j - n_i + 1)$ clockwise circles around $z_{im}$ followed by a half-clockwise monodromy around $z_{ij}$, i.e., $\sqrt{C_{ij}} \cdot C_{im}^{n_j-n_i+1} \cdot \sqrt{C_{ij}}$.

21. $n_n \geq n_i \geq n_j \geq n_m$, $\quad n_n > n_j$

In this case we first move $\omega_n \to \omega_n - \pi$. It crosses 2 past and 1 future light cones which as per rule (3) of §6.2 gives $A_{in}$. We then move $\omega_n \to \omega_n - (n_n - 1)\pi$. It crosses 3 future light cones which as per rule (1) gives no monodromy, i.e. $\phi$. We then move $\omega_i \to \omega_i - n_i\pi$. It crosses 1 past and 2 future light cones which as per rule (2) gives $C_{in}^{n_i}$. We then move $\omega_j \to \omega_j - n_j\pi$. It crosses 2 past and 1 future light cones which as per rule (3) of §6.2 gives $A_{jm}^{n_j} \equiv A_{in}^{n_j}$. Finally we move $\omega_m \to \omega_m - n_m\pi$. It crosses 3 past light cones and hence as per rule (1) does nothing.

In summary the configuration described in this item is located in cross-ratio space as follows: we start on the Euclidean sheet and make $(n_i - n_j - 1)$ clockwise rotations around $z_{in}$, i.e., $C_{in}^{n_i-n_j-1}$.

22. $n_m \geq n_n \geq n_i \geq n_j, \quad n_n > n_j$

   Once again, the relabeling $(j, m, n) \leftrightarrow (n, m, j)$ turns this case to the time reversal of item §21. The final result in this case is to start from the Euclidean sheet and make $(-n_i - (-n_n)) - 1 = (n_n - n_i - 1)$ clockwise circles around $z_{ij}$, i.e., $C_{ij}^{n_n - n_i - 1}$.

23. $n_n \geq n_i \geq n_m > n_j, \quad n_n > n_m$

   In this case we first move $\omega_n \to \omega_n - \pi$. It crosses 2 past and 1 future light cones which as per rule (3) of §6.2 gives $A_{in}$. We then move $\omega_n \to \omega_n - (n_n - 1)\pi$. It crosses 3 future light cones which as per rule (1) gives no monodromy, i.e. $\phi$. We then move $\omega_i \to \omega_i - n_i \pi$. It crosses 1 past and 2 future light cones which as per rule (2) gives $C_{in}^{n_i}$. We then move $\omega_m \to \omega_m - \pi$. It crosses 3 past light cones which as per rule (1) gives $\phi$. We then move $\omega_m \to \omega_m - (n_m - 1)\pi$. It crosses 1 future and 2 past light cones which as per rule (3) of §6.2 gives $A_{jm}^{n_m - 1} \equiv A_{in}^{n_m - 1}$. Finally we move $\omega_j \to \omega_j - n_j \pi$. It crosses 3 past light cones and hence as per rule (1) does nothing.

   In summary the configuration described in this item is located in cross-ratio space as follows: we start on the Euclidean sheet and make $(n_i - n_m)$ clockwise rotations around $z_{in}$, i.e., $C_{in}^{n_i - n_m}$.

24. $n_n > n_m \geq n_i \geq n_j, \quad n_m > n_j$

   Once again, the relabeling $(j, m, n) \leftrightarrow (n, m, j)$ turns this case to the time reversal of item §23. The final result in this case is to start from the Euclidean sheet and make $(-n_i - (-n_m)) = (n_m - n_i)$ clockwise circles around $z_{ij}$, i.e., $C_{ij}^{n_m - n_i}$.

   As a check, let's see this case explicitly:-

   We first move $\omega_n \to \omega_n - \pi$. It crosses 2 past and 1 future light cones which according to rule (3) of §6.2 gives $A_{in}$. We then move $\omega_n \to \omega_n - (n_n - 1)\pi$ which crosses 3 future light cones and according to rule (1) of §6.2 gives no monodromy. We then move $\omega_m \to \omega_m - \pi$. It crosses past, future, past lightcone configuration which as per rule (3) gives $\sqrt{C_{mj}} \cdot A_{mi} \cdot \sqrt{A_{mj}} \equiv \sqrt{C_{in}} \cdot A_{im} \cdot \sqrt{A_{in}}$.

   We then move $\omega_m \to \omega_m - (n_m - 1)\pi$. It crosses 2 future and 1 past light cones which according to rule (2) gives $C_{mn}^{n_m - 1} \equiv C_{ij}^{n_m - 1}$. We now move $\omega_i \to \omega_i - n_i \pi$. It crosses 2 past and 1 future light cones which as per rule (3) gives $A_{ij}^{n_i}$. Finally we move $\omega_j \to \omega_j - n_j \pi$. It crosses 3 past light cones and hence as per rule (1) does nothing.

$$A_{in} \cdot \sqrt{C_{in}} \cdot A_{im} \cdot \sqrt{A_{in}} \cdot C_{ij}^{n_m - 1} \cdot A_{ij}^{n_i}$$
$$= \sqrt{A_{in}} \cdot A_{im} \cdot \sqrt{C_{im}} \cdot \sqrt{C_{ij}} \cdot C_{ij}^{n_m - n_i - 1}$$
$$= \sqrt{A_{in}} \cdot \sqrt{A_{im}} \cdot \sqrt{C_{ij}} \cdot C_{ij}^{n_m - n_i - 1}$$
$$= \sqrt{C_{ij}} \cdot C_{ij}^{n_m - n_i - \frac{1}{2}}$$
$$= C_{ij}^{n_m - n_i}.$$

   So in this configuration we start from the Euclidean sheet and do $(n_m - n_i)$ clockwise circles around $z_{ij}$, i.e., $C_{ij}^{n_m - n_i}$.

## F.5 *E* type configurations

*E* type configurations are extremely similar to *D* type configurations (in fact they are related to the latter by a parity shift). The analysis of the previous subsection carries over without any change to these configurations - all final results for *E* type configurations are identical to those for the *D* type configurations discussed in the previous sub-subsection.

### F.6  *F* type configurations

In this subsubsection we study the configurations that are obtained by the moves $\omega_i \to \omega_i - n_i\pi$, starting with configurations of the form depicted in Fig. 8f.

We also need to comment on the time reversal. Interchanging indices $(i \leftrightarrow n)$ and $(j \leftrightarrow m)$ along with $n_a \to -n_a$ gives us the time reversal cases of this configuration.

Consider the top and bottom points $(i, n)$ to be red balls (R) and the middle two points $(j, m)$ to be blue balls (B). Whenever a case has same coloured balls in the extremities like RBBR or BRRB type structure, then that case will be its own time reversal. Example: $(n_i \geq n_j \geq n_m \geq n_n)$, $(n_j > n_i \geq n_n > n_m)$ and $(n_n > n_j \geq n_m > n_i)$ map to themselves under time reversal.

But with such an analogy with subsection F.7 we should not think that there is a $\mathbb{Z}_2$ symmetry, i.e., $(i, j) \leftrightarrow (n, m)$ is not a symmetry because they are timelike separated. It is not a symmetry as can be seen in case 3 and case 22.

1. $n_i \geq n_j \geq n_m \geq n_n$

   The final monodromies associated with this case are precisely those of Euclidean - A case 1. The answer is $C_{ij}^{n_j-n_m}$.

2. $n_i \geq n_j \geq n_n > n_m$

   In this case we first move $\omega_i \to \omega_i - n_i\pi$. It crosses 3 future light cones which as per rule (1) of §6.2 gives no monodromy, i.e. $\phi$. We then move $\omega_j \to \omega_j - n_j\pi$. It crosses 1 past and 2 future light cones which as per rule (2) gives $C_{ij}^{n_j}$.

   We then move $\omega_n \to \omega_n - \pi$. It crosses 3 past light cones which as per rule (1) gives $\phi$. We then move $\omega_n \to \omega_n - (n_n - 1)\pi$. It crosses 1 future and 2 past light cones which as per rule (3) of §6.2 gives $A_{mn}^{n_n-1} \equiv A_{ij}^{n_n-1}$. Finally we move $\omega_m \to \omega_m - n_m\pi$. It crosses 3 past light cones and hence as per rule (1) does nothing.

   In summary the configuration described is located in cross-ratio space as follows: we start on the Euclidean sheet and make $(n_j - n_n + 1)$ clockwise rotations around $z_{ij}$, i.e., $C_{ij}^{n_j-n_n+1}$.

3. $n_i \geq n_m > n_j \geq n_n$

   In this case we first move $\omega_i \to \omega_i - n_i\pi$. It crosses 3 future light cones which as per rule (1) of §6.2 gives $\phi$. Then we move $\omega_m \to \omega_m - \pi$. It crosses 2 past and 1 future light cones which as per rule (3) gives $A_{mn} \equiv A_{ij}$. Then we move $\omega_m \to \omega_m - (n_m - 1)\pi$. It crosses future, past, future lightcone configuration which according to rule (2) of §6.2 gives $\sqrt{A_{jm}} \cdot C_{im}^{n_m-1} \cdot \sqrt{C_{jm}} \equiv \sqrt{A_{in}} \cdot C_{im}^{n_m-1} \cdot \sqrt{C_{in}}$.

   Now we move $\omega_j \to \omega_j - n_j\pi$. It crosses past, future, past lightcone configuration which as per rule (3) of §6.2 gives $\sqrt{C_{ij}} \cdot A_{jn}^{n_j} \cdot \sqrt{A_{ij}} \equiv \sqrt{C_{ij}} \cdot A_{im}^{n_j} \cdot \sqrt{A_{ij}}$.

   Finally we move $\omega_n \to \omega_n - n_n\pi$. It crosses 3 past light cones and as per rule (1) gives $\phi$.

   $$
   \begin{aligned}
   A_{ij} &\cdot \sqrt{A_{in}} \cdot C_{im}^{n_m-1} \cdot \sqrt{C_{in}} \cdot \sqrt{C_{ij}} \cdot A_{im}^{n_j} \cdot \sqrt{A_{ij}} \\
   &= A_{ij} \cdot \sqrt{C_{ij}} \cdot \sqrt{C_{im}} \cdot C_{im}^{n_m-1} \cdot \sqrt{A_{im}} \cdot A_{im}^{n_j} \cdot \sqrt{A_{ij}} \\
   &= \sqrt{A_{ij}} \cdot C_{im}^{n_m-n_j-1} \cdot \sqrt{A_{ij}} \\
   &= \sqrt{C_{in}} \cdot \sqrt{C_{im}} \cdot C_{im}^{n_m-n_j-1} \cdot \sqrt{C_{im}} \cdot \sqrt{C_{in}} \\
   &= \sqrt{C_{in}} \cdot C_{im}^{n_m-n_j} \cdot \sqrt{C_{in}}.
   \end{aligned}
   $$

So in this configuration we start from the Euclidean sheet and first do a half-clockwise monodromy around $z_{in}$, then do $(n_m - n_j)$ clockwise circles around $z_{im}$ followed by a half-clockwise monodromy around $z_{in}$, i.e., $\sqrt{C_{in}} \cdot C_{im}^{n_m - n_j} \cdot \sqrt{C_{in}}$.

4. $n_i \geq n_m \geq n_n > n_j$

In this case we first move $\omega_i \to \omega_i - n_i \pi$. It crosses 3 future light cones which gives $\phi$. Then we move $\omega_m \to \omega_m - \pi$. It crosses 2 past and 1 future light cones which as per rule (3) of §6.2 gives $A_{mn} \equiv A_{ij}$. Then we move $\omega_m \to \omega_m - (n_m - 1)\pi$. It crosses future, past, future lightcone configuration which according to rule 2 of §6.2 gives $\sqrt{A_{jm}} \cdot C_{im}^{n_m - 1} \cdot \sqrt{C_{jm}} \equiv \sqrt{A_{in}} \cdot C_{im}^{n_m - 1} \cdot \sqrt{C_{in}}$.

Now we move $\omega_n \to \omega_n - \pi$. It crosses 3 past light cones which as per rule (1) of §6.2 gives $\phi$. Then we move $\omega_n \to \omega_n - (n_n - 1)\pi$. It crosses past, future, past lightcone configuration which according to rule 3 of §6.2 gives $\sqrt{C_{nm}} \cdot A_{nj}^{n_n - 1} \cdot \sqrt{A_{nm}} \equiv \sqrt{C_{ij}} \cdot A_{im}^{n_n - 1} \cdot \sqrt{A_{ij}}$.

Finally we move $\omega_j \to \omega_j - n_j \pi$. It crosses 3 past light cones and gives $\phi$.

$$
\begin{aligned}
A_{ij} \cdot & \sqrt{A_{in}} \cdot C_{im}^{n_m - 1} \cdot \sqrt{C_{in}} \cdot \sqrt{C_{ij}} \cdot A_{im}^{n_n - 1} \cdot \sqrt{A_{ij}} \\
&= A_{ij} \cdot \sqrt{C_{ij}} \cdot \sqrt{C_{im}} \cdot C_{im}^{n_m - 1} \cdot \sqrt{A_{im}} \cdot A_{im}^{n_n - 1} \cdot \sqrt{A_{ij}} \\
&= \sqrt{A_{ij}} \cdot C_{im}^{n_m - n_n} \cdot \sqrt{A_{ij}} \\
&= \sqrt{C_{in}} \cdot \sqrt{C_{im}} \cdot C_{im}^{n_m - n_n} \cdot \sqrt{C_{im}} \cdot \sqrt{C_{in}} \\
&= \sqrt{C_{in}} \cdot C_{im}^{n_m - n_n + 1} \cdot \sqrt{C_{in}}.
\end{aligned}
$$

So in this configuration we start from the Euclidean sheet and first do a half-clockwise monodromy around $z_{in}$, then do $(n_m - n_n + 1)$ clockwise circles around $z_{im}$ followed by a half-clockwise monodromy around $z_{in}$, i.e., $\sqrt{C_{in}} \cdot C_{im}^{n_m - n_n + 1} \cdot \sqrt{C_{in}}$.

5. $n_i \geq n_n > n_j \geq n_m$

In this case we first move $\omega_i \to \omega_i - n_i \pi$. It crosses 3 future light cones which as per rule (1) of §6.2 gives no monodromy, i.e., $\phi$. We then move $\omega_n \to \omega_n - \pi$. It crosses 3 past light cones which as per rule (1) gives $\phi$. We then move $\omega_n \to \omega_n - (n_n - 1)\pi$. It crosses 2 future and 1 past light cones which as per rule (2) gives $C_{in}^{n_n - 1}$.

We then move $\omega_j \to \omega_j - n_j \pi$. It crosses 2 past and 1 future light cones which as per rule (3) of §6.2 gives $A_{jm}^{n_j} \equiv A_{in}^{n_j}$. Finally we move $\omega_m \to \omega_m - n_m \pi$. It crosses 3 past light cones and hence as per rule (1) does nothing.

In summary the configuration described in this item is located in cross-ratio space as follows: we start on the Euclidean sheet and make $(n_n - n_j - 1)$ clockwise rotations around $z_{in}$, i.e., $C_{in}^{n_n - n_j - 1}$.

6. $n_i \geq n_n > n_m > n_j$

In this case we first move $\omega_i \to \omega_i - n_i \pi$. It crosses 3 future light cones which as per rule (1) of §6.2 gives no monodromy, i.e., $\phi$. We then move $\omega_n \to \omega_n - \pi$. It crosses 3 past light cones which as per rule (1) gives $\phi$. We then move $\omega_n \to \omega_n - (n_n - 1)\pi$. It crosses 2 future and 1 past light cones which as per rule (2) gives $C_{in}^{n_n - 1}$.

We then move $\omega_m \to \omega_m - \pi$. It crosses 3 past light cones which as per rule (1) gives $\phi$. We then move $\omega_m \to \omega_m - (n_m - 1)\pi$. It crosses 1 future and 2 past light cones which as per rule (3) of §6.2 gives $A_{jm}^{n_m - 1} \equiv A_{in}^{n_m - 1}$. Finally we move $\omega_j \to \omega_j - n_j \pi$. It crosses 3 past light cones and hence as per rule (1) does nothing.

In summary the configuration described in this item is located in cross-ratio space as follows: we start on the Euclidean sheet and make $(n_n - n_m)$ clockwise rotations around $z_{in}$, i.e., $C_{in}^{n_n - n_m}$.

7. $n_j > n_i \geq n_m \geq n_n$

In this case we first move $\omega_j \to \omega_j - \pi$. It crosses 1 past and 2 future light cones which according to rule (2) of §6.2 gives $C_{ij}$. We then move $\omega_j \to \omega_j - (n_j - 1)\pi$ which crosses 3 future light cones and according to rule (1) gives no monodromy. We then move $\omega_i \to \omega_i - n_i \pi$. It crosses 2 future and 1 past light cones which according to rule (2) of §6.2 gives $C_{ij}^{n_i}$.

We then move $\omega_m \to \omega_m - n_m \pi$. It crosses 2 past and 1 future light cones which according to rule (3) of §6.2 gives $A_{mn}^{n_m} \equiv A_{ij}^{n_m}$. Finally we move $\omega_n \to \omega_n - n_n \pi$. It crosses 3 past light cones and hence as per rule (1) does nothing.

So in this configuration we start from the Euclidean sheet and do $(n_i - n_m + 1)$ clockwise circles around $z_{ij}$, i.e., $C_{ij}^{n_i - n_m + 1}$.

8. $n_j > n_i \geq n_n > n_m$

In this case we first move $\omega_j \to \omega_j - \pi$. It crosses 1 past and 2 future light cones which according to rule (2) of §6.2 gives $C_{ij}$. We then move $\omega_j \to \omega_j - (n_j - 1)\pi$ which crosses 3 future light cones and according to rule (1) gives no monodromy. We then move $\omega_i \to \omega_i - n_i \pi$. It crosses 2 future and 1 past light cones which according to rule (2) of §6.2 gives $C_{ij}^{n_i}$.

We now move $\omega_n \to \omega_n - \pi$. It crosses 3 past light cones which gives $\phi$. We then move $\omega_n \to \omega_n - (n_n - 1)\pi$. It crosses 1 future and 2 past light cones which according to rule (3) of §6.2 gives $A_{mn}^{n_n - 1} \equiv A_{ij}^{n_n - 1}$. Finally we move $\omega_m \to \omega_m - n_m \pi$. It crosses 3 past light cones and hence as per rule (1) does nothing.

So in this configuration we start from the Euclidean sheet and do $(n_i - n_n + 2)$ clockwise circles around $z_{ij}$, i.e., $C_{ij}^{n_i - n_n + 2}$.

9. $n_j \geq n_m > n_i \geq n_n$

In this case we first move $\omega_j \to \omega_j - \pi$. It crosses 1 past and 2 future light cones which according to rule (2) of §6.2 gives $C_{ij}$. We then move $\omega_j \to \omega_j - (n_j - 1)\pi$ which crosses 3 future light cones and according to rule (1) gives no monodromy. We then move $\omega_m \to \omega_m - \pi$. It crosses 2 past and 1 future light cones which according to rule (3) of §6.2 gives $A_{mn} \equiv A_{ij}$. We then move $\omega_m \to \omega_m - (n_m - 1)\pi$. It crosses 1 past and 2 future light cones which according to rule (2) gives $C_{jm}^{n_m - 1} \equiv C_{in}^{n_m - 1}$.

We now move $\omega_i \to \omega_i - n_i \pi$. It crosses 1 future and 2 past light cones which as per rule (3) gives $A_{in}^{n_i}$. Finally we move $\omega_n \to \omega_n - n_n \pi$. It crosses 3 past light cones and hence as per rule (1) does nothing.

So in this configuration we start from the Euclidean sheet and do $(n_m - n_i - 1)$ clockwise circles around $z_{in}$, i.e., $C_{in}^{n_m - n_i - 1}$.

10. $n_j \geq n_m \geq n_n > n_i$

In this case we first move $\omega_j \to \omega_j - \pi$. It crosses 1 past and 2 future light cones which according to rule (2) of §6.2 gives $C_{ij}$. We then move $\omega_j \to \omega_j - (n_j - 1)\pi$ which crosses 3 future light cones and according to rule (1) gives no monodromy. We then move $\omega_m \to \omega_m - \pi$. It crosses 2 past and 1 future light cones which according to rule (3) of §6.2 gives $A_{mn} \equiv A_{ij}$. We then move $\omega_m \to \omega_m - (n_m - 1)\pi$. It crosses 1 past and 2 future light cones which according to rule (2) gives $C_{jm}^{n_m - 1} \equiv C_{in}^{n_m - 1}$.

We now move $\omega_n \to \omega_n - \pi$. It crosses 3 past light cones which gives $\phi$. We then move $\omega_n \to \omega_n - (n_n - 1)\pi$. It crosses 2 past and 1 future light cones which according to rule (3) of §6.2 gives $A_{in}^{n_n-1}$. Finally we move $\omega_i \to \omega_i - n_i\pi$. It crosses 3 past light cones and hence as per rule (1) does nothing.

So in this configuration we start from the Euclidean sheet and do $(n_m - n_n)$ clockwise circles around $z_{in}$, i.e., $\boldsymbol{C_{in}^{n_m-n_n}}$.

11. $\boldsymbol{n_j \geq n_n > n_i \geq n_m}$

In this case we first move $\omega_j \to \omega_j - \pi$. It crosses 1 past and 2 future light cones which as per rule (2) of §6.2 gives $C_{ij}$. We then move $\omega_j \to \omega_j - (n_j - 1)\pi$. It crosses 3 future light cones which gives $\phi$. Then we move $\omega_n \to \omega_n - \pi$. It crosses 3 past light cones which gives $\phi$. Then we move $\omega_n \to \omega_n - (n_n - 1)\pi$. It crosses future, past, future lightcone configuration which according to rule 2 of §6.2 gives $\sqrt{A_{nm}} \cdot C_{jn}^{n_n-1} \cdot \sqrt{C_{mn}} \equiv \sqrt{A_{ij}} \cdot C_{im}^{n_n-1} \cdot \sqrt{C_{ij}}$.

Now we move $\omega_i \to \omega_i - n_i\pi$. It crosses past, future, past lightcone configuration which according to rule 3 gives $\sqrt{C_{in}} \cdot A_{im}^{n_i} \cdot \sqrt{A_{in}}$.

Finally we move $\omega_m \to \omega_m - n_m\pi$. It crosses 3 past light cones and as per rule (1) gives $\phi$.

$$C_{ij} \cdot \sqrt{A_{ij}} \cdot C_{im}^{n_n-1} \cdot \sqrt{C_{ij}} \cdot \sqrt{C_{in}} \cdot A_{im}^{n_i} \cdot \sqrt{A_{in}}$$
$$= \sqrt{C_{ij}} \cdot C_{im}^{n_n-1} \cdot \sqrt{A_{im}} \cdot A_{im}^{n_i} \cdot \sqrt{C_{im}} \cdot \sqrt{C_{ij}}$$
$$= \sqrt{C_{ij}} \cdot C_{im}^{n_n-n_i-1} \cdot \sqrt{C_{ij}} .$$

So in this configuration we start from the Euclidean sheet and first do a half-clockwise monodromy around $z_{ij}$, then do $(n_n - n_i - 1)$ clockwise circles around $z_{im}$ followed by a half-clockwise monodromy around $z_{ij}$, i.e., $\boldsymbol{\sqrt{C_{ij}} \cdot C_{im}^{n_n-n_i-1} \cdot \sqrt{C_{ij}}}$.

12. $\boldsymbol{n_j \geq n_n > n_m > n_i}$

In this case we first move $\omega_j \to \omega_j - \pi$. It crosses 1 past and 2 future light cones which as per rule (2) of §6.2 gives $C_{ij}$. We then move $\omega_j \to \omega_j - (n_j - 1)\pi$. It crosses 3 future light cones which gives $\phi$. Then we move $\omega_n \to \omega_n - \pi$. It crosses 3 past light cones which as per rule (1) gives $\phi$. Then we move $\omega_n \to \omega_n - (n_n - 1)\pi$. It crosses future, past, future lightcone configuration which according to rule 2 of §6.2 gives $\sqrt{A_{nm}} \cdot C_{jn}^{n_n-1} \cdot \sqrt{C_{mn}} \equiv \sqrt{A_{ij}} \cdot C_{im}^{n_n-1} \cdot \sqrt{C_{ij}}$.

Now we move $\omega_m \to \omega_m - \pi$. It crosses 3 past light cones which as per rule (1) gives $\phi$. Then we move $\omega_m \to \omega_m - (n_m - 1)\pi$. It crosses past, future, past lightcone configuration which according to rule 3 of §6.2 gives $\sqrt{C_{jm}} \cdot A_{im}^{n_m-1} \cdot \sqrt{A_{jm}} \equiv \sqrt{C_{in}} \cdot A_{im}^{n_m-1} \cdot \sqrt{A_{in}}$.

Finally we move $\omega_i \to \omega_i - n_i\pi$. It crosses 3 past light cones and as per rule (1) gives $\phi$.

$$C_{ij} \cdot \sqrt{A_{ij}} \cdot C_{im}^{n_n-1} \cdot \sqrt{C_{ij}} \cdot \sqrt{C_{in}} \cdot A_{im}^{n_m-1} \cdot \sqrt{A_{in}}$$
$$= \sqrt{C_{ij}} \cdot C_{im}^{n_n-1} \cdot \sqrt{A_{im}} \cdot A_{im}^{n_m-1} \cdot \sqrt{C_{im}} \cdot \sqrt{C_{ij}}$$
$$= \sqrt{C_{ij}} \cdot C_{im}^{n_n-n_m} \cdot \sqrt{C_{ij}} .$$

So in this configuration we start from the Euclidean sheet and first do a half-clockwise monodromy around $z_{ij}$, then do $(n_n - n_m)$ clockwise circles around $z_{im}$ followed by a half-clockwise monodromy around $z_{ij}$, i.e., $\boldsymbol{\sqrt{C_{ij}} \cdot C_{im}^{n_n-n_m} \cdot \sqrt{C_{ij}}}$.

13. $n_m > n_i \geq n_j \geq n_n$

   In this case we first move $\omega_m \to \omega_m - \pi$. It crosses 2 past and 1 future light cones which as per rule (3) of §6.2 gives $A_{mn} \equiv A_{ij}$. We then move $\omega_m \to \omega_m - (n_m - 1)\pi$. It crosses 3 future light cones which as per rule (1) gives $\phi$. Then we move $\omega_i \to \omega_i - n_i\pi$. It crosses future, past, future lightcone configuration which according to rule 2 of §6.2 gives $\sqrt{A_{in}} \cdot C_{im}^{n_i} \cdot \sqrt{C_{in}}$.

   Now we move $\omega_j \to \omega_j - n_j\pi$. It crosses past, future, past lightcone configuration which according to rule 3 gives $\sqrt{C_{ij}} \cdot A_{im}^{n_j} \cdot \sqrt{A_{ij}}$.

   Finally we move $\omega_n \to \omega_n - n_n\pi$. It crosses 3 past light cones and as per rule (1) gives $\phi$.

$$\begin{aligned}
A_{ij} &\cdot \sqrt{A_{in}} \cdot C_{im}^{n_i} \cdot \sqrt{C_{in}} \cdot \sqrt{C_{ij}} \cdot A_{im}^{n_j} \cdot \sqrt{A_{ij}} \\
&= A_{ij} \cdot \sqrt{C_{ij}} \cdot \sqrt{C_{im}} \cdot C_{im}^{n_i} \cdot \sqrt{A_{im}} \cdot A_{im}^{n_j} \cdot \sqrt{C_{im}} \cdot \sqrt{C_{in}} \\
&= \sqrt{A_{ij}} \cdot C_{im}^{n_i - n_j + \frac{1}{2}} \cdot \sqrt{C_{in}} \\
&= \sqrt{C_{in}} \cdot \sqrt{C_{im}} \cdot C_{im}^{n_i - n_j + \frac{1}{2}} \cdot \sqrt{C_{in}} \\
&= \sqrt{C_{in}} \cdot C_{im}^{n_i - n_j + 1} \cdot \sqrt{C_{in}}.
\end{aligned}$$

   So in this configuration we start from the Euclidean sheet and first do a half-clockwise monodromy around $z_{in}$, then do $(n_i - n_j + 1)$ clockwise circles around $z_{im}$ followed by a half-clockwise monodromy around $z_{in}$, i.e., $\sqrt{C_{in}} \cdot C_{im}^{n_i - n_j + 1} \cdot \sqrt{C_{in}}$.

14. $n_m > n_i \geq n_n > n_j$

   In this case we first move $\omega_m \to \omega_m - \pi$. It crosses 2 past and 1 future light cones which as per rule (3) of §6.2 gives $A_{mn} \equiv A_{ij}$. We then move $\omega_m \to \omega_m - (n_m - 1)\pi$. It crosses 3 future light cones which as per rule (1) gives $\phi$. Then we move $\omega_i \to \omega_i - n_i\pi$. It crosses future, past, future lightcone configuration which according to rule (2) of §6.2 gives $\sqrt{A_{in}} \cdot C_{im}^{n_i} \cdot \sqrt{C_{in}}$.

   Now we move $\omega_n \to \omega_n - \pi$. It crosses 3 past light cones and as per rule (1) gives $\phi$. Then we move $\omega_n \to \omega_n - (n_n - 1)\pi$. It crosses past, future, past lightcone configuration which according to rule (3) of §6.2 gives $\sqrt{C_{nm}} \cdot A_{nj}^{n_n - 1} \cdot \sqrt{A_{nm}} \equiv \sqrt{C_{ij}} \cdot A_{im}^{n_n - 1} \cdot \sqrt{A_{ij}}$.

   Finally we move $\omega_j \to \omega_j - n_j\pi$. It crosses 3 past light cones and as per rule (1) gives $\phi$.

$$\begin{aligned}
A_{ij} &\cdot \sqrt{A_{in}} \cdot C_{im}^{n_i} \cdot \sqrt{C_{in}} \cdot \sqrt{C_{ij}} \cdot A_{im}^{n_n - 1} \cdot \sqrt{A_{ij}} \\
&= A_{ij} \cdot \sqrt{C_{ij}} \cdot \sqrt{C_{im}} \cdot C_{im}^{n_i} \cdot \sqrt{A_{im}} \cdot A_{im}^{n_n - 1} \cdot \sqrt{C_{im}} \cdot \sqrt{C_{in}} \\
&= \sqrt{A_{ij}} \cdot C_{im}^{n_i - n_n + \frac{3}{2}} \cdot \sqrt{C_{in}} \\
&= \sqrt{C_{in}} \cdot \sqrt{C_{im}} \cdot C_{im}^{n_i - n_n + \frac{3}{2}} \cdot \sqrt{C_{in}} \\
&= \sqrt{C_{in}} \cdot C_{im}^{n_i - n_n + 2} \cdot \sqrt{C_{in}}.
\end{aligned}$$

   So in this configuration we start from the Euclidean sheet and first do a half-clockwise monodromy around $z_{in}$, then do $(n_i - n_n + 2)$ clockwise circles around $z_{im}$ followed by a half-clockwise monodromy around $z_{in}$, i.e., $\sqrt{C_{in}} \cdot C_{im}^{n_i - n_n + 2} \cdot \sqrt{C_{in}}$.

15. $n_m > n_j > n_i \geq n_n$

In this case we first move $\omega_m \to \omega_m - \pi$. It crosses 2 past and 1 future light cones which according to rule (3) of §6.2 gives $A_{mn} \equiv A_{ij}$. We then move $\omega_m \to \omega_m - (n_m - 1)\pi$ which crosses 3 future light cones and according to rule (1) gives no monodromy. We then move $\omega_j \to \omega_j - \pi$. It crosses past, future, past lightcone configuration which as per rule (3) of §6.2 gives $\sqrt{C_{ij}} \cdot A_{jn} \cdot \sqrt{A_{ij}} \equiv \sqrt{C_{ij}} \cdot A_{im} \cdot \sqrt{A_{ij}}$.

We then move $\omega_j \to \omega_j - (n_j - 1)\pi$. It crosses 2 future and 1 past light cones which according to rule (2) gives $C_{jm}^{n_j-1} \equiv C_{in}^{n_j-1}$. We now move $\omega_i \to \omega_i - n_i\pi$. It crosses 1 future and 2 past light cones which as per rule (3) gives $A_{in}^{n_i}$. Finally we move $\omega_n \to \omega_n - n_n\pi$. It crosses 3 past light cones and hence as per rule (1) does nothing.

$$
A_{ij} \cdot \sqrt{C_{ij}} \cdot A_{im} \cdot \sqrt{A_{ij}} \cdot C_{in}^{n_j-1} \cdot A_{in}^{n_i}
$$
$$
= \sqrt{A_{ij}} \cdot A_{im} \cdot \sqrt{C_{im}} \cdot \sqrt{C_{in}} \cdot C_{in}^{n_j-n_i-1}
$$
$$
= \sqrt{A_{ij}} \cdot \sqrt{A_{im}} \cdot C_{in}^{n_j-n_i-\frac{1}{2}}
$$
$$
= \sqrt{C_{in}} \cdot C_{in}^{n_j-n_i-\frac{1}{2}}
$$
$$
= C_{in}^{n_j-n_i} .
$$

So in this configuration we start from the Euclidean sheet and do $(n_j - n_i)$ clockwise circles around $z_{in}$, i.e., $C_{in}^{n_j-n_i}$.

16. $n_m > n_j \geq n_n > n_i$

In this case we first move $\omega_m \to \omega_m - \pi$. It crosses 2 past and 1 future light cones which according to rule (3) of §6.2 gives $A_{mn} \equiv A_{ij}$. We then move $\omega_m \to \omega_m - (n_m - 1)\pi$ which crosses 3 future light cones and according to rule (1) gives no monodromy. We then move $\omega_j \to \omega_j - \pi$. It crosses past, future, past lightcone configuration which as per rule (3) of §6.2 gives $\sqrt{C_{ij}} \cdot A_{jn} \cdot \sqrt{A_{ij}} \equiv \sqrt{C_{ij}} \cdot A_{im} \cdot \sqrt{A_{ij}}$.

We then move $\omega_j \to \omega_j - (n_j - 1)\pi$. It crosses 2 future and 1 past light cones which according to rule (2) gives $C_{jm}^{n_j-1} \equiv C_{in}^{n_j-1}$. We now move $\omega_n \to \omega_n - \pi$. It crosses 3 past light cones which as per rule (1) gives $\phi$. We then move $\omega_n \to \omega_n - (n_n - 1)\pi$. It crosses 2 past and 1 future light cones which according to rule (3) gives $A_{in}^{n_n-1}$.

Finally we move $\omega_i \to \omega_i - n_i\pi$. It crosses 3 past light cones and hence as per rule (1) does nothing.

$$
A_{ij} \cdot \sqrt{C_{ij}} \cdot A_{im} \cdot \sqrt{A_{ij}} \cdot C_{in}^{n_j-1} \cdot A_{in}^{n_n-1}
$$
$$
= \sqrt{A_{ij}} \cdot A_{im} \cdot \sqrt{C_{im}} \cdot \sqrt{C_{in}} \cdot C_{in}^{n_j-n_n}
$$
$$
= \sqrt{A_{ij}} \cdot \sqrt{A_{im}} \cdot \sqrt{C_{in}} \cdot C_{in}^{n_j-n_n}
$$
$$
= \sqrt{C_{in}} \cdot \sqrt{C_{in}} \cdot C_{in}^{n_j-n_n}
$$
$$
= C_{in}^{n_j-n_n+1} .
$$

So in this configuration we start from the Euclidean sheet and do $(n_j - n_n + 1)$ clockwise circles around $z_{in}$, i.e., $C_{in}^{n_j-n_n+1}$.

17. $n_m \geq n_n > n_i \geq n_j$ In this case we first move $\omega_m \to \omega_m - \pi$. It crosses 2 past and 1 future light cones which according to rule (3) of §6.2 gives $A_{mn} \equiv A_{ij}$. We then move

$\omega_m \rightarrow \omega_m - (n_m - 1)\pi$ which crosses 3 future light cones and according to rule (1) gives no monodromy. We then move $\omega_n \rightarrow \omega_n - \pi$. It crosses 3 past light cones which as per rule (1) gives $\phi$. We then move $\omega_n \rightarrow \omega_n - (n_n - 1)\pi$. It crosses 1 past and 2 future light cones which according to rule (2) gives $C_{mn}^{n_n - 1} \equiv C_{ij}^{n_n - 1}$.

We now move $\omega_i \rightarrow \omega_i - n_i \pi$. It crosses 2 past and 1 future light cones which as per rule (3) gives $A_{ij}^{n_i}$. Finally we move $\omega_j \rightarrow \omega_j - n_j \pi$. It crosses 3 past light cones and hence as per rule (1) does nothing.

So in this configuration we start from the Euclidean sheet and do $(n_n - n_i - 2)$ clockwise circles around $z_{ij}$, i.e., $C_{ij}^{n_n - n_i - 2}$.

18. $n_m \geq n_n > n_j > n_i$

In this case we first move $\omega_m \rightarrow \omega_m - \pi$. It crosses 2 past and 1 future light cones which according to rule (3) of §6.2 gives $A_{mn} \equiv A_{ij}$. We then move $\omega_m \rightarrow \omega_m - (n_m - 1)\pi$ which crosses 3 future light cones and according to rule (1) gives no monodromy. We then move $\omega_n \rightarrow \omega_n - \pi$. It crosses 3 past light cones which as per rule (1) gives $\phi$. We then move $\omega_n \rightarrow \omega_n - (n_n - 1)\pi$. It crosses 1 past and 2 future light cones which according to rule (2) gives $C_{mn}^{n_n - 1} \equiv C_{ij}^{n_n - 1}$.

We now move $\omega_j \rightarrow \omega_j - \pi$. It crosses 3 past light cones which as per rule (1) gives $\phi$. We then move $\omega_j \rightarrow \omega_j - (n_j - 1)\pi$. It crosses 1 future and 2 past light cones which according to rule (3) pf §6.2 gives $A_{ij}^{n_j - 1}$. Finally we move $\omega_i \rightarrow \omega_i - n_i \pi$. It crosses 3 past light cones and hence as per rule (1) does nothing.

So in this configuration we start from the Euclidean sheet and do $(n_n - n_j - 1)$ clockwise circles around $z_{ij}$, i.e., $C_{ij}^{n_n - n_j - 1}$.

19. $n_n > n_i \geq n_j \geq n_m$

In this case we first move $\omega_n \rightarrow \omega_n - \pi$. It crosses 3 past light cones which according to rule (1) of §6.2 gives $\phi$. We then move $\omega_n \rightarrow \omega_n - (n_n - 1)\pi$ which crosses 3 future light cones and according to rule (1) gives no monodromy. We then move $\omega_i \rightarrow \omega_i - n_i \pi$. It crosses 1 past and 2 future light cones which according to rule (2) of gives $C_{in}^{n_i}$.

We then move $\omega_j \rightarrow \omega_j - n_j \pi$. It crosses 2 past and 1 future light cones which as per rule (3) §6.2 gives $A_{jm}^{n_j} \equiv A_{in}^{n_j}$. Finally we move $\omega_m \rightarrow \omega_m - n_m \pi$. It crosses 3 past light cones and hence as per rule (1) does nothing.

So in this configuration we start from the Euclidean sheet and do $(n_i - n_j)$ clockwise circles around $z_{in}$, i.e., $C_{in}^{n_i - n_j}$.

20. $n_n > n_i \geq n_m > n_j$

In this case we first move $\omega_n \rightarrow \omega_n - \pi$. It crosses 3 past light cones which according to rule (1) of §6.2 gives $\phi$. We then move $\omega_n \rightarrow \omega_n - (n_n - 1)\pi$ which crosses 3 future light cones and according to rule (1) gives no monodromy. We then move $\omega_i \rightarrow \omega_i - n_i \pi$. It crosses 1 past and 2 future light cones which according to rule (2) gives $C_{in}^{n_i}$.

We now move $\omega_m \rightarrow \omega_m - \pi$. It crosses 3 past light cones which as per rule (1) gives $\phi$. We then move $\omega_m \rightarrow \omega_m - (n_m - 1)\pi$. It crosses 1 future and 2 past light cones which according to rule (3) of §6.2 gives $A_{jm}^{n_m - 1} \equiv A_{in}^{n_m - 1}$. Finally we move $\omega_j \rightarrow \omega_j - n_j \pi$. It crosses 3 past light cones and hence as per rule (1) does nothing.

So in this configuration we start from the Euclidean sheet and do $(n_i - n_m + 1)$ clockwise circles around $z_{in}$, i.e., $C_{in}^{n_i - n_m + 1}$.

21. $n_n > n_j > n_i \geq n_m$

In this case we first move $\omega_n \to \omega_n - \pi$. It crosses 3 past light cones which as per rule (1) of §6.2 gives $\phi$. We then move $\omega_n \to \omega_n - (n_n - 1)\pi$. It crosses 3 future light cones which again as per rule (1) gives $\phi$. Then we move $\omega_j \to \omega_j - \pi$. It crosses 2 past and 1 future light cones which as per rule (3) of §6.2 gives $A_{jm} \equiv A_{in}$. Then we move $\omega_j \to \omega_j - (n_j - 1)\pi$. It crosses future, past, future lightcone configuration which according to rule 2 of §6.2 gives $\sqrt{A_{ij}} \cdot C_{jn}^{n_j - 1} \cdot \sqrt{C_{ij}} \equiv \sqrt{A_{ij}} \cdot C_{im}^{n_j - 1} \cdot \sqrt{C_{ij}}$.

Now we move $\omega_i \to \omega_i - n_i \pi$. It crosses past, future, past lightcone configuration which according to rule 3 of §6.2 gives $\sqrt{C_{in}} \cdot A_{im}^{n_i} \cdot \sqrt{A_{in}}$.

Finally we move $\omega_m \to \omega_m - n_m \pi$. It crosses 3 past light cones and gives $\phi$.

$$
\begin{aligned}
& A_{in} \cdot \sqrt{A_{ij}} \cdot C_{im}^{n_j - 1} \cdot \sqrt{C_{ij}} \cdot \sqrt{C_{in}} \cdot A_{im}^{n_i} \cdot \sqrt{A_{in}} \\
& = A_{in} \cdot \sqrt{C_{in}} \cdot \sqrt{C_{im}} \cdot C_{im}^{n_j - 1} \cdot \sqrt{A_{im}} \cdot A_{im}^{n_i} \cdot \sqrt{C_{im}} \cdot \sqrt{C_{ij}} \\
& = \sqrt{A_{in}} \cdot C_{im}^{n_j - n_i - \frac{1}{2}} \cdot \sqrt{C_{ij}} \\
& = \sqrt{C_{ij}} \cdot \sqrt{C_{im}} \cdot C_{im}^{n_j - n_i - \frac{1}{2}} \cdot \sqrt{C_{ij}} \\
& = \sqrt{C_{ij}} \cdot C_{im}^{n_j - n_i} \cdot \sqrt{C_{ij}}.
\end{aligned}
$$

So in this configuration we start from the Euclidean sheet and first do a half-clockwise monodromy around $z_{ij}$, then do $(n_j - n_i)$ clockwise circles around $z_{im}$ followed by a half-clockwise monodromy around $z_{ij}$, i.e., $\sqrt{C_{ij}} \cdot C_{im}^{n_j - n_i} \cdot \sqrt{C_{ij}}$.

22. $n_n > n_j \geq n_m > n_i$

In this case we first move $\omega_n \to \omega_n - \pi$. It crosses 3 past light cones which as per rule (1) of §6.2 gives $\phi$. We then move $\omega_n \to \omega_n - (n_n - 1)\pi$. It crosses 3 future light cones which again as per rule (1) gives $\phi$. Then we move $\omega_j \to \omega_j - \pi$. It crosses 2 past and 1 future light cones which as per rule (3) gives $A_{jm} \equiv A_{in}$. Then we move $\omega_j \to \omega_j - (n_j - 1)\pi$. It crosses future, past, future lightcone configuration which according to rule 2 of §6.2 gives $\sqrt{A_{ij}} \cdot C_{jn}^{n_j - 1} \cdot \sqrt{C_{ij}} \equiv \sqrt{A_{ij}} \cdot C_{im}^{n_j - 1} \cdot \sqrt{C_{ij}}$.

Now we move $\omega_m \to \omega_m - \pi$. It crosses 3 past light cones which as per rule (1) gives $\phi$. Then we move $\omega_m \to \omega_m - (n_m - 1)\pi$. It crosses past, future, past lightcone configuration which according to rule (3) of §6.2 gives $\sqrt{C_{mj}} \cdot A_{im}^{n_m - 1} \cdot \sqrt{A_{mj}} \equiv \sqrt{C_{in}} \cdot A_{im}^{n_m - 1} \cdot \sqrt{A_{in}}$.

Finally we move $\omega_i \to \omega_i - n_i \pi$. It crosses 3 past light cones and as per rule (1) gives $\phi$.

$$
\begin{aligned}
& A_{in} \cdot \sqrt{A_{ij}} \cdot C_{im}^{n_j - 1} \cdot \sqrt{C_{ij}} \cdot \sqrt{C_{in}} \cdot A_{im}^{n_m - 1} \cdot \sqrt{A_{in}} \\
& = A_{in} \cdot \sqrt{C_{in}} \cdot \sqrt{C_{im}} \cdot C_{im}^{n_j - 1} \cdot \sqrt{A_{im}} \cdot A_{im}^{n_m - 1} \cdot \sqrt{C_{im}} \cdot \sqrt{C_{ij}} \\
& = \sqrt{A_{in}} \cdot C_{im}^{n_j - n_m + \frac{1}{2}} \cdot \sqrt{C_{ij}} \\
& = \sqrt{C_{ij}} \cdot \sqrt{C_{im}} \cdot C_{im}^{n_j - n_m + \frac{1}{2}} \cdot \sqrt{C_{ij}} \\
& = \sqrt{C_{ij}} \cdot C_{im}^{n_j - n_m + 1} \cdot \sqrt{C_{ij}}.
\end{aligned}
$$

So in this configuration we start from the Euclidean sheet and first do a half-clockwise monodromy around $z_{ij}$, then do $(n_j - n_m + 1)$ clockwise circles around $z_{im}$ followed by a half-clockwise monodromy around $z_{ij}$, i.e., $\sqrt{C_{ij}} \cdot C_{im}^{n_j - n_m + 1} \cdot \sqrt{C_{ij}}$.

23. $n_n > n_m > n_i \geq n_j$

In this case we first move $\omega_n \rightarrow \omega_n - \pi$. It crosses 3 past light cones which according to rule (1) of §6.2 gives $\phi$. We then move $\omega_n \rightarrow \omega_n - (n_n - 1)\pi$ which crosses 3 future light cones and according to rule (1) gives no monodromy. We then move $\omega_m \rightarrow \omega_m - \pi$. It crosses 3 past light cones which as per rule (1) gives $\phi$. We then move $\omega_m \rightarrow \omega_m - (n_m - 1)\pi$. It crosses 2 future and 1 past light cones which according to rule (2) gives $C_{mn}^{n_m-1} \equiv C_{ij}^{n_m-1}$.

We then move $\omega_i \rightarrow \omega_i - n_i \pi$. It crosses 2 past and 1 future light cones which as per rule (3) of §6.2 gives $A_{ij}^{n_i}$. Finally we move $\omega_j \rightarrow \omega_j - n_j \pi$. It crosses 3 past light cones and hence as per rule (1) does nothing.

So in this configuration we start from the Euclidean sheet and do $(n_m - n_i - 1)$ clockwise circles around $z_{ij}$, i.e., $C_{ij}^{n_m-n_i-1}$.

24. $n_n > n_m > n_j > n_i$

In this case we first move $\omega_n \rightarrow \omega_n - \pi$. It crosses 3 past light cones which according to rule (1) of §6.2 gives $\phi$. We then move $\omega_n \rightarrow \omega_n - (n_n - 1)\pi$ which crosses 3 future light cones and according to rule (1) gives no monodromy. We then move $\omega_m \rightarrow \omega_m - \pi$. It crosses 3 past light cones which as per rule (1) gives $\phi$. We then move $\omega_m \rightarrow \omega_m - (n_m - 1)\pi$. It crosses 2 future and 1 past light cones which according to rule (2) gives $C_{mn}^{n_m-1} \equiv C_{ij}^{n_m-1}$.

We now move $\omega_j \rightarrow \omega_j - \pi$. It crosses 3 past light cones which as per rule (1) gives $\phi$. We then move $\omega_j \rightarrow \omega_j - (n_j - 1)\pi$. It crosses 1 future and 2 past light cones which according to rule (3) of §6.2 gives $A_{ij}^{n_j-1}$. Finally we move $\omega_i \rightarrow \omega_i - n_i \pi$. It crosses 3 past light cones and hence as per rule (1) does nothing.

So in this configuration we start from the Euclidean sheet and do $(n_m - n_j)$ clockwise circles around $z_{ij}$, i.e., $C_{ij}^{n_m-n_j}$.

## F.7 Scattering configurations

In the diagram Fig. 14a we listed a configuration that lies within the Minkowski diamond, has $z = \bar{z}$, but does not lie on the Euclidean Sheet. In fact this configuration lies on the "scattering sheet" (obtained starting from the Euclidean sheet and performing a single clockwise monodromy around $z_{ij}$).

In this subsection we describe the sheet structure of all configurations that can be brought to a scattering configuration by making $\pi$ shifts of the $\omega$ coordinates of the various insertions. As in the previous subsection, it is useful to fix on a convention. We denote the insertion labels for the top two operators in Fig. §16 as $m$ and $n$ (it does not matter which is which), but denote the insertion labels of the bottom two operators in Fig. §16 as $i$ and $j$, as shown in the figure. Starting with this configuration, we then move to new configurations by making the shifts $\omega_a \rightarrow \omega_a - n_a \pi$ for $a = i, j, m, n$.

Since $(i \sim j)$ and $(m \sim n)$ are spacelike separated pairs, and both pairs are timelike separated with each other, the 4! cases are related to each other by the symmetries $i \leftrightarrow j$ and (independently) $m \leftrightarrow n$. There are six inequivalent cases, which can be characterized by introducing two pieces of notation.

First, if $n_a > n_b > n_c > n_d$, we say that our configuration is in the ordering (abcd).[62] Given an ordering $(abcd)$ we call $a$ and $d$ the extremities of our ordering. Let us also choose to call

---

[62]If two $n's$ are equal, we choose the ordering to ensure that if $a$ is in the causal future of $b$ then $a$ lies to the left of $b$. If two $a$ and $b$ are spacelike related, but have the same $n$ then we are free to choose the relative orderings of $a$ and $b$ arbitrarily.

the top two insertions $(m, n)$ (in the starting Poincare diamond configuration) "red insertions" (R) and the bottom two insertions $(i, j)$ "blue insertions" (B). The six inequivalent cases are, respectively, the orderings (BBRR), (BRBR), (RBRB), (RRBB), (BRRB), (RBBR). We have only 6 inequivalent (rather than 24 inequivalent cases) because the two blue and two red insertions are equivalent.

Let us now study the action of time reversal on our 6 classes of configurations. The pair $(m, n)$ were distinguished from the pair $(i, j)$ because each of $(m, n)$ was to the future of each of $(i, j)$. Clearly time reversal interchanges $(m, n)$ and $(i, j)$, and so turns a "blue" operator red and a "red" operator blue. In addition, time reversal switches the order of operator insertions (abcd) goes to (dcba).

Let us, for example consider the action of time reversal on the ordering (*RBRB*).[63] The action of time reversal first reverses the order (i.e. yields (*BRBR*)), and then turns every blue to red and red to blue, i.e. yields (*RBRB*). We see that time reversal maps an (*RBRB*) configuration to a configuration of the same sort. The reader can easily check that the same is true of all orderings with opposite colours in the extremities, irrespective of the middle two orderings, i.e. for the ordering (BBRR, BRBR, RRBB) in addition to (*RBRB*) (see cases studied in 1, 2, 5, 6 below). On the other hand consider (*RBBR*). Reversing the order of time takes this to (*RBBR*). Then performing the interchange $R \leftrightarrow B$ takes this configuration to (*BRRB*). We see, consequently, that under time reversal BRRB $\leftrightarrow$ RBBR (see the cases studied in 3 and 4 below).

In the rest of this subsection we give a detailed derivation for the final branch structure obtained, starting from a scattering type configuration, and making the moves $\omega_a \rightarrow \omega_a - n_a \pi$. Once again, this subsection is lengthy as we have presented all details of the derivation. The reader who is interested only in final results is invited to skip over to the next subsection.

1. **$n_i \geq n_j > n_m \geq n_n$**

   In this case we first move $\omega_i \rightarrow \omega_i - \pi$. It crosses 2 past and 1 future light cones which according to rule (3) of §6.2 gives $A_{ij}$. We then move $\omega_i \rightarrow \omega_i - (n_i - 1)\pi$ which crosses 3 future light cones and according to rule (1) gives no monodromy. We then move $\omega_j \rightarrow \omega_j - \pi$. It crosses 3 past light cones which as per rule (1) gives $\phi$. We then move $\omega_j \rightarrow \omega_j - (n_j - 1)\pi$. It crosses 1 past and 2 future light cones which according to rule (2) gives $C_{ij}^{n_j-1}$.

   We then move $\omega_m \rightarrow \omega_m - n_m \pi$. It crosses 2 past and 1 future light cones which as per rule (3) gives $A_{mn}^{n_m} \equiv A_{ij}^{n_m}$. Finally we move $\omega_n \rightarrow \omega_n - n_n \pi$. It crosses 3 past light cones and hence as per rule (1) does nothing.

   Notice that the starting configuration is obtained from the Euclidean sheet by making a single clockwise monodromy around $z_{ij}$. So in this configuration we start from the Euclidean sheet and do $(n_j - n_m - 1)$ clockwise circles around $z_{ij}$, i.e., $C_{ij}^{n_j - n_m - 1}$.

   Time reversal maps this set of moves to itself. Time reversal consists of the interchange $(m, n) \leftrightarrow (i, j)$ together with all $n's$ flipping sign. This combined operation leaves our final answer for the monodromy unchanged, as we had expected.

2. **$n_i > n_m \geq n_j > n_n$**

   In this case we first move $\omega_i \rightarrow \omega_i - \pi$. It crosses 2 past and 1 future light cones which according to rule (3) of §6.2 gives $A_{ij}$. We then move $\omega_i \rightarrow \omega_i - (n_i - 1)\pi$ which crosses 3 future light cones and according to rule (1) gives no monodromy. We then

---

[63]By this we mean that the operator with the largest value of $n$ is a red operator (e.g. $m$). The operator with the second largest value of $n$ is a blue operator (e.g. $i$). And so on.

move $\omega_m \to \omega_m - n_m \pi$. It crosses future, past, future light cones configuration which according to rule (2) of §6.2 gives $\sqrt{A_{jm}} \cdot C_{im}^{n_m} \cdot \sqrt{C_{jm}}$.

We then move $\omega_j \to \omega_j - \pi$. It crosses 3 past light cones which as per rule (1) gives $\phi$. We then move $\omega_j \to \omega_j - (n_j - 1)\pi$. It crosses past, future, past lightcone configuration which according to rule (3) gives $\sqrt{C_{ij}} \cdot A_{jn}^{n_j - 1} \cdot \sqrt{A_{ij}} \equiv \sqrt{C_{ij}} \cdot A_{im}^{n_j - 1} \cdot \sqrt{A_{ij}}$.

Finally we move $\omega_n \to \omega_n - n_n \pi$. It crosses 3 past light cones and hence as per rule (1) does nothing. Notice that the starting configuration is obtained from the Euclidean sheet by making a single clockwise monodromy around $z_{ij}$.

$$C_{ij} \cdot A_{ij} \cdot \sqrt{A_{in}} \cdot C_{im}^{n_m} \cdot \sqrt{C_{in}} \cdot \sqrt{C_{ij}} \cdot A_{im}^{n_j - 1} \cdot \sqrt{A_{ij}}$$
$$= \sqrt{C_{ij}} \cdot \sqrt{C_{im}} \cdot C_{im}^{n_m} \cdot \sqrt{A_{im}} \cdot A_{im}^{n_j - 1} \cdot \sqrt{A_{ij}}$$
$$= \sqrt{C_{ij}} \cdot C_{im}^{n_m - n_j + 1} \cdot \sqrt{A_{ij}}.$$

So in this configuration we start from the Euclidean sheet and first do a half-clockwise monodromy around $z_{ij}$, then do $(n_m - n_j + 1)$ clockwise circles around $z_{im}$ followed by a half-anticlockwise monodromy around $z_{ij}$, i.e., $\sqrt{C_{ij}} \cdot C_{im}^{n_m - n_j + 1} \cdot \sqrt{A_{ij}}$.

Time reversal maps this operation to itself. As above, the action of time reversal is the interchange $(m, n) \leftrightarrow (i, j)$ together with all $n's$ flipping sign. This combined operation leaves our final answer for the monodromy unchanged, as we had expected.

3. $n_i > n_m \geq n_n \geq n_j$

In this case we first move $\omega_i \to \omega_i - \pi$. It crosses 2 past and 1 future light cones which according to rule (3) of §6.2 gives $A_{ij}$. We then move $\omega_i \to \omega_i - (n_i - 1)\pi$ which crosses 3 future light cones and according to rule (1) gives no monodromy. We then move $\omega_m \to \omega_m - n_m \pi$. It crosses future, past, future lightcone configuration which according to rule (2) of §6.2 gives $\sqrt{A_{jm}} \cdot C_{im}^{n_m} \cdot \sqrt{C_{jm}} \equiv \sqrt{A_{in}} \cdot C_{im}^{n_m} \cdot \sqrt{C_{in}}$.

We then move $\omega_n \to \omega_n - n_n \pi$. It crosses past, future, past lightcone configuration which as per rule (3) gives $\sqrt{C_{ij}} \cdot A_{im}^{n_n} \cdot \sqrt{A_{ij}}$.

Finally we move $\omega_j \to \omega_j - n_j \pi$. It crosses 3 past light cones and hence as per rule (1) does nothing. Notice that the starting configuration is obtained from the Euclidean sheet by making a single clockwise monodromy around $z_{ij}$.

$$C_{ij} \cdot A_{ij} \cdot \sqrt{A_{in}} \cdot C_{im}^{n_m} \cdot \sqrt{C_{in}} \cdot \sqrt{C_{ij}} \cdot A_{im}^{n_n} \cdot \sqrt{A_{ij}}$$
$$= \sqrt{C_{ij}} \cdot \sqrt{C_{im}} \cdot C_{im}^{n_m} \cdot \sqrt{A_{im}} \cdot A_{im}^{n_n} \cdot \sqrt{A_{ij}}$$
$$= \sqrt{C_{ij}} \cdot C_{im}^{n_m - n_n} \cdot \sqrt{A_{ij}}.$$

So in this configuration we start from the Euclidean sheet and first do a half-clockwise monodromy around $z_{ij}$, then do $(n_m - n_n)$ clockwise circles around $z_{im}$ followed by a half-anticlockwise monodromy around $z_{ij}$, i.e., $\sqrt{C_{ij}} \cdot C_{im}^{n_m - n_n} \cdot \sqrt{A_{ij}}$.

Time reversal maps this set of moves to the case studied in §4. As above , time reversal consists of the interchange $(m, n) \leftrightarrow (i, j)$ together with all $n's$ flipping sign. This combined operation indeed maps our final result for the monodromy to that in §4 as expected.

4. $n_m \geq n_i \geq n_j > n_n$

In this case we first move $\omega_m \to \omega_m - n_m \pi$. It crosses 3 future light cones which according to rule (1) of §6.2 gives $\phi$. We then move $\omega_i \to \omega_i - \pi$. It crosses 2 past and 1 future

light cones which as per rule (3) of §6.2 gives $A_{ij}$. We then move $\omega_i \to \omega_i - (n_i - 1)\pi$. It crosses future, past, future lightcone configuration which according to rule (2) gives $\sqrt{A_{in}} \cdot C_{im}^{n_i-1} \cdot \sqrt{C_{in}}$.

We then move $\omega_j \to \omega_j - \pi$. It crosses 3 past light cones which as per rule (1) gives $\phi$. We then move $\omega_j \to \omega_j - (n_j - 1)\pi$. It crosses past, future, past lightcone configuration which as per rule (3) gives $\sqrt{C_{ij}} \cdot A_{jn}^{n_j-1} \cdot \sqrt{A_{ij}} \equiv \sqrt{C_{ij}} \cdot A_{im}^{n_j-1} \cdot \sqrt{A_{ij}}$.

Finally we move $\omega_n \to \omega_n - n_n\pi$. It crosses 3 past light cones and hence as per rule (1) does nothing. Notice that the starting configuration is obtained from the Euclidean sheet by making a single clockwise monodromy around $z_{ij}$.

$$
C_{ij} \cdot A_{ij} \cdot \sqrt{A_{in}} \cdot C_{im}^{n_i-1} \cdot \sqrt{C_{in}} \cdot \sqrt{C_{ij}} \cdot A_{im}^{n_j-1} \cdot \sqrt{A_{ij}}
$$
$$
= \sqrt{C_{ij}} \cdot \sqrt{C_{im}} \cdot C_{im}^{n_i-1} \cdot \sqrt{A_{im}} \cdot A_{im}^{n_j-1} \cdot \sqrt{A_{ij}}
$$
$$
= \sqrt{C_{ij}} \cdot C_{im}^{n_i-n_j} \cdot \sqrt{A_{ij}}.
$$

So in this configuration we start from the Euclidean sheet and first do a half-clockwise monodromy around $z_{ij}$, then do $(n_i - n_j)$ clockwise circles around $z_{im}$ followed by a half-anticlockwise monodromy around $z_{ij}$, i.e., $\sqrt{C_{ij}} \cdot C_{im}^{n_i-n_j} \cdot \sqrt{A_{ij}}$.

5. $n_m \geq n_i > n_n \geq n_j$

In this case we first move $\omega_m \to \omega_m - n_m\pi$. It crosses 3 future light cones which according to rule (1) of §6.2 gives $\phi$. We then move $\omega_i \to \omega_i - \pi$. It crosses 2 past and 1 future light cones which as per rule (3) of §6.2 gives $A_{ij}$. We then move $\omega_i \to \omega_i - (n_i - 1)\pi$. It crosses future, past, future lightcone configuration which according to rule (2) of §6.2 gives $\sqrt{A_{in}} \cdot C_{im}^{n_i-1} \cdot \sqrt{C_{in}}$.

We then move $\omega_n \to \omega_n - n_n\pi$. It crosses past, future, past lightcone configuration which as per rule (3) gives $\sqrt{C_{mn}} \cdot A_{jn}^{n_n} \cdot \sqrt{A_{mn}} \equiv \sqrt{C_{ij}} \cdot A_{im}^{n_n} \cdot \sqrt{A_{ij}}$.

Finally we move $\omega_j \to \omega_j - n_j\pi$. It crosses 3 past light cones and hence as per rule (1) does nothing. Notice that the starting configuration is obtained from the Euclidean sheet by making a single clockwise monodromy around $z_{ij}$.

$$
C_{ij} \cdot A_{ij} \cdot \sqrt{A_{in}} \cdot C_{im}^{n_i-1} \cdot \sqrt{C_{in}} \cdot \sqrt{C_{ij}} \cdot A_{im}^{n_n} \cdot \sqrt{A_{ij}}
$$
$$
= \sqrt{C_{ij}} \cdot \sqrt{C_{im}} \cdot C_{im}^{n_i-1} \cdot \sqrt{A_{im}} \cdot A_{im}^{n_n} \cdot \sqrt{A_{ij}}
$$
$$
= \sqrt{C_{ij}} \cdot C_{im}^{n_i-n_n-1} \cdot \sqrt{A_{ij}}.
$$

So in this configuration we start from the Euclidean sheet and first do a half-clockwise monodromy around $z_{ij}$, then do $(n_i - n_n - 1)$ clockwise circles around $z_{im}$ followed by a half-anticlockwise monodromy around $z_{ij}$, i.e., $\sqrt{C_{ij}} \cdot C_{im}^{n_i-n_n-1} \cdot \sqrt{A_{ij}}$.

Time reversal maps this operation to itself. As above, the action of time reversal is the interchange $(m, n) \leftrightarrow (i, j)$ together with all $n's$ flipping sign. This combined operation leaves our final answer for the monodromy unchanged, as we had expected.

6. $n_m \geq n_n \geq n_i \geq n_j$

In this case we first move $\omega_m \to \omega_m - n_m\pi$. It crosses 3 future light cones which according to rule (1) of §6.2 gives $\phi$. We then move $\omega_n \to \omega_n - n_n\pi$. It crosses 1 past and 2 future light cones which as per rule (2) of §6.2 gives $C_{mn}^{n_n} \equiv C_{ij}^{n_n}$.

We then move $\omega_i \to \omega_i - n_i\pi$. It crosses 2 past and 1 future light cones which as per rule (3) gives $A_{ij}^{n_i}$. Finally we move $\omega_j \to \omega_j - n_j\pi$. It crosses 3 past light cones and hence as per rule (1) does nothing.

Notice that the starting configuration is obtained from the Euclidean sheet by making a single clockwise monodromy around $z_{ij}$. So in this configuration we start from the Euclidean sheet and do $(n_n - n_i + 1)$ clockwise circles around $z_{ij}$, i.e., $C_{ij}^{n_n - n_i + 1}$.

Time reversal maps this operation to itself. As above, the action of time reversal is the interchange $(m, n) \leftrightarrow (i, j)$ together with all $n's$ flipping sign. This combined operation leaves our final answer for the monodromy unchanged, as we had expected.

### F.8 Regge configurations

In the diagram Fig. 14b, we listed a second configuration that lies within the Minkowski diamond, has $z = \bar{z}$, but does not lie on the Euclidean Sheet. In fact this configuration lies on the "Regge sheet" (obtained starting from the Euclidean sheet and performing a single anticlockwise monodromy around $z_{ij}$).

In this subsection we describe the sheet structure of all configurations that can be brought to such a Regge configuration by making $\pi$ shifts of the $\omega$ coordinates of the various insertions. As in the previous subsection, it is useful to fix on a convention. As in Fig. §15, the insertions corresponding to one pair of timelike separated operators as $i$ and $j$ with $j$ to the future of $i$, and denote the second pair of timelike separated operators by $m$ and $n$ (with $n$ to the future of $m$).

Starting with this configuration, we then move to new configurations by making the shifts $\omega_a \to \omega_a - n_a\pi$ for $a = i, j, m, n$. There are inequivalent cases which we take up in turn. The 24 possible $n_a$ orderings are related to each other under the $\mathbb{Z}_2$ symmetry operation $(i, j) \leftrightarrow (m, n)$. Consequently we need to consider 12 inequivalent orderings of the $n_a$. 12 different cases are (here $\sim$ means related by $(i, j) \leftrightarrow (m, n)$)

$$
\begin{aligned}
&1.\ (ijmn) \sim (mnij), &&2.\ (ijnm) \sim (mnji), \\
&3.\ (imjn) \sim (minj), &&4.\ (imnj) \sim (mijn), \\
&5.\ (injm) \sim (mjni), &&6.\ (inmj) \sim (mjin), \\
&7.\ (jimn) \sim (nmij), &&8.\ (jinm) \sim (nmji), \\
&9.\ (jmin) \sim (nimj), &&10.\ (jmni) \sim (nijm), \\
&11.\ (jnim) \sim (njmi), &&12.\ (jnmi) \sim (njim).
\end{aligned}
\tag{F.2}
$$

Time reversal in this configuration consists of the interchange $(i, m) \leftrightarrow (j, n)$ together with all $n_a \to -n_a$ (reversal of the (abcd) ordering). It is not difficult to convince oneself that under time reversal (and modulo the $Z_2$ interchange)

$$
\begin{aligned}
&1.\ (ijmn) \to (mnij) = case\ 1, \\
&2.\ (ijnm) \to (nmij) = case\ 7, \\
&3.\ (imjn) \to (minj) = case\ 3, \\
&4.\ (imnj) \to (imnj) = case\ 4, \\
&5.\ (injm) \to (nimj) = case\ 9, \\
&6.\ (inmj) \to (injm) = case\ 6, \\
&7.\ (jimn) \to (mnji) = case\ 2, \\
&8.\ (jinm) \to (nmji) = case\ 8, \\
&9.\ (jmin) \to (injm) = case\ 5,
\end{aligned}
\tag{F.3}
$$

$$10.\ (jmni) \to (jmni) = case\ 10\,,$$
$$11.\ (jnim) \to (njmi) = case\ 11\,,$$
$$12.\ (jnmi) \to (jnmi) = case\ 12\,.$$

1. $n_i > n_j \geq n_m > n_n$

   In this case we first move $\omega_i \to \omega_i - \pi$. It crosses 1 past and 2 future light cones which according to rule (2) of §6.2 gives $C_{ij}$. We then move $\omega_i \to \omega_i - (n_i - 1)\pi$ which crosses 3 future light cones and according to rule (1) gives no monodromy. We then move $\omega_j \to \omega_j - n_j \pi$. It crosses 2 future and 1 past light cones which according to rule (2) of §6.2 gives $C_{ij}^{n_j}$.

   We then move $\omega_m \to \omega_m - \pi$. It crosses 3 past light cones which as per rule (1) gives $\phi$. We then move $\omega_m \to \omega_m - (n_m - 1)\pi$. It crosses 1 future and 2 past light cones which according to rule (3) gives $A_{mn}^{n_m - 1} \equiv A_{ij}^{n_m - 1}$. Finally we move $\omega_n \to \omega_n - n_n \pi$. It crosses 3 past light cones and hence as per rule (1) does nothing.

   Notice that the starting configuration is obtained from the Euclidean sheet by making a single anticlockwise monodromy around $z_{ij}$. So in this configuration we start from the Euclidean sheet and do $(n_j - n_m + 1)$ clockwise circles around $z_{ij}$, i.e., $C_{ij}^{n_j - n_m + 1}$.

2. $n_i > n_j \geq n_n \geq n_m$

   In this case we first move $\omega_i \to \omega_i - \pi$. It crosses 1 past and 2 future light cones which according to rule (2) of §6.2 gives $C_{ij}$. We then move $\omega_i \to \omega_i - (n_i - 1)\pi$ which crosses 3 future light cones and according to rule (1) gives no monodromy. We then move $\omega_j \to \omega_j - n_j \pi$. It crosses 2 future and 1 past light cones which according to rule (2) of §6.2 gives $C_{ij}^{n_j}$.

   We then move $\omega_n \to \omega_n - n_n \pi$. It crosses 2 past and 1 future light cones which according to rule (3) gives $A_{mn}^{n_n} \equiv A_{ij}^{n_n}$. Finally we move $\omega_m \to \omega_m - n_m \pi$. It crosses 3 past light cones and hence as per rule (1) does nothing.

   Notice that the starting configuration is obtained from the Euclidean sheet by making a single anticlockwise monodromy around $z_{ij}$. So in this configuration we start from the Euclidean sheet and do $(n_j - n_n)$ clockwise circles around $z_{ij}$, i.e., $C_{ij}^{n_j - n_n}$.

3. $n_i \geq n_m \geq n_j \geq n_n :\ n_i > n_j$ **and** $n_m > n_n$

   In this case we first move $\omega_i \to \omega_i - \pi$. It crosses 1 past and 2 future light cones which according to rule (2) of §6.2 gives $C_{ij}$. We then move $\omega_i \to \omega_i - (n_i - 1)\pi$ which crosses 3 future light cones and according to rule (1) gives no monodromy. We then move $\omega_m \to \omega_m - \pi$. It crosses 2 past and 1 future light cones which as per rule (3) gives $A_{jm} \equiv A_{in}$.

   We then move $\omega_m \to \omega_m - (n_m - 1)\pi$. It crosses future, past, future lightcone configuration which according to rule (2) of §6.2 gives $\sqrt{A_{mn}} \cdot C_{mi}^{n_m - 1} \cdot \sqrt{C_{mn}} \equiv \sqrt{A_{ij}} \cdot C_{im}^{n_m - 1} \cdot \sqrt{C_{ij}}$.

   We then move $\omega_j \to \omega_j - n_j \pi$. It crosses past, future, past lightcone configuration which according to rule (3) gives $\sqrt{C_{jm}} \cdot A_{jn}^{n_j} \cdot \sqrt{A_{jm}} \equiv \sqrt{C_{in}} \cdot A_{im}^{n_j} \cdot \sqrt{A_{in}}$.

   Finally we move $\omega_n \to \omega_n - n_n \pi$. It crosses 3 past light cones and hence as per rule (1) does nothing. Notice that the starting configuration is obtained from the Euclidean

sheet by making a single anticlockwise monodromy around $z_{ij}$.

$$
\begin{aligned}
A_{ij} \cdot C_{ij} \cdot A_{in} & \cdot \sqrt{A_{ij}} \cdot C_{im}^{n_m-1} \cdot \sqrt{C_{ij}} \cdot \sqrt{C_{in}} \cdot A_{im}^{n_j} \cdot \sqrt{A_{in}} \\
&= A_{in} \cdot \sqrt{C_{in}} \cdot \sqrt{C_{im}} \cdot C_{im}^{n_m-1} \cdot \sqrt{A_{im}} \cdot A_{im}^{n_j} \cdot \sqrt{C_{im}} \cdot \sqrt{C_{ij}} \\
&= \sqrt{A_{in}} \cdot C_{im}^{n_m-n_j-\frac{1}{2}} \cdot \sqrt{C_{ij}} \\
&= \sqrt{C_{ij}} \cdot \sqrt{C_{im}} \cdot C_{im}^{n_m-n_j-\frac{1}{2}} \cdot \sqrt{C_{ij}} \\
&= \sqrt{C_{ij}} \cdot C_{im}^{n_m-n_j} \cdot \sqrt{C_{ij}}.
\end{aligned}
$$

So in this configuration we start from the Euclidean sheet and first do a half-clockwise monodromy around $z_{ij}$, then do $(n_m - n_j)$ clockwise circles around $z_{im}$ followed by a half-clockwise monodromy around $z_{ij}$, i.e., $\sqrt{C_{ij}} \cdot C_{im}^{n_m-n_j} \cdot \sqrt{C_{ij}}$.

4. $n_i \geq n_m > n_n \geq n_j$

   In this case we first move $\omega_i \to \omega_i - \pi$. It crosses 1 past and 2 future light cones which according to rule (2) of §6.2 gives $C_{ij}$. We then move $\omega_i \to \omega_i - (n_i - 1)\pi$ which crosses 3 future light cones and according to rule (1) gives no monodromy.

   We then move $\omega_m \to \omega_m - \pi$. It crosses 2 past and 1 future light cones which as per rule (3) gives $A_{jm} \equiv A_{in}$. We then move $\omega_m \to \omega_m - (n_m - 1)\pi$. It crosses future, past, future lightcone configuration which according to rule (2) of §6.2 gives $\sqrt{A_{mn}} \cdot C_{mi}^{n_m-1} \cdot \sqrt{C_{mn}} \equiv \sqrt{A_{ij}} \cdot C_{im}^{n_m-1} \cdot \sqrt{C_{ij}}$.

   We then move $\omega_n \to \omega_n - n_n\pi$. It crosses past, future, past lightcone configuration which according to rule (3) gives $\sqrt{C_{ni}} \cdot A_{nj}^{n_n} \cdot \sqrt{A_{ni}} \equiv \sqrt{C_{in}} \cdot A_{im}^{n_n} \cdot \sqrt{A_{in}}$.

   Finally we move $\omega_j \to \omega_j - n_j\pi$. It crosses 3 past light cones and hence as per rule (1) does nothing. Notice that the starting configuration is obtained from the Euclidean sheet by making a single anticlockwise monodromy around $z_{ij}$.

$$
\begin{aligned}
A_{ij} \cdot C_{ij} \cdot A_{in} & \cdot \sqrt{A_{ij}} \cdot C_{im}^{n_m-1} \cdot \sqrt{C_{ij}} \cdot \sqrt{C_{in}} \cdot A_{im}^{n_n} \cdot \sqrt{A_{in}} \\
&= A_{in} \cdot \sqrt{C_{in}} \cdot \sqrt{C_{im}} \cdot C_{im}^{n_m-1} \cdot \sqrt{A_{im}} \cdot A_{im}^{n_n} \cdot \sqrt{C_{im}} \cdot \sqrt{C_{ij}} \\
&= \sqrt{A_{in}} \cdot C_{im}^{n_m-n_n-\frac{1}{2}} \cdot \sqrt{C_{ij}} \\
&= \sqrt{C_{ij}} \cdot \sqrt{C_{im}} \cdot C_{im}^{n_m-n_n-\frac{1}{2}} \cdot \sqrt{C_{ij}} \\
&= \sqrt{C_{ij}} \cdot C_{im}^{n_m-n_n} \cdot \sqrt{C_{ij}}.
\end{aligned}
$$

   So in this configuration we start from the Euclidean sheet and first do a half-clockwise monodromy around $z_{ij}$, then do $(n_m - n_n)$ clockwise circles around $z_{im}$ followed by a half-clockwise monodromy around $z_{ij}$, i.e., $\sqrt{C_{ij}} \cdot C_{im}^{n_m-n_n} \cdot \sqrt{C_{ij}}$.

5. $n_i \geq n_n \geq n_j \geq n_m : n_i > n_j$

   In this case we first move $\omega_i \to \omega_i - \pi$. It crosses 1 past and 2 future light cones which according to rule (2) of §6.2 gives $C_{ij}$. We then move $\omega_i \to \omega_i - (n_i - 1)\pi$ which crosses 3 future light cones and according to rule (1) gives no monodromy. We then move $\omega_n \to \omega_n - n_n\pi$. It crosses 1 past and 2 future light cones which according to rule (2) of §6.2 gives $C_{in}^{n_n}$.

   We then move $\omega_j \to \omega_j - n_j\pi$. It crosses 1 future and 2 past light cones which according to rule (3) gives $A_{jm}^{n_j} \equiv A_{in}^{n_j}$. Finally we move $\omega_m \to \omega_m - n_m\pi$. It crosses 3 past light cones and hence as per rule (1) does nothing.

Notice that the starting configuration is obtained from the Euclidean sheet by making a single anticlockwise monodromy around $z_{ij}$. So in this configuration we start from the Euclidean sheet and do $(n_n - n_j)$ clockwise circles around $z_{in}$, i.e., $C_{in}^{n_n - n_j}$.

6. $n_i \geq n_n \geq n_m \geq n_j : n_i > n_j$

In this case we first move $\omega_i \to \omega_i - \pi$. It crosses 1 past and 2 future light cones which according to rule (2) of §6.2 gives $C_{ij}$. We then move $\omega_i \to \omega_i - (n_i - 1)\pi$ which crosses 3 future light cones and according to rule (1) gives no monodromy. We then move $\omega_n \to \omega_n - n_n \pi$. It crosses 1 past and 2 future light cones which according to rule (2) of §6.2 gives $C_{in}^{n_n}$.

We then move $\omega_m \to \omega_m - n_m \pi$. It crosses 2 past and 1 future light cones which according to rule (3) gives $A_{jm}^{n_m} \equiv A_{in}^{n_m}$. Finally we move $\omega_j \to \omega_j - n_j \pi$. It crosses 3 past light cones and hence as per rule (1) does nothing.

Notice that the starting configuration is obtained from the Euclidean sheet by making a single anticlockwise monodromy around $z_{ij}$. So in this configuration we start from the Euclidean sheet and do $(n_n - n_m)$ clockwise circles around $z_{in}$, i.e., $C_{in}^{n_n - n_m}$.

7. $n_j \geq n_i \geq n_m > n_n$

In this case we first move $\omega_j \to \omega_j - n_j \pi$. It crosses 3 future light cones and according to rule (1) of §6.2 gives no monodromy. We then move $\omega_i \to \omega_i - n_i \pi$. It crosses 1 past and 2 future light cones which according to rule (2) of §6.2 gives $C_{ij}^{n_i}$.

We then move $\omega_m \to \omega_m - \pi$. It crosses 3 past light cones which as per rule (1) gives $\phi$. We then move $\omega_m \to \omega_m - (n_m - 1)\pi$. It crosses 1 future and 2 past light cones which according to rule (3) of §6.2 gives $A_{mn}^{n_m - 1} \equiv A_{ij}^{n_m - 1}$. Finally we move $\omega_n \to \omega_n - n_n \pi$. It crosses 3 past light cones and hence as per rule (1) does nothing.

Notice that the starting configuration is obtained from the Euclidean sheet by making a single anticlockwise monodromy around $z_{ij}$. So in this configuration we start from the Euclidean sheet and do $(n_i - n_m)$ clockwise circles around $z_{ij}$, i.e., $C_{ij}^{n_i - n_m}$.

8. $n_j \geq n_i \geq n_n \geq n_m$

In this case we first move $\omega_j \to \omega_j - n_j \pi$. It crosses 3 future light cones and according to rule (1) of §6.2 gives no monodromy. We then move $\omega_i \to \omega_i - n_i \pi$. It crosses 1 past and 2 future light cones which according to rule (2) of §6.2 gives $C_{ij}^{n_i}$.

We then move $\omega_n \to \omega_n - n_n \pi$. It crosses 2 past and 1 future light cones which according to rule (3) of §6.2 gives $A_{mn}^{n_n} \equiv A_{ij}^{n_n}$. Finally we move $\omega_m \to \omega_m - n_m \pi$. It crosses 3 past light cones and hence as per rule (1) does nothing.

Notice that the starting configuration is obtained from the Euclidean sheet by making a single anticlockwise monodromy around $z_{ij}$. So in this configuration we start from the Euclidean sheet and do $(n_i - n_n - 1)$ clockwise circles around $z_{ij}$, i.e., $C_{ij}^{n_i - n_n - 1}$.

9. $n_j \geq n_m \geq n_i \geq n_n : n_m > n_n$

In this case we first move $\omega_j \to \omega_j - n_j \pi$. It crosses 3 future light cones and according to rule (1) of §6.2 gives no monodromy. We then move $\omega_m \to \omega_m - \pi$. It crosses past, future, past lightcone configuration which according to rule (3) of §6.2 gives $\sqrt{C_{mn}} \cdot A_{im} \cdot \sqrt{A_{mn}} \equiv \sqrt{C_{ij}} \cdot A_{im} \cdot \sqrt{A_{ij}}$.

We then move $\omega_m \to \omega_m - (n_m - 1)\pi$. It crosses 2 future and 1 past light cones which according to rule (2) of §6.2 gives $C_{jm}^{n_m - 1} \equiv C_{in}^{n_m - 1}$.

We then move $\omega_i \to \omega_i - n_i \pi$. It crosses 2 past and 1 future light cones which according to rule (3) of §6.2 gives $A_{in}^{n_i}$. Finally we move $\omega_n \to \omega_n - n_n \pi$. It crosses 3 past light cones and hence as per rule (1) does nothing.

Notice that the starting configuration is obtained from the Euclidean sheet by making a single anticlockwise monodromy around $z_{ij}$.

$$
\begin{aligned}
A_{ij} \cdot \sqrt{C_{ij}} \cdot A_{im} \cdot & \sqrt{A_{ij}} \cdot C_{in}^{n_m-1} \cdot A_{in}^{n_i} \\
&= \sqrt{A_{ij}} \cdot A_{im} \cdot \sqrt{C_{im}} \cdot \sqrt{C_{in}} \cdot C_{in}^{n_m-n_i-1} \\
&= \sqrt{A_{ij}} \cdot \sqrt{A_{im}} \cdot C_{in}^{n_m-n_i-\frac{1}{2}} \\
&= \sqrt{C_{in}} \cdot C_{in}^{n_m-n_i-\frac{1}{2}} \\
&= C_{in}^{n_m-n_i} .
\end{aligned}
$$

So in this configuration we start from the Euclidean sheet and do $(n_m - n_i)$ clockwise circles around $z_{in}$, i.e., $C_{in}^{n_m-n_i}$.

10. $n_j \geq n_m > n_n \geq n_i$

In this case we first move $\omega_j \to \omega_j - n_j \pi$. It crosses 3 future light cones and according to rule (1) of §6.2 gives no monodromy. We then move $\omega_m \to \omega_m - \pi$. It crosses past, future, past lightcone configuration which according to rule (3) of §6.2 gives $\sqrt{C_{mn}} \cdot A_{im} \cdot \sqrt{A_{mn}} \equiv \sqrt{C_{ij}} \cdot A_{im} \cdot \sqrt{A_{ij}}$.

We then move $\omega_m \to \omega_m - (n_m - 1)\pi$. It crosses 2 future and 1 past light cones which according to rule (2) of §6.2 gives $C_{jm}^{n_m-1} \equiv C_{in}^{n_m-1}$.

We then move $\omega_n \to \omega_n - n_n \pi$. It crosses 1 future and 2 past light cones which according to rule (3) of §6.2 gives $A_{in}^{n_n}$. Finally we move $\omega_i \to \omega_i - n_i \pi$. It crosses 3 past light cones and hence as per rule (1) does nothing.

Notice that the starting configuration is obtained from the Euclidean sheet by making a single anticlockwise monodromy around $z_{ij}$.

$$
\begin{aligned}
A_{ij} \cdot \sqrt{C_{ij}} \cdot A_{im} \cdot & \sqrt{A_{ij}} \cdot C_{in}^{n_m-1} \cdot A_{in}^{n_n} \\
&= \sqrt{A_{ij}} \cdot A_{im} \cdot \sqrt{C_{im}} \cdot \sqrt{C_{in}} \cdot C_{in}^{n_m-n_n-1} \\
&= \sqrt{A_{ij}} \cdot \sqrt{A_{im}} \cdot C_{in}^{n_m-n_n-\frac{1}{2}} \\
&= \sqrt{C_{in}} \cdot C_{in}^{n_m-n_n-\frac{1}{2}} \\
&= C_{in}^{n_m-n_n} .
\end{aligned}
$$

So in this configuration we start from the Euclidean sheet and do $(n_m - n_n)$ clockwise circles around $z_{in}$, i.e., $C_{in}^{n_m-n_n}$.

11. $n_j \geq n_n \geq n_i \geq n_m$

In this case we first move $\omega_j \to \omega_j - n_j \pi$. It crosses 3 future light cones and according to rule (1) of §6.2 gives no monodromy. We then move $\omega_n \to \omega_n - n_n \pi$. It crosses future, past, future lightcone configuration which according to rule (2) of §6.2 gives $\sqrt{A_{ni}} \cdot C_{nj}^{n_n} \cdot \sqrt{C_{ni}} \equiv \sqrt{A_{in}} \cdot C_{im}^{n_n} \cdot \sqrt{C_{in}}$.

We then move $\omega_i \to \omega_i - n_i \pi$. It crosses past, future, past lightcone configuration which according to rule (3) of §6.2 gives $\sqrt{C_{ij}} \cdot A_{im}^{n_i} \cdot \sqrt{A_{ij}}$.

Finally we move $\omega_m \to \omega_m - n_m\pi$. It crosses 3 past light cones and hence as per rule (1) does nothing. Notice that the starting configuration is obtained from the Euclidean sheet by making a single anticlockwise monodromy around $z_{ij}$.

$$
\begin{aligned}
A_{ij} \cdot \sqrt{A_{in}} \cdot C_{im}^{n_n} \cdot \sqrt{C_{in}} \cdot \sqrt{C_{ij}} \cdot A_{im}^{n_i} \cdot \sqrt{A_{ij}} \\
= A_{ij} \cdot \sqrt{C_{ij}} \cdot \sqrt{C_{im}} \cdot C_{im}^{n_n} \cdot \sqrt{A_{im}} \cdot A_{im}^{n_i} \cdot \sqrt{A_{ij}} \\
= \sqrt{A_{ij}} \cdot C_{im}^{n_n - n_i} \cdot \sqrt{A_{ij}} \\
= \sqrt{C_{in}} \cdot \sqrt{C_{im}} \cdot C_{im}^{n_n - n_i} \cdot \sqrt{C_{im}} \cdot \sqrt{C_{in}} \\
= \sqrt{C_{in}} \cdot C_{im}^{n_n - n_i + 1} \cdot \sqrt{C_{in}}.
\end{aligned}
$$

So in this configuration we start from the Euclidean sheet and first do a half-clockwise monodromy around $z_{in}$, then do $(n_n - n_i + 1)$ clockwise circles around $z_{im}$ followed by a half-clockwise monodromy around $z_{in}$, i.e., $\sqrt{C_{in}} \cdot C_{im}^{n_n - n_i + 1} \cdot \sqrt{C_{in}}$.

12. $n_j \geq n_n \geq n_m \geq n_i$

In this case we first move $\omega_j \to \omega_j - n_j\pi$. It crosses 3 future light cones and according to rule (1) of §6.2 gives no monodromy. We then move $\omega_n \to \omega_n - n_n\pi$. It crosses future, past, future lightcone configuration which according to rule (2) of §6.2 gives $\sqrt{A_{ni}} \cdot C_{nj}^{n_n} \cdot \sqrt{C_{ni}} \equiv \sqrt{A_{in}} \cdot C_{im}^{n_n} \cdot \sqrt{C_{in}}$.

We then move $\omega_m \to \omega_m - n_m\pi$. It crosses past, future, past lightcone configuration which according to rule (3) of §6.2 gives $\sqrt{C_{mn}} \cdot A_{mi}^{n_m} \cdot \sqrt{A_{mn}} \equiv \sqrt{C_{ij}} \cdot A_{im}^{n_m} \cdot \sqrt{A_{ij}}$.

Finally we move $\omega_i \to \omega_i - n_i\pi$. It crosses 3 past light cones and hence as per rule (1) does nothing. Notice that the starting configuration is obtained from the Euclidean sheet by making a single anticlockwise monodromy around $z_{ij}$.

$$
\begin{aligned}
A_{ij} \cdot \sqrt{A_{in}} \cdot C_{im}^{n_n} \cdot \sqrt{C_{in}} \cdot \sqrt{C_{ij}} \cdot A_{im}^{n_m} \cdot \sqrt{A_{ij}} \\
= A_{ij} \cdot \sqrt{C_{ij}} \cdot \sqrt{C_{im}} \cdot C_{im}^{n_n} \cdot \sqrt{A_{im}} \cdot A_{im}^{n_m} \cdot \sqrt{A_{ij}} \\
= \sqrt{A_{ij}} \cdot C_{im}^{n_n - n_m} \cdot \sqrt{A_{ij}} \\
= \sqrt{C_{in}} \cdot \sqrt{C_{im}} \cdot C_{im}^{n_n - n_m} \cdot \sqrt{C_{im}} \cdot \sqrt{C_{in}} \\
= \sqrt{C_{in}} \cdot C_{im}^{n_n - n_m + 1} \cdot \sqrt{C_{in}}.
\end{aligned}
$$

So in this configuration we start from the Euclidean sheet and first do a half-clockwise monodromy around $z_{in}$, then do $(n_n - n_m + 1)$ clockwise circles around $z_{im}$ followed by a half-clockwise monodromy around $z_{in}$, i.e., $\sqrt{C_{in}} \cdot C_{im}^{n_n - n_m + 1} \cdot \sqrt{C_{in}}$.

## G  Tables summarizing the results of appendix F

Table 9: Single branch point towers with $q \geq -1$.

| $q \geq -1$ | Monodromy | Configuration | Condition |
|---|---|---|---|
| 1 | $C_{ij}^{n_i - n_n - 1}$ | Regge case 8 | $n_i \geq n_n$ |
| 2 | $C_{ij}^{n_n - n_i - 2}$ | Euclidean - F case 17 | $n_n > n_i$ |
| 3 | $C_{ij}^{n_n - n_i - 1}$ | Euclidean - D/E case 22 | $n_n \geq n_i, n_n > n_j$ |
| 4 | $C_{in}^{n_i - n_j - 1}$ | Euclidean - D/E case 21 | $n_i \geq n_j, n_n > n_j$ |

Table 10: Single branch point towers with $q \geq 0$.

| $q \geq 0$ | Monodromy | Configuration | Condition |
|---|---|---|---|
| 1 | $C_{ij}^{n_i - n_m}$ | Euclidean - A case 7 | $n_i \geq n_m$ |
| | | Euclidean - D/E case 13 | Same as above |
| | | Regge case 7 | Same as above |
| 2 | $C_{ij}^{n_i - n_n}$ | Euclidean - A case 8 | $n_i \geq n_n$ |
| 3 | $C_{ij}^{n_j - n_m - 1}$ | Scattering case 1 | $n_j > n_m$ |
| 4 | $C_{ij}^{n_j - n_m}$ | Euclidean - A case 1 | $n_j \geq n_m$ |
| | | Euclidean - B case 1 | Same as above |
| | | Euclidean - D/E case 1 | Same as above |
| | | Euclidean - F case 1 | Same as above |
| 5 | $C_{ij}^{n_j - n_n}$ | Euclidean - A case 2 | $n_j \geq n_n$ |
| | | Euclidean - B case 2 | Same as above |
| | | Regge case 2 | Same as above |
| 6 | $C_{ij}^{n_m - n_i - 1}$ | Euclidean - B case 23 | $n_m > n_i$ |
| | | Euclidean - C case 2 | Same as above |
| | | Euclidean - F case 23 | Same as above |
| 7 | $C_{ij}^{n_m - n_i}$ | Euclidean - A case 23 | $n_m \geq n_i$ |
| | | Euclidean - D/E case 24 | $n_m \geq n_i, n_m > n_j$ |
| 8 | $C_{ij}^{n_m - n_j}$ | Euclidean - A case 24 | $n_m \geq n_j$ |
| | | Euclidean - B case 24 | Same as above |
| | | Euclidean - C case 1 | Same as above |
| 9 | $C_{ij}^{n_n - n_i - 1}$ | Euclidean - B case 17 | $n_n > n_i$ |
| 10 | $C_{ij}^{n_n - n_i}$ | Euclidean - A case 17 | $n_n \geq n_i$ |
| 11 | $C_{ij}^{n_n - n_j - 1}$ | Euclidean - D/E case 10 | $n_n > n_j$ |
| | | Euclidean - F case 18 | Same as above |
| 12 | $C_{ij}^{n_n - n_j}$ | Euclidean - A case 18 | $n_n \geq n_j$ |
| | | Euclidean - B case 18 | Same as above |
| 13 | $C_{in}^{n_i - n_j}$ | Euclidean - A case 19 | $n_i \geq n_j$ |
| | | Euclidean - F case 19 | Same as above |
| 14 | $C_{in}^{n_i - n_m}$ | Euclidean - A case 20 | $n_i \geq n_m$ |
| | | Euclidean - D/E case 23 | $n_i \geq n_m, n_n > n_m$ |
| 15 | $C_{in}^{n_j - n_i - 1}$ | Euclidean - B case 15 | $n_j > n_i$ |
| 16 | $C_{in}^{n_j - n_i}$ | Euclidean - A case 15 | $n_j \geq n_i$ |
| 17 | $C_{in}^{n_j - n_n}$ | Euclidean - A case 16 | $n_j \geq n_n$ |
| | | Euclidean - B case 16 | Same as above |
| 18 | $C_{in}^{n_m - n_i - 1}$ | Euclidean - B case 9 | $n_m > n_i$ |
| | | Euclidean - F case 9 | Same as above |
| 19 | $C_{in}^{n_m - n_i}$ | Euclidean - A case 9 | $n_m \geq n_i$ |
| | | Euclidean - D/E case 14 | $n_m \geq n_i$ |
| | | Regge case 9 | $n_m \geq n_i, n_m > n_n$ |
| 20 | $C_{in}^{n_m - n_n}$ | Euclidean - A case 10 | $n_m \geq n_n$ |
| | | Euclidean - B case 10 | Same as above |
| | | Euclidean - D/E case 2 | Same as above |
| | | Euclidean - F case 10 | Same as above |
| 21 | $C_{in}^{n_n - n_j - 1}$ | Euclidean - D/E case 9 | $n_n > n_j$ |
| | | Euclidean - F case 5 | Same as above |
| 22 | $C_{in}^{n_n - n_j}$ | Euclidean - A case 5 | $n_n \geq n_j$ |
| | | Euclidean - B case 5 | Same as above |
| | | Regge case 5 | $n_n \geq n_j, n_i > n_j$ |
| 23 | $C_{in}^{n_n - n_m}$ | Euclidean - A case 6 | $n_n \geq n_m$ |
| | | Euclidean - B case 6 | Same as above |
| | | Euclidean - C case 4 | $n_n \geq n_m$ |
| | | Regge case 6 | $n_n \geq n_m, n_i > n_j$ |

Table 11: Single branch point towers with $q \geq 1$.

| $q \geq 1$ | Monodromy | Configuration | Condition |
|---|---|---|---|
| 1 | $C_{ij}^{n_i - n_m + 1}$ | Euclidean - B case 7 | $n_i \geq n_m$ |
| | | Euclidean - F case 7 | Same as above |
| 2 | $C_{ij}^{n_i - n_n + 1}$ | Euclidean - B case 8 | $n_i \geq n_n$ |
| | | Euclidean - D/E case 15 | Same as above |
| 3 | $C_{ij}^{n_j - n_m + 1}$ | Regge case 1 | $n_j \geq n_m$ |
| 4 | $C_{ij}^{n_j - n_n + 1}$ | Euclidean - D/E case 5 | $n_j \geq n_n$ |
| | | Euclidean - F case 2 | Same as above |
| 5 | $C_{ij}^{n_m - n_j}$ | Euclidean - D/E case 12 | $n_m > n_j$ |
| | | Euclidean - F case 24 | Same as above |
| 6 | $C_{ij}^{n_n - n_i + 1}$ | Scattering case 6 | $n_n \geq n_i$ |
| 7 | $C_{in}^{n_i - n_j + 1}$ | Euclidean - B case 19 | $n_i \geq n_j$ |
| 8 | $C_{in}^{n_i - n_m + 1}$ | Euclidean - B case 20 | $n_i \geq n_m$ |
| | | Euclidean - C case 3 | Same as above |
| | | Euclidean - F case 20 | Same as above |
| 9 | $C_{in}^{n_j - n_i}$ | Euclidean - F case 15 | $n_j > n_i$ |
| 10 | $C_{in}^{n_j - n_i + 1}$ | Euclidean - D/E case 16 | $n_j \geq n_i$ |
| 11 | $C_{in}^{n_j - n_n + 1}$ | Euclidean - D/E case 6 | $n_j \geq n_n$ |
| | | Euclidean - F case 16 | Same as above |
| 12 | $C_{in}^{n_m - n_n}$ | Regge case 10 | $n_m > n_n$ |
| 13 | $C_{in}^{n_n - n_m}$ | Euclidean - D/E case 11 | $n_n > n_m$ |
| | | Euclidean - F case 6 | Same as above |

Table 12: Single branch point towers with $q \geq 2$.

| $q \geq 2$ | Monodromy | Configuration | Condition |
|---|---|---|---|
| 1 | $C_{ij}^{n_i - n_n + 2}$ | Euclidean - F case 8 | $n_i \geq n_n$ |

Table 13: Double branch point towers with $q \geq 0$.

| $q \geq 0$ | Monodromy | Configuration | Condition |
|---|---|---|---|
| 1 | $\sqrt{C_{ij}} \cdot C_{im}^{n_i-n_j} \cdot \sqrt{A_{ij}}$ | Euclidean - A case 13 | $n_i \geq n_j$ |
| | | Scattering case 4 | Same as above |
| 2 | $\sqrt{C_{ij}} \cdot C_{im}^{n_i-n_n-1} \cdot \sqrt{A_{ij}}$ | Scattering case 5 | $n_i > n_n$ |
| 3 | $\sqrt{C_{ij}} \cdot C_{im}^{n_i-n_n} \cdot \sqrt{A_{ij}}$ | Euclidean - A case 14 | $n_i \geq n_n$ |
| 4 | $\sqrt{C_{ij}} \cdot C_{im}^{n_m-n_j} \cdot \sqrt{A_{ij}}$ | Euclidean - A case 3 | $n_m \geq n_j$ |
| | | Euclidean - B case 3 | Same as above |
| 5 | $\sqrt{C_{ij}} \cdot C_{im}^{n_m-n_n} \cdot \sqrt{A_{ij}}$ | Euclidean - A case 4 | $n_m \geq n_n$ |
| | | Euclidean - B case 4 | Same as above |
| | | Scattering case 3 | Same as above |
| 6 | $\sqrt{C_{in}} \cdot C_{im}^{n_j-n_i-1} \cdot \sqrt{A_{in}}$ | Euclidean - B case 21 | $n_j > n_i$ |
| 7 | $\sqrt{C_{in}} \cdot C_{im}^{n_j-n_i} \cdot \sqrt{A_{in}}$ | Euclidean - A case 21 | $n_j \geq n_i$ |
| 8 | $\sqrt{C_{in}} \cdot C_{im}^{n_j-n_m} \cdot \sqrt{A_{in}}$ | Euclidean - A case 22 | $n_j \geq n_m$ |
| | | Euclidean - B case 22 | Same as above |
| 9 | $\sqrt{C_{in}} \cdot C_{im}^{n_n-n_i-1} \cdot \sqrt{A_{in}}$ | Euclidean - B case 11 | $n_n > n_i$ |
| 10 | $\sqrt{C_{in}} \cdot C_{im}^{n_n-n_i} \cdot \sqrt{A_{in}}$ | Euclidean - A case 11 | $n_n \geq n_i$ |
| 11 | $\sqrt{C_{in}} \cdot C_{im}^{n_n-n_m} \cdot \sqrt{A_{in}}$ | Euclidean - A case 12 | $n_n \geq n_m$ |
| | | Euclidean - B case 12 | Same as above |
| 12 | $\sqrt{C_{ij}} \cdot C_{im}^{n_m-n_j} \cdot \sqrt{C_{ij}}$ | Regge case 3 | $n_m \geq n_j$ |
| 13 | $\sqrt{C_{ij}} \cdot C_{im}^{n_n-n_i-1} \cdot \sqrt{C_{ij}}$ | Euclidean - F case 11 | $n_n > n_i$ |
| 14 | $\sqrt{C_{ij}} \cdot C_{im}^{n_n-n_i} \cdot \sqrt{C_{ij}}$ | Euclidean - D/E case 18 | $n_n \geq n_i$ |
| 15 | $\sqrt{C_{in}} \cdot C_{im}^{n_i-n_j} \cdot \sqrt{C_{in}}$ | Euclidean - D/E case 17 | $n_i \geq n_j$ |

Table 14: Double branch point towers with $q \geq 1$.

| $q \geq 1$ | Monodromy | Configuration | Condition |
|---|---|---|---|
| 1 | $\sqrt{C_{ij}} \cdot C_{im}^{n_i - n_j + 1} \cdot \sqrt{A_{ij}}$ | Euclidean - B case 13 | $n_i \geq n_j$ |
| 2 | $\sqrt{C_{ij}} \cdot C_{im}^{n_i - n_n + 1} \cdot \sqrt{A_{ij}}$ | Euclidean - B case 14 | $n_i \geq n_n$ |
| 3 | $\sqrt{C_{ij}} \cdot C_{im}^{n_m - n_j + 1} \cdot \sqrt{A_{ij}}$ | Scattering case 2 | $n_m \geq n_j$ |
| 4 | $\sqrt{C_{ij}} \cdot C_{im}^{n_j - n_i} \cdot \sqrt{C_{ij}}$ | Euclidean - F case 21 | $n_j > n_i$ |
| 5 | $\sqrt{C_{ij}} \cdot C_{im}^{n_j - n_i + 1} \cdot \sqrt{C_{ij}}$ | Euclidean - D/E case 20 | $n_j \geq n_i$ |
| 6 | $\sqrt{C_{ij}} \cdot C_{im}^{n_j - n_m + 1} \cdot \sqrt{C_{ij}}$ | Euclidean - D/E case 8 | $n_j \geq n_m$ |
|   |   | Euclidean - F case 22 | Same as above |
| 7 | $\sqrt{C_{ij}} \cdot C_{im}^{n_m - n_n} \cdot \sqrt{C_{ij}}$ | Regge case 4 | $n_m > n_n$ |
| 8 | $\sqrt{C_{ij}} \cdot C_{im}^{n_n - n_m} \cdot \sqrt{C_{ij}}$ | Euclidean - D/E case 4 | $n_n > n_m$ |
|   |   | Euclidean - F case 12 | Same as above |
| 9 | $\sqrt{C_{in}} \cdot C_{im}^{n_i - n_j + 1} \cdot \sqrt{C_{in}}$ | Euclidean - F case 13 | $n_i \geq n_j$ |
| 10 | $\sqrt{C_{in}} \cdot C_{im}^{n_i - n_n + 1} \cdot \sqrt{C_{in}}$ | Euclidean - D/E case 19 | $n_i \geq n_n$ |
| 11 | $\sqrt{C_{in}} \cdot C_{im}^{n_m - n_j} \cdot \sqrt{C_{in}}$ | Euclidean - D/E case 3 | $n_m > n_j$ |
|   |   | Euclidean - F case 3 | Same as above |
| 12 | $\sqrt{C_{in}} \cdot C_{im}^{n_m - n_n + 1} \cdot \sqrt{C_{in}}$ | Euclidean - D/E case 7 | $n_m \geq n_n$ |
|   |   | Euclidean - F case 4 | Same as above |
| 13 | $\sqrt{C_{in}} \cdot C_{im}^{n_n - n_i + 1} \cdot \sqrt{C_{in}}$ | Regge case 11 | $n_n \geq n_i$ |
| 14 | $\sqrt{C_{in}} \cdot C_{im}^{n_n - n_m + 1} \cdot \sqrt{C_{in}}$ | Regge case 12 | $n_n \geq n_m$ |

Table 15: Double branch point towers with $q \geq 2$.

| $q \geq 2$ | Monodromy | Configuration | Condition |
|---|---|---|---|
| 1 | $\sqrt{C_{in}} \cdot C_{im}^{n_i - n_n + 2} \cdot \sqrt{C_{in}}$ | Euclidean - F case 14 | $n_i \geq n_n$ |

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
