# Peer review of "Monodromies of CFT correlators on the Lorentzian Cylinder"

_SciPost Physics, doi:SciPost Phys. 19, 162 (2025)_

## Round 1 · Referee Report · Anonymous (Referee 1) · 2025-8-2

Report

In this paper, the authors studied the branch structure of time-ordered four-point functions in CFTs. Such structures are usually quite intricate as there are infinitely many sheets for Lorentzian correlators and a complete understanding is still lacking. In the simplest case of $1+1$ dimensions, where the spacetime can be maximally analytically continued to a Lorentzian cylinder, the authors presented a complete classification of the sheets. They also constructively provided a physical interpretation for infinitely many of them. The techniques developed in this paper provide useful tools for systematically studying the multi-sheet structure of CFT correlators also in higher dimensions. This paper makes significant progress in understanding the multi-valued structure of Lorentzian correlators and I recommend the paper for publication on SciPost. I also have the following comments. The authors may see if they agree with these comments and choose to incorporate a subset of them.

  1. The paper has a large number of footnotes. This sometimes interrupts the logic flow of the paper and can be distracting for the reader. The authors may consider reorganizing and incorporating some of the footnotes into the main text (at least for the introductory section).

  2. Regarding new features in higher dimensions mentioned in the discussion, it seems the difference should become more evident when going to higher-point correlators. The number of cross ratios are different, i.e., $n-2$ v.s. $n(n-3)/2$ for $n\leq d+2$. Therefore, the analytic structure of correlators presumably will be very different.

  3. As an initial step of higher dimensional exploration, the authors considered $D$-functions in footnote 63 and Appendix $D$. However, this example might be a bit too special because $D$-functions are almost ignorant of the the AdS dimension (only appearing as an overall factor). This comment also applies to exchange Witten diagrams which can be written as sums of $D$-functions. Perhaps one should also look at one-loop correlators for the effect of higher dimensions to be potentially nontrivial.

  4. Related to this, the comment that the factorized structure (1) does not apply does not seem very accurate. It depends on whether one is okay with decomposing the higher dimensional correlator into two dimensional conformal blocks. The decomposition is always possible by using the dimensional reduction formula for conformal blocks. Then one would always find such a factorized structure.

  5. Another interesting extension of the authors' analysis is correlators in CFTs with defects. When the defect has co-dimension greater than 1, the defect two-point function also has two cross ratios which can be identified to $z$, $\bar{z}$ of the defect-free four-point function after a conformal map. The generalization can be made more or less straightforwardly (at least similar to defect-free four-point functions in higher dimensions), but also with new features introduced by the presence of the defect. Understanding the branch structure in this case is also very important as it is relevant for using the dispersion relations or applying the flat-space limit formula. The authors may optionally consider mentioning this in the outlook to further extend the scope of this paper.

Recommendation

Publish (meets expectations and criteria for this Journal)

  • validity: -
  • significance: -
  • originality: -
  • clarity: -
  • formatting: -
  • grammar: -

Author:  Abhishek Navhal  on 2025-10-14  [id 5924]

(in reply to Report 1 on 2025-08-02)
Category:
answer to question

We would like to thank the referee for their careful reading of the manuscript and their useful comments and suggestions, and for pointing out directions for further development. Below, we provide point-by-point responses to the referee's comments.

Referee Comment 1: Footnotes
``The paper has a large number of footnotes. This sometimes interrupts the logic flow of the paper and can be distracting for the reader. The authors may consider reorganizing and incorporating some of the footnotes into the main text (at least for the introductory section).''

Response: We thank the referee for this feedback. We have modified the file accordingly. In particular, we have incorporated most of the footnotes (from the Introduction) to the main text. In particular we have merged two of the larger footnotes as a separate subsection 1.5. We hope this is now satisfactory.

Referee Comment 2: Higher-dimensional Features
``Regarding new features in higher dimensions mentioned in the discussion, it seems the difference should become more evident when going to higher-point correlators. The number of cross ratios are different, i.e., $n-2$ v.s. $n(n-3)/2$ for $n \leq d+2$. Therefore, the analytic structure of correlators presumably will be very different.''

Response: We thank the referee for making this important point. We have added a paragraph in the Conclusion section (see the second last paragraph), and have thanked the referee for making this point.

Referee Comment 3: $D$-functions in Higher Dimensions
``As an initial step of higher dimensional exploration, the authors considered $D$-functions in footnote 63 and Appendix D. However, this example might be a bit too special because $D$-functions are almost ignorant of the AdS dimension (only appearing as an overall factor). This comment also applies to exchange Witten diagrams which can be written as sums of $D$-functions. Perhaps one should also look at one-loop correlators for the effect of higher dimensions to be potentially nontrivial.''

Response: We thank the referee for making this comment. We have added a paragraph at the end of appendix D:
"We parenthetically note that the $D$ function is the result of a correlator computed using contact diagram in the bulk of AdS (via the AdS/CFT correspondence). It would be interesting to perform an analysis similar to this appendix, on the more complicated analytic structures that arise out of exchange or loop computations in the bulk."

Referee Comment 4: Factorized Structure
``Related to this, the comment that the factorized structure (1) does not apply does not seem very accurate. It depends on whether one is okay with decomposing the higher dimensional correlator into two dimensional conformal blocks. The decomposition is always possible by using the dimensional reduction formula for conformal blocks. Then one would always find such a factorized structure.''

Response: We agree with the Referee that the comments on this footnote 48 are potentially confusing. In response to the referees comments, we have truncated the footnote, omitting the statement the comment about factorization.

Referee Comment 5: Defects and Future Extensions
``Another interesting extension of the authors' analysis is correlators in CFTs with defects. When the defect has co-dimension greater than 1, the defect two-point function also has two cross ratios which can be identified to $z$, $\bar{z}$ of the defect-free four-point function after a conformal map. The generalization can be made more or less straightforwardly (at least similar to defect-free four-point functions in higher dimensions), but also with new features introduced by the presence of the defect. Understanding the branch structure in this case is also very important as it is relevant for using the dispersion relations or applying the flat-space limit formula. The authors may optionally consider mentioning this in the outlook to further extend the scope of this paper.''

Response: We thank the referee for making this interesting point. We have added a short paragraph (third last paragraph in the conclusions section) making this point and thanking the referee for the suggestion.

Once again, we thank the referee for their detailed and thoughtful feedback which has improved our paper. We hope the paper is now suitable for publication.

Sincerely,
Suman Kundu, Shiraz Minwalla, Abhishek Navhal

Attachment:

SciPost_Reply1.pdf

---

## Round 1 · Referee Report · Anonymous (Referee 2) · 2025-9-2

Report

The paper discusses the branch structure of the Lorentzian four point function of a CFT on the Lorentzian cylinder. The Lorentzian four point function has an infinite number of branches and the authors provide physical interpretation of an infinite family of them. The paper is well written and reasonably easy to understand. I have enjoyed reading the paper. The technique they have developed is interesting and useful. I recommend the paper for publication in SciPost.

I have some comments which the authors may or may not choose to address in this paper.

1) This is a very interesting mathematical problem in its own right. However, it will be good if the authors can add few sentences to motivate this from a physical point of view. I have a feeling that this is somewhat lacking in the paper.

2) In section-2 that authors show that "path-independence" leads to the condition that the spin of all the inserted operators should be integers. This condition can also be obtained from the analysis of the Euclidean CFT correlation functions. Do the authors expect that a similar analysis of higher point Lorentzian correlators will give rise to other constraints on the CFT data which may not be (easily) accessible from Euclidean CFT?

Recommendation

Publish (meets expectations and criteria for this Journal)

  • validity: -
  • significance: -
  • originality: -
  • clarity: -
  • formatting: -
  • grammar: -

Author:  Abhishek Navhal  on 2025-10-14  [id 5925]

(in reply to Report 2 on 2025-09-02)
Category:
answer to question

We would like to thank the referee for their careful reading of the manuscript and their useful comments and suggestions, and for pointing out directions for further development. Below, we provide point-by-point responses to the referee's comments.

Referee Comment 1
``This is a very interesting mathematical problem in its own right. However, it will be good if the authors can add few sentences to motivate this from a physical point of view. I have a feeling that this is somewhat lacking in the paper.''

Response: We thank the referee for making this point. We have added the following lines at the end of the second and third paragraphs of section 1.3 (in the introduction) to highlight some of the physical implications of our results.

Line added at the end of the second para in section 1.3:
{As correlation functions capture the response of a theory to sources, our results are needed (together, of course, with knowledge of the correlators as analytic function of $z$ and ${\bar z}$), to determine the physical response of a CFT on $S^1 \times $ time to arbitrary sources as a function of angle and time. }

Line added at the end of the third para in section 1.3:
{It is physically interesting that several causally distinct configurations sometimes land on the same sheet of the correlator. This tells us that different (and symmetry unrelated) physical experiments sometimes have identical answers. We leave the interesting problem of understanding this observation from a physical viewpoint to future work. }

We hope this addition goes some way to ameliorating the referee's concern on this point.

Referee Comment 2
`` In section-2 that authors show that "path-independence" leads to the condition that the spin of all the inserted operators should be integers. This condition can also be obtained from the analysis of the Euclidean CFT correlation functions. Do the authors expect that a similar analysis of higher point Lorentzian correlators will give rise to other constraints on the CFT data which may not be (easily) accessible from Euclidean CFT?''

Response: We thank the referee for this important question. We had initially hoped it would turn out that the physical requirement - that different paths leading to the same final insertion location must yield the same final correlator - would lead to nontrivial new constraints on four-point functions. However, we regard our proof that four-point functions are automatically path independent (as long as all insertions have integer spins) as negating our initial hope; path independence imposes no nontrivial constraints on four-point functions. While it is possible that one might encounter a surprise while investigating higher point functions, we no longer expect this to be the case.

In response to this question, we have added the following new statement at the end of subsection 1.2:
``It is also a disappointment because it tells us that the requirement of `path independence' is automatic, and imposes no new general constraints on four-point functions of operators with integer spins.''

Once again, we thank the referee for their thoughtful feedback which has improved our paper. We hope the paper is now suitable for publication.

Sincerely,
Suman Kundu, Shiraz Minwalla, Abhishek Navhal

Attachment:

SciPost_Reply2.pdf

---

## Round 1 · Referee Report · Anonymous (Referee 3) · 2025-9-10

Report

The paper deals with understanding the branch structure of Lorentzian correlators in conformal field theories, in particular four point correlation functions.

The paper is extremely interesting and it presents a method to study properties of Lorentzian correlators in a solid framework. I believe that the study of higher dimensional CFTs with these techniques can shed light on pressing questions in the context of holographic theories.

The paper deserves to be published in SciPost, I have some comments that the authors can address:

1) How is it possible to use crossing symmetry in this context? In particular, can this help to understand the analytic properties of conformal blocks?

2) I find appendix D very interesting. Can this analysis be extended to the case in which Log^2 z or higher trascendental functions appears? (For instance derivatives of D functions). What are the challenges to extend it?

Recommendation

Publish (easily meets expectations and criteria for this Journal; among top 50%)

  • validity: -
  • significance: -
  • originality: -
  • clarity: -
  • formatting: -
  • grammar: -

Author:  Abhishek Navhal  on 2025-10-14  [id 5927]

(in reply to Report 3 on 2025-09-10)
Category:
answer to question

We would like to thank the referee for their careful reading of the manuscript and their useful comments and suggestions, and for pointing out directions for further development. Below, we provide point-by-point responses to the referee's comments.

Referee Comment 1
``How is it possible to use crossing symmetry in this context? In particular, can this help to understand the analytic properties of conformal blocks?''

Response: We thank the referee for this interesting question. We do not (yet) know the answer to this question, but have added the following paragraph to the conclusion section as a tentative suggestion for future work:

{The braiding, and fusions matrices that characterize holomorphic conformal blocks of rational CFTs are well known to obey nontrivial pentagon and hexagon identities [see G. W. Moore and N. Seiberg, Lectures on RCFT]. While the discussion of subsection 3.3 below touched on (analogues of) these matrices, we never had occasion to make nontrivial use of the identities these objects obey. It would be interesting to explore the interplay (if any) of these identities with the study of Lorentzian correlators, along the lines of this paper. }

Referee Comment 2
``I find Appendix D very interesting. Can this analysis be extended to the case in which $Log^2 z$ or higher transcendental functions appear? (For instance, derivatives of D functions). What are the challenges to extend it?''

Response: We thank the referee for their appreciation and for making this point. We agree it would be interesting to study more complicated structures like derivatives of D functions. We do not see a clear obstruction. As our paper focuses mainly on 2d theories, our comments on higher dimensional theories were incidental; we have not performed a systematic analysis of structures more complicated than the simple D functions. We feel this could be an interesting topic for further work, and have added a line at the end of Appendix D, making this point:
{We parenthetically note that the $Li_2(z)$ shares some of the properties of the function ($Log^2 z$), as a single monodromy operation around each of these functions produces a $Log \;z$.}

Once again, we thank the referee for their detailed and thoughtful feedback which has improved our paper. We hope the paper is now suitable for publication.

Sincerely,
Suman Kundu, Shiraz Minwalla, Abhishek Navhal

Attachment:

SciPost_Reply3.pdf

---

## Round 2 · List of Changes

1. Referee 1 comment 1: We have modified the file accordingly. In particular, we have incorporated most of the footnotes (from the Introduction) to the main text. In particular we have merged two of the larger footnotes as a separate subsection 1.5.

  2. Referee 1 comment 2: We have added a paragraph in the Conclusion section (see the second last paragraph), and have thanked the referee for making this point.

  3. Referee 1 comment 3: We have added a paragraph at the end of appendix D: "We parenthetically note that the $D$ function is the result of a correlator computed using contact diagram in the bulk of AdS (via the AdS/CFT correspondence). It would be interesting to perform an analysis similar to this appendix, on the more complicated analytic structures that arise out of exchange or loop computations in the bulk."

  4. Referee 1 comment 4: We agree with the Referee that the comments on this footnote 48 are potentially confusing. In response to the referees comments, we have truncated the footnote, omitting the statement the comment about factorization.

  5. Referee 1 comment 5: We have added a short paragraph (third last paragraph in the conclusions section) making this point and thanking the referee for the suggestion.

  6. Referee 2 comment 1: We have added the following lines at the end of the second and third paragraphs of section 1.3 (in the introduction) to highlight some of the physical implications of our results.

Line added at the end of the second para in section 1.3: {As correlation functions capture the response of a theory to sources, our results are needed (together, of course, with knowledge of the correlators as analytic function of $z$ and ${\bar z}$), to determine the physical response of a CFT on $S^1 \times $ time to arbitrary sources as a function of angle and time. }

Line added at the end of the third para in section 1.3:

  1. Referee 2 comment 2: In response to this question, we have added the following new statement at the end of subsection 1.2: `It is also a disappointment because it tells us that the requirement ofpath independence' is automatic, and imposes no new general constraints on four-point functions of operators with integer spins.''

  2. Referee 3 comment 1: We have added the following paragraph to the conclusion section as a tentative suggestion for future work:

{The braiding, and fusions matrices that characterize holomorphic conformal blocks of rational CFTs are well known to obey nontrivial pentagon and hexagon identities [see G. W. Moore and N. Seiberg, Lectures on RCFT]. While the discussion of subsection 3.3 below touched on (analogues of) these matrices, we never had occasion to make nontrivial use of the identities these objects obey. It would be interesting to explore the interplay (if any) of these identities with the study of Lorentzian correlators, along the lines of this paper. }

  1. Referee 3 comment 2: We feel this could be an interesting topic for further work, and have added a line at the end of Appendix D, making this point: {We parenthetically note that the $Li_2(z)$ shares some of the properties of the function ($Log^2 z$), as a single monodromy operation around each of these functions produces a $Log \;z$.}

---

## Editorial Decision

published